

# Climate Assessment Platform of Different Aircraft Routing Strategies in the Chemistry-Climate Model EMAC 2.41: AirTraf 1.0

Hiroshi Yamashita[1], Volker Grewe[1,2], Patrick Jöckel[1], Florian Linke[3], Martin Schaefer[4], and Daisuke Sasaki[5]

[1]Deutsches Zentrum für Luft- und Raumfahrt, Institut für Physik der Atmosphäre, Oberpfaffenhofen, Germany
[2]also at: Delft University of Technology, Aerospace Engineering, Delft, The Netherlands
[3]Deutsches Zentrum für Luft- und Raumfahrt, Institut für Lufttransportsysteme, Hamburg, Germany
[4]Deutsches Zentrum für Luft- und Raumfahrt, Institut für Antriebstechnik, Köln, Germany, present affiliation: Bundesministerium für Verkehr und digitale Infrastruktur
[5]Kanazawa Institute of Technology, Department of Aeronautics, Hakusan, Japan

*Correspondence to:* H. Yamashita (hiroshi.yamashita@dlr.de)

**Abstract.** Aviation contributes to anthropogenic climate impact through various emissions. Mobility becomes more and more important to society and hence air transportation is expected to grow further over the next decades. Reducing the climate impact from aviation emissions and building a climate-friendly air transportation system are required for a sustainable development of commercial

aviation. A climate optimized routing, which avoids climate sensitive regions by re-routing horizontally and vertically, is an important approach for climate impact reduction. The idea includes a number of different routing strategies (routing options) and shows a great potential for the reduction. To evaluate this, the impact of not only $CO_2$ but also non-$CO_2$ emissions must be considered. $CO_2$ is a long-lived and stable gas, while non-$CO_2$ emissions are short-lived and vary regionally. This

study introduces AirTraf (version 1.0) for climate impact evaluations that performs global air traffic simulations on long time scales, including effects of local weather conditions on the emissions. Air-Traf was developed as a new submodel of the ECHAM5/MESSy Atmospheric Chemistry (EMAC) model. Air traffic information comprises Eurocontrol's Base of Aircraft Data (BADA Revision 3.9) and International Civil Aviation Organization (ICAO) engine performance data. Fuel use and emis-

sions were calculated by the total energy model based on the BADA methodology and DLR fuel flow method. The flight trajectory optimization was performed by a Genetic Algorithm (GA) with respect to routing options. In the model development phase, two benchmark tests were performed for great circle and flight time routing options. The first test showed that the great circle calculations were accurate to within $\pm 0.05$ %, compared to those calculated by other published code. The

second test showed that the optimal solution sufficiently converged to the theoretical true-optimal



solution. The difference in flight time between the two solutions is less than 0.01 %. The dependence of optimal solutions on initial populations was analyzed. We found that the influence was small (around 0.01 %). The trade-off between the accuracy of GA optimizations and the number of function evaluations is clarified and the appropriate population and generation sizing is discussed.

The results showed that a large reduction in number of function evaluations of around 90 % can be achieved with only a small decrease in the accuracy of less than 0.1 %. Finally, one-day AirTraf simulations are demonstrated with the great circle and the flight time routing options for a specific winter day. 103 trans-Atlantic flight plans were used, assuming an Airbus A330-301 aircraft. The results confirmed that AirTraf simulates the air traffic properly for the two options. In addition, the GA

successfully found the time-optimal flight trajectories for all airport pairs, reflecting local weather conditions. The consistency check for the one-day AirTraf simulations verified that calculated flight time, fuel consumption, $NO_x$ emission index and aircraft weights are comparable to reference data.

## 1 Introduction

World air traffic has grown significantly over the past 20 years. With increasing the number of air-

35 craft, the air traffic's contribution to climate change becomes a major problem. At present, aircraft emission impacts (this includes still uncertain aviation-induced cirrus cloud effects) contribute approximately 4.9 % (with a range of 2-14 %, which is a 90 % likelihood range) of total anthropogenic radiative forcing (Lee et al., 2009, Lee et al., 2010, Burkhardt and Kärcher, 2011). An Airbus forecast shows that the world air traffic will grow at an average annual rate of 4.6 % over the next 20 years

(2015-2034, Airbus, 2015), while Boeing forecasts the value of 4.9 % over the same period (Boeing, 2015). This indicates a further increase of aircraft emissions and therefore environmental impacts from aviation increase. Reducing the impacts and building a climate-friendly air transportation system are required for a sustainable development of commercial aviation. The emissions induced by air traffic primarily comprise carbon dioxide ($CO_2$), nitrogen oxides ($NO_x$), water vapor ($H_2O$), carbon

monoxide, unburned hydrocarbons and soot. They lead to changes in the atmospheric composition, thereby changing the greenhouse gas concentrations of $CO_2$, ozone ($O_3$), $H_2O$ and methane ($CH_4$). The emissions also induce cloudiness via the formation of contrail, contrail-cirrus and soot cirrus (Penner et al., 1999).

The climate impact induced by aircraft emissions depends on local weather conditions: it depends

on geographic location (latitude and longitude) and altitude at which the emissions are released (except for $CO_2$) and time. In addition, the impact has different timescales: chemical effects induced by the aircraft emissions have a range of life-times and affect the atmosphere from minutes to centuries. $CO_2$ has a long perturbation life-time in the order of decades to centuries. The atmosphere-ocean system responds to the change in the radiation fluxes in the order of 30 years. $NO_x$, released in

the upper troposphere and lower stratosphere, has a different life-time ranging from a few days to



several weeks, depending on atmospheric transport and chemical background conditions. In some regions, which experience a downward motion, e.g. ahead of a high pressure system, $NO_x$ has short life-times and is converted to $HNO_3$ and then rapidly washed out (Matthes et al., 2012, Grewe et al., 2014b). The most localized and short-lived effect is contrail formation with typical life-times from

minutes to hours. Persistent contrails only form in ice supersaturated regions (Schumann, 1995) and extend a few 100 m vertically and around 150 km horizontally with a large spatial and temporal variability (Spichtinger et al., 2003).

There are two approaches to counteract the climate impact induced by aircraft emissions: technological and operational approaches, as summarized by Irvine et al. (2013). The former includes

aerodynamic improvements of aircraft (Blended-Wing-Body aircraft, laminar flow control, etc.), more efficient engines and alternative fuels (liquid hydrogen, bio-fuels). The latter includes efficient air traffic control (reduced holding time, more direct flights, etc.), efficient flight-profiles (continuous descent approach) and climate-optimized routing. Nowadays, flight trajectories are optimized with respect to time and economic costs (fuel, crew, operating costs) primarily by taking advantage of tail

winds, e.g. jet streams, while the climate-optimized routing should optimize flight trajectories such that released aircraft emissions lead to a minimum climate impact. Earlier studies investigated the effect of systematic routing changes, i.e. flight altitude changes, on the climate impact (Koch et al., 2011, Schumann et al., 2011, Frömming et al., 2012 and Søvde et al., 2014). They confirmed that the changed altitude has a strong effect on the reduction of climate impact. A number of studies have

investigated the potential of applying the climate-optimized routing for real flight data. Matthes et al. (2012) and Sridhar et al. (2013) addressed weather-dependent trajectory optimization using real flight routes and showed a large potential of the climate-optimized routing. As the climate impact of aircraft emissions depends on local weather conditions, Grewe et al. (2014a) optimized flight trajectories by considering regions described as the climate-sensitive regions and showed a trade-off

between climate impact and economic costs. This study reported: "large reductions in the climate impact of up to 25 % can be achieved by only small increase in economic costs of less than 0.5 %." The climate-optimized routing therefore seems to be a useful routing option, however, this option is unused in today's flight planning yet.

This study aims to investigate how much the climate impact of aircraft emissions can be reduced

by aircraft routing. Here, we present a new assessment platform AirTraf (version 1.0, Yamashita et al., 2015) that is a global air traffic submodel coupled to the Chemistry-Climate model EMAC (Jöckel et al., 2010). Figure 1 shows the research road map for this study (Grewe et al., 2014b). The first step is to investigate specific past weather situations, in particular the climate impact of locally released aircraft emissions (Matthes et al., 2012, Grewe et al., 2014b). The resulting data are called climate

cost functions (CCFs, Frömming et al., 2013, Grewe et al., 2014a, Grewe et al., 2014b) that identify climate sensitive regions with respect to $CO_2$, $O_3$, $CH_4$, $H_2O$ and contrails. They are specific climate metrics, i.e. climate impacts per unit emission, and are used for optimal aircraft routings. In a further



step, weather proxies are identified for the specific weather situations, which correlate the intensity of the climate sensitive regions with meteorological data. The proxies will be available from numer-
ical weather forecasts, like temperature, precipitation, ice supersaturated regions, vertical motions or weather patterns in general. These proxies are then used to optimize air traffic with respect to the climate impact expressed by the CCFs. An assessment platform is required to validate the optimization strategy based on the proxies in multi-annual (long-term) simulations and to evaluate the total mitigation gain of the climate impact — one important objective of the AirTraf development.

This paper is organized as follows. Section 2 presents the model description and calculation procedures of AirTraf. Section 3 describes aircraft routing methodologies for great circle and flight time routing options. A benchmark test for the great circle routing option is performed and resulting great circle distances are compared to those calculated by other published code. Another benchmark test is also performed for the flight time routing option. The optimal solution is compared to the true-
optimal solution. The dependence of optimal solutions on initial populations is examined and the appropriate population and generation sizing is discussed. In Sect. 4, one-day AirTraf simulations are demonstrated for the two options and the results are discussed. Section 5 verifies the consistency for the AirTraf simulations and Sect. 6 states the code availability. Finally, Sect. 7 concludes the study.

## 2   AirTraf: air traffic in a climate model

### 2.1   Overview

AirTraf was developed as a submodel of EMAC (Jöckel et al., 2010). This is reasonable, because we perform global air traffic simulations on long time scales considering local weather conditions. Geographic location and altitude at which emissions are released should be also considered. In addition,
various submodels of EMAC can be used to evaluate climate impacts. Therefore, EMAC is a well suited development environment for AirTraf.

Figure 2 shows the flowchart of the AirTraf submodel. First, air traffic data and AirTraf entries are read in `messy_initialize`, which is one of the main entry points of the Modular Earth Submodel System (MESSy, Fig. 2, dark blue). Second, all entries are distributed in parallel following a
distributed memory approach (`messy_init_memory`, Fig. 2, blue): AirTraf is parallelized using the message passing interface (MPI) standard. As shown in Fig. 3, the one-day flight plan is decomposed for a number of processing elements (PEs), here PE is synonym to MPI task, so that each PE has a similar work load, while a whole flight trajectory between an airport pair is handled by the same PE. Third, a global air traffic simulation (AirTraf integration, Fig. 2, light blue) is performed
in `messy_global_end`, i.e. at the end of the time loop of EMAC. Thus, naturally short-term and long-term simulations consider the local weather conditions for every flight in EMAC (AirTraf continuously treats overnight flights). This AirTraf integration is linked to several modules: the aircraft



routing module (Fig. 2, light green) and the fuel/emissions calculation module (Fig. 2, light orange). The former is also linked to the flight trajectory optimization module (Fig. 2, dark green) to calcu-

late flight trajectories corresponding to a selected routing option. The latter calculates fuel use and emissions on the calculated trajectories. Finally, the calculated flight trajectories and global fields are output (Fig. 2, rose red). The results are gathered from all PEs for output of global fields. Other evaluation models, e.g. climate metric models, can easily be integrated into AirTraf and hence the output is used to evaluate the reduction potential of the routing option on the climate impact.

The following assumptions are made in AirTraf (version 1.0): a spherical Earth is assumed (radius is $R_E = 6,371$ km). The aircraft performance model of Eurocontrol's Base of Aircraft Data (BADA Revision 3.9, Eurocontrol, 2011) is used with a constant Mach number $M$ (the Mach number is a velocity divided by a speed of sound). Therefore, true air speed $V_{TAS}$ and ground speed $V_{ground}$ vary along flight trajectories corresponding to a given latitude, longitude, altitude and time. Only

the cruise flight phase is considered, while ground operations, take off, landing and any other flight phases are unconsidered. Potential conflicts of flight trajectories and operational constraints from air traffic control, such as the semi-circular rule and limits rates of aircraft climb and descent, are disregarded. However, a sector demand analysis can be performed on the basis of the output data. The following sections mention the used models briefly, while characteristic procedures of AirTraf

are described in detail.

## 2.2  Chemistry-climate model EMAC

The ECHAM/MESSy Atmospheric Chemistry (EMAC) model is a numerical chemistry and climate simulation system that includes submodels describing tropospheric and middle atmosphere processes and their interaction with oceans, land and human influences (Jöckel et al., 2010). It uses

the second version of the MESSy (i.e. MESSy2) to link multi-institutional computer codes. The core atmospheric model is the $5^{th}$ generation European Centre Hamburg general circulation model (ECHAM5, Roeckner et al., 2006). For the present study we applied EMAC (ECHAM5 version 5.3.02, MESSy version 2.41) in the T42L31ECMWF-resolution, i.e. with a spherical truncation of T42 (corresponding to a quadratic Gaussian grid of approximately 2.8 by 2.8 degrees in latitude and

longitude) with 31 vertical hybrid pressure levels up to 10 hPa (middle of uppermost layer). MESSy provides interfaces (Fig. 2, yellow) to couple various submodels. Further information about MESSy, including the EMAC model system, is available from http://www.messy-interface.org.

## 2.3  Air traffic data

The air traffic data (Fig. 2, dark blue) consist of a one-day flight plan, aircraft and engine perfor-

mance data. Table 1 lists the primary data of A330-301 used for this study. The flight plan includes flight connection information consisting of departure/arrival airport codes, latitude/longitude of the airports, and a departure time. The latitude and longitude coordinates are given as values $[-90, 90]$



and $[-180, 180]$, respectively. Any arbitrary number of flight plans is applicable to AirTraf. The aircraft performance data are provided by BADA Revision 3.9 (Eurocontrol, 2011); the data are re-
quired to calculate the aircraft's fuel flow. As for the engine performance data, four data pairs of reference fuel flows $f_{ref}$ (in $\mathrm{kg(fuel)s^{-1}}$) and corresponding NO$_x$ emission index $\mathrm{EINO}_{x,ref}$ (in $\mathrm{g(NO_x)(kg(fuel))^{-1}}$) at take off, climb out, approach and idle conditions are taken from the ICAO engine emissions databank (ICAO, 2005). An overall weight load factor is also provided by ICAO (Anthony, 2009).

**2.4  Calculation procedures of the AirTraf submodel**

The calculation procedures in the AirTraf integration are described step by step. As shown in Fig. 2 (light blue), a flight status of all flights is initialized as 'non-flight' at the first time step of EMAC. The departure check is then performed at the beginning of every time step. When a flight gets to the time for departure in the time loop of EMAC, its flight status changes into 'in-flight.' The time step
index of EMAC $t$ is introduced here. The index is assigned $t = 1$ to the flight at the departure time. Thereafter the flight moves to flying process (dashed box in Fig. 2, light blue), which mainly comprises four steps (bold-black boxes in Fig. 2, light blue): flight trajectory calculation, fuel/emissions calculation, moving aircraft position and gathering global emissions. The following parts of this section describe these four steps and Figs. 4a to 4d illustrate the respective steps.
The flight trajectory calculation linked to the aircraft routing module (Fig. 2, light green) calculates a flight trajectory corresponding to a routing option. AirTraf will provide seven routing options: great circle (minimum flight distance), flight time (time-optimal), NO$_x$, H$_2$O, fuel (might differ to H$_2$O, if alternative fuel options can be used), contrail and CCF (Frömming et al., 2013, Grewe et al., 2014b). In AirTraf (version 1.0), the great circle and the flight time routing options can be used. The
great circle option is a basis for the other routing options and the module calculates a great circle by analytical formulae, assuming constant flight altitude. In contrast to this, for the other six options, a single-objective minimization problem is solved for a selected option by the linked flight trajectory optimization module (Fig. 2, dark green); this module comprises the Genetic Algorithm (GA, Holand, 1975, Goldberg, 1989) and finds an optimal flight trajectory including altitude changes. For
example, if the flight time routing option is selected, the flight trajectory optimization is applied to all flights taking into account the individual departure times. Generally, a wind-optimal route means an economically optimal flight route taking the most advantageous wind pattern into account. This route minimizes total costs with respect to time and economic costs (fuel, crew and others), i.e. it has multiple objectives. On the other hand, AirTraf distinguishes between the flight time and the
fuel routing options to investigate trade-offs (conflicting scenarios) among different routing options. Thus, the time-optimal route is not always the same as the wind-optimal route. With the contrail option, the best trajectory for contrail avoidance will be found. The CCF is provided by the EU FP7 Project REACT4C (Reducing Emissions from Aviation by Changing Trajectories for the benefit of




Climate, REACT4C, 2014) and estimates total climate impacts due to some aviation emissions (see

Sect. 1). Thus, the best trajectory for minimum CCF will be calculated.

For all routing options, local weather conditions provided by EMAC at $t = 1$ (i.e. at the departure day and time of the aircraft) are used to calculate the flight trajectory. The conditions are assumed to be constant during the flight trajectory calculation. No weather forecasts (or weather archives) are used. Once an optimal flight trajectory is calculated, it is not re-optimized in subsequent time

steps ($t \geq 2$). The detailed flight trajectory calculation methodologies for the great circle and the flight time routing options are described in Sect. 3. After the flight trajectory calculation, the trajectory consists of waypoints generated at even intervals along the trajectory, and flight segments (Fig. 4a). In addition, a number of flight properties are available corresponding to the waypoints, flight segments and the whole trajectory, as listed in Table 2. Here, the waypoint index $i$ is introduced

($i = 1, 2, \cdots, n_{wp}$); $n_{wp}$ is the number of waypoints arranged from the departure airport ($i = 1$) to the arrival airport ($i = n_{wp}$). $i$ is also used as the flight segment index ($i = 1, 2, \cdots, n_{wp-1}$).

Next, the fuel/emissions calculation linked to the fuel/emissions calculation module (Fig. 2, light orange) calculates fuel use, $NO_x$ and $H_2O$ emissions by using a total energy model based on the BADA methodology (Schaefer, 2012) and the DLR fuel flow method (Deidewig et al., 1996, see

Sects. 2.5 and 2.6 for more details). After this calculation, additional flight properties are newly available (see Fig. 4b and Table 2). Note, the flight trajectory calculation described above and this fuel/emissions calculation are performed only once at $t = 1$.

The next step is advancing the aircraft positions along the flight trajectory corresponding to time steps of EMAC (Fig. 4c). Here, aircraft position parameters $pos_{new}$ and $pos_{old}$ are introduced to

indicate a present and previous position of the aircraft along the flight trajectory. They are expressed by real numbers of the waypoint index $i$ (integers), i.e. $real(1, 2, \cdots, n_{wp})$. At $t = 1$, the aircraft is set at the first waypoint ($pos_{new} = pos_{old} = 1.0$). As the time loop of EMAC progresses, the aircraft moves along the trajectory referring to the Estimated Time Over (ETO, Table 2). For example, Fig. 4c shows $pos_{new} = 2.3$ and $pos_{old} = 1.0$ at $t = 2$. This means that the aircraft moves 30 % of the

distance between $i = 2$ and $i = 3$ in one time step. $pos_{new}$ and $pos_{old}$ are stored in the memory and the aircraft continues the flight from $pos_{new} = 2.3$ at the next time step. After the aircraft moves to a new position, the arrival check is performed (dashed box in Fig. 2, light blue). If $pos_{new} \geq real(n_{wp})$, the flight status changes into 'arrived.'

Finally, the individual aircraft's emissions corresponding to the flight path in one time step are

gathered into a global field (three-dimensional Gaussian grid). This step is applied for all flights with 'in-flight' or 'arrived' status. As shown in Fig. 4d, for example, the released $NO_x$ emission along a flight segment $i$ ($NO_{x,i}$ or the fraction of it) is mapped onto the nearest grid point of the global field. For this $NO_{x,i}$, the coordinates of the $(i+1)^{th}$ waypoint is used to find the nearest grid point. In this way, AirTraf calculates the global fields of $NO_x$ and $H_2O$ emissions, fuel use and flight distance for

output. After this step, the flight status check is performed at the end of the flying process. If the



status is 'arrived,' the flight quits the flying process and its status is reset into 'non-flight.' Therefore, the flight status becomes either 'in-flight' or 'non-flight' after the flying process. If $t \geq 2$ of the day (i.e. once the status becomes 'in-flight'), the departure check is false and the aircraft moves to the new aircraft position without re-calculating flight trajectory and fuel/emissions (Fig. 2, light blue).

For more than two consecutive days simulations, the same flight plan is reused: the departure time is automatically updated to the next day and the calculation procedures start from the departure check.

### 2.5 Fuel calculation

The calculation methodologies of the fuel/emissions calculation module (Fig. 2, light orange) are described. Fuel use, $NO_x$ and $H_2O$ emissions are calculated along the flight trajectory obtained from

245 the flight trajectory calculation. A total energy model based on the BADA methodology and the DLR fuel flow method are used. The fuel use calculation consists of the following two steps: the first rough trip fuel estimation and the second fuel calculation (dashed boxes in Fig. 2, light orange). The former estimates an aircraft weight at the last waypoint ($m_{n_{wp}}$), while the latter calculates fuel use for every flight segment and aircraft weights at any waypoint by backward calculation along the

250 flight trajectory, using the $m_{n_{wp}}$ as initial condition.

First, a trip fuel ($FUEL_{trip}$) required for a flight between a given airport pair is roughly estimated:

$$FUEL_{trip} = F_{BADA}FT \tag{1}$$

where $FT$ is the estimated flight time (Table 2) and $F_{BADA}$ is the fuel flow. The BADA performance table provides cruise fuel flow data at specified flight altitudes for three different weights (low, nom-

255 inal and high) under international standard atmosphere conditions. Hence, $F_{BADA}$ is calculated by interpolating the BADA data (assuming nominal weight) to the mean flight altitude of the flight ($\overline{h}$, Table 2). Next, $m_{n_{wp}}$ is estimated by

$$m_{n_{wp}} = OEW + MPL \times OLF + r_{fuel}FUEL_{trip} \tag{2}$$

where OEW, MPL and OLF are given in Table 1. The last term represents the sum of an alternate

fuel, reserve fuel and extra fuel. It is assumed as 3 % of the $FUEL_{trip}$ ($r_{fuel} = 0.03$). The burn-off fuel required to fly from $i = 1$ to $i = n_{wp}$ and contingency fuel are assumed to be consumed during the flight and hence they are not included in $m_{n_{wp}}$. While the 3 % estimation is probably not far from reality for long-range flights, it is worth noting that typical reserve fuel quantities may amount to higher values depending on the exact flight route. Airlines have their own fuel strategy and

information about actual onboard fuel quantities are generally unavailable. A refined fuel estimation will be employed for calculating $m_{n_{wp}}$ in future.

Second, the burn-off fuel is calculated for every flight segment and the aircraft weights are estimated at all waypoints (the contingency fuel is disregarded in AirTraf (version 1.0)). With the BADA total energy model (Revision 3.9), the rate of work done by forces acting on the aircraft is equated





to the rate of increase in potential and kinetic energy:

$$(Thr - D)V_{TAS} = mg\frac{dh}{dt} + mV_{TAS}\frac{dV_{TAS}}{dt} \tag{3}$$

where $Thr$ and $D$ are thrust and drag forces, respectively. $m$ is the aircraft weight, $g$ is the gravity acceleration, $h$ is the flight altitude and $dh/dt$ is the rate-of-climb (or descent). In AirTraf (version 1.0), $dh/dt = 0$ is assumed and $V_{TAS}$ is calculated at every waypoint (Table 2). For an aircraft in

cruise, Eq. (3) becomes $Thr_i = D_i$ at waypoint $i$. To calculate $Thr_i$, the $D_i$ is calculated:

$$C_{L,i} = \frac{2m_i g}{\rho_i V_{TAS,i}^2 S \cos\varphi_i} \tag{4}$$

$$C_{D,i} = C_{D0} + C_{D2}C_{L,i}^2 \tag{5}$$

$$D_i = \frac{1}{2}\rho_i V_{TAS,i}^2 C_{D,i} S \tag{6}$$

where $C_{L,i}$ and $C_{D,i}$ are lift and drag coefficients, respectively. The performance parameters ($S$,

$C_{D0}$ and $C_{D2}$) and the air density $\rho_i$ are given in Tables 1 and 2. The bank angle $\varphi_i$ is assumed to be zero. The thrust specific fuel consumption (TSFC) $\eta_i$ and a fuel flow of the aircraft $F_{cr,i}$ are then calculated assuming a cruise flight for jet aircraft:

$$\eta_i = C_{f1}(1 + \frac{V_{TAS,i}}{C_{f2}}) \tag{7}$$

$$F_{cr,i} = \eta_i Thr_i C_{fcr} \tag{8}$$

where $C_{f1}$, $C_{f2}$ and $C_{fcr}$ are given in Table 1. The fuel use in the $i^{th}$ flight segment ($FUEL_i$) is calculated as

$$FUEL_i = F_{cr,i}(ETO_{i+1} - ETO_i)Oneday, \tag{9}$$

where $ETO_i$ at the $i^{th}$ waypoint (in Julian date) is converted into seconds by multiplying $Oneday$ (Table 2). The $FUEL_i$ reflects the tail/head winds effect on $V_{ground}$ through ETO. The relation be-

tween the $FUEL_i$ and the aircraft weight ($m$) is obtained regarding the $i^{th}$ and $(i+1)^{th}$ waypoints:

$$m_{i+1} = m_i - FUEL_i. \tag{10}$$

Given $m_{n_{wp}}$ by Eq. (2), the fuel use for the last flight segment $FUEL_{n_{wp}-1}$ and the aircraft weight next to the last waypoint $m_{n_{wp}-1}$ can be calculated. This calculation is performed iteratively in

reverse order from the last to the first waypoint using Eqs. (3) to (10). Finally, the aircraft weight at the first waypoint $m_1$ is obtained. As the aircraft weight is pre-calculated in this module, it reduces during the flight as fuel is burnt, corresponding to the time steps of EMAC.

## 2.6  Emission calculation

$NO_x$ and $H_2O$ emissions are calculated after the fuel calculations. $NO_x$ emission under the actual

flight conditions is calculated by the DLR fuel flow method (Deidewig et al., 1996). It depends on the





engine type, the power setting of the engine and atmospheric conditions. The calculation procedure follows four steps: first, the reference fuel flow of an engine under sea level conditions, $f_{ref,i}$, is calculated from the actual fuel flow at altitude, $f_{a,i}$ ($= F_{cr,i}/$(number of engines), see Eq. (8)):

$$f_{ref,i} = \frac{f_{a,i}}{\delta_{total,i}\sqrt{\theta_{total,i}}} \tag{11}$$

$$\delta_{total,i} = \frac{P_{total,i}}{P_0} \tag{12}$$

$$\theta_{total,i} = \frac{T_{total,i}}{T_0} \tag{13}$$

where $\delta_{total,i}$ and $\theta_{total,i}$ are correction factors. $P_{total}$ (in Pa) and $T_{total}$ (in K) are the total pressure

and total temperature at the engine air intake, respectively, and $P_0$ and $T_0$ are the corresponding sea level values (Table 1). $P_{total}$ and $T_{total}$ are calculated as

$$P_{total,i} = P_{a,i}(1 + 0.2M^2)^{3.5} \tag{14}$$
$$T_{total,i} = T_{a,i}(1 + 0.2M^2) \tag{15}$$

where $P_{a,i}$ (in Pa) and $T_{a,i}$ (in K) are the static pressure and temperature under actual flight condi-

315 tions at the altitude $h_i$ (Table 2). Here, $h_i$ is the altitude of the $i^{th}$ waypoint above the sea level (the geopotential altitude is used to calculate $h_i$). The cruise Mach number $M$ is given in Table 1.

Second, the reference emission index under sea level conditions, $EINO_{x,ref,i}$, is calculated using the ICAO engine emissions databank (ICAO, 2005) and the calculated reference fuel flow, $f_{ref,i}$ (Eq. 11). Four data pairs of reference fuel flows $f_{ref}$, and corresponding $EINO_{x,ref}$, are tabulated in

the ICAO databank for a specific engine under sea level conditions. Therefore, $EINO_{x,ref,i}$ values, corresponding to $f_{ref,i}$, are calculated by a Least Squares interpolation ($2^{nd}$-order).

Third, the emission index under actual flight conditions, $EINO_{x,a,i}$ is calculated from the $EINO_{x,ref,i}$:

$$EINO_{x,a,i} = EINO_{x,ref,i} \, \delta_{total,i}^{0.4} \, \theta_{total,i}^3 \, H_{c,i} \tag{16}$$
$$H_{c,i} = e^{(-19.0(q_i - 0.00634))} \tag{17}$$
$$q_i = 10^{-3}e^{(-0.0001426(h_i - 12,900))} \tag{18}$$

where $\delta_{total,i}$ and $\theta_{total,i}$ are defined by Eqs. (12) and (13), respectively. $H_{c,i}$ is the humidity correction factor (dimensionless number) and $q_i$ is the specific humidity at $h_i$ (the unit ft is used here).

Finally, $NO_x$ and $H_2O$ emissions under actual flight conditions are calculated for the $i^{th}$ flight segment using the pre-calculated $FUEL_i$ (Eq. (9)):

$$NO_{x,i} = FUEL_i \, EINO_{x,a,i} \tag{19}$$
$$H_2O_i = FUEL_i \, EIH_2O \tag{20}$$

where the $H_2O$ emission index is $EIH_2O = 1,230 \, g(H_2O)(kg(fuel))^{-1}$ (Penner et al., 1999). The $H_2O$ emission is proportional to the fuel use, assuming an ideal combustion of jet fuel. The $NO_x$ and $H_2O$ emissions are included in the flight properties (Table 2).



With regard to the reliability of the fuel/emissions calculation using these methods, Schulte et al. (1997) showed a comparison of measured and calculated $EINO_x$ for some aircraft/engine combinations. The study gave some confidence in the prediction abilities of the DLR method, although it showed that the calculated values from the DLR method underestimated the measured values on average by 12 %. In Section 5 we verify the methods, using one-day AirTraf simulation results. De-

tailed descriptions of the total energy model and the DLR fuel flow method can be found elsewhere (Eurocontrol, 2011, Deidewig et al., 1996).

## 3 Aircraft routing methodologies

The current aircraft routing module (Fig. 2, light green) works with respect to the great circle and flight time routing options. These routing methodologies are described in Sects. 3.1 and 3.2. Bench-

345 mark tests are performed off-line (without EMAC) to verify the accuracy of the methodologies.

### 3.1 Great circle routing option

#### 3.1.1 Formulation of great circles

AirTraf calculates a great circle at any arbitrary flight altitude with the great circle routing option. First, the coordinates of the waypoints are calculated. For the $i^{th}$ and $(i+1)^{th}$ waypoints, the central

angle $\Delta\hat{\sigma}_i$ ($i = 1, 2, \cdots, n_{wp} - 1$) is calculated by the Vincenty formula (Vincenty, 1975):

$$\Delta\hat{\sigma}_i = arctan\left(\frac{\sqrt{(cos\phi_{i+1}sin(\Delta\lambda_i))^2 + (cos\phi_i sin\phi_{i+1} - sin\phi_i cos\phi_{i+1}cos(\Delta\lambda_i))^2}}{sin\phi_i sin\phi_{i+1} + cos\phi_i cos\phi_{i+1}cos(\Delta\lambda_i)}\right) \quad (21)$$

where $\phi_i$ (in rad) is the latitude of the $i^{th}$ waypoint and $\Delta\lambda_i$ (in rad) is the difference in longitude between the $i^{th}$ and $(i+1)^{th}$ waypoints. The Vincenty formula was set as the default method, while optionally the spherical law of cosines or the Harvesine formula can be used in AirTraf to calculate

$\Delta\hat{\sigma}$ (unshown). With Eq. (21), the great circle distance for the $i^{th}$ flight segment $d_i$ is calculated:

$$d_i = (R_E + h_i)\Delta\hat{\sigma}_i \quad (22)$$

or

360 $$d_i = \sqrt{(R_E + h_i)^2 + (R_E + h_{i+1})^2 - 2(R_E + h_i)(R_E + h_{i+1})cos(\Delta\hat{\sigma}_i)}. \quad (23)$$

For the great circle routing option, flight altitudes at all waypoints are set as $h_i$ = constant for $i = 1, 2, \cdots, n_{wp}$ (km in Eqs. (22) and (23)) and either Eq. (22) or Eq. (23) is used to calculate $d_i$. Equation (22) calculates $d_i$ by an arc and hence the great circle distance between airports, i.e. the $\sum_{i=1}^{n_{wp}-1} d_i$ is independent of $n_{wp}$. On the other hand, Eq. (23) calculates $d_i$ by linear interpolation

based on Polar coordinates. Therefore, $\sum_{i=1}^{n_{wp}-1} d_i$ depends on $n_{wp}$; the sum becomes close to that calculated from Eq. (22) with increasing $n_{wp}$. If AirTraf simulation results with the great circle





option are compared to those with other routing options, Eq. (23) should be used for the comparison with the same $n_{wp}$. In addition, Eq. (23) is used for the flight trajectory optimization (see Sect. 3.2), because it is necessary to calculate $d_i$ including altitude changes.

Next, the true air speed $V_{TAS}$ and the ground speed $V_{ground}$ of the $i^{th}$ waypoint are calculated:

$$V_{TAS,i} = Ma_i = M\sqrt{\gamma RT_i} \tag{24}$$

$$V_{ground,i} = V_{TAS,i} + V_{wind,i} \tag{25}$$

where $M$ is Mach number, $\gamma$ is the adiabatic gas constant and $R$ is the gas constant for dry air
(Table 1). Temperature $T_i$ and three dimensional wind components $(u_i, v_i, w_i)$ of the $i^{th}$ waypoint are available from the EMAC model fields at $t = 1$; the local speed of sound $a_i$ is then calculated (Table 2). The flight direction is calculated for every flight segment by using the three dimensional coordinates of the $i^{th}$ and $(i+1)^{th}$ waypoints. Thereafter, $V_{TAS,i}$, $V_{wind,i}$ and $V_{ground,i}$ (scalar values) corresponding to the flight direction are calculated. As shown in Eq. (25), the influence
of tail/head winds on ground speed is considered. In AirTraf, $M$ was set constant as default. It is also possible to perform AirTraf simulations with different options, such as $V_{TAS,i} = $ constant and $V_{wind,i} = 0$. Finally, $ETO_i$ (in Julian date) and $FT$ (in s) are calculated as

$$ETO_i = ETO_{i-1} + \frac{d_{i-1}}{V_{ground,i-1} \times Oneday} \ (i = 2, 3, \cdots, n_{wp}) \tag{26}$$

$$FT = (ETO_{n_{wp}} - ETO_1) \times Oneday \tag{27}$$

where $ETO_1$ is the departure time of the flight and $ETO_i$ reflects the influence of tail/head winds on the flight.

### 3.1.2 Benchmark test on great circle calculations

A benchmark test of the great circle routing option was performed to confirm the accuracy of the
great circle distance calculation. Great circles were calculated for the five representative routes without EMAC (off-line). Table 3 shows the information for the five routes (the locations are shown in Fig. 5). The characteristics of the routes were as follows: R1 consisted of an airport pair in the northern hemisphere (MUC-JFK) and the difference in longitude between them was $\Delta\lambda_{airport} < 180$ (in deg); R2 consisted of an airport pair in the northern hemisphere (HND-JFK) with $\Delta\lambda_{airport} > 180$
(discontinuous longitude values due to the definition of the longitude range $[-180, 180]$); R3 consisted of an airport pair in the northern and southern hemispheres (MUC-SYD); R4 was a special route, where $\Delta\lambda_{airport} = 0$ and the difference in latitude was $\Delta\phi_{airport} \neq 0$ deg; and R5 was another special route with $\Delta\lambda_{airport} \neq 0$ and $\Delta\phi_{airport} = 0$. Other calculation conditions were set as follows: $M = 0.82$; $h_i = 0$, $a_i = 304.5 \ \text{ms}^{-1}$ and $V_{TAS,i} = V_{ground,i} = 249.7 \ \text{ms}^{-1}$ (under no-
wind conditions, i.e. $V_{wind,i} = 0$) for $i = 1, 2, \cdots, n_{wp}$. The great circle distances $\sum_{i=1}^{n_{wp}-1} d_i$ were each calculated by Eqs. (22) and (23), and were compared to that calculated with the Movable type





script (MTS, Movable type script, 2014). In addition, the sensitivity of the great circle distance with respect to $n_{wp}$ was analyzed varying $n_{wp}$ in $[2, 100]$.

Table 4 shows the calculated great circle distances by Eqs. (22) and (23) and the MTS. The columns 5 to 7 show the difference in the distance among them (see caption of Table 4 for more details). The results showed that $\Delta d_{eq23,eq22}$, $\Delta d_{eq23,MTS}$ and $\Delta d_{eq22,MTS}$ varied between $-0.0036$ and $-0.0008$ %, between $-0.0435$ and $0.0054$ %, and between $-0.0463$ and $0.0046$ %, respectively. The great circle distances calculated by Eqs. (22) and (23) were accurate to within $\pm 0.05$ % and hence this routing option works properly. Figure 6 shows the result of the sensitivity analysis of $n_{wp}$ on the great circle distance. The results showed that the distance calculated by Eq. (22) (open circle) has no dependence on $n_{wp}$ as noted in Sect. 3.1.1, whereas that by Eq. (23) (closed circle) depends on $n_{wp}$ and converged with increasing $n_{wp}$: the accuracy of the results by Eq. (23) decreased when using fewer $n_{wp}$, as a result of the linear interpolation. For $n_{wp} \geq 20$, the results of Eqs. (22) and (23) were almost the same. Therefore, $n_{wp} \geq 20$ is practically desired for the use of Eq. (23).

### 3.2 Flight time routing option

#### 3.2.1 Overview of the Genetic Algorithm

The flight trajectory optimization with respect to the flight time routing option was performed using GA (Holand, 1975, Goldberg, 1989), which is a stochastic optimization algorithms. The Aircraft routing module (Fig. 2, light green) is linked to the flight trajectory optimization module (Fig. 2, dark green); this optimization module consists of the Adaptive Range Multi-Objective Genetic Algorithm (ARMOGA version 1.2.0) developed by D. Sasaki and S. Obayashi (Sasaki et al., 2002, Sasaki and Obayashi, 2004, Sasaki and Obayashi, 2005). The ARMOGA will be implemented as part of the MESSy infrastructure in the next version of MESSy so that it can be used for optimization problems by other submodels as well. With a routing option except for the great circle routing option, a single-objective optimization problem on the selected routing option is solved. The main advantage of GA is that GA requires neither the computation of derivatives or gradients of functions, nor the continuity of functions. Therefore, various objective functions can easily be adapted to GA. As for the working principle of GA, a random initial population is created and the population evolves over generations to adapt to an environment by the genetic operators: evaluation, selection, crossover and mutation. When this biological evolutionary concept is applied for design optimizations, fitness, individuals and genes correspond to an objective function, solutions and design variables, respectively. A solution found in GA is called optimal solution, whereas a solution having the theoretical-optimum of the objective function is called true-optimal solution. If GA works properly, it is expected that the optimal solution converges to the true-optimal solution. On the other hand, the main disadvantage of GA is that GA is computationally expensive. The flight trajectory optimization is applied for all flights and therefore a user has to choose appropriate GA parame-





### 3.2.2 Formulation of flight trajectory optimization

The flight trajectory optimization is described focusing on geometry definitions of the flight trajectory, the definition of objective function and the genetic operators. There exists a number of selection, crossover and mutation operators in ARMOGA. Therefore, the genetic operators employed in this study are described here.

A solution $x$ (the term is used interchangeably to mean a flight trajectory) is a vector of $n_{dv}$ design
variables: $x = (x_1, x_2, \cdots, x_{n_{dv}})^T$. Using the design variable index $j$ $(j = 1, 2, \cdots, n_{dv}; n_{dv} = 11)$, the $j^{th}$ design variable varies in lower/upper bounds $[x_j^l, x_j^u]$. GA searches for the optimal solution, corresponding to the routing option, around the great circle of an airport pair including altitude changes. Figure 7 shows the geometry definition of a flight trajectory from MUC to JFK as an example: the geographic location (bottom) with three control points (CPs, black circles) and the
longitude vs altitude (top) with five CPs. The coordinates of the airports were given from a flight plan (Fig. 2, dark blue) and were fixed (the coordinates of MUC and JFK are shown in Table 5).

Six design variables $x_j(j = 1, 2, \cdots, 6)$ were used for location, as shown in Fig. 7 (bottom). To create three rectangular domains for the design variables (dashed boxes), central points of the domains (diamond symbols) were calculated. The points are located on the great circle, dividing the
455 longitude distance between MUC and JFK ($\Delta\lambda_{airport}$) into four equal parts. After that, the three domains centering around the central points were created. The domain size was set to $0.1 \times \Delta\lambda_{airport}$ (short-side) and $0.3 \times \Delta\lambda_{airport}$ (long-side). This procedure calculates the lower/upper bounds of the six design variables, i.e. $[x_j^l, x_j^u]$ $(j = 1, 2, \cdots, 6)$, and Table 6 lists these values. GA provided the values for $x_1$ to $x_6$ within the respective bounds (i.e. the values were generated within the rectangu-
460 lar domains) and the coordinates of the three CPs were determined: CP1 $(x_1, x_2)$, CP2 $(x_3, x_4)$ and CP3 $(x_5, x_6)$. Here $x_1, x_3$ and $x_5$ indicate longitudes, while $x_2, x_4$ and $x_6$ indicate latitudes. A flight trajectory is represented by a B-spline curve with the three CPs as location (bold solid line, Fig. 7 bottom) and then any arbitrary number of waypoints is generated along the trajectory. To generate the waypoints at even intervals, $n_{wp}$ was calculated as $mod(n_{wp} - 1, n_{CP_{loc}} + 1) = 0$, where the
465 number of CPs was $n_{CP_{loc}} = 3$.

For the altitude direction, five design variables $x_j(j = 7, 8, \cdots, 11)$ were used (Fig. 7, top). With the lower $h^l$ and the upper $h^u$ variable bound parameters, the bounds of the five design variables were determined by $x_j^l = h^l$ and $x_j^u = h^u$ for $j = 7, 8, \cdots, 11$. In this study, $h^l = $ FL290 and $h^u = $ FL410, as listed in Table 6 ('FL290' stands for a flight level at 29,000 ft). These altitudes correspond
to a general cruise flight altitude range of commercial aircraft (Sridhar et al., 2013). GA provided the values of $x_7$ to $x_{11}$ in [FL290, FL410] and the coordinates of the five CPs were determined: CP4 $(x_7)$, CP5 $(x_8)$, CP6 $(x_9)$, CP7 $(x_{10})$ and CP8 $(x_{11})$. Here $x_7$ to $x_{11}$ indicate altitude values. The



longitude-coordinates of the five CPs were pre-calculated to divide the $\Delta\lambda_{airport}$ into six equal parts. The altitude of the airports were fixed at $h^l$ (= FL290). A flight trajectory is also represented by a B-
spline curve with the five CPs as longitude vs altitude (bold solid line, Fig. 7 top) and then waypoints are generated along the trajectory. Note, GA creates trajectories represented by two B-splines, one latitude vs longitude and one longitude vs altitude, where longitude-coordinate of waypoints is the same for the two curves.

The initial population operator (Fig. 2, dark green) provides initial values of the eleven design vari-
ables by random numbers, thereby creating solutions. The operator creates diverse solutions defined by a fixed population size $n_p$ and GA starts its search with a random set of solutions (population-approach). To evaluate the solutions, the objective function $f$ was calculated for each of the solutions by summing the flight time for flight segments (Fig. 2, dark green). The single-objective optimization solved here is as follows:

$$\left.\begin{array}{l} Minimize \ \ f = \displaystyle\sum_{i=1}^{n_{wp}-1} \frac{d_i}{V_{ground,i}} \\[2mm] Subject \ to \ \ x_j^l \le x_j \le x_j^u, \ \ j = 1, 2, \cdots, n_{dv} \end{array}\right\}, \tag{28}$$

where $d_i$ and $V_{ground,i}$ are calculated by Eqs. (23) and (25), respectively ($V_{TAS,i}$ and $V_{wind,i}$ are calculated as described in Sect. 3.1.1). No constraint function is used in AirTraf (version 1.0).

Good solutions are identified in the population by the Fonseca and Fleming's pareto ranking method (Fonseca et al., 1993), although the single-objective optimization is solved here. A $rank$
of a solution was assigned proportional to the number of solutions that dominate it, and a fitness value of a solution was computed by $1/rank$ (no fitness sharing was used). A solution with higher fitness value (i.e. smaller $rank$ value) has a higher probability of being copied into a mating pool. The Stochastic Universal Sampling Selection (Baker, 1985) made duplicates of good solutions in the mating pool at the expense of bad solutions based on cumulative probability values, while keeping
the size of $n_p$.

To create a new solution, the Blend crossover (BLX-$\alpha$) operator (Eshelman, 1993) was applied to the population in the mating pool. Two solutions (parent solutions) were picked from the mating pool at random and the operator created two new solutions (child solutions):

$$\left.\begin{array}{l} x_{j,c1} = \gamma x_{j,p1} + (1-\gamma)x_{j,p2} \\[2mm] x_{j,c2} = (1-\gamma)x_{j,p1} + \gamma x_{j,p2} \end{array}\right\} \tag{29}$$

with $\gamma = (1+2\alpha)u_1 - \alpha$ and $j$ varies in $[1, n_{dv}]$. This operator was applied to each design variable; $n_{dv} = 11$. $x_{j,c1}$ and $x_{j,c2}$ denote the $j^{th}$ design variable of the child solutions, and $x_{j,p1}$ and $x_{j,p2}$ denote the $j^{th}$ design variables of the parent solutions (the mated pair of the old generation). $\alpha$ is an user-specified crossover parameter and $u_1$ is a random number between zero and one.

Thereafter, the mutation operator added a disturbance to the child solution by the revised polyno-
mial mutation operator (Deb and Agrawal, 1999) with a mutation rate $r_m$. A polynomial probability





distribution was used and the mutated design variable was created. The parameter $\delta_q$ is first calculated as

$$\delta_q = \begin{cases} [2u_2 + (1-2u_2)(1-\delta)^{\eta_m+1}]^{\frac{1}{\eta_m+1}} - 1, & \text{if } u_2 \le 0.5, \\ 1 - [2(1-u_2) + 2(u_2-0.5)(1-\delta)^{\eta_m+1}]^{\frac{1}{\eta_m+1}}, & \text{if } u_2 > 0.5, \end{cases} \tag{30}$$

where $\delta = min[(x_{j,c} - x_j^l), (x_j^u - x_{j,c})]/(x_j^u - x_j^l)$. The $j^{th}$ design variable varies in $[x_j^l, x_j^u]$. $u_2$ is a
random number between zero and one, and $\eta_m$ is an external parameter controlling the shape of the probability distribution. The mutated design variable (mutated child solution) $x_{j,mc}$ is calculated as follows:

$$x_{j,mc} = x_{j,c} + \delta_q(x_j^u - x_j^l), \quad j = 1, 2, \cdots, n_{dv}. \tag{31}$$

Using the genetic operators above, it is expected that the population of the solutions is improved and a new and better population is created in subsequent generations. When the evolution is computed for a fixed generation number $n_g$, GA quits the optimization and an optimal solution is output corresponding to the routing option. The optimal solution has the best combination of the eleven design variables $\boldsymbol{x} = (x_1, x_2, \cdots, x_{11})^T$ to minimize $f$. Naturally, the flight properties of the optimal
solution are available (ETO, $\overline{h}$, $FT$, etc. listed in the first and the second groups (divided by rows) of Table 2). The flight trajectory optimization methodology described here can be applied to any routing option (except for the great circle routing option). In that case, the objective function $f$ given by Eq. (28) needs to be reformulated corresponding to the selected routing option.

### 3.2.3 Benchmark test on flight trajectory optimization with flight time routing option

To quantify the performance of GA, there is a need to choose an appropriate benchmark test of the flight trajectory optimization, where the true-optimal solution $f_{true}$ of the test is known. Here, the single-objective optimization for minimization of flight time from MUC to JFK was solved without EMAC (off-line), that is, the optimization problem defined in Sect. 3.2.2 was solved. Calculation conditions for the test are summarized in Table 5. As $V_{TAS}$ and $V_{ground}$ were set to 898.8 $\text{kmh}^{-1}$
(constant) under no-wind conditions, the $f_{true}$ equals the flight time along the great circle from MUC to JFK at FL290: $f_{true} = 25,994.0$ s calculated by Eq. (23) with $h_i = $ FL290 for $i = 1, 2, \cdots, 101$. With regard to the dependence of the optimal solutions on initial populations, 10 independent GA simulations from different initial populations were performed. In these simulations, both $n_p$ and $n_g$ were set to 100, while other calculation conditions were set as shown in Table 5. In the same way,
to discuss an appropriate $n_p$ and $n_g$ sizing, 10 independent GA simulations from different initial populations were performed for each combination of $n_p$ $(10, 20, \cdots, 100)$ and $n_g$ $(10, 20, \cdots, 100)$, i.e. total 1,000 independent GA simulations were performed. Other calculation conditions were also set as shown in Table 5.



### 3.2.4 Optimization results

The influence of the population size $n_p$ and the generation number $n_g$ on the convergence properties of GA was confirmed. Figure 8 shows the optimal solutions varying with $n_g$ for a number of fixed $n_p$. The results confirmed that the optimal solutions sufficiently come close to the $f_{true}$ with increasing $n_p$ and $n_g$. The optimal solution showing the closest flight time to the $f_{true}$ was obtained for $n_p = 100$ and $n_g = 100$. This solution is called best solution in this study and its flight time was $f_{best} =$

$25,996.6$ s. The difference in flight time between the $f_{best}$ and the $f_{true}$ was $\Delta f < 3.0$ s (less than $0.01$ %).

   To confirm the diversity of GA optimization, we focus on the optimization results, which found the best solution ($n_p = 100$ and $n_g = 100$). Figure 9 shows all the solutions explored by GA as longitude vs altitude (top) and as location (bottom). It is clear that GA explored diverse solutions from MUC to JFK including altitude changes and found the best solution. As shown in Fig. 9, the best solution

(red line) overlapped with the true-optimal solution, i.e. great circle at FL290 (dashed line, black). To confirm the difference between the solutions, the comparison of trajectories for the best solution and the true-optimal solution as longitude vs altitude are plotted in Fig. 10. The maximum difference in altitude is less than 1 m. Therefore, GA is adequate for finding an optimal solution with sufficient

accuracy.

### 3.2.5 Dependence of initial populations

   To confirm the dependence of optimal solutions on initial populations, Fig. 11 shows the best-of-generation flight time vs the number of objective function evaluations ($= n_p \times n_g$) corresponding to the 10 independent GA simulations with $n_p = 100$ and $n_g = 100$. Figure 11 shows that the 10

solutions converged in early generations and gradually continued to converge to $f_{true}$ with increasing number of function evaluations. The convergence behavior is similar among the 10 simulations, regardless of the initial population. Table S1 in the Supplement shows a summary of the 10 optimal solutions. As indicated in Table S1, there is a small degree of variation in the objective function $f$ (= flight time). $\Delta f (= f - f_{true})$ ranged from 2.5 to 3.7 s, which is approximately 0.01 % of $f_{true}$. In

addition, the mean value of the 10 objective functions was $\overline{\Delta f} = 2.9$ s (0.01 % of the $f_{true}$) and the standard deviation was $s_{\Delta f} = 0.4$ s (0.001 % of the $f_{true}$). Therefore, the variation in the objective function with different initial populations is small.

### 3.2.6 Poplulation and generation sizing

   With an increase in number of $n_p$ and $n_g$, GA can discover an improved solution. It is important to

note that the required size of $n_p$ and $n_g$ is problem-dependent, e.g. weather situations, and therefore estimating appropriate $n_p$ and $n_g$ could be different. However, following a simple initial guess for $n_p$ and $n_g$ is a good starting point for their sizing.



The influence of $n_p$ and $n_g$ on the accuracy of GA optimizations and on the variation in the optimal solution due to different initial populations were analyzed. Figure 12 shows the calculated $\overline{\Delta f}$ and $s_{\Delta f}$ for all the combinations of $n_p$ and $n_g$. The results confirm that $\overline{\Delta f}$ and $s_{\Delta f}$ decrease with an increase of $n_p$ and $n_g$. That is, the optimal solution converges to the true-optimal solution (the accuracy increases) and the variation in the optimal solution due to different initial populations decreases (the dependency decreases).

On the other hand, computational costs also should be kept as low as possible for practical use of EMAC/AirTraf (on-line) applied to long-term global air traffic simulations. Figure 13 shows the variation of the $\overline{\Delta f}$ and the $s_{\Delta f}$ for all combinations of $n_p$ and $n_g$ with respect to the number of function evaluations. The symbols and error bars in the figure correspond to the $\overline{\Delta f}$ and $s_{\Delta f}$, respectively (Table S2 in the Supplement lists these values). The results showed that there is a trade-off between the accuracy of GA optimizations and the number of function evaluations (i.e. computing time). The figure also shows the power function (red line) fitted to the results by using the standard least-squares algorithm (see caption in Fig. 13 for more details). As shown in the enlarged drawing in Fig. 13, the large reduction in number of function evaluations of 92 % can be achieved, keeping $\overline{\Delta f}$ less than 0.05 % ($s_{\Delta f} \approx 0.02$ %), compared to the optimal solution obtained by 10,000 function evaluations ($n_p = 100$ and $n_g = 100$). Similarly, that reduction of 97 % can be achieved, keeping $\overline{\Delta f}$ less than 0.1 % ($s_{\Delta f} \approx 0.04$ %). Therefore, computational costs can be reduced drastically by selecting $n_p$ and $n_g$ for different purposes.

## 4 Demonstration of a one-day AirTraf simulation

The aircraft routing methodologies corresponding to the great circle and flight time routing options were verified in Sect. 3. Here, one-day AirTraf simulations were performed in EMAC (on-line) with the respective routing options for demonstrations.

### 4.1 Calculation conditions

We focus on the trans-Atlantic region for the demonstration, because the optimization potential is possibly large for this region. Table 7 lists the calculation conditions for the one-day simulations. The simulation was performed for one specific winter day in the T42L31ECMWF-resolution. The weather situation on that day showed a typical weather pattern for winter characterized by westerly jet streams in the North-Atlantic region. The number of trans-Atlantic flights in the region was 103 (52 eastbound flights and 51 westbound flights). We assumed that all flights were operated by A330-301 aircraft with CF6-80E1A2 (2GE051) engines. Thus, the data shown in Table 1 were used. Four one-day simulations were separately performed for the great circle routing option at fixed altitudes FL290, FL330, FL370 and FL410 (see Sect. 3.1.1). On the other hand, a single one-day simulation was performed for the flight time routing option including altitude changes in [FL290,



FL410] (see Sect. 3.2.2). For the two options, the Mach number was set to $M = 0.82$ and therefore $V_{TAS}$ and $V_{ground}$ varied along the waypoints (Eqs. (24) and (25)). The number of waypoints was set to $n_{wp} = 101$. As described in Sect. 3.1.1, the flight distance was calculated by Eq. (23) for the
610 two routing options. The optimization parameters were set as follows: $n_p = 100$, $n_g = 100$ and other GA parameters were the same as those used in the benchmark test in Sect. 3.2.3.

The one-day simulation was parallelized on 4 PEs of Fujitsu Esprimo P900 (Intel Core i5-2500CPU with 3.30 GHz; 4 GB of memory; peak performance of $105.6 \times 4$ GFLOPS) at the Institute of Atmospheric Physics, German Aerospace Center. The one-day simulation required approximately 15
615 min for a great circle case, while it took approximately 20 hours for a time-optimal case. Most of the computational time is consumed by the trajectory optimizations. Therefore it can be reduced by choosing all GA parameters right, using more PEs, or decreasing $n_p$ and $n_g$. As discussed in Sect. 3.2.6, a large reduction in computing time of roughly 90 % can be achieved by a small number of $n_p$ and $n_g$ with sufficient accuracy of the optimizations.

**4.2 Optimal solutions for three selected airport pairs**

The one-day simulation results for the flight time routing option confirmed that the optimized flight trajectories showed a large altitude variation. To give an overview of the optimizations, we classified the trajectories according to their altitude changes into three categories. Type I: east- and westbound time-optimal flight trajectories showed little altitude changes, Type II: eastbound time-optimal flight
trajectory showed little altitude changes, while westbound time-optimal flight trajectory showed distinct altitude changes, and Type III: east- and westbound time-optimal flight trajectories showed distinct altitude changes. We have selected the three airport pairs of each type and Table 8 shows the details of them. Here, we mainly discuss the selected solution of Type II, which were east- and westbound flights between Minneapolis (MSP) and Amsterdam (AMS).

We examined first the optimal flight trajectories between MSP and AMS. Figure 14 shows all trajectories explored by GA (black lines) and the time-optimal flight trajectories for east- and westbound flights (red and blue lines). Figures 14a and 14b show that GA explored diverse trajectories properly considering altitude changes in [FL290, FL410]. Similar results were obtained when calculating for the selected solutions of Type I and III, as shown in Figs. S1 and S2 in the Supplements.
In addition, the eastbound time-optimal flight trajectory was located at FL290, while that for westbound showed large altitude changes, i.e. it climbed, descended and climbed again. The mean flight altitude of these trajectories were $\overline{h} = 8,839$ m and $\overline{h} = 10,002$ m. These time-optimal flight trajectories were compared to the prevailing wind fields. To calculate tail/head winds in east and west directions, the major wind component is shown in Fig. 15. The contours represent the zonal wind
speed ($u$); black arrows show the wind speed ($\sqrt{u^2 + v^2}$) and direction at the departure time at the $\overline{h}$. Figures 15a and 15b show that the eastbound time-optimal flight trajectory (red line) was located to the south of the great circle (black line) to take advantage from the tail winds of the westerly





jet stream (red region), while the westbound time-optimal flight trajectory (blue line) was located to the north of the great circle to avoid the head winds (red region). Similar comparisons for the selected solutions of Type I and III showed that the obtained optimal flight trajectories effectively take advantages of the wind fields (see Supplements, Figs. S3 and S4).

To understand the behavior of altitude changes of the optimal flight trajectories, Fig. 16 plots the altitude distribution of the true air speed ($V_{TAS}$) and the tail wind indicator ($V_{ground}/V_{TAS}$) along the time-optimal flight trajectories. The indicator was calculated by Eq. (25) transformed into $V_{ground}/V_{TAS} = 1 + V_{wind}/V_{TAS}$; this means tail winds ($\geq 1.0$) and head winds ($< 1.0$) to the flight direction. Figure 16c shows that the core tail winds region was located at 8.5 km and the tail winds were most beneficial for the eastbound flight trajectory. On the other hand, the westbound flight trajectory went through the regions where $V_{TAS}$ was high, as shown in Fig. 16b. In addition, Fig. 16d shows that the descent at a flight time of 16,000 s was effective to counteract the head winds. These results confirm that GA correctly reflects the weather conditions and finds the appropriate flight trajectories corresponding to the flight direction. Similar results were obtained for the solutions of Type I and III (see Supplements, Figs. S5 and S6).

Next, we confirmed the resulting flight time quantitatively for the selected solutions. Table 8 shows the obtained flight times for the time-optimal and the great circle cases. As indicated in Table 8, the flight time decreased for the time-optimal case compared to the great circle cases. In addition, the flight time decreased for the eastbound time-optimal flight trajectories compared to that for the westbound time-optimal flight trajectories. This supports the observation that GA correctly reflects weather conditions for the trajectory optimization. With regard to the convergence behavior of the optimization, Fig. 17 shows the best-of-generation flight time vs the number of objective function evaluations corresponding to the GA simulations for the three selected airport pairs. As expected, the solutions sufficiently converged to each optimal solution. Thus, GA successfully found the time-optimal flight trajectories for the three airport pairs. It is also clear from Fig. 17 that the reduction in computing time can be achieved by sizing $n_p$ and $n_g$, although the solutions converged more slowly under the wind conditions than those under no-wind conditions (Fig. 13).

### 4.3 One-day simulation results for all flights

The one-day AirTraf simulations for 103 trans-Atlantic flights are discussed. Figure 18 shows the obtained flight trajectories for the flight time and great circle routing options. Figures 18a and 18c show that many eastbound time-optimal flight trajectories congregated around 50°N over the trans-Atlantic Ocean to take advantage from the tail winds in the westerly jet stream. On the other hand, the westbound time-optimal flight trajectories were located to the north and south of the region to avoid head winds (as shown in Figs. 18b and 18d). In addition, Figs. 18a and 18b show that only 5 of 52 eastbound time-optimal flight trajectories showed large altitude changes, in comparison to 35



of 51 westbound time-optimal flight trajectories. The mean flight altitude for the 52 eastbound, 51 westbound and total 103 flights were $\overline{h} = 9{,}029$ m, 9,517 m and 9,271 m, respectively.

As shown in Fig. 16, altitude changes were due to variations of $V_{TAS}$ and prevailing winds. We now confirm this behavior, focusing on the results for all flights. Figures 19a and 19b plot the values of $V_{TAS}$ and $V_{ground}/V_{TAS}$ at waypoints for the time-optimal and the great circle flights, with linear fitted lines. Figure 19a shows that $V_{TAS}$ increased at low altitudes. From Eq. (25), high $V_{TAS}$ values increase $V_{ground}$ values, thereby minimizing flight time. The mean $V_{TAS}$ for the time-

optimal and the great circle cases are shown in Table 9. The mean $V_{TAS}$ value (column 4) for the time-optimal case is 245.1 ms$^{-1}$, while that for the great circle cases ranges from 241.2 to 244.9 ms$^{-1}$, although the mean flight altitude for the time-optimal case is $\overline{h} = 9{,}271$ m, which is higher than FL290 $(= 8{,}839$ m$)$. GA successfully found the flight trajectories, which had high $V_{TAS}$ values, as time-optimal flights.

With regard to the wind effects, Fig. 19b shows that the fitted line for the eastbound time-optimal case (solid line, red) increases between FL290 $(= 8{,}839$ m$)$ and 9,500 m compared to that for the eastbound great circle case (dashed line, red). These altitude bounds are effective under the present weather condition to take advantage of tail winds for the eastbound flights. Thus, almost all the eastbound time-optimal flight trajectories were located at FL290, as shown in Fig. 18a (top). On

the other hand, the fitted line for the westbound time-optimal case (solid line, blue) is distributed widely in altitude and increases between FL290 $(= 8{,}839$ m$)$ and 12,000 m compared to that for the westbound great circle case (dashed line, blue). The westbound time-optimal flight trajectories certainly mitigated the head winds effect. Thus, many westbound time-optimal flight trajectories showed large altitude changes, as shown in Fig. 18b (top). The similar plot of $V_{ground}$ is shown in

the Supplement (Fig. S7), which reflects the influences of both $V_{TAS}$ and winds; the plot indicates similar trends as shown in Fig. 19b. Table 9 also shows that the mean $V_{ground}$ value (column 7) for the time-optimal case is 250.2 ms$^{-1}$, while that for the great circle cases ranges from 241.1 to 244.7 ms$^{-1}$. Therefore, GA correctly selected the airspace by altitude changes, where $V_{ground}$ values increased.

This behavior of altitude changes affects the variation in fuel consumptions (the terms are used interchangeably to mean fuel flows). Figure 20 shows the mean fuel consumption (in kg(fuel)min$^{-1}$) vs altitude for the time-optimal and the great circle flights. The results show that the fuel consumption increases at low altitudes due to the increased aerodynamic drag (i.e. increased air density). In addition, the mean value for the time-optimal case is high, due to its low mean flight altitude

$(\overline{h} = 9{,}271$ m, which is between FL290 $(= 8{,}839$ m$)$ and FL330 $(= 10{,}058$ m$))$. Table 10 lists the mean fuel consumptions for the different cases. In the great circle cases, the mean value for the eastbound cases is lower than that for the westbound cases (columns 2 and 3 of Table 10), because the eastbound flights benefit from the tail winds of the westerly jet stream. On the other hand, the mean value for the eastbound time-optimal case increases owing to its low mean flight altitude $(\overline{h} = 9{,}029$



m) compared to that for the westbound case ($\overline{h} = 9,517$ m). Note, the fuel consumption was not regarded as the objective function (Eq. (28)).

We also compared the total flight time, fuel use, $NO_x$ and $H_2O$ emissions for the time-optimal and the great circle cases. Figure 21 shows the flight time corresponding to individual flights (the similar figures for the fuel use, $NO_x$ and $H_2O$ emissions are shown in the Supplement (Fig. S8)). The results showed that all symbols lay in the right-hand domain. That is, the flight time for the time-optimal flights decreased for all airport pairs compared to that for the great circle flights. Table 11 shows the total flight time simulated by AirTraf for eastbound, westbound and total flights. The total value was certainly minimal for the time-optimal case, while in relative terms the value increased by $+1.5$ %, $+2.5$ %, $+2.9$ % and $+2.9$ % for the great circle cases at FL290, FL330, FL370 and FL410, respectively. Regarding the total value of fuel use, Table 11 indicates that the value increased by $+5.4$ % for the great circle case at FL290 when compared with the value of the time-optimal case. To confirm this intuitively, Fig. 22 shows the global distribution maps of the fuel use (in $\mathrm{kg(fuel)box^{-1}s^{-1}}$, 2 hour averages) for these cases. The maps show that the time-optimal case has low values of the fuel use. On the other hand, Table 11 indicates that the fuel use decreased by $-5.8$ %, $-14.9$ % and $-20.8$ % for the great circle cases at FL330, FL370 and FL410, respectively. The total values of $NO_x$ and $H_2O$ emissions show a similar trend: the total value of $NO_x$ emission increased by $+5.2$ % for the great circle at FL290, while it decreased by $-12.9$ %, $-24.9$ % and $-29.4$ % for the great circle cases at FL330, FL370 and FL410, respectively. The changes in total $H_2O$ emission were the same as those of the total fuel use, because $\mathrm{EIH_2O} = 1,230\ \mathrm{g(H_2O)(kg(fuel))^{-1}}$ was used. Figure 20 already shows that the mean fuel consumption for the time-optimal case is high, owing to the low mean flight altitude. Thus, the total amount of fuel use increased for this case, which increased total $NO_x$ and $H_2O$ emissions. It is important to note that the variations in the flight time, fuel use, $NO_x$ and $H_2O$ emissions are not representative for all seasons and the whole world's air traffic, because they have been obtained under the specific winter conditions using the trans-Atlantic flight plans.

## 5 Consistency check for the AirTraf simulations

To verify the consistency for AirTraf simulations, the one-day simulation results described in Sect. 4 were compared to reference data of flight time, fuel consumption, $\mathrm{EINO}_x$ and aircraft weight. The data obtained under similar conditions (aircraft/engine types, flight conditions, weather situations, etc.) were selected for the comparison, although they are not completely the same as the calculation conditions for the one-day simulations. Note, the verification of the aircraft weight is related to that of the fuel use calculations, because the aircraft weight was calculated by adding the amount of fuel use (Eq. (10)). In addition, $H_2O$ emission is proportional to the fuel use assuming ideal combustion. Thus, its verification would be redundant.



First, Table 12 shows a comparison of the flight time between the seven time-optimal flight trajec-
tories simulated by AirTraf and three reference data (the seven airport pairs are geographically close
to those of the reference data). Sridhar et al. (2014) simulated the wind-optimal flight trajectory from
Newark (EWR) to Frankfurt (FRA) using a specific winter day and the flight time was 22,980 s. The
flight time of the time-optimal flight trajectory from JFK to FRA simulated by AirTraf was 22,955
s. This agrees well with the value reported by Sridhar et al. (2014). Irvine et al. (2013) analyzed
the variation in flight time of time-optimal flight trajectories between JFK and London (LHR) using
weather data for three winters. The results showed that the flight time for east- and westbound flights
ranged from approximately 18,000 to 22,200 s, and from 21,600 to 27,000 s, respectively (see Fig.
3 in the literature). This indicated that the flight time increased for westbound flights on the trans-
Atlantic region in winter due to westerly jet streams. In addition, Grewe et al. (2014a) optimized the
trans-Atlantic one-day air traffic (for winter) with respect to air traffic climate impacts and economic
costs to investigate routing options for minimizing the impacts. The results showed that the mean
flight time of the air traffic ranged from 26,136 to 27,792 s (eastbound), while it ranged from 29,664
to 31,788 s (westbound), depending on the degree of climate impact reduction (see Tables 2 and 3
in the literature). This also indicated the increased flight time for westbound trans-Atlantic flights in
winter due to westerly jets streams. The magnitude in flight times of the seven airport pairs is close
and the variation shows good agreement with the trend of the flight time for westbound flights in
winter, as indicated from the reference data.

Second, the fuel consumption was verified using the mean fuel consumption value of 103 flights
and reference data, as shown in columns 4 to 7 of Table 10. Note, the AirTraf simulations were per-
formed under the specific winter conditions (Table 7), while the reference data show the estimated
values under international standard atmosphere conditions. Table 10 shows that the mean fuel con-
sumption values for the time-optimal and the great circle cases (column 4) were comparable to those
of the reference data corresponding to low and nominal weights (columns 5 and 6). In the AirTraf
simulations, the load factor of the worldwide air traffic indication in 2008 was used (Table 1). If a
specific load factor of A330-301 for international flights is available, the value is possibly higher
than 0.62 and the corresponding mean fuel consumption values are expected to increase.

Third, the mean $EINO_x$ (in $g(NO_x)(kg(fuel))^{-1}$) simulated by AirTraf were compared to the
six reference data. Table 13 shows that the obtained mean $EINO_x$ value decreased at high altitudes
and it ranged from 10.8 to 12.2 $g(NO_x)(kg(fuel))^{-1}$. These values are in the same range as the
reference data. Note, the reference data provided by Sutkus et al. (2001) show higher $EINO_x$ values.
They correspond to the values for the CF6-80E1A2 (1GE033) engine instead of the CF6-80E1A2
(2GE051) engine used in our simulations. $NO_x$ of aircraft engines, in general, decrease owing to
an installation of a new combustor. The 2GE051 installed the new 1862M39 combustor, which is
known as a low-emissions combustor. Thus, the reference $EINO_x$ value of 2GE051 will be lower
than that of the 1GE033.





Finally, the aircraft weights simulated by AirTraf were verified to make sure whether the fuel use calculations were performed properly. AirTraf duplicates real fuel consumptions under cruising flight, i.e. the aircraft weight reduces from the first waypoint ($m_1$) to the last waypoint ($m_{n_{wp}}$) as fuel is burnt (as described in Sect. 2.5). Thus, $m_1$ and $m_{n_{wp}}$ correspond to the maximum and minimum

aircraft weights, respectively. Here the obtained $m_1$ and $m_{n_{wp}}$ for 103 flights were compared with three structural limit weights (MTOW, MLW and MZFW), which are commonly used to provide safety flight operations, and one specified limit weight (MLOW) of the A330-301 aircraft. Table 14 shows the designated constraints among the $m_1$, $m_{n_{wp}}$ and the four limit weights. Note, no model that constrains to the structural limit weights was included in AirTraf.

As indicated in Table 14, the first constraint is on Maximum Take-off Weight (MTOW). The MTOW is limited for the aircraft not to cause structural damage to the airframe during take off. Figure 23 shows a comparison of $m_1$ and $m_{n_{wp}}$ with the limit weights (MTOW, MLW and MLOW). The results showed that almost all the $m_1$ (closed circle) were less than the MTOW. The only 15 of 515 flights (total of the time-optimal and the great circle cases: 5 cases × 103 flights) exceeded the

MTOW. For these 15 flights, actual flight planning data probably indicate altitude changes (generally higher flight altitudes) to increase a fuel mileage, which decreases $m_1$. The second constraint is on Maximum Landing Weight (MLW). To prevent the structural damage to the landing gear and the fuselage, aircraft has to reduce the total weight below MLW prior to landing. Figure 23 shows that all the $m_{n_{wp}}$ (open circle) were certainly less than MLW. The third constraint is on Maximum

Zero Fuel Weight (MZFW), which corresponds to the maximum operational weight of the aircraft without usable fuel. The MZFW of A330-301 is 164,000 kg (EASA, 2013), while the calculated zero fuel weight (ZFW) was 154,798 kg for all flights. This always satisfies the third constraint ZFW ≦ MZFW. Note, the ZFW is calculated as ZFW = OEW + MPL × OLF and hence it depends only on the aircraft type and the load factor (Table 1). In addition, the fourth constraint is on the

approximately minimum operational weight of A330-301 in the international standard atmosphere (MLOW). The MLOW is used here as a measure of validity of fuel use calculations and is not a strict constraint. As shown in Fig. 23, all the $m_{n_{wp}}$ (open circle) were more than the MLOW. As a result, almost all the $m_1$ and $m_{n_{wp}}$ simulated by AirTraf satisfied the four constraints. Thus, AirTraf simulates fairly good fuel use calculations.

**6   Code availability**

AirTraf is published for the first time as an submodel of Modular Earth Submodel System (MESSy). The MESSy is continuously further developed and applied by a consortium of institutions. The usage of MESSy and access to the source code is licenced to all affiliates of institutions which are members of the MESSy Consortium. Institutions can become a member of the MESSy Consortium by signing

the MESSy Memorandum of Understanding. More information can be found on the MESSy Consor-



tium Website (http://www.messy-interface.org). The version presented here corresponds to AirTraf 1.0. Some improvements will be performed and AirTraf 1.0 will be updated for the latest version of the code. For example, evaluation functions corresponding to the $NO_x$, $H_2O$, fuel, contrail and CCF routing options will be added. The status information for AirTraf including the licence conditions

will be available at the website.

## 7   Conclusions

This study presents the global air traffic submodel AirTraf (version 1.0) of EMAC. The great circle and flight time routing options can be used in AirTraf 1.0. Two benchmark tests were performed without EMAC (off-line). First, the benchmark test was performed for the great circle routing option

using five representative routes. The results showed that the routing methodology works properly and the great circle distances showed quantitatively good agreement with those calculated by other published code. The accuracy of the results was within $\pm 0.05$ %. Second, the benchmark test was performed for the flight time routing option by GA, focusing on a flight from MUC to JFK. The results showed that GA explored diverse solutions and successfully found the time-optimal solution.

The difference in flight time between the solution and its true-optimal solution was less than 0.01 %. The dependence of the optimal solutions on initial populations was investigated by 10 independent GA simulations from different initial populations. The obtained 10 optimal solutions slightly varied, however the variability was sufficiently small (approximately 0.01 %). In addition, the population and generation sizing for the trajectory optimization was examined by 1,000 independent GA sim-

ulations. The results show that there is a clear trade-off between the accuracy of GA optimizations and the number of function evaluations (i.e. computational costs). The present results indicate that a large reduction in number of function evaluations of around 92 %-97 % can be achieved with only a small decrease in the accuracy of optimizations of around 0.05 %-0.1 %.

AirTraf simulations were demonstrated in EMAC (on-line) for a specific winter day by using 103

trans-Atlantic flight plans of an A330 aircraft. Four one-day simulations were separately performed with the great circle routing option at FL290, FL330, FL370 and FL410, while a single one-day simulation was performed with the flight time routing option allowing altitude changes. The results confirmed that AirTraf correctly works on-line for the two options. Specifically, we verified that GA successfully found time-optimal flight trajectories for all airport pairs. A comparison of the simu-

lations showed that the total flight time was minimal for the time-optimal case, while it increased ranging from +1.5 % to +2.9 % for the great circle cases. On the other hand, the total fuel use, $NO_x$ and $H_2O$ emissions increased for the time-optimal case compared to the great circle cases at FL330, FL370 and FL410. Compared to the time-optimal case, the total fuel use and $H_2O$ emission increased by +5.4 % for the great circle case at FL290, while they decreased by $-5.8$ %, $-14.9$ %

and $-20.8$ % for the great circle cases at FL330, FL370 and FL410, respectively. Similarly, the total



$NO_x$ emission increased by $+5.2$ % for the great circle case at FL290, while it decreased by $-12.9$ %, $-24.9$ % and $-29.4$ % for the great circle cases at FL330, FL370 and FL410, respectively. Note, the changes are confined to the specific weather conditions and the changes can vary on longer time scales.

The consistency of the one-day simulations was verified with reference data of flight time, fuel consumption, $EINO_x$ and aircraft weight (i.e. fuel use). Comparison of the flight time between the selected trajectories and the reference data showed that the values were close and indicated the similar trend: an increased flight time for westbound flights on the trans-Atlantic region in winter. The mean fuel consumption values simulated by AirTraf were comparable to the reference values of

BADA corresponding to low and nominal weights. The mean $EINO_x$ values were in the same range as the reference data. Finally, obtained maximum and minimum aircraft weights were compared to the three structural limit weights and one specified limit weight of the A330-301 aircraft. Almost all the values satisfied the four limit weights and only 15 of 515 flights exceeded the Maximum Take-off Weight. Thus, AirTraf comprises a sufficiently good fuel use calculation model.

The fundamental framework of AirTraf has been developed to perform fairly realistic air traffic simulations. AirTraf 1.0 is sufficient to investigate a reduction potential of aircraft routings on air traffic climate impacts. AirTraf is coupled with various submodels of EMAC to evaluate the impacts, and objective functions corresponding to other routing options will be integrated soon.

*Acknowledgements.* This work was supported by the DLR Project WeCare. The authors wish to thank Prof.

Dr. Shigeru Obayashi of the Institute of Fluid Science, Tohoku University for his invaluable comments on this work. The authors wish to acknowledge our colleagues, especially Prof. Dr. Robert Sausen for his support of this project. The authors also wish to thank an internal reviewer for helpful reviews of and comments on this work. The authors would like to express their gratitude to anonymous reviewers for their helpful comments and discussions.



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

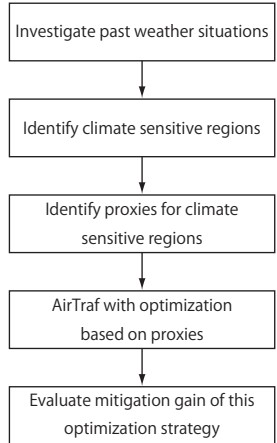

**Figure 1.** Road map for climate-optimized routing.





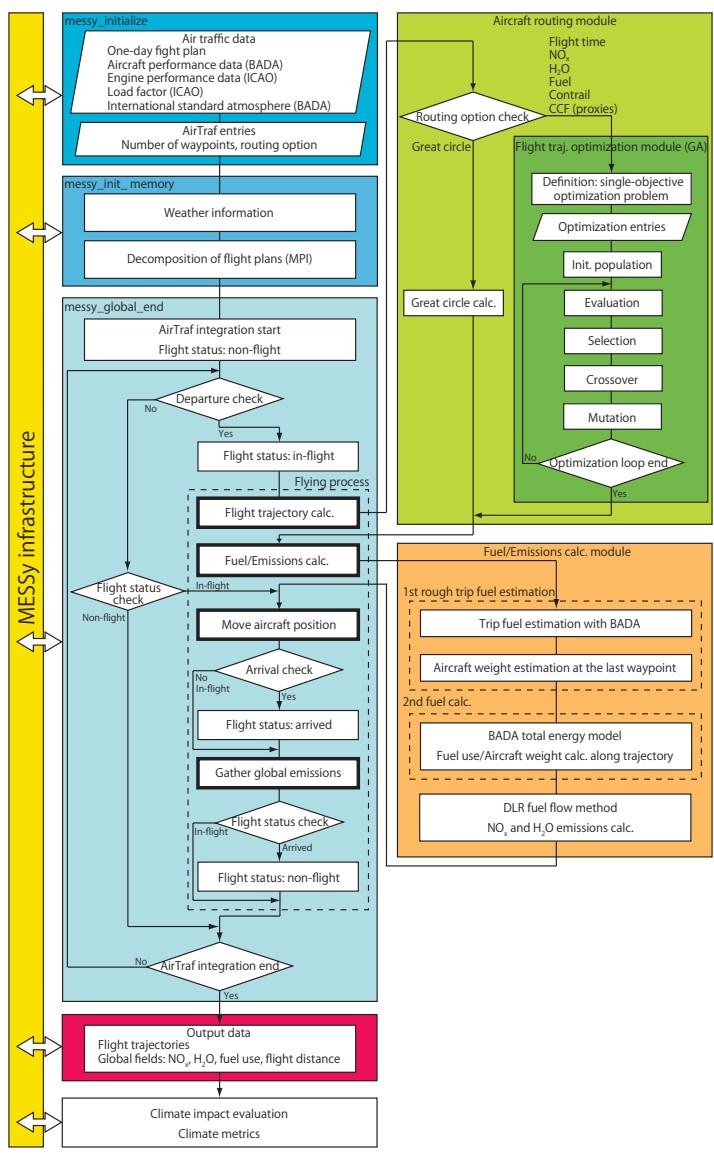

**Figure 2.** Flowchart of EMAC/AirTraf. MESSy as part of EMAC provides interfaces (yellow) to couple various submodels for data exchange, run control and data input/output. Air traffic data and AirTraf entries are input in the initialization phase (`messy_initialize`, dark blue). AirTraf includes the flying process in `messy_global_end` (dashed box, light blue), which comprises four main computation procedures (bold-black boxes). The detailed procedures are described in Sect. 2.4 and are illustrated in Fig. 4. AirTraf is linked to three modules: the aircraft routing module (light green), the flight trajectory optimization module (dark green), and the fuel/emissions calculation module (light orange). Resulting flight trajectories and global fields are calculated for output (rose red). Various submodels of EMAC can be linked to evaluate climate impacts on the basis of the output.





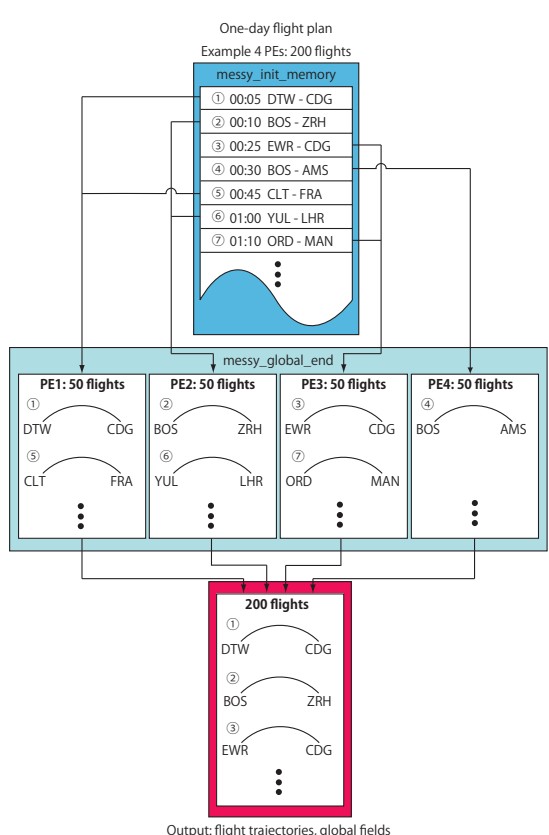

**Figure 3.** Decomposition of global flight plans in a parallel environment of EMAC/AirTraf. A one-day flight plan is distributed among many processing elements (PEs) in `messy_init_memory` (blue), while a whole trajectory of an airport pair is handled by the same PE in the time loop of EMAC (`messy_global_end`, light blue). Finally, results are gathered from all the PEs for output (rose red).





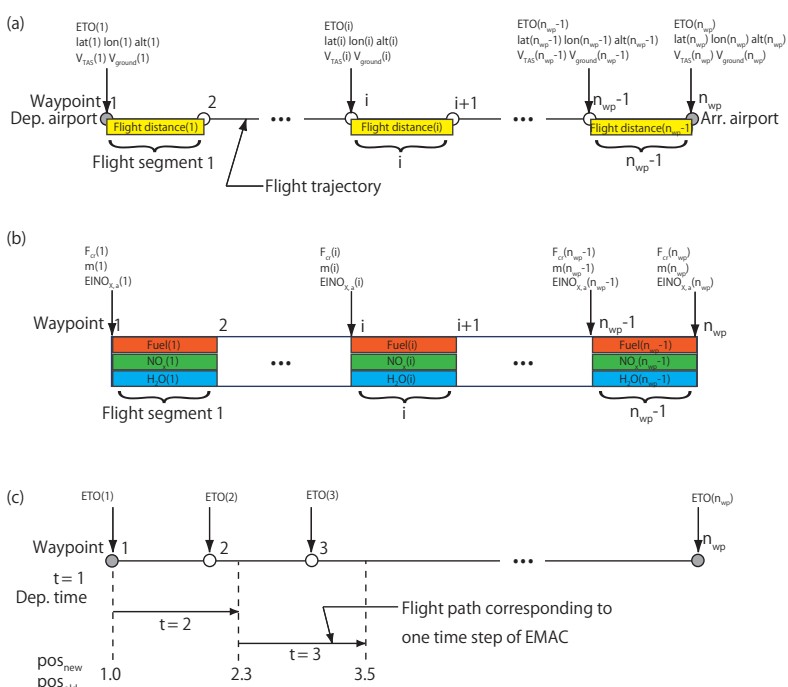

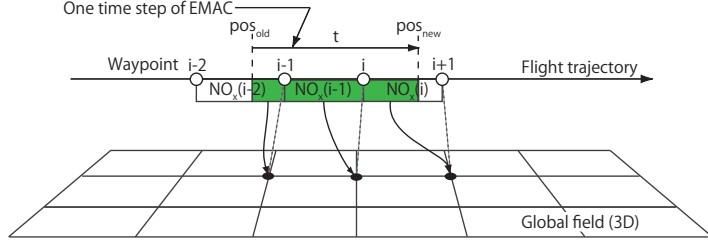

**Figure 4.** Illustration of the flying process of AirTraf (dashed box in Fig. 2, light blue). (a) Flight trajectory calculation. (b) Fuel/emissions calculation. (c) Moving aircraft position. (d) Gathering global emissions; the fraction of $NO_{x,i}$ corresponding to the EMAC grid box is mapped onto the nearest grid point (closed circle) relative to the $(i+1)^{th}$ waypoint (open circle). ETO: Estimated Time Over; $F_{cr}$: fuel flow of an aircraft; m: aircraft weight; t: time step index of EMAC. The detailed calculation procedures are described in Sect. 2.4.



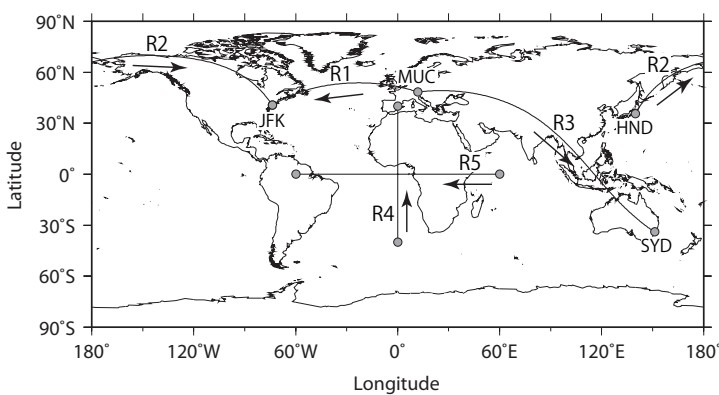

**Figure 5.** Five representative routes for the great circle benchmark test. The details of locations are listed in Table 3.

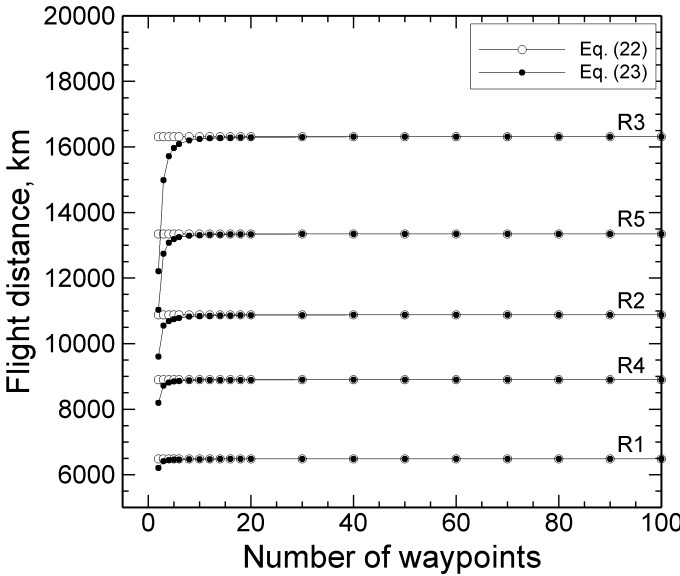

**Figure 6.** Comparison of the flight distance for the five representative routes. ○: great circle distance calculated by Eq. (22), ●: great circle distance calculated by Eq. (23).





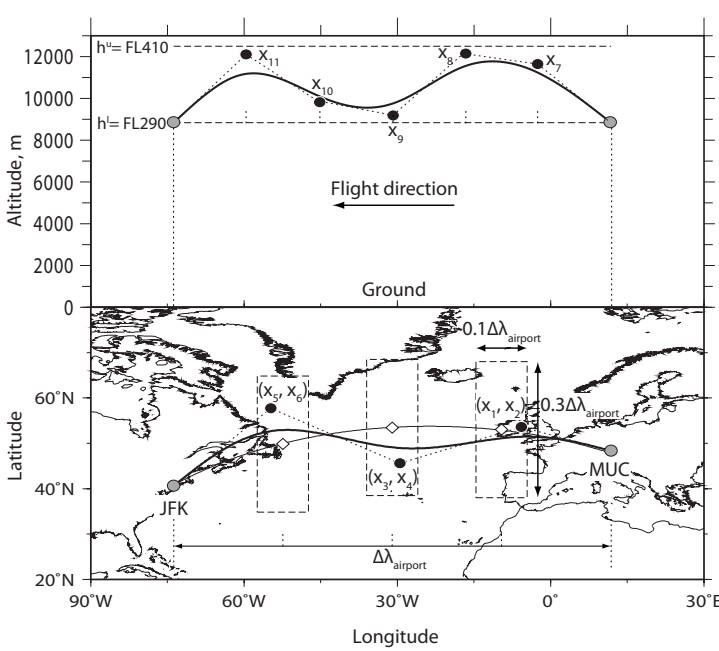

**Figure 7.** Geometry definition of flight trajectory as longitude vs altitude (top) and as geographic location (bottom). The bold solid line indicates a trajectory from MUC to JFK. •: control points consisting of design variables $x = (x_1, x_2, \cdots, x_{11})^T$. The lower/upper bounds of the eleven design variables are shown in Table 6. Bottom: the dashed boxes show rectangular domains of three control points. ◇: central points of the domains are calculated on the great circle (thin solid line), which divide the $\Delta\lambda_{airport}$ into four equal parts. Top: the dashed lines show the lower/upper variable bounds in altitude. 'FL290' stands for a flight level at 29,000 ft. Longitude-coordinates for $x_7, x_8, \cdots, x_{11}$ are pre-calculated; the coordinates divide the $\Delta\lambda_{airport}$ into six equal parts.





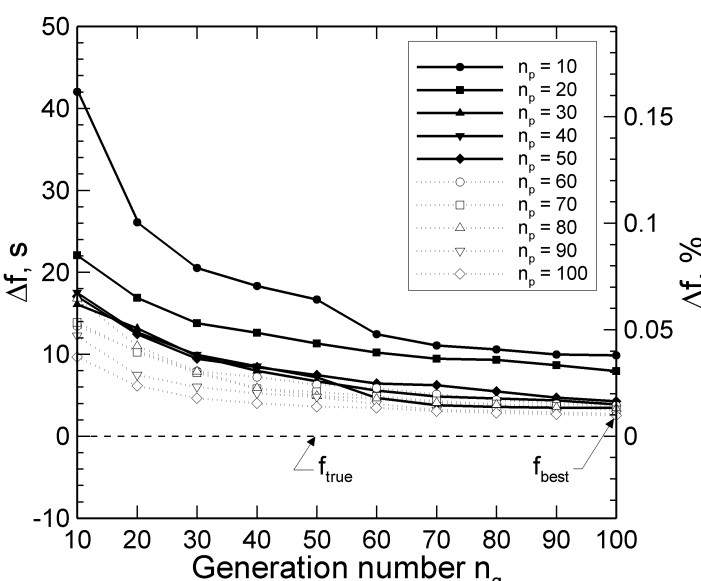

**Figure 8.** Optimal solutions are shown varying with the population size $n_p$ and generation number $n_g$. $\Delta f$ means the difference in flight time between the optimal solution $f$ and the true-optimal solution $f_{true}$. The $\Delta f$ (in %) is calculated as $(\Delta f / f_{true}) \times 100$. GA discovers the solutions as close to the $f_{true}$ (= 25,994.0 s) with increasing $n_p$ and $n_g$. For each $n_p$, the optimal solution shows minimum flight time for $n_g = 100$. For each $n_g$, the optimal solution shows minimum flight time for $n_p = 100$. The flight time of the best solution is $f_{best} = 25,996.6$ s (for $n_p = 100$ and $n_g = 100$, $\Delta f < 3.0$ s (less than 0.01 %)).





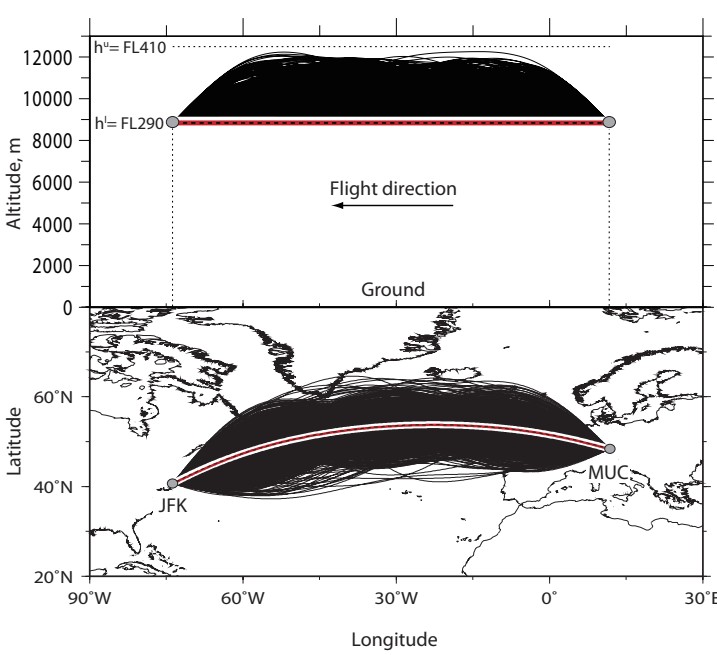

**Figure 9.** 1,000 explored trajectories (solid line, black) from MUC to JFK as longitude vs altitude (top) and as location (bottom). The population size $n_p = 100$ and the generation number $n_g = 100$. The best solution (red line) overlaps with the true-optimal solution (dashed line, black), i.e. the great circle at FL290. The flight time of the best solution is 25,996.6 s, while that of the true-optimal solution is 25,994.0 s.





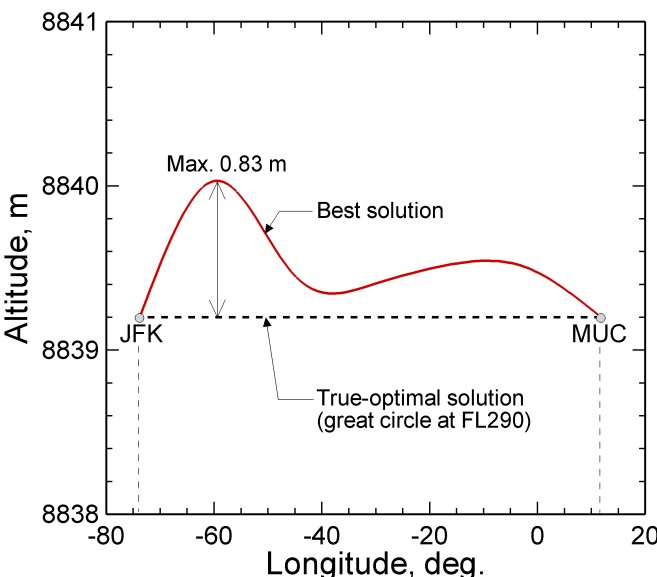

**Figure 10.** Comparison of trajectories for the best solution (red line) and the true-optimal solution (dashed line, black). This shows the enlarged drawing of Fig. 9 (top). The maximum difference in altitude is 0.83 m.




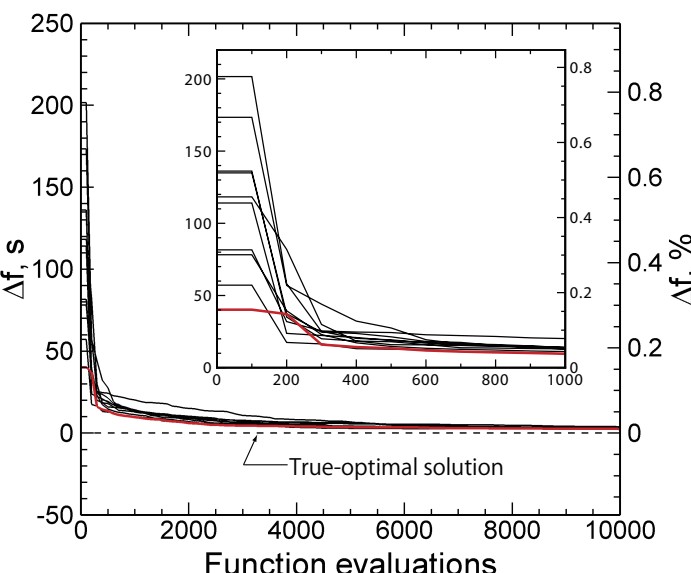

**Figure 11.** Best-of-generation flight time vs function evaluations ($= n_p \times n_g$), including the enlarged drawing in the early 1,000 evaluations. The population size $n_p = 100$ and the generation number $n_g = 100$. $\Delta f$ means the difference in flight time between the solution $f$ and the true-optimal solution $f_{true}$ ($= 25,994.0\,\text{s}$). The $\Delta f$ (in %) is calculated as $(\Delta f/f_{true}) \times 100$. The solution shown as red line corresponds to the best solution in Figs. 8 to 10. Table S1 summarizes the 10 optimal solutions in detail.




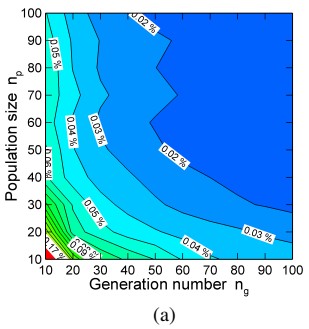
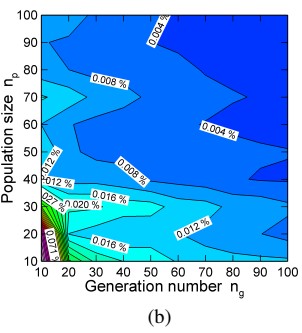

(a)                                    (b)

**Figure 12.** Variation of the mean value of the difference in flight time between the true-optimal solution
($f_{true} = 25,994.0$ s) and the optimal solution $\overline{\Delta f}$ (a), and the standard deviation of $\Delta f$ ($s_{\Delta f}$, b) are shown
varying with the population size $n_p$ and the generation number $n_g$. The variation was calculated by 10 inde-
pendent GA simulations from different initial populations for each combination of $n_p$ and $n_g$: totally 1,000
independent simulations. On the $\overline{\Delta f}$ and $s_{\Delta f}$: $\overline{\Delta f} = \frac{1}{n}\sum_{i=1}^{n}\Delta f_i$, $s_{\Delta f} = \sqrt{\frac{1}{n-1}\sum_{i=1}^{n}(\Delta f_i - \overline{\Delta f})^2}$, where
$n = 10$. $\overline{\Delta f}$ and $s_{\Delta f}$ (in %) relative to the true-optimal solution are calculated as $(\overline{\Delta f}/f_{true}) \times 100$ and
$(s_{\Delta f}/f_{true}) \times 100$, respectively.





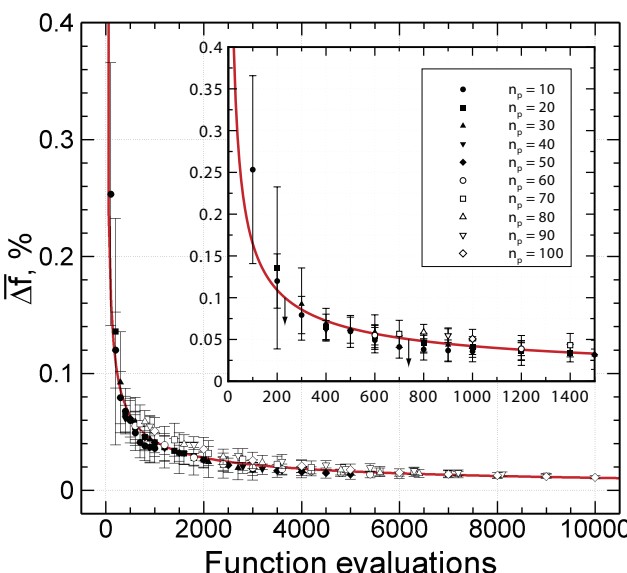

**Figure 13.** Chart for finding the appropriate number of function evaluations ($= n_p \times n_g$), including the enlarged drawing in the early 1,500 evaluations. The symbols with error bars correspond to $\overline{\Delta f} \pm s_{\Delta f}$ (in %); their definitions are given in the caption in Fig. 12. The fitted curve (power function, red line) to $\overline{\Delta f}$ is $y = e^{0.92} x^{-0.59}$, where $x$ are the function evaluations and $y$ is $\overline{\Delta f}$ (in %); $R^2 = 0.89$. The fitted curve to $s_{\Delta f}$ is calculated similarly: $y = e^{0.67} x^{-0.73}$, where $R^2 = 0.71$ (unshown).





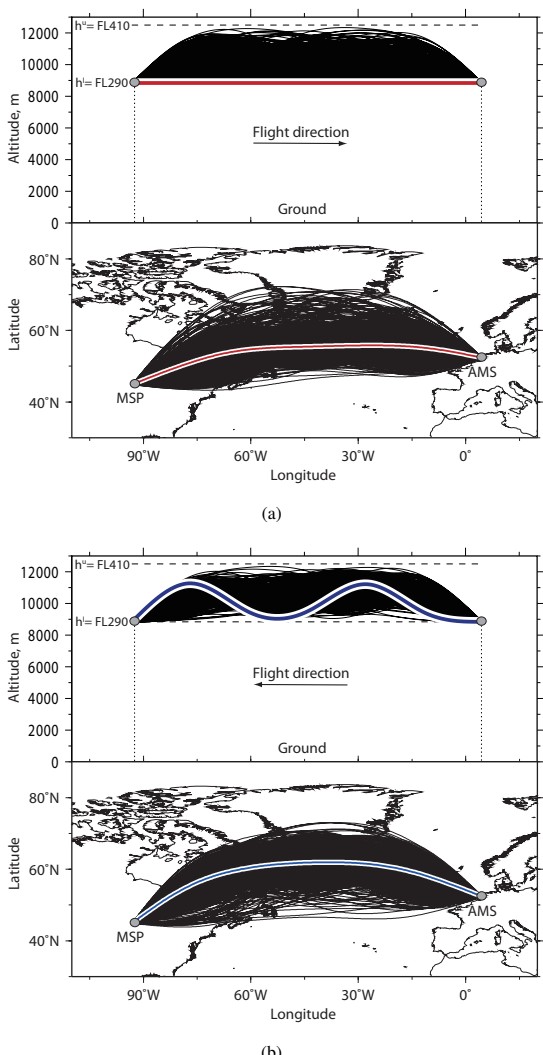

**Figure 14.** 1,000 explored trajectories (black lines) between MSP and AMS as longitude vs altitude (top) and as location (bottom), including time-optimal flight trajectories (red and blue lines). (a) The eastbound flight from MSP to AMS. (b) The westbound flight from AMS to MSP.





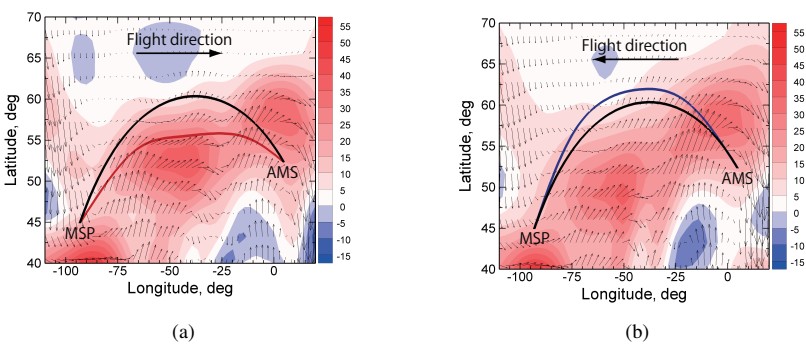

(a)                (b)

**Figure 15.** Comparison of trajectories for the time-optimal (red and blue lines) and the great circle cases (black lines) between MSP and AMS. The contours show the zonal wind speed ($u$); arrows (black) show the wind speed ($\sqrt{u^2 + v^2}$) and direction. (a) The eastbound flight from MSP to AMS with the wind field at $\overline{h} = 8,839$ m at 21:35:00 UTC. (b) The westbound flight from AMS to MSP with the wind field at $\overline{h} = 10,002$ m at 12:50:00 UTC.





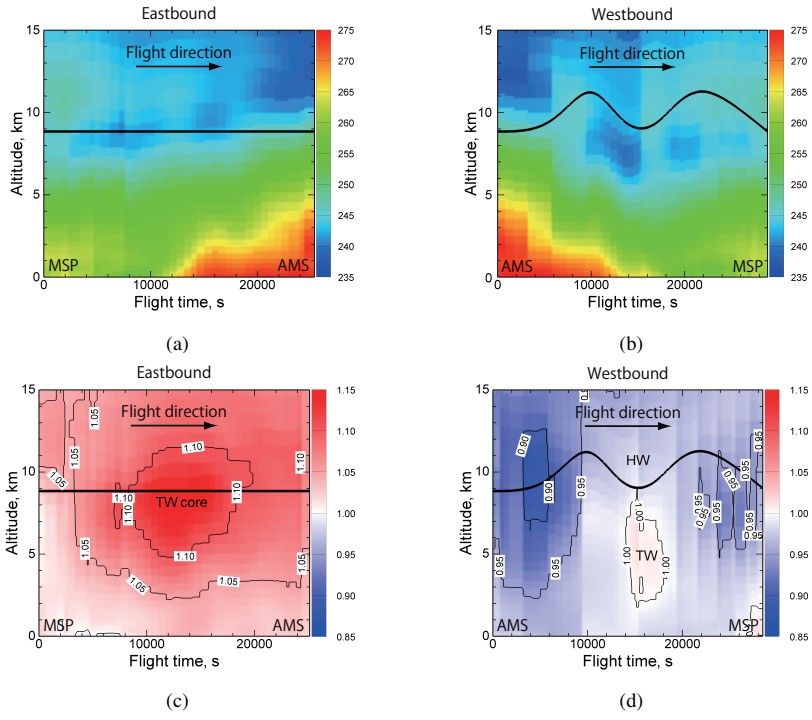

**Figure 16.** Altitude distributions of the true air speed $V_{TAS}$ (a and b) and the tail wind indicator $V_{ground}/V_{TAS}$ (c and d) along the time-optimal flight trajectories (black line) between MSP and AMS. Note, $(V_{ground}/V_{TAS}) \geq 1.0$ means tail winds (TW, red), while $(V_{ground}/V_{TAS}) < 1.0$ means head winds (HW, blue) to the flight direction. The contours were obtained at the departure time: 21:35:00 UTC (eastbound, a and c); 12:50:00 UTC (westbound, b and d).





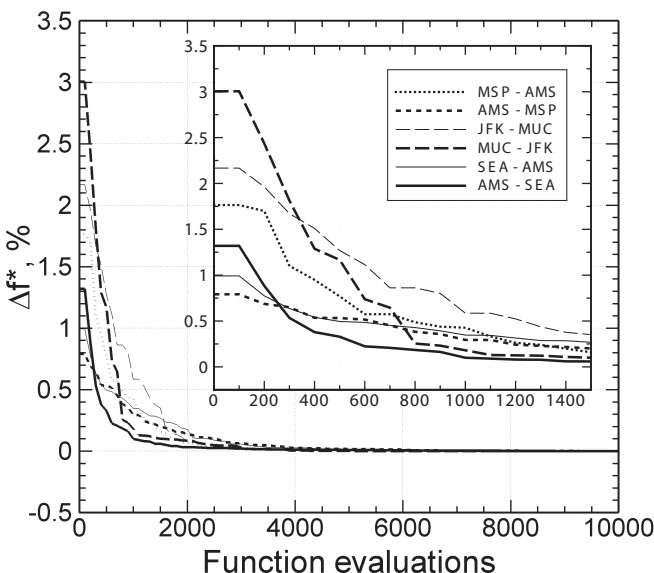

**Figure 17.** Best-of-generation flight time (in %) vs function evaluations ($= n_p \times n_g$) for three selected airport pairs, including the enlarged drawing in the early 1,500 evaluations. Population size $n_p = 100$ and generation number $n_g = 100$. $\Delta f^*$ means the difference in flight time between the solution $f$ and the obtained optimal solution $f_{opt}$, which was finally obtained after 10,000 function evaluations. This was chosen because $f_{true}$ for the six flights are unknown. The $f_{opt}$ for each flight corresponds to the flight time for the time-optimal case (column 7, Table 8). The $\Delta f^*$ (in %) is calculated as $(\Delta f^*/f_{opt}) \times 100$.



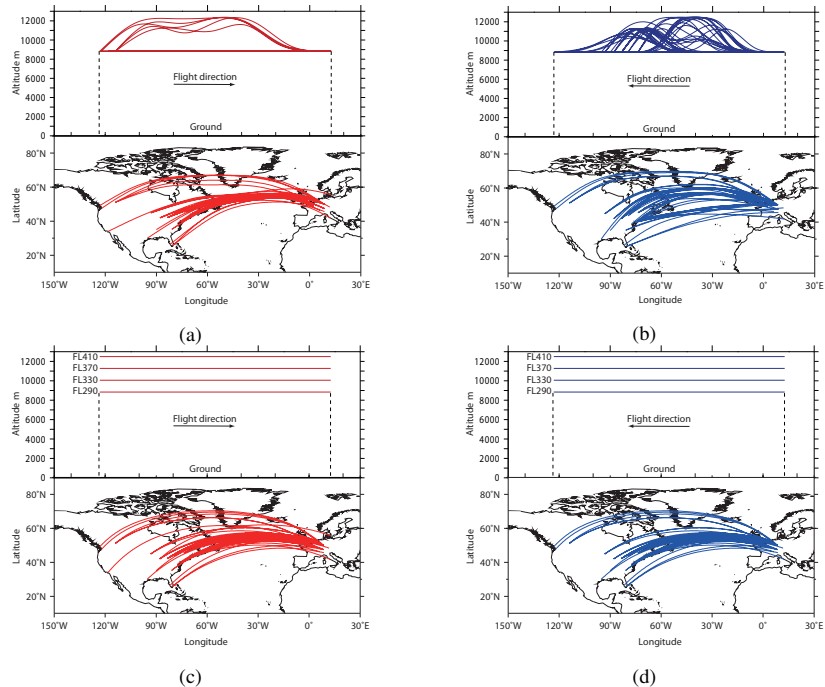

**Figure 18.** Obtained flight trajectories from one-day AirTraf simulations corresponding to the time-optimal case including altitude changes in [FL290, FL410] (a and b) and the great circle cases at FL290, FL330, FL370 and FL410 (c and d). For each figure, the trajectories as longitude vs altitude (top) and as location (bottom). The one-day flights comprise 52 eastbound (red lines) and 51 westbound flights (blue lines).



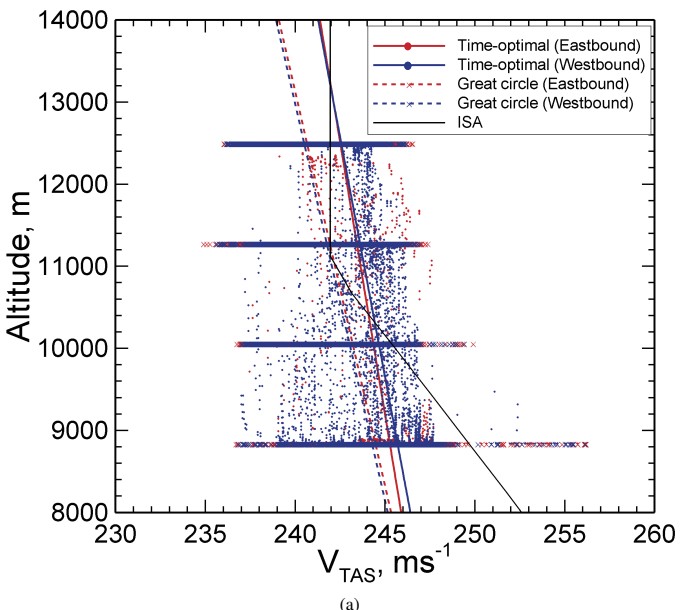

(a)

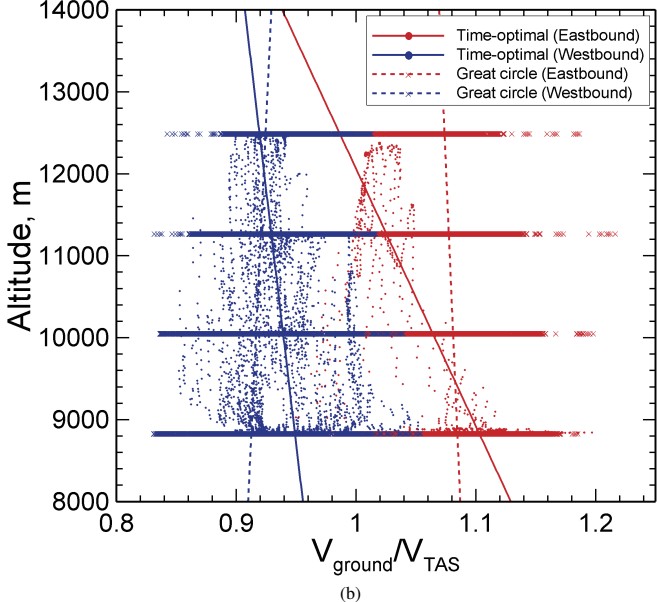

(b)

**Figure 19.** Values of the true air speed $V_{TAS}$ (a) and the tail wind indicator $V_{ground}/V_{TAS}$ (b) at waypoints for the time-optimal and the great circle flights. Linear fits of the time-optimal (solid line, red (eastbound) and blue (westbound)) and that of the great circle cases (dashed line, red (eastbound) and blue (westbound)) are included. $V_{TAS}$ of the international standard atmosphere (ISA) is given in (a) (solid line, black) provided by the BADA atmosphere table (Eurocontrol, 2010).





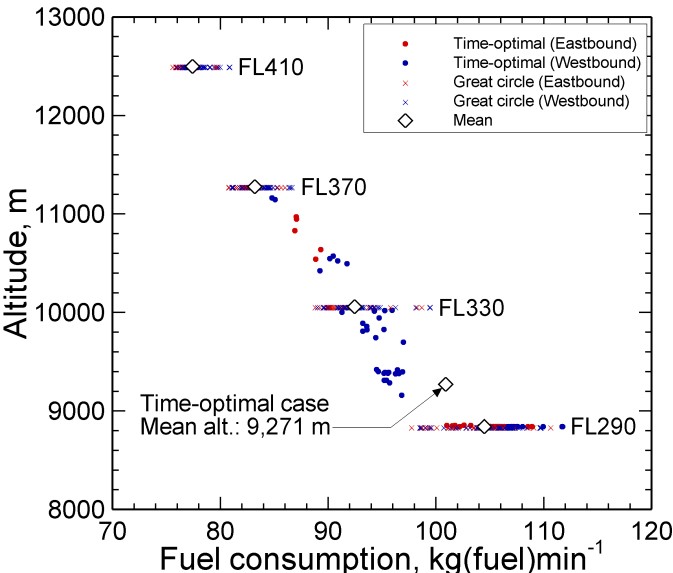

**Figure 20.** Mean fuel consumption (in $\mathrm{kg(fuel)min}^{-1}$) vs altitude for the time-optimal and the great circle flights. ◊: mean value of all 103 flights; these values are shown in column 4 of Table 10.





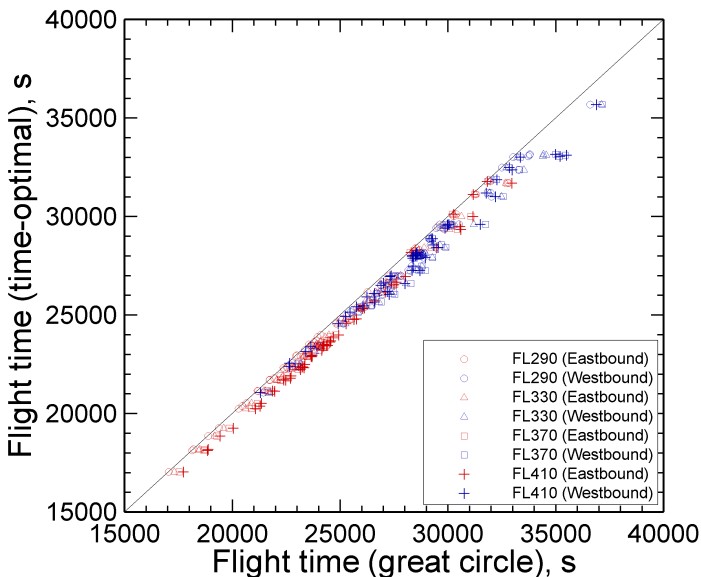

**Figure 21.** Comparison of the flight time for individual flights. A symbol indicates the value for one airport pair, corresponding to the time-optimal and the great circle flight. If the value for the time-optimal flight is the same as that of the great circle flight, the symbol lies on the 1:1 solid line.





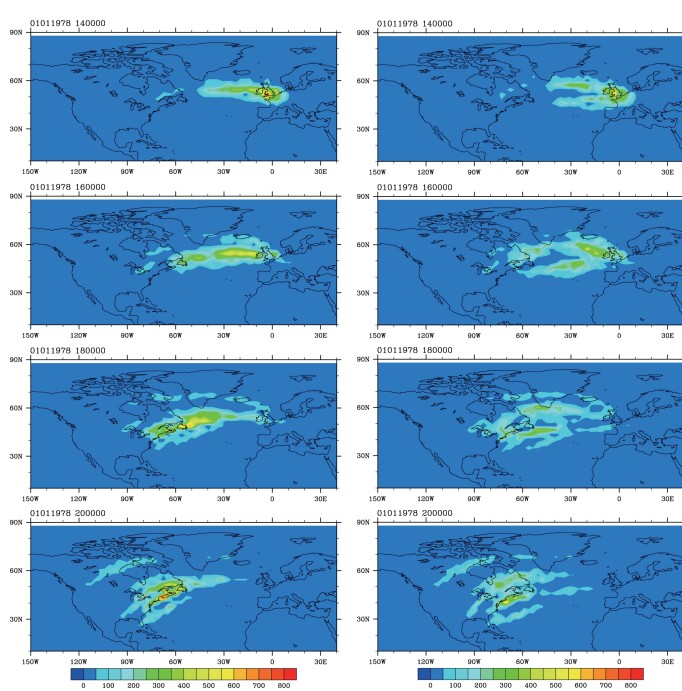

**Figure 22.** Global, vertically integrated, distribution of the fuel use (in $\mathrm{kg(fuel)box^{-1}s^{-1}}$): 2 hour averages simulated by EMAC/AirTraf from 1 January 1978 00:00:00 to 2 January 1978 00:00:00 UTC. Left: great circle case at FL290. Right: time-optimal case. The maps, beginning at the top, correspond to the results at 14:00:00; 16:00:00; 18:00:00; and 20:00:00 UTC.





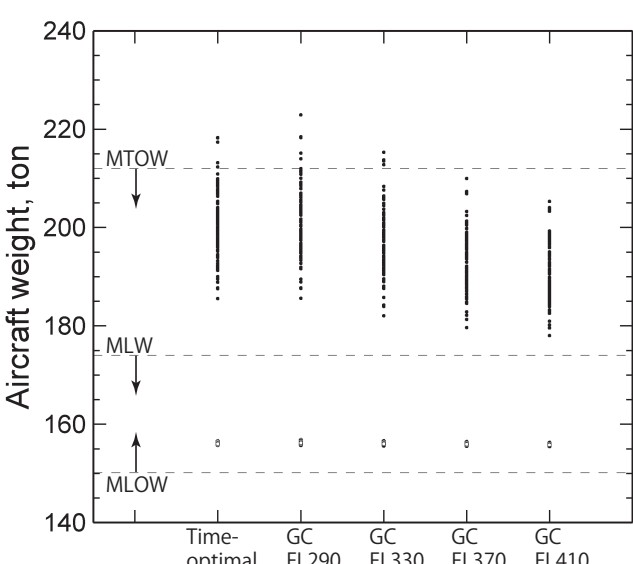

**Figure 23.** Comparison of aircraft weights with structural limit weights (MTOW and MLW) and one specified limit weight (MLOW). The aircraft weights of the 103 flights for the time-optimal and the great circle cases are plotted. ○: aircraft weight at the last waypoint ($m_{n_{wp}}$). ●: aircraft weight at the first waypoint ($m_1$). The description of the limits is shown in Table 14.





**Table 1.** Primary data of A330-301 and constant parameters used in AirTraf simulations.

| Parameter | Value | Unit | Description |
|---|---|---|---|
| OEW | 125,100 | kg | Operational empty weight[a] |
| MPL | 47,900 | kg | Maximum payload[a] |
| $S$ | 361.6 | $m^2$ | Reference wing surface area[a] |
| $C_{D0}$ | 0.019805 | – | Parasitic drag coefficient (cruise)[a] |
| $C_{D2}$ | 0.031875 | – | Induced drag coefficient (cruise)[a] |
| $C_{f1}$ | 0.61503 | $kg\,min^{-1}kN^{-1}$ | First thrust specific fuel consumption (TSFC) coefficient (jet)[a] |
| $C_{f2}$ | 919.03 | kt | Second TSFC coefficient[a] |
| $C_{fcr}$ | 0.93655 | – | Cruise fuel flow correction coefficient[a] |
| $M$ | 0.82 | – | Cruise Mach number[a] |
| $f_{ref}$ | 0.228; 0.724; 2.245; 2.767 | $kg(fuel)s^{-1}$ | Reference fuel flow at take off, climb out, approach and idle conditions (sea level). CF6-80E1A2 (2GE051)[b] |
| $EINO_{x,ref}$ | 4.88; 12.66; 22.01; 28.72 | $g(NO_x)(kg(fuel))^{-1}$ | Reference $NO_x$ emission index at take off, climb out, approach and idle conditions (sea level). CF6-80E1A2 (2GE051)[b] |
| $EIH_2O$ | 1,230 | $g(H_2O)(kg(fuel))^{-1}$ | $H_2O$ emission index[c] |
| OLF | 0.62 | – | ICAO overall weight load factor[d] |
| $Oneday$ | 86,400 | s | $60 \times 60 \times 24 = 86,400$ seconds in a day. Time (Julian date) $\times Oneday$ = Time (s) |
| $g$ | 9.8 | $ms^{-2}$ | Gravity acceleration |
| $\gamma$ | 1.4 | – | Adiabatic gas constant |
| $P_0$ | 101.325 | Pa | Total pressure (sea level) |
| $R$ | 287.05 | $JK^{-1}kg^{-1}$ | Gas constant for dry air |
| $T_0$ | 288.15 | K | Total temperature (sea level) |

[a] Eurocontrol, 2011;   [b] ICAO, 2005;   [c] Penner et al., 1999;   [d] Anthony, 2009.



**Table 2.** Properties assigned to a flight trajectory. The properties of the three groups (divided by rows) are obtained from the nearest grid point of EMAC, the flight trajectory calculation (Fig. 4a), and the fuel/emissions calculation (Fig. 4b), respectively. The attribute type indicates where the values of properties are allocated. "W", "S" and "T" stand for waypoints ($i = 1, 2, \cdots, n_{wp}$), flight segments ($i = 1, 2, \cdots, n_{wp-1}$) and a whole flight trajectory in column 3, respectively.

| Property | Unit | Attribute type | Description |
|---|---|---|---|
| $P$ | Pa | W | Pressure |
| $T$ | K | W | Temerature |
| $\rho$ | $\mathrm{kgm}^{-3}$ | W | Air density |
| $u, v, w$ | $\mathrm{ms}^{-1}$ | W | Three dimensional wind components |
| $\phi$ | deg | W | Latitude |
| $\lambda$ | deg | W | Longitude |
| $h$ | m | W | Altitude |
| ETO | Julian date | W | Estimated time over |
| $a$ | $\mathrm{ms}^{-1}$ | W | Speed of sound |
| $V_{TAS}$ | $\mathrm{ms}^{-1}$ | W | True air speed |
| $V_{ground}$ | $\mathrm{ms}^{-1}$ | W | Ground speed |
| $d$ | m | S | Flight distance |
| $\overline{h}$ | m | T | Mean flight altitude. $\overline{h} = 1/n_{wp} \sum_{i=1}^{n_{wp}} h_i$ with waypoint number $n_{wp}$. |
| $FT$ | s | T | Flight time. $FT = (\mathrm{ETO}_{n_{wp}} - \mathrm{ETO}_1) \times Oneday$ |
| $F_{cr}$ | $\mathrm{kg(fuel)s}^{-1}$ | W | Fuel flow of an aircraft (cruise) |
| $m$ | kg | W | Aircraft weight |
| $\mathrm{EINO}_{x,a}$ | $\mathrm{g(NO_x)(kg(fuel))}^{-1}$ | W | $NO_x$ emission index |
| $FUEL$ | kg | S | Fuel use |
| $NO_x$ | $\mathrm{g(NO_x)(kg(fuel))}^{-1}$ | S | $NO_x$ emission |
| $H_2O$ | $\mathrm{g(H_2O)(kg(fuel))}^{-1}$ | S | $H_2O$ emission |



**Table 3.** Information for the five representative routes of the great circle benchmark test.

| Route | Departure airport | Latitude | Longitude | Arrival airport | Latitude | Longitude |
|-------|-------------------|----------|-----------|-----------------|----------|-----------|
| R1 | Munich (MUC) | 48.35°N | 11.79°E | New York (JFK) | 40.64°N | 73.78°W |
| R2 | Tokyo Haneda (HND) | 35.55°N | 139.78°E | New York (JFK) | 40.64°N | 73.78°W |
| R3 | Munich (MUC) | 48.35°N | 11.79°E | Sydney (SYD) | 33.95°S | 151.18°E |
| R4 | – | 40.0°S | 0 | – | 40.0°N | 0 |
| R5 | – | 0 | 60.0°E | – | 0 | 60.0°W |





**Table 4.** Great circle distance ($d$) of the five representative routes calculated with different calculation methods. Column 2 ($d_{eq22}$) corresponds to the result calculated by Eq. (22); column 3 ($d_{eq23}$) corresponds to the result calculated by Eq. (23) with $n_{wp} = 100$; column 4 ($d_{MTS}$) shows the result calculated with the Movable type scripts (MTS), which output only integer values. On columns 5 to 7: $\Delta d_{eq23,eq22} = \frac{(d_{eq23}-d_{eq22})}{d_{eq22}} \times 100$, $\Delta d_{eq23,MTS} = \frac{(d_{eq23}-d_{MTS})}{d_{MTS}} \times 100$, $\Delta d_{eq22,MTS} = \frac{(d_{eq22}-d_{MTS})}{d_{MTS}} \times 100$.

| Route | $d_{eq22}$, km | $d_{eq23}$, km | $d_{MTS}$, km | $\Delta d_{eq23,eq22}$, % | $\Delta d_{eq23,MTS}$, % | $\Delta d_{eq22,MTS}$, % |
|---|---|---|---|---|---|---|
| R1 | 6,481.1 | 6,481.0 | 6,481 | −0.0005 | −0.0004 | −0.0009 |
| R2 | 10,875.0 | 10,874.7 | 10,870 | −0.0028 | −0.0435 | −0.0463 |
| R3 | 16,312.1 | 16,311.5 | 16,310 | −0.0036 | −0.0091 | −0.0127 |
| R4 | 8,895.6 | 8,895.5 | 8,896 | −0.0008 | 0.0054 | 0.0046 |
| R5 | 13,343.4 | 13,343.1 | 13,340 | −0.0019 | −0.0236 | −0.0254 |





**Table 5.** Calculation conditions for the benchmark test on flight trajectory optimizations.

| Parameter | |
|---|---|
| Objective function | Minimize flight time |
| Design variable, $n_{dv}$ | 11 |
| Number of waypoints, $n_{wp}$ | 101 |
| Departure airport | MUC (lat. = 48.35°N, lon. = 11.79°E, alt. = FL290) |
| Arrival airport | JFK (lat. = 40.64°N, lon. = 73.78°W, alt. = FL290) |
| $V_{TAS}, V_{ground}$ | 898.8 kmh$^{-1}$ (constant) |
| $V_{wind}$ | 0 (no-wind) |
| Optimizer | Real-coded GA[a] |
| Population size | $10, 20, \cdots, 100$ |
| Generation number | $10, 20, \cdots, 100$ |
| Selection | Stochastic universal sampling |
| Crossover | Blend crossover BLX-0.2 ($\alpha = 0.2$) |
| Mutation | Revised polynomial mutation ($r_m = 0.1; \eta_m = 5.0$) |

[a] Sasaki et al., 2002 and Sasaki and Obayashi, 2004.

**Table 6.** Lower/Upper bounds of the eleven design variables.

| Design variable | Dimension | Unit | Lower value | Upper value |
|---|---|---|---|---|
| $x_1$ | Longitude | °W | 14.6 | 4.6 |
| $x_2$ | Latitude | °N | 38.0 | 68.0 |
| $x_3$ | Longitude | °W | 36.0 | 26.0 |
| $x_4$ | Latitude | °N | 38.5 | 68.5 |
| $x_5$ | Longitude | °W | 57.4 | 47.4 |
| $x_6$ | Latitude | °N | 34.9 | 64.9 |
| $x_7, x_8, \cdots, x_{11}$ | Altitude | ft | FL290 | FL410 |





**Table 7.** Calculation conditions for AirTraf one-day simulations.

| Parameter | Routing option | |
|---|---|---|
| | Great circle | Flight time |
| ECHAM5 resolution | T42L31ECMWF (2.8° by 2.8°) | |
| Duration of simulation | 1 January 1978 00:00:00 - 2 January 1978 00:00:00 UTC | |
| Time step of EMAC | 12 min | |
| Flight plan | 103 trans-Atlantic flights (eastbound 52/westbound 51)[a] | |
| Aircraft type | A330-301 | |
| Engine type | CF6-80E1A2, 2GE051 (with 1862M39 combustor) | |
| Flight altitude changes | Fixed FL290, FL330, FL370, FL410 | [FL290, FL410] |
| Mach number | 0.82 | |
| Wind effect | Three-dimensional components ($u$, $v$, $w$) | |
| Number of waypoints, $n_{wp}$ | 101 | |
| Optimization | – | Minimize flight time |
| Design variable, $n_{dv}$ | – | 11 (location 6/altitude 5) |
| Population size, $n_p$ | – | 100 |
| Generation number, $n_g$ | – | 100 |
| Selection | – | Stochastic universal sampling |
| Crossover | – | Blend crossover BLX-0.2 ($\alpha = 0.2$) |
| Mutation | – | Revised polynomial mutation ($r_m = 0.1$; $\eta_m = 5.0$) |

[a] REACT4C, 2014.




**Table 8.** Information for the trajectories of the three selected airport pairs; they were extracted from the one-day AirTraf simulations. Columns 7 to 11 show the obtained flight times for the flight time and the great circle routing options. GC FL290: great circle at 29,000 ft.

| Type | Departure airport | Arrival airport | Flight direction | Departure time, UTC | Mean flight altitude $\bar{h}$, m | Flight time $FT$, s | | | | | |
|---|---|---|---|---|---|---|---|---|---|---|---|
| | | | | | | Time-optimal | GC FL290 | GC FL330 | GC FL370 | GC FL410 | |
| I | New York (JFK) | Munich (MUC) | Eastbound | 01:30:00 | 8,841 | 23,986.2 | 24,100.1 | 24,472.1 | 24,772.6 | 24,931.9 | |
| | Munich (MUC) | New York (JFK) | Westbound | 14:27:00 | 8,839 | 28,429.0 | 29,417.3 | 29,856.7 | 29,899.0 | 29,538.5 | |
| II | Minneapolis (MSP) | Amsterdam (AMS) | Eastbound | 21:35:00 | 8,839 | 25,335.6 | 25,958.4 | 25,957.9 | 25,989.3 | 26,043.9 | |
| | Amsterdam (AMS) | Minneapolis (MSP) | Westbound | 12:50:00 | 10,002 | 28,869.5 | 29,117.0 | 29,211.7 | 29,292.6 | 29,219.1 | |
| III | Seattle (SEA) | Amsterdam (AMS) | Eastbound | 21:05:00 | 10,829 | 31,784.6 | 31,962.9 | 31,943.5 | 31,841.9 | 31,825.4 | |
| | Amsterdam (AMS) | Seattle (SEA) | Westbound | 12:30:00 | 9,311 | 33,010.5 | 33,026.2 | 33,230.5 | 33,342.6 | 33,354.1 | |





**Table 9.** The mean value of $V_{TAS}$ and $V_{ground}$ for the time-optimal and the great circle cases. The mean values were calculated using $V_{TAS}$ and $V_{ground}$ values at all waypoints. Eastbound: mean value of 52 eastbound flights; Westbound: that of 51 westbound flights; and Total: that of 103 flights.

| Case | $V_{TAS}$, ms$^{-1}$ | | | $V_{ground}$, ms$^{-1}$ | | |
| --- | --- | --- | --- | --- | --- | --- |
| | Eastbound | Westbound | Total | Eastbound | Westbound | Total |
| Time-optimal | 245.1 | 245.1 | 245.1 | 268.7 | 231.2 | 250.2 |
| GC FL290 | 245.0 | 244.8 | 244.9 | 265.3 | 223.7 | 244.7 |
| GC FL330 | 242.8 | 242.6 | 242.7 | 262.7 | 222.0 | 242.6 |
| GC FL370 | 241.3 | 241.1 | 241.2 | 260.4 | 221.7 | 241.2 |
| GC FL410 | 241.2 | 241.1 | 241.2 | 258.7 | 223.1 | 241.1 |

**Table 10.** The mean fuel consumption (in kg(fuel)min$^{-1}$) for the time-optimal and the great circle cases. Eastbound: mean value of 52 eastbound flights; Westbound: that of 51 westbound flights; and Total: that of 103 flights. Columns 5 to 7 show the reference cruise fuel consumption (in kg(fuel)min$^{-1}$) for three different weights (low, nominal and high) in the international standard atmosphere. BADA provides the reference data at specific flight altitudes. Therefore, the reference values for the time-optimal case in parentheses were estimated from the reference data at FL290 and FL330 by linear interpolation (the mean flight altitude of the time-optimal case was $\overline{h} = 9,271$ m, which is the medium value between FL290 ($= 8,839$ m) and FL330 ($= 10,058$ m)).

| Case | Simulation | | | Reference data[a] | | |
| --- | --- | --- | --- | --- | --- | --- |
| | Eastbound | Westbound | Total | Low | Nominal | High |
| Time-optimal | 103.6 | 98.2 | 100.9 | (99.8) | (104.0) | (111.9) |
| GC FL290 | 104.1 | 104.9 | 104.5 | 104.8 | 108.7 | 116.0 |
| GC FL330 | 92.1 | 92.9 | 92.5 | 90.8 | 95.5 | 104.3 |
| GC FL370 | 82.8 | 83.6 | 83.2 | 79.9 | 85.5 | 96.1 |
| GC FL410 | 77.1 | 77.8 | 77.4 | 72.2 | 79.0 | 91.9 |

[a] Eurocontrol, 2011.



**Table 11.** Sum of flight time, fuel use, $NO_x$ and $H_2O$ emissions for the time-optimal and the great circle cases obtained from one-day AirTraf simulations. Eastbound: sum of 52 eastbound flights; Westbound: that of 51 westbound flights; and Total: that of 103 flights. Changes (in %) relative to the time-optimal case are given in parentheses.

| Case | Flight time, h | | |
|---|---|---|---|
| | Eastbound | Westbound | Total |
| Time-optimal | 348.2 | 395.9 | 744.1 |
| GC FL290 | 351.2 (+0.9) | 404.4 (+2.2) | 755.6 (+1.5) |
| GC FL330 | 354.4 (+1.8) | 408.0 (+3.1) | 762.4 (+2.5) |
| GC FL370 | 357.4 (+2.7) | 408.5 (+3.2) | 765.9 (+2.9) |
| GC FL410 | 359.7 (+3.3) | 405.6 (+2.5) | 765.3 (+2.9) |
| Case | Fuel use, ton | | |
| | Eastbound | Westbound | Total |
| Time-optimal | 2,155.4 | 2,339.1 | 4,494.5 |
| GC FL290 | 2,190.1 (+1.6) | 2,545.1 (+8.8) | 4,735.2 (+5.4) |
| GC FL330 | 1,958.4 (−9.1) | 2,275.7 (−2.7) | 4,234.1 (−5.8) |
| GC FL370 | 1,776.4 (−17.6) | 2,049.9 (−12.4) | 3,826.3 (−14.9) |
| GC FL410 | 1,665.5 (−22.7) | 1,894.7 (−19.0) | 3,560.2 (−20.8) |
| Case | $NO_x$ emission, ton | | |
| | Eastbound | Westbound | Total |
| Time-optimal | 26.5 | 28.7 | 55.2 |
| GC FL290 | 26.8 (+1.4) | 31.2 (+8.8) | 58.1 (+5.2) |
| GC FL330 | 22.2 (−16.0) | 25.8 (−10.1) | 48.1 (−12.9) |
| GC FL370 | 19.3 (−27.1) | 22.2 (−22.8) | 41.5 (−24.9) |
| GC FL410 | 18.3 (−31.0) | 20.7 (−28.0) | 39.0 (−29.4) |
| Case | $H_2O$ emission, ton | | |
| | Eastbound | Westbound | Total |
| Time-optimal | 2,651.1 | 2,877.0 | 5,528.2 |
| GC FL290 | 2,693.8 (+1.6) | 3,130.5 (+8.8) | 5,824.3 (+5.4) |
| GC FL330 | 2,408.9 (−9.1) | 2,799.1 (−2.7) | 5,208.0 (−5.8) |
| GC FL370 | 2,185.0 (−17.6) | 2,521.4 (−12.4) | 4,706.4 (−14.9) |
| GC FL410 | 2,048.5 (−22.7) | 2,330.5 (−19.0) | 4,379.0 (−20.8) |

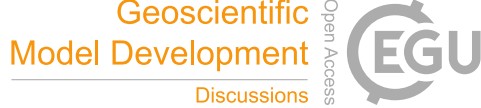



**Table 12.** Comparison of the flight time for time-optimal flight trajectories from one-day AirTraf simulations and optimal trajectories from earlier studies. The original units of the flight time of the studies are converted into seconds.

| Airport pair | Flight time, s | | Detailed information |
| --- | --- | --- | --- |
| | Eastbound | Westbound | |
| New York (JFK) - Shannon (SNN) | 18,187.4 | 22,389.1 | These seven time-optimal flight trajectories in this first group (divided by rows) were simulated by AirTraf. |
| New York (JFK) - Dublin (DUB) | 18,853.2 | 23,150.6 | |
| Newark (EWR) - Amsterdam (AMS) | 21,705.9 | 26,512.3 | |
| Newark (EWR) - Paris (CDG) | 21,790.9 | 25,668.3 | |
| New York (JFK) - Frankfurt (FRA) | 22,955.2 | 27,261.5 | |
| New York (JFK) - Zurich (ZRH) | 23,450.9 | 27,246.7 | |
| New York (JFK) - Munich (MUC) | 23,986.2 | 28,429.0 | |
| Newark (EWR) - Frankfurt (FRA)[a] | 22,980 | – | Wind-optimal flight trajectory (constant flight altitude, cruise phase) |
| New York (JFK) - London (LHR)[b] | 18,000 - 22,200 | 21,600 - 27,000 | Time-optimal flight trajectory (constant $V_{TAS}$, constant flight altitude, cruise phase, including wind fields) |
| Trans-Atlantic air traffic[c] | 26,136 - 27,792 | 29,664 - 31,788 | Climate-optimal/Economic-optimal flight trajectories for real air traffic flight plans: 391 eastbound flights with 28 aircraft types/394 westbound flights with 30 aircraft types. |

[a] Sridhar et al., 2014;     [b] Irvine et al., 2013;     [c] Grewe et al., 2014a.





**Table 13.** The mean value of EINO$_x$ (in g(NO$_x$)(kg(fuel))$^{-1}$) for 103 flights. Some reference data of EINO$_x$ are provided by the literature in the table.

| Case | EINO$_x$, g(NO$_x$)(kg(fuel))$^{-1}$ | Detailed information |
|---|---|---|
| Time-optimal | 12.2 | These values in this first group (divided by rows) were simulated by AirTraf. |
| GC FL290 | 12.2 | |
| GC FL330 | 11.3 | |
| GC FL370 | 10.8 | |
| GC FL410 | 10.9 | |
| Sutkus Jr et al., 2001 | 21.8 | Airbus A330-301 CF6-80E1A2, 1GE033 (1-9 km altitude band) |
| | 13.9 | (10-13 km altitude band) |
| Jelinek et al., 2004 | 11.33 | A330 (mean of 1318 flights, no profile completion option) |
| | 11.53 | A330 (mean of 1318 flights, complete all operations option) |
| Penner et al., 1999 | 7.9 - 11.9 | Typical emission for short haul |
| | 11.1 - 15.4 | Typical emission for long haul |





**Table 14.** Constraints on the structural limit weights (MTOW, MLW and MZFW) and one specified limit weight (MLOW) of A330-301 aircraft. $m_1$ and $m_{n_{wp}}$ correspond to the aircraft weight at the first and the last waypoints, respectively. OEW and MPL are given in Table 1.

| Constraint | Limit weight, kg | Description |
|---|---|---|
| $m_1 \leqq$ MTOW | 212,000 | Maximum take-off weight. A330-301 (weight variant 000 BASIC)[a] |
| $m_{n_{wp}} \leqq$ MLW | 174,000 | Maximum landing weight. A330-301 (weight variant 000 BASIC)[a] |
| Zero fuel weight $\leqq$ MZFW | 164,000 | Maximum zero fuel weight. MZFW = OEW + MPL. A330-301 (weight variant 000 BASIC)[a] |
| $m_{n_{wp}} \geqq$ MLOW | 150,120 | Planned minimum operational weight in the international standard atmosphere. MLOW = 1.2× OEW. A330-301[b] |

[a] EASA, 2013;  [b] Eurocontrol, 2011.