# Peer review of "Air Traffic Simulation in the Chemistry-Climate Model EMAC 2.41: AirTraf 1.0"

_Geoscientific Model Development, 2015_

## Referee Comment (RC1) · Anonymous Referee #1 · 1 Mar 2016

Summary This paper describes a new model which will eventually be used in calculating the climate impact of aircraft routes. There are several different parts to the model, which are detailed in the paper, including generating the route either by calculating a great circle or time-optimal route (the two constraints which are described and tested here), calculating the fuel use along the route, and the emissions for example of water vapour and NOx along the route. A thorough assessment is made of the model and its ability to generate the routes and calculate the various parameters; the model performs well and appears to be fit for purpose. The paper is generally clear and the different components of the model are well-described. My only major concern regards the vertical flight profiles, please see the major comment below. I recommend the paper for

publication after revision.

General comment In the calculation of the time-optimal flights, the flight altitude is allowed to vary freely between FL290 and FL410. Some of the resulting time-optimal flight profiles display significant altitude changes during the flight, as shown in Figure 14 (b), where the flight altitude profile along the flight is 'm' shaped (i.e. increases, decreases, increases and then decreases again). This is in contrast to the familiar stepped profiles, where the aircraft altitude increases are done as step climbs when enough fuel has been burned off, or alternatively a gradual increase in height to a maximum cruising altitude, followed by a descent. It is difficult to see how (or why) an aircraft would do this 'm' profile in real life, given air traffic constraints, for example. Given how unusual these profiles are, some justification or explanation for why these profiles are allowed in this study should be given, as well as a comment on how realistic it would be for an aircraft to fly this profile.

Minor comments 1. p3 L61 – the Spichtinger et al (2003) study referenced by the authors analyses the vertical distribution of ice-supersaturated regions. The mean length of 150 km is from Gierens and Spichtinger (2000), as stated in the Spichtinger paper.

2. p3, final paragraph (L84 - 99). As I understand it, the aim of the study presented in the paper is to introduce, describe and validate the AirTraf model, not to investigate 'how much the climate impact . . .can be reduced by aircraft routing' – that is a separate study which would use AirTraf. This paragraph is therefore confusing to the reader, and there is extra detail here which is not all necessary to understand this paper. Please re-phrase the aims of the study to be consistent with what is presented in the paper, remove unnecessary detail about future studies and I also suggest removing Figure 1 which is not needed here.

3. p4 L121, p5 L159 and caption of Figure 3 - "one-day flight plan". It is not clear what you mean by this phrase (it sounds like you are referring to a single flight on a

single day, rather than many flights on a single day). It would be helpful to give a short description the first time you use the phrase.

4. p4 L126 – "AirTraf continuously treats overnight flights". What does this mean?

5. p7 L201 – "local weather conditions provided by EMAC". Specifically, the wind field is used?

6. p8 L260 – You assume that the sum of the alternate, reserve and extra fuel is 3% of the total fuel. Is there any justification for this number? I acknowledge that this kind of data is almost impossible to get from airlines, but have other studies used a similar number, for instance?

7. p20 L647 – 656. The explanation of why the flight altitude profiles are optimal is that the flight changes altitude to benefit from changes to the true airspeed and to increase its tailwind or reduce its headwind. The argument is currently not well supported by the figures (Figure 16, and S5 and S6) which show the altitude distribution of the true airspeed and tailwind indicator. The variations in these quantities at flight altitude are hard to see, since the vertical scale on the plots is 0 – 15 km. The case might be made much clearer simply by re-plotting these figures with a limited altitude range (i.e. only plot the range of altitudes relevant to the aircraft), and re-scaling the colour bar.

8. p22 L727 – 729 and Figure 22. "The maps show the time-optimal case has low values of the fuel use" (compared to the great circle case). The great circle case at FL290 clearly has a lower fuel use, as shown in Table 11. However, I do not think this is clear from Figure 22; the flights in the time optimal case are spread over a larger area than in the great circle case therefore it is difficult to assess objectively whether the fuel use is higher or lower in the time-optimal case. I do not think that this figure adds any weight to your argument. I suggest removing it.

9. p25 L824. I cannot find AirTraf or any status information for it on the list of submodels on the MESSy website (accessed on 24/02/2016).

10. Figure 15, 16, S4 – S6 – Please add units to the colour bar and/or text.

11. Table 8. It is difficult to compare the flight time for the time-optimal with the great circle at different altitudes, since the mean flight altitude of the time-optimal flights is given in m and the altitude of the great circle flights in feet. Please add either the mean flight altitude in feet for the time-optimal flights, or the flight altitude in m for the great circle flights to aid the comparison.

12. Table 11, Caption. 'sum of flight time, fuel use, NOx and H2O emissions....'. This implies that the table shows the quantity flight time + fuel use + NOx + H2O, when in fact they are displayed separately. Please re-phrase.

---

## Referee Comment (RC2) · Anonymous Referee #2 · 7 Mar 2016

Review on "Climate Assessment Platform of Different Aircraft Routing Strategies in the Chemistry-Climate Model EMAC 2.41: AirTraf 1.0" by Hiroshi Yamashita et al.

This paper presents a development of "module" adapted to the climate chemistry model ECHAM5/MESSY in order to calculate the climate impact of aircraft routes. Only one part of the module needed has been included in the model and presented in this paper: the part generating the route and only in the case of great circle (simple) or time-optimal route (optimisation). From these two routing the module calculates fuel use, and some emissions ($H_2O$ and $NO_x$ only), these parameter are assessed with real data. The module is tested over one winter day data over the North Atlantic corridor. In its present form I unfortunately cannot recommend the publication of the paper in

[Figure]

Geoscientific Model Development for several reasons that I will be listing. I would strongly recommend the editor to request a severe revision before publication. The time optimal calculation module may be of interest for modellers. The optimisation module description as well as the size of the population to be included in the optimisation to converge toward optimal time may be presented in a revised paper.

1) My first problem is the presentation of the subject within most of the article (title, abstract and even structure of the manuscript). The focus seems to be in the "optimal routing for climate impact reduction" when you check the paper, however the reader is disappointed as the presented module is not doing that at all – only optimising for travel time. The manuscript needs to be reshaped completely to acknowledge that fact. 2) I am also extremely disappointed in the fact that a part of the paper is dedicated in presenting and comparing "great circle routing" calculations. This is nothing new, and no advance in modelling or science presented. This part should be cut down and re-moved from the discussion. The more important difference could come from the fact the earth is not a perfect sphere or maybe taking into account flight altitude. The table 4 is comparing calculation with decimal and no-decimal data when the difference is in the decimal value. 3) Concerning the "optimisation routing" for flying time the validation over the North Atlantic is interesting but what would happen with a case of congested space or restricted space (military)? Please do tests in different part of the world or at different season. 4) Moreover I am unsure of the complete philosophy of the inclu-sion of the "optimisation" module in the ECHAM5/MESSY model. I understand well the impact of local weather and composition on the impact the aircraft routing will have on climate change. However I am short in understanding the need of the online opti-misation as I don't see the effect of "climate optimal routing" on the climate model – would a simple offline calculation not enough to determine this potential "climate opti-mal routing" (the day the full module will be ready) as well as making the "optimisation" easier to be adapted to other climate-chemistry model output? 5) Finally I am unhappy with the fact that the only simple "time optimal routing" (optimising only for one vari-able) the weather situation if fixed for the entire flight. What would happen in the case

of multi optimisation when you have to trade-off between time, fuel use, and different emissions? Could you comment on the impact on contrail formation from long flights? "-For all routing options, local weather conditions provided by EMAC at t = 1 (i.e. at the departure day and time of the aircraft) are used to calculate the flight trajectory. The conditions are assumed to be constant during the flight trajectory calculation-"making the model as simple as an offline module but complicated as an inside module of an already complex model?

---

## Referee Comment (RC3) · Anonymous Referee #3 · 7 Mar 2016

**Review of**
**Climate assessment platform of different aircraft routing strategies in the chemistry-climate model EMAC2.41 : AirTraf 1.0**

March 7, 2016

**1  Introduction**

The manuscript is well structured, and different aspects of AirTraf are explained by a nice equilibrium of description and examples. The motivation of the work is reasonably well explained. Figures and tables are informative. There is a substantial comparison with results from other studies to give confidence in the results obtained here.

Implementing aircraft routing strategies in a general circulation model or a numerical weather prediction model is not an easy task. Arriving at the status as described here in the manuscript is already a considerable achievement. However, as the tool is not finished, one wonders whether it is useful to describe the tool in its current status (with only 2 of the 7 optimization options implemented, fuel consumption due to climbing not included, the meteorological fields in the optimization are the ones at the start of the flight, ...).

Publishing the manuscript now shows the status of the work. It makes clear that for specific options the optimization works, and it can trigger discussion with other researchers/institutes on the approaches chosen (is the optimization working well, could other optimization routines be faster, ...).

I think the manuscript is worth publishing, but is should be considerably improved in several ways. A list of principal remarks is given below, followed by a list of more specific comments. I hope the authors will take them into consideration, and if not give a sound argumentation why they do not.

**2  Principal remarks**

**Work in progress** The manuscript describes a submodel in MESSy which works, but is not finished yet (only 2 of the 7 optimization options are in place). Why not waiting until all the work is finished? One has to guarantee that this manuscript remains valid and worth all the work once the remaining parts come into place, and that this document is therefore worth publishing.

**Language** There is a lot of improvement needed for the language. The use of articles (a/an/the/none) should be improved. Specific expressions (e.g., "trajectories as longitude vs altitude, trajectories as location" or "number of $n_p$", ...) should be modified.

**CP in trajectories** Concerning the treatment of CP points, I have several questions.

- As an example, 3 CPs have been used for the geographical location, and 5 for the altitude. Is this fixed? Do all flights use the same number of CPs?

- For the 103 flights, which were primarily zonal, rectangles around the CPs could be described by using a range in latitude and longitude. How is the choice around the CPs when flights cross the equator, e.g., at an angle of 45°? What if flights go from low to high latitudes and defining regions whit fixed ranges in longitude makes them very different in size?

- For a given trajectory (which is a B-spline curve), how are the waypoints found? Are they equally spaced along that trajectory between the CPs? I am wondering whether it is possible to find explicit expressions for equidistant waypoints on a B-spline curve?

- In the example used, 3 CPs were used for the geographical location, 5 CPs for for the altitude, and 101 waypoints. However, the condition $(101-1)\mathrm{modulo}(5+1) = 0$ is not fulfilled. One also gets the impression

that the waypoints for the altitude and longitude are not located at the same place (although the manuscript confirms it actually is). Could this be clarified?

**GA algorithm** This algorithm is explained to some detail, but I suggest that all terms used should be explained to some extent (e.g., mating pool). One should also be informed on how the final solution is derived from the population in the last generation. Finally, the abstract uses some terminology related to the optimization routine (e.g., population), which are too technical to be mentioned in the abstract.

**Abstract, introduction, conclusion** The abstract is sometimes too much a summing up of what has been done, with vocabulary/terms which have no concrete meaning without a concrete context. There is also much more overlap between these three parts (abstract, introduction, and conclusion) than needed. The abstract should be written differently, and considerably improved.

**Sensitivity** In the approach followed here, quite some assumptions and simplifications are introduced. It would be useful to give the reader an idea of the impact of these assumptions on the results. A list of some of the assumptions is :

- line 274 : $dh(t)/dt = 0$ in Eq. (270).
- $M$ is set constant. Can this be varied slightly? Or have pilots only a very small envelope of allowed or possible speeds?
- What if weather not just from $t = 1$ is taken, but from the whole period of the flight?
- Leaving out the ascent and descend phase of the flight : how does this impact the optimization?

**Mathematical formulas** The mathematical expressions should be improved.

- In mathematical formulas, variables longer than one letter should be written straight.
- A lot of indices should be straight letters : $V_{\text{ground}}$, $V_{\text{wind}}$, ...
- After every equation, there should be a ”,” or ”.”, depending on the function of the equation in the sentence.
- Names of trigonometric formulas should not be italic : sin, cos, ...

**Climate model, long/short time scales** Why is this tool implemented in a climate model? To my opinion, the tool could also have been build such that it uses off-line 3-hourly meteo fields over the range of time it has flights which should be optimized : one thinks over a range of 1 to 10 days. The meteo data might come from a NWP, or a climate model.

Maybe the authors want to show that it is possible to have such a tool on-line in a NWP or GCM. However, in that case, I would have chosen for a NWP as that is the place where, if the tool is operationally used, might be most appropriate. What was the reason that the authors made the choice of implementing it in a climate model?

A reason I can imagine is that one could do tests like : how would the optimal routing be in a year 2100 climate, when global climate is considerably different from nowadays?

**Benchmarks** Is proving that the great circle option works well worth publishing and/or mentioning in an abstract? In addition, I think that the word benchmark puts more importance on a test than it actually deserves.

**Size of the document** The files are so large (30 MB) that people will have problems printing the documents. To my opinion it is mainly related to the figures which show different flight trajectories. I assume that the figures contain all the information from all trajectories, while a large central part of the figure is just black. These figures should be made in such a way that they become much smaller in size, without loosing their precision.

**3 Comments on the text**

**Page 1**

**p 1, l 1–5** : The sequence of the first three sentences is a bit strange. I would even skip the first sentence (as it says the same as the first 7 words of sentence 3).

**p 1, l 3–6** : ”building a climate-friendly”, ”for a sustainable development”, ”is an important approach”. It makes me wonder whether this is not a too optimistic view on aviation.

**p 1, l 9** : "stable" gas. This is not precise enough.

**p 1, l 9** : "vary regionally". I would rather use something like "inhomogeneous distribution".

**p 1, l 11** : "on long time scales". I assume that the tool takes into account climate impacts on long time scale, via e.g. the CCFs. However, the tool itself is an optimization of only the flights planned within the next few days. There should be no confusion about these very different aspects.

**p 1, l 15** : were → are (because you describe the functioning of a tool).

**p 1, l 15** : DLR. This abbreviation should be explained.

**p 1, l 16–17** : "with respect to routing options" : vague.

**p 1, l 17–18** : "two benchmark tests ... for great circle and time routing options" : sounds a bit strange → "benchmark tests ... for the great circle and time routing options".

**p 1, l 19** : "by other published code" : vague, and inappropriate language for an abstract.

**p 1, l 20** : "optimal solution" → "optimal solution found by the algorithm" (distinguish whether it relates to the real optimal solution, or to the best estimate found by the optimization routine).

**Page 2**

**p 2, l 22** : "initial population" : as such, this is too technical for an abstract. I suggest to skip this from the abstract, or one could also choose to describe a bit better the optimization algorithm/methodology in the abstract.

**p 2, l 22–23** : "We found that the influence was small (around 0.01 %)" : I suggest to combine this into one sentence with the former sentence.

**p 2, l 24** : "function evaluations", "generation sizing" : too technical for an abstract.

**p 2, l 27** "one-day AirTraf simulations are demonstrated ..." : vague.

**p 2, l 27** : specific winter day → typical winter day.

**p 2, l 29** : "for the two options" : it is a long time ago that these were mentioned. So maybe express them explicitly again.

**p 2, l 30** : for all airport pairs : too vague for an abstract.

**p 2, l 30–31** : "reflecting" local weather → taking into account (?).

**p 2, l 31** : verified → confirmed.

**p 2, l 32** : "comparable to reference data" : too vague.

**p 2, l 34** : "with increasing the number " → "with the increasing number".

**p 2, l 35** : "a major problem" : too vague.

**p 2, l 35** : "At present" → Nowadays, currently, ... .

**p 2, l 35–37** : aircraft emission impacts contribute 4.9% of total anthropogenic radiative forcing : skip "impacts", as radiative forcing is an impact; 4.9 → to 4.9 ; of total → "of the total".

**p 2, l 39** : will grow → might grow.

**p 2, l 40** : the value of 4.9% → a value of 4.9%.

**p 2, l 41** : indicates → implies.

**p 2, l 41–42** : "and therefore environmental impacts from aviation increase" : try to avoid to have twice "increase" in this sentence.

**p 2, l 42–43** : This sentence sounds more positive than one can possibly defend.

**p 2, l 47** : contrail → contrails.

**p 2, l 49** : depends → depends partially.

**p 2, l 49–51** : What follows behind the ":" is not an explanation from what is said before ":".

**p 2, l 50** : geographic → geographical (both are possible).

**p 2, l 51 – p3, l 59** : "... and affect the atmosphere from minutes to centuries." Minutes probably refers to the time scale for disappearance of some chemical perturbations. However, every appearance (even if it is only a few minutes) of a GHG, has a century-timescale effect. Although I think I understand what the authors want to say, I think that the whole paragraph is rather inaccurate, and should be rewritten more precisely.

**Page 3**

**p 3, l 61** : "150 km horizontally" : maybe distinguish two directions (is it perpendicular to the flight path, or along the flight path). Isn't this 150 km much too specific? Isn't there a very broad spectrum?

**p 3, l 63** : There "are" two options ... : this sounds very optimistic.

**p 3, l 64** : "approaches" → measures.

**p 3, l 69** : "... are optimized with respect to time and economic costs." : if both are taken into account, how are they weighted?

**p 3, l 69** : "fuel, crew, operating costs" : isn't fuel part of the operating costs?

**p 3, l 72** : "systematic routing changes" : reading this, one gets the impression that there are different options. However, later it is reduced to just "i.e. flight altitude change". I suggest to just say "systematic flight altitude changes".

**p 3, 74** : has a strong effect on the reduction of the climate impact → has a strong impact on climate. (From the original formulation it is not clear whether the increase or the decrease in flight altitude leads to a reduction of the climate impact.)

**p 3, l 74–77** : "the" climate-optimized routing → climate-optimized routing.

**p 3, l 79** : "the" climate sensitive regions → climate-sensitive regions.

**p 3, l 80** : "This study" → "That study".

**p 3, l 81** : by only small increase → by only a small increase.

**p 3, l 80–81** : This study reported: "large reductions ..." → That study reported that large reductions ...

**p 3, l 82** : useful : is useful what one wants to express?

**p 3, l 85–86** : The current study wants apparently to investigate something (how much the climate impact of aircraft emissions can be reduced) that already has been investigated before (see lines 80–81 : large reductions in the climate impact of up to 25% can be achieved). One should be more specific of what the current study will do extra with respect to the former study.

**p 3, l 84–87** : Do you mean by "this study" = "this manuscript"? Or is "this study" broader? After reading the manuscript, I have the impression that line 84–85 is not what is answered by this manuscript.

**p 3, l 87** : The first step "is" → The first step "was".

**p 3, l 87–89** : The first step is to investigate specific past weather situations, in particular the climate impact of locally released aircraft emissions → The first step was to investigate the influence of specific weather situations on the climate impact of aircraft emissions.

**p 3, l 89** : "The resulting data are ..." : too vague. Maybe one could say : "This results in climate cost functions ...".

**p 3, l 91** : Why is $CO_2$ in this list? I can understand that the impact of adding $CO_2$ depends on the altitude, but this comes a bit unexpected after formulating earlier that $CO_2$ is well-mixed.

**p 3, l 91** : "They are specific climate metrics, i.e. climate impact per unit of emission" → "per unit amount of emission."

**p 3, l 92** : "and are used ..." → "will/might be used".

**Page 4**

**p 4, l 92** : "In a further step, weather proxies are identified for the specific weather situations." It is not clear whether this has been done.

**p 4, l 102–104** : "A benchmark test for the great circle routing option is performed and ..." : the part before and after the "and" actually express more or less the same.

**p 4, l 103** : "by other published code" : too vague.

**p 4, l 103–104** : "Another ... also ..." : I suggest to skip one of these words.

**p 4, l 103–105** : I would transform this into one sentence.

**p 4, l 105–106** : This sentence is too technical with "population" and "generation sizing".

**p 4, l 107** : "consistency" is too general. One has not enough background information at this point in the text to understand this.

**p 4, l 108** : "states" : I suggest to use another word.

**p 4, l 112–116** : This paragraph should be rewritten.

**p 4, l 112** : "reasonable" : I think this is not enough as a motivation.

**p 4, l 113** : "because we perform global air traffic simulations on long time scales considering local weather conditions." : I think this is a vague argumentation.

**p 4, l 114** : "geographic location and altitude at which emissions are released should be also considered" : vague.

**p 4, l 115** : This is maybe the main reason why the effort is done to implement AirTraf in a climate model, and not just in a NWP, or using off-line available weather forecasts. So make this more explicit, and give examples of which climate impacts can be evaluated.

**p 4, l 117** : Explain what "entries" are.

**p 4, l 121–124** : This sentence should be improved. You have to put "here PE is synonym to MPI task" possibly between brackets. I am also not sure whether "while" is the most appropriate word to use here.

**p 4, l 125** : I think one should be more specific about what a "time loop" is : isn't rather meant "time step"?

**p 4, l 125–126** : Thus, naturally short-term and long-term simulations consider the local weather conditions for every flight in EMAC. I think this should be explained more clearly.

**p 4, l 126–127** : "(AirTraf continuously treats overnight flights)" : this is not logically related to the sentence it is attached to. What is meant by this? Because the weather patterns used in AirTraf are the ones at the time of take-off, it seems to me that there is no large complexity about it. Is it therefore still worth mentioning?

**Page 5**

**p 5, l 131–132** : What is meant by these "global fields"? Give examples.

**p 5, l 132–134** : What is meant by the sentence "Other evaluation models ... on the climate impact"? I suggest to make this more concrete.

**p 5, l 135–136** : "$R_E$=6371 km" : I don't know whether this level of detail should be mentioned in the manuscript.

**p 5, l 137–138** : The Mach number is a ($\rightarrow$ "the") velocity divided by a ($\rightarrow$ "the") speed of sound.

**p 5, l 138** : "true air speed" $\rightarrow$ "the true air speed". Maybe add to the sentence : "When an aircraft flies at a constant Mach number". Isn't "vary along flight trajectories" enough? I don't think that "latitude, longitude, altitude and time" should be added. If one really wants to be more specific, I would rather add temperature and wind speed as factors modifying the true air speed and ground speed.

**p 5, l 142** : limits rates $\rightarrow$ limit rates.

**p 5, l 142** : Explain "semi-circular rule", and "sector demand analysis".

**p 5, l 144** : "mention" : I do not think this is the appropriate wording.

**p 5, l 149** : What is meant by "interactions with human influences"?

**p 5, l 153** : T42L31ECMWF-resolution $\rightarrow$ T42L31ECMWF resolution

**p 5, l 159** : Can it exist out of more than one day? On page 6, line 163 : "Any arbitrary number of flight plans is applicable to AirTraf". So one can give flight plans for many days at once?

**p 5, l 160** : of A330-301 $\rightarrow$ of an A330-301 aircraft.

**p 5, l 162** : a departure time $\rightarrow$ the departure time.

**p 5, l 162** : as values [-90,90] $\rightarrow$ as values in the range [-90,90].

**Page 6**

**p 6, l 164** : the data are required $\rightarrow$ these data are required.

**p 6, l 165** : "As for ..." $\rightarrow$ "Concerning ...".

**p 6, l 166** : flows (plural) while index (singular).

**p 6, l 168** : What is meant by an "overall" weight factor?

**p 6, l 171** : are described "here" step by step.

**p 6, l 172** : a flight status $\rightarrow$ the flight status.

**p 6, l 178** : moving aircraft position $\rightarrow$ aircraft position calculation.

**p 6, l 182–183** : differ to $\rightarrow$ differ from.

**p 6, l 184** : can be used $\rightarrow$ can currently be used.

**p 6, l 187** : for a selected option $\rightarrow$ for the selected option.

**p 6, l 191–194** : Why adding these sentences? It makes the text confusing. In addition, it is not well defined how an optimization might work when one optimizes according to two criteria (time and cost). One should also mention then how to weight or compare both (trade-off between them).

**p 6, l 197** : The CCF is $\rightarrow$ The CCFs are.

**p 6, l 199** : "total" climate impacts versus "some" aviation emissions : this sounds strange.

**Page 7**

**p 7, l 211** : $n_{wp-1} \to n_{wp} - 1$.

**p 7, l 212–213** : calculation/calculation/calculate : try to vary the wording more.

**p 7, l 218–219** : corresponding to time steps → corresponding to "the" time steps.

**p 7, l 219–220** : "present" and "previous" is a bit vague : isn't it the position at the beginning of a time step of EMAC, and at the end of a time step?

**p 7, l 220** : "a" present and previous position → "the" present and previous position.

**p 7, l 221** : by real numbers of the waypoint index → by real numbers as a function of the waypoint index.

**p 7, l 224** : I would rather say : "This means that the aircraft moves 100% of the distance between $i = 1$ and $i = 2$, and 30% of the distance between $i = 2$ and $i = 3$ in one time step."

**p 7, l 233** : is used → are used.

**p 7, l 233** : This is a little bit inaccurate (see also Fig. 4). Assess the impact of this inaccuracy.

**Page 8**

**p 8, l 237** : "If $t \geq 2$ of the day" : express this better.

**p 8, l 239** : without recalculating flight trajectory and fuel emissions → without recalculating the flight trajectory or fuel emissions.

**p 8, l 240–241** : "For more than two consecutive days simulations" → "For simulations longer than two days".

**p 8, l 243** : Twice "calculation".

**p 8, l 246** : are used → is used.

**p 8, l 246–247** : the first trip fuel estimation → a first trip fuel estimation.

**p 8, l 247** : the second fuel calculation : a bit vague. Maybe mention that it is more detailed.

**p 8, l 256** : mean flight altitude of the flight → mean altitude of the flight.

**p 8, l 260** : it is assumed as → it is assumed to be.

**Page 9**

**p 9, l 274–275** : "For an aircraft in cruise ..." : express this better.

**p 9, line 276–278** : One should have a "," or a "." after most of the formula.

**p 9, line 280** : The numerical value of $\rho_i$ is not given in Table (2) (as for $S$, $C_{D0}$ and $C_{D1}$ in Table 1).

**p 9, l 281** : a fuel flow → the fuel flow.

**p 9, l 282** : I suggest to skip "for jet aircraft".

**p 9, l 283–284** : "," after the equations.

**p 9, l 287** : Oneday : I suggest to find another name for this variable in the manuscript. In addition, its units in Table 1 should be "sec day$^{-1}$".

**p 9, l 289** : "reflects" → "incorporates" or "is impacted by".

**p 9, l 290** : $(m) \to (m_i)$.

**p 9, l 294** : next to the last → at the one but last.

**p 9, l 296–297** : I do not think this last sentence gives new information. Or formulate it nicer.

**Page 10**

**p 10, l 302** : first → First.

**p 10, l 310–311** : corresponding sea level values → corresponding values at sea level.

**p 10, l 314–315** : ”,” after equations.

**p 10, l 327** : ”... and $q_i$ is the specific humidity at $h_i$” : mention units of $q_i$ ($\mathrm{kg\,kg^{-1}}$, $\mathrm{g\,kg^{-1}}$, ...).

**p 10, l 329** : pre-calculated → calculated.

**p 10, l 330–331** : ”,” after equations. I do not think it is a good idea to have variables whit names as $NO_{x,i}$ and $H_2O_i$. I would rather use names like $m_{NO_x}$.

**Page 11**

**p 11, l 339** : one-day → one day of.

**p 11, l 343** : works → works only.

**p 11, l 351** : arctan, sin, cos, ... should not be italic.

**p 11, l 351** : ”,” after equation.

**p 11, l 362** : Why mentioning ”km” here? Better to write on line 355 : $d_i$ (km).

**p 11, l 363** : i.e. the → i.e.

**p 11, l 365** : ”based on Polar coordinates”? Explain this better.

**p 11, l 365** : therefore → in that case.

**Page 12**

**p 12, l 370** : of the $i$-th waypoint → at the $i$-th waypoint.

**p 12, l 371–372** : ”,” after equations.

**p 12, l 374** : where $M$ is ”the” Mach number.

**p 12, l 378–379** : Although it is mentioned that $V_{TAS}$, $V_{wind}$ and $V_{ground}$ are scalars, Eq. (25) on line 372 is actually a vector equation.

**p 12, l 386** : ”reflects” : this is not the only aspect which is reflected. I suggest to use ”incorporates”.

**p 12, l 390** : for the five → for five.

**p 12, l 393–395** : 180 → 180° (while ”deg” on line 397).

**p 12, l 398** : Missing deg?

**p 12, l 399** : ”;” → ”,”.

**Page 13**

**p 13, l 403** : varying $n_{\text{wp}}$ in "the range" [2, 100].

**p 13, l 404** : and the MTS $\rightarrow$ and MTS.

**p 13, l 406** : I do not think that $\Delta d_{\text{eq23,eq22}}$, etc. are appropriate choices for variable names. As these are difference, I think they should not not have a specific variable name attributed.

**p 13, l 409–410** : "shows" versus "showed".

**p 13, l 413** : I would not call it linear interpolation : one goes straight whereas the other follows an arc. Shouldn't you also add that $n_{wp}$ maybe should depend on the length of the flight?

**p 13, l 417** : with respect to the flight time routing option $\rightarrow$ with respect to the flight time.

**p 13, l 418** : algorithms $\rightarrow$ algorithm.

**p 13, l 422** : The ARMOGA $\rightarrow$ ARMOGA.

**p 13, l 424–425** : With a routing option $\rightarrow$ For each routing option (except ...). I also suggest to skip "on the selected routing" in the second part of the sentence.

**p 13, l 427** : Explain what an objective function in this context is.

**p 13, l 432-433** : "Is called "an" optimal solution" and "is called "the" true-optimal solution".

**p 13, l 434** : Say what is meant by converge : larger initial population, or just more generations?

**p 13, l 435** : Will every flight have the same size for its initial population, and the same number of generations? Is that independent of the length of the flight?

**Page 14**

**p 14, l 440–441** : I do not think that "definitions" is the appropriate word to be used here.

**p 14, l 441** : of objective functions $\rightarrow$ of the objective function.

**p 14, l 444** : used interchangeably to mean a flight trajectory $\rightarrow$ used interchangeably to flight trajectory.

**p 14, l 445** : $n_{dv} = 11$ should not be here.

**p 14, l 456** : centering $\rightarrow$ centered.

**p 14, l 463–464** : how are these waypoints calculated? Will the arc lengths be equal?

**p 14, l 458–459 and 470–471** : "GA provided the values" : Do you mean already the final optimal values?

**p 14, l 462** : Explain a little bit more a B-spline curve.

**p 14, l 464** : Are the waypoints on the B-spline curve still equidistant?

**p 14, l 461 and 472** : "Here $x_1$, ... indicate longitudes/latitudes/altitude values". Shouldn't this be mentioned earlier in the paragraphs, i.e. on lines 452 and 466?

**Page 15**

**p 15, 477** : where longitude-coordinate of waypoints → where "the" longitude of the waypoints.

**p 15, l 476–478** "where longitude-coordinate of waypoints is the same for the two curves." Is this true in the example here? The lon-lat curve contains 3 CPs and thus 4 intervals. The the lon-altitude curve contains 5 CPs and 6 intervals. The number of waypoints is 101, so 100 intervals. This is however not a multiple of 6, so I don't see that the longitude of the waypoints for both B-spline curves are automatically identical.

**p 15, l 479** : "provides initial values by random numbers" : this is too cryptic.

**p 15, l 481** : "The operator creates divers solutions defined by a fixed population size $n_p$." : This is a complicated way to say : "The operator creates $n_p$ different solutions (where $n_p$ is the population size)."

**p 15, l 481** : "a random set" : do you mean the random set which is just described (then I suggest to use "the"), or is it even another random set? I would put the sentence "GA starts it search with a random set of solutions (population approach)" at the beginning of the paragraph.

**p 15, l 483** : By summing the flight time for flight segments → by summing the flight time over all flight segments.

**p 15, l 483–484** : "The .. optimization solved here" : too cryptic and vague.

**p 15, l 485** : "Minimize" and "Subject to" should not be italic.

**p 15, l 490** : What is meant by "solutions that dominate it"?

**p 15, l 489–491** : Why is "rank" written in italic, but "fitness" not?

**p 15, l 493** : made → makes (because "are identified" on line 488).

**p 15, l 492** : What is meant by a "mating pool"?

**p 15, l 500** : "This operator was applied to each design value." : Isn't this said already in the sentence before?

**p 15, l 504** : "added a disturbance to the child solution." : It does if for both child solutions I presume.

**Page 16**

**p 16, l 515** : the population of "the" solutions → the population of solutions.

**p 16, l 517** : "an optimal solution is output." : How is that solution found based on the last generation?

**p 16, l 518** : "corresponding to the routing option" : I don't think this has to be repeated here.

**p 16, l 518** : "the best" : one cannot guarantee that it is the best I think.

**p 16, l 519** : "naturally" : is this the appropriate wording?

**p 16, l 521–522** : can be applied to any routing option (I thought that was not possible yet in version 1.0?) → could.

**p 16, l 529** : "As $V_{\text{TAS}}$ and $V_{\text{ground}}$ were set to $898.8\,\text{km}\,\text{h}^{-1}$" : Isn't it better to mention first explicitly that we have set $V_{\text{wind}} = 0$, and from that it follows that $V_{\text{TAS}}$ and $V_{\text{ground}}$ are $898.8\,\text{km}\,\text{h}^{-1}$ (and not set).

**p 16, l 531** : Maybe one should say why flying at FL290 will be faster than at other altitudes. I assume that this depends on the value of $T$. Are the initial and final points at FL290? Mention that $M = 0.82$.

**p 16, l 537** : total 1000 independent → a total of 1000 independent.

**p 16, l532–538** : Isn't the first experiment also included in the second setup?

**Page 17**

**p 17, l 540** : generation number $n_g$ → number of generations $n_g$.

**p 17, l 541** : Is "confirmed" the appropriate wording?

**p 17, l 542** : sufficiently come close → come sufficiently close.

**p 17, l 542, 543, 545** : the $f_{true}$ → $f_{\text{true}}$.

**p 17, l 545** : $\Delta f$ : you do not need an extra variable name for something you express only once.

**p 17, l 547** : What is meant by "diversity" of GA optimization?

**p 17, l 547–548** : we focus on the optimization results, which found the best solution → we focus on the optimization setup which gave the best solution.

**p 17, l 548** : "all the solutions" : Are these the $100 \times 100 = 10000$?

**p 17, l 548–549** : solutions explored by GA as longitude vs altitude (top) and as location. This should be worded correctly.

**p 17, l 552** : "To confirm the difference" : I don't think confirm is appropriate to be used here.

**p 17, l 554–555** : Isn't this conclusion too fast? What if the trajectory is not so zonal, but the trajectory crosses the equator at an angle of 45° : how would the CPs and regions around be defined?

**p 17 , l 552** : "confirm" is not appropriate here.

**p 17, l 552** : To confirm the dependence of optimal solutions on initial populations → To "analyze" the dependence of "the" optimal solution on "the" initial population, ...

**p 17, l 552–553** : I don't think one should use words like "best-of-generation".

**p 17, l 558–559** : corresponding to → for.

**p 17, l 653** : "there is a small degree of variation in the objective function". Stated like this, it gives the impression that a different objective function is used. Probably, what is meant is that the value of the objective function for the final flight is different.

**p 17, l 564** : Writing $f - f_{\text{true}}$ is a bit strange. For me, $f$ and $f_{\text{true}}$ are solutions, i.e. flights defined by $x_1$, ... $x_{11}$. Here, $f$ and $f_{\text{true}}$ seem to indicate the value of the flight time.

**p 17, l 569 and 570** : "number of $n_p$ and $n_g$" and "size of $n_p$ and $n_g$". One should use : "the value of $n_p$", or "the size of the population", not something hybrid like "the number of $n_p$".

**p 17, l 569** : "discover" : I suggest to use a different word.

**p 17, l 570** : "is problem dependent, e.g. weather situations" : this should be formulated properly.

**p 17, l 571** : "estimating appropriate $n_p$ and $n_g$ could be different" : I suggest to formulate this differently.

**Page 18**

**p 18, l 573–574** : unclear sentence. What is, e.g., the difference between accuracy of GA optimizations and variation in the optimal solutions? I also had the impression that the impact of the initial population was already studied in Sect. 3.2.5.

**p 18, l 574** : Skip "calculated".

**p 18, l 581** : the variation of the $\Delta f$ and the $s_{\Delta f}$ → Skip "the".

**p 18, l 582** : the $\Delta f$ → Skip "the".

**p 18, l 589** : that reduction → a reduction.

**p 18, l 591** : "by selecting $n_p$ and $n_g$ for different purposes." This should be formulated differently.

**p 18, l 595** : for demonstrations → for demonstration.

**p 18, l 596, 598** : Calculation conditions : too vague.

**p 18, l 598–599** : simulation"s" and simulation.

**p 18, l 605** : "On the other hand" → "In addition".

**p 18, l 606 − p19, l 607** : in [FL290, FL410] → in the range of ..

**p 18, l 607** : "and therefore" : I think $V_{\text{ground}}$ also varies for other reasons, e.g., due to varying wind speed and direction.

**Page 19**

**p 19, l 615** : Does "case" refers to just one flight, or to all 103 flights together?

**p 19, l 616** : It is initially unclear what "it" refers to.

**p 19, l 617** : "right" : This is maybe not the most appropriate wording.

**p 19, l 618** : by a small → by "using" a small.

**p 19, l 618** : "a small number of $n_p$" → "a small value of $n_p$", or "a small population size"

**p 19, l 619** : with sufficient accuracy → with "still" sufficient accuracy.

**p 19, l 620** : I think the title of Sect. 4.2 does not describe well the content : only one airport pair is discussed (Amsterdam - Minneapolis) really in depth. I suggest something more general.

**p 19, l 623** : trajectories : Is meant the final trajectories?

**p 19, l 627** : we have selected "the" three airport pairs → we have selected three airport pairs.

**p 19, l 633** : in [FL290,FL410] → in the range of [FL290,FL410].

**p 19, l 633–634** : "when calculating for the selected solutions" : This should be formulated better.

**p 19, l 634** : in the supplements → in the supplementary material.

**p 19, l 638–639** : east and west direction → eastern and western directions.

**p 19, l 639** : major wind component : What is meant by this?

**p 19, l 640–641** : at the $h$ → at $h$.

**Page 20**

**p 20, l 646** : Supplements → Supplementary material.

**p 20, l 647** : the behaviour of altitude changes → the behaviour of the altitude changes.

**p 20, l 647** : Fig. 16 plots → Fig. 16 shows.

**p 20, l 650–651** : this means tail winds ($\geq 1.0$) and head winds ($< 1.0$) to the flight direction : Formulate better.

**p 20, l 655, 662** : "reflects" → "takes into account", or "accounts for".

**p 20, l 658** : confirmed → compared. Skip "quantitatively".

**p 20, l 659** : as indicated → as shown.

**p 20, l 659–662** : decreased → is lower.

**p 20, l 664** : "sufficiently" : I think this is a bit vague.

**p 20, l 667** : that the reduction → a reduction.

**p 20, l 668** : "sizing" → "reducing" or "choosing properly".

**p 20, l 671** : This is not a nice first sentence for a paragraph.

**p 20, l 673–674** : trans-Atlantic Ocean → Atlantic ocean.

**p 20, l 675** : of "the" region → of "that" region.

**Page 21**

**p 21, l 681** : plot → show.

**p 21, l 683** : with linear fitted lines : be more precise.

**p 21, l 683** : increased → is higher.

**p 21, l 688–689** : which had high $V_{\text{TAS}}$ values → with high $V_{\text{TAS}}$ values.

**p 21, l 691** : increases → is larger.

**p 21, l 696** : increases → is larger.

**p 21, l 700** : Supplement → Supplementary material.

**p 21, l 703** : correctly selected the airspace : improve this formulation.

**p 21, l 705** : This behaviour of altitude changes → These altitude changes.

**p 21, l 705** : affects the variation in fuel consumptions → affects the fuel consumption.

**p 21, l 705** : the terms are used interchangeably to mean fuel flows : improve the formulation.

**p 21, l 708** : increases → is higher.

**p 21, l 708** : the mean value → the mean value of the fuel consumption.

**p 21, l 714** : increases → is higher.

**Page 22**

**p 22, l 718** : corresponding to "the 103" individual flights.

**p 22, l 718–719** : the similar figures → similar figures.

**p 22, l 720** : showed → show.

**p 22, l 720** : in the right-hand domain : choose a better expression.

**p 22, l 721** : decreases → is lower. Put "for all airport pairs" at the end of the sentence.

**p 22, l 723 and 725** : increased → increases.

**p 22, l 740–741** : "Consistency" : just by reading the section title, it is not clear what is meant by this.

**p 22, l 742** : were → are.

**p 22, l 742–743** : The data → Data.

**p 22, l 744** : "they" is ambiguous.

**Page 23**

**p 23, l 723** : I would not say explicitly that the table shows "a comparison".

**p 23, l 758 and 764** : literature → write the correct reference.

**p 23, l 758** : indicated → indicates.

**p 23, l 758–759 / 764–765** : Is it worth mentioning this?

**p 23, l 764** : "indicate" : I don't think this is the appropriate word.

**p 23, l 765** : "close" → "close to".

**p 23. l 769** : reference data → the reference data.

**p 23, l 774** : indication : shouldn't one use a different word?

**p 23, l 778** : decreased → is lower.

**p 23, l 783** : installed → contains.

**Page 24**

**p 24, l 787** : "duplicates" : What is meant by this?

**p 24, l 790** : for 103 flights → for "the" 103 flights.

**p 24, l 792** : safety flight operations → flight operations safety.

**p 24, l 794** : constrains to → constraints

**p 24, l 800–801** : This sentence should be improved.

**p 24, l 802** : to prevent "the" structural damage → to prevent structural damage.

**p 24, l 803** : "aircraft has" → "aircraft have" or "an aircraft has".

**p 24, l 803** : "to reduce below" → "to reduce until" or "to bring below".

**p 24, l 808** : Why not using $\leq$?

**p 24, l 806, 810** : of A330-301 → of an A330-301 aircraft.

**p 24, l 812** : more → higher.

**p 24, l 814** : Skip "calculations".

**p 24, l 816** : an submodel → a submodel.

**p 24, l 817** : "applied" : shouldn't it be "used"?

**Page 25**

**p 25, l 823–824** : What is meant by this sentence?

**p 25, l 829** : the benchmark test → a benchmark test.

**p 25, l 831–832** : by other published code : this is too vague.

**p 25, l 832** : the benchmark test → a benchmark test.

**p 25, l 836** : dependence on the initial population.

**p 25, l 835 and 838** : The fact that both values are 0.01 % is maybe not a good sign. I would think that you want the second one to be much smaller than the first one.

**Page 26**

**p 26, l 860 and 866** : Please be more specific about what "reference data" is.

**p 26, l 862** : close → (very) similar.

**p 26, l 869** : fuel use calculation model → fuel use model.

**p 26, l 871** : "is sufficient" : But some things do not work yet?

**p 26, l 871** : "a" reduction potential → "the" reduction potential.

**4   Remarks on figures**

**Figure 1** : I presume parts of this are already done in other optimized studies. Mention what is already done, what is part of this manuscript, and what shall be done in the future.

**Figure 7** : Bizarre first sentence in caption. Consisting of → determined by. $\Delta\lambda_{airport} \to \Delta\lambda_{\mathrm{airport}}$.

**Figure 8** : Conclusions/observations/interpretations should not be written in figure captions. I would not use the word "discovers".

**Figure 9** : Change the first sentence. "The population size $n_p = 100$ ..." : This is not a good sentence. Replace "=" by "is".

**Figure 10** : Skip "Comparison of".

**Figure 11** : Don't use expressions like "Best-of-generation". "vs function evaluations" → "vs number of function evaluations". $f_{true} \to f_{\mathrm{true}}$. Change "On the ... and ...".

**Figure 17** : Don't use expressions like "Best-of-generation".

**Figure 22** : Shouldn't one have as unit for the emissions : $\mathrm{kg(fuel)\,m^{-2}\,s^{-1}}$? The figures are 2-hourly averages. However, the ranges are not clear from just mentioning 14:00:00, 16:00:00, 18:00:00, 20:00:00. Is it 14:00:00–16:00:00, 16:00:00–18:00:00, ..., or rather 12:00:00–14:00:00, 14:00:00–16:00:00, ...

**5   Comments on tables**

**Table 1** 101.325 → 101,325. Why is there "(jet)" at the end of the line with $C_{f1}$? There should be a small space between "kg" and "min". I would not give a variable the name "Oneday". $P_0$ and $T_0$ are not total pressure or temperature, but reference pressure and temperature.

**Table 2** : $n_{wp-1} \to n_{wp} - 1$.

**Table 4** : I think it makes no sense to introduce all these new variable names. Put in the heading (first row) of the table just : "Eq. (22)", "Eq. (23)", ...

**Table 5** : For population size and generation number : "$\cdots$" → "$\ldots$".

**Table 9** : "that of" → "average of". Why "medium"?

**Table 12** : Skip "Comparison of".

**Table 14** : "Constraints on" → "Constraints from". Why not just using $\geq$ and $\leq$? Why have on all the four lines A330-301 after some "." at the end of the line?

**6   Supplementary material**

Fig. S1 and S2 : including "the" time-optimal flight trajectories.

Fig. S3 and S4 : Skip "Comparison of".

Fig. S7 : Skip "that".

---

## Author Comment (AC1) · 17 May 2016

We are most grateful to the referee #1 for the very helpful and encouraging comments on the original version of our manuscript. Here are our replies:

- Summary: This paper describes a new model which will eventually be used in calculating the climate impact of aircraft routes. There are several different parts to the model, which are detailed in the paper, including generating the route either by calculating a great circle or time-optimal route (the two constraints which are described and tested here), calculating the fuel use along the route, and the emissions for example of water vapour and $NO_x$ along the route. A thorough assessment is made of the model and its ability to generate the routes and calculate the various parameters; the model performs well and appears to be fit for purpose. The paper is generally clear and the different components of the model are well-described. My only major concern regards the vertical flight profiles, please see the major comment below. I recommend the paper for publication after revision.

  Reply: We thank the referee #1 for these positive comments. We will reply to your major concern regarding the vertical flight profiles in the "General comment" section.

- General comment: In the calculation of the time-optimal flights, the flight altitude is allowed to vary freely between FL290 and FL410. Some of the resulting time-optimal flight profiles display significant altitude changes during the flight, as shown in Figure 14 (b), where the flight altitude profile along the flight is 'm' shaped (i.e. increases, decreases, increases and then decreases again). This is in contrast to the familiar stepped profiles, where the aircraft altitude increases are done as step climbs when enough fuel has been burned off, or alternatively a gradual increase in height to a maximum cruising altitude, followed by a descent. It is difficult to see how (or why) an aircraft would do this 'm' profile in real life, given air traffic constraints, for example. Given how unusual these profiles are, some justification or explanation for why these profiles are allowed in this study should be given, as well as a comment on how realistic it would be for an aircraft to fly this profile.

  Reply: In this paper, we have confirmed that the 'm' shaped flight profile effectively takes advantages of the wind fields and leads to the time-optimal solution (please see on page 20 line 647 – 657).

  As the referee #1 pointed out, AirTraf allows aircraft to vary flight altitudes freely between FL290 and FL410. Here, the AirTraf submodel is used to investigate an optimization strategy of aircraft routing for minimizing the climate impact of aircraft emissions and to show its mitigation gain for the future. The regions with high climate impacts, e.g. regions where contrail form, are often very shallow (vertically). In order to investigate how such regions can be avoided more flexibility in the routing options is required. Hence, in this approach it is necessary for aircraft to have a high flexibility for flight profiles to explore widely the possibility of minimizing climate impact by aircraft routing.

  If the optimization strategy is found, it will be proven by a more realistic air traffic simulation model, considering realistic air traffic constraints. The "m" shaped flight profile will be modified to adapt to the constraints (probably stepped profiles). The development of the realistic air traffic simulation model is addressed by research groups of DLR-Hamburg and DLR-Braunschweig in the DLR Project WeCare.

  We will add this information in the revised manuscript: on page 14 line 472, "Here $x_7$ to $x_{11}$ indicate altitude values**; these values vary freely between FL290 and FL410 to explore widely the possibility of minimizing climate impact by aircraft routing.**"

  Further, we will add the text: on page 19 line 635, "..., while that for west-bound showed large altitude changes, i.e. it climbed, descended, climbed and **then descended** again."

- Minor comments:
  (1) p3 L61 – the Spichtinger et al (2003) study referenced by the authors analyses the vertical distribution of ice-supersaturated regions. The mean length of 150 km is from Gierens and Spichtinger (2000), as stated in

the Spichtinger paper.

Reply: Thank you very much. We will refer the paper in the revised manuscript: on page 3 line 61, "... extend a few 100 m vertically and around 150 km horizontally with a large spatial and temporal variability (**Gierens et al., 2000,** Spichtinger et al., 2003)." We will also add the paper to References in the revised manuscript: on page 27, "**Gierens, K., and Spichtinger, P.: On the size distribution of ice-supersaturated regions in the upper troposphere and lowermost stratosphere, Annales Geophysicae, vol. 18, No. 4, 499–504, 2000.**"

- (2) p3, final paragraph (L84 – 99). As I understand it, the aim of the study presented in the paper is to introduce, describe and validate the AirTraf model, not to investigate 'how much the climate impact … can be reduced by aircraft routing' – that is a separate study which would use AirTraf. This paragraph is therefore confusing to the reader, and there is extra detail here which is not all necessary to understand this paper. Please rephrase the aims of the study to be consistent with what is presented in the paper, remove unnecessary detail about future studies and I also suggest removing Figure 1 which is not needed here.

Reply: As the referee #1 noted, this paragraph is confusing. On the other hand, we think that the information of this paragraph is helpful for readers to understand the motivation and background for the AirTraf development. To improve the manuscript, we will remove Fig. 1 and rephrase the aims of this study: on page 3, final paragraph (line 84 – 99),

"**This paper presents a new assessment platform AirTraf (version 1.0, Yamashita et al., 2015) that is a global air traffic submodel coupled to the Chemistry-Climate model EMAC (Jöckel et al., 2010). Here, we describe the AirTraf in detail and validate it. The AirTraf will eventually be used to investigate how much the climate impact of aircraft emissions can be reduced by aircraft routing. The research road map for our study is as follows (Grewe et al., 2014b).** The first step is to investigate ...".

- (3) p4 L121, p5 L159 and caption of Figure 3 – "one-day flight plan". It is not clear what you mean by this phrase (it sounds like you are referring to a single flight on a single day, rather than many flights on a single day). It would be helpful to give a short description the first time you use the phrase.

Reply: We will add the text in the revised manuscript: on page 4 line 121, "As shown in Fig. 3, the one-day flight plan**, which includes the flights of a single day,** is decomposed for a number of processing elements (PEs)."

- (4) p4 L126 – "AirTraf continuously treats overnight flights". What does this mean?

Reply: Some international (long-distance) flights fly over two days. For example, NH215 departs at MUC on 21:35 and arrives at Tokyo on 15:50 + 1day. AirTraf can simulate the flight correctly. We will rewrite the text in the revised manuscript: on page 4 line 125, "Thus, naturally short-term and long-term simulations consider the local weather conditions for every flight in EMAC (AirTraf continuously treats overnight flights **with arrival on the next day**)."

Further, from the referee #3 comment on "p 4, l 126 – 127", the text of the sentence "(AirTraf continuously treats overnight flights **with arrival on the next day**)" will be moved from the current position to an appropriate position in the manuscript, which is related logically: finally, on page 4 line 125, "Thus, naturally short-term and long-term simulations consider the local weather conditions for every flight in EMAC "; and on page 7 line 225, "$pos_{new}$ and $pos_{old}$ are stored in the memory and the aircraft continues the flight from $pos_{new}$ = 2.3 at the next time step **(AirTraf continuously treats overnight flights with arrival on the next day)**."

- (5) p7 L201 – "local weather conditions provided by EMAC". Specifically, the wind field is used?

Reply: Specifically, temperature and wind fields are used here to calculate a flight trajectory. On pages 6 – 8

in section 2.4, we describe the overview of calculation procedures briefly. Thus, we describe on page 7 line 201 as, "For all routing options, local weather conditions provided by EMAC at $t = 1$ (i.e. at the departure day and time of the aircraft) are used to calculate the flight trajectory."

In the following section, this trajectory calculation method is described in detail. For great circle routing option, on page 12 line 375 in section 3.1.1, "Temperature $T_i$ and three dimensional wind components ($u_i$, $v_i$, $w_i$) of the $i^{th}$ waypoint are available from the EMAC model fields at $t = 1$." For the time-optimal routing option, on page 15 line 487, "... where $d_i$ and $V_{ground,i}$ are calculated by Eqs. (23) and (25), respectively ($V_{TAS,i}$ and $V_{wind,i}$ are calculated as described in Sect. 3.1.1)."

- (6) p8 L260 – You assume that the sum of the alternate, reserve and extra fuel is 3% of the total fuel. Is there any justification for this number? I acknowledge that this kind of data is almost impossible to get from airlines, but have other studies used a similar number, for instance?

  Reply: According to general fuel planning regulations, e.g. JAR-OPS 1.255[1], an additional 3% of the total fuel is considered as contingency fuel in the fuel planning assuming an en-route alternate aerodrome can be found on any mission whereas alternate, final reserve, additional and extra fuel are neglected as their contri bution to the overall fuel amount is very small on long-haul flights. Although the fuel planning process of Air Traf, which is described on page 8 – 9 in section 2.5, is not exactly the same as JAR-OPS 1.255, the 3% as sumption (calculated by Eq. (2) on page 8) as the entire reserve fuel is not far from reality.

  Further, we will delete the sentence related to this matter: on page 8 line 265, "**A refined fuel estimation will be employed for calculating m$_{nwp}$ in future.**" will be deleted in the revised manuscript, since the sentence is not necessary for our argument here.

  [1] The Joint Aviation Authorities Committee, "Joint Aviation Requirements: JAR-OPS 1, Commercial Air Transportation (Aeroplanes)", 1-D-4.

- (7) p20 L647 – 656. The explanation of why the flight altitude profiles are optimal is that the flight changes altitude to benefit from changes to the true airspeed and to increase its tailwind or reduce its headwind. The argument is currently not well supported by the figures (Figure 16, and S5 and S6) which show the altitude distribution of the true airspeed and tailwind indicator. The variations in these quantities at flight altitude are hard to see, since the vertical scale on the plots is 0 – 15 km. The case might be made much clearer simply by re-plotting these figures with a limited altitude range (i.e. only plot the range of altitudes relevant to the aircraft), and re-scaling the colour bar.

  Reply: We think that the referee's suggestion is right. On this matter, we have a reason why we used the vertical scale on the plots as 0 – 15 km. In Figs. 16, S5 and S6, we would like to show clearly that we start with the trajectory at FL290 and concentrate on the cruise mission only. In fact, we optimize flight trajectories within the general cruise flight altitude of commercial aircraft in [FL290, FL410], as shown in Fig. 7 (top), and the altitude of the airports are located at FL290 (not ground at 0 ft). We have seen situations many times that people assumed the start/end point of the time-optimal flight trajectories (in Fig. 16) as "the ground at 0 ft," when we plotted the same results in the range of altitude relevant to the aircraft. To avoid this situation, we plotted these figures in 0 – 15 km including the ground (just like Figs. 9, 14 and 18).

- (8) p22 L727 – 729 and Figure 22. "The maps show the time-optimal case has low values of the fuel use" (compared to the great circle case). The great circle case at FL290 clearly has a lower fuel use, as shown in Table 11. However, I do not think this is clear from Figure 22; the flights in the time optimal case are spread over a larger area than in the great circle case therefore it is difficult to assess objectively whether the fuel use is higher or lower in the time-optimal case. I do not think that this figure adds any weight to your argument. I suggest removing it.

  Reply: As the referee #1 suggested, we will remove Fig. 22 in the revised manuscript. In addition, we

will remove the sentences related to Fig. 22: on page 22 line 726 – 729, "**To confirm this intuitively, Fig. 22 shows the global distribution maps of the fuel use (in kg(fuel)box−1s−1, 2 hour averages) for these cases. The maps show that the time-optimal case has low values of the fuel use.**"

- (9) p25 L824. I cannot find AirTraf or any status information for it on the list of submodels on the MESSy website (accessed on 24/02/2016).

  Reply: On the basis of the MESSy Consortium Steering Group Policy, a status information for a new submodel is generally provided on the MESSy website after its publication. Nevertheless, we have provided the status information for AirTraf on the website. In the revised manuscript, we will rephrase the sentence related to this matter: on page 25 line 824, " The status information for AirTraf including the licence conditions **is**  available at the website."

- (10) Figure 15, 16, S4 – S6 – Please add units to the colour bar and/or text.

  Reply: Thank you very much. We will add units in the captions for Figs. 15, 16, S3 – S6. In Figs. 15, S3 and S4, we will add the unit in the captions as, "The contours show the zonal wind speed ($u$ **in ms$^{-1}$**)." In Figs. 16, S5 and S6, we will add the unit in the captions as, "Altitude distributions of the true air speed $V_{TAS}$ **in ms$^{-1}$** (a and b)." The wind indicator is dimensionless quantity.

- (11) Table 8. It is difficult to compare the flight time for the time-optimal with the great circle at different altitudes, since the mean flight altitude of the time-optimal flights is given in m and the altitude of the great circle flights in feet. Please add either the mean flight altitude in feet for the time-optimal flights, or the flight altitude in m for the great circle flights to aid the comparison.

  Reply: Thank you very much. In the revised manuscript, we will add the mean flight altitude in feet for the time-optimal flights on column 6 in Table 8: "Mean flight altitude h, m **(in ft)**; 8,841 **(29,005)**; 8,839 **(29,000)**; 8,839 **(29,000)**; 10,002 **(32,815)**; 10,829 **(35,527)**; 9,311 **(30,546)**."

- (12) Table 11, Caption. 'sum of flight time, fuel use, NO$_x$ and H$_2$O emissions...'. This implies that the table shows the quantity flight time + fuel use + NO$_x$ + H$_2$O, when in fact they are displayed separately. Please rephrase.

  Reply: Thank you very much. In the revised manuscript, we will remove the word "**Sum of**" from the caption: on page 59 in Table 11, we will rewrite the caption as "**Flight time, fuel use, NO$_x$ and H$_2$O emissions for the time-optimal and the great circle cases**…".

---

## Author Comment (AC2) · 3 Jun 2016

We are most grateful to the referee #2 for the very helpful and encouraging comments on the original version of our manuscript. Here are our replies:

- This paper presents a development of "module" adapted to the climate chemistry model ECHAM5/MESSY in order to calculate the climate impact of aircraft routes. Only one part of the module needed has been included in the model and presented in this paper: the part generating the route and only in the case of great circle (simple) or time-optimal route (optimisation). From these two routing the module calculates fuel use, and some emissions ($H_2O$ and $NO_x$ only), these parameter are assessed with real data. The module is tested over one winter day data over the North Atlantic corridor. In its present form I unfortunately cannot recommend the publication of the paper in Geoscientific Model Development for several reasons that I will be listing. I would strongly recommend the editor to request a severe revision before publication. The time-optimal calculation module may be of interest for modellers. The optimisation module description as well as the size of the population to be included in the optimisation to converge toward optimal time may be presented in a revised paper.

    Reply: We are grateful to the referee #2 for the critical comments and useful suggestions that have helped us to improve our manuscript. As indicated in the responses that follow, we have addressed all the comments and suggestions. We now state in the introduction that this development is a prerequisite for the investigation of climate-optimal routings. So that the motivation for this development is clear. And we are deleting this overall objective from other text passages, since we agree that they are misleading. We will reply to this point in the following (1). As the referee #2 noted, the descriptions of the time-optimal calculation module and the population sizing are included in the revised manuscript, as originally described.

- (1) My first problem is the presentation of the subject within most of the article (title, abstract and even structure of the manuscript). The focus seems to be in the "optimal routing for climate impact reduction" when you check the paper, however the reader is disappointed as the presented module is not doing that at all – only optimising for travel time. The manuscript needs to be reshaped completely to acknowledge that fact.

    Reply: As the referee #2 pointed out, the subject of this paper seems to be confusing. We should make clear that this paper introduces AirTraf submodel in its basic version, technically describes and validates the various components for first, simple aircraft routings (great circle and time-optimal). Eventually, we are aiming at an optimal routing for climate impact reduction. This will be a separate study, which requires a couple of developments beforehand, amongst which the present study is one of them. Here, we would like to make clear that the final purpose of the AirTraf is not to find "fastest routes." For this, an Earth System Model (ESM) is not necessary. There are even better tools to answer this question. However, to find climate-optimal routes, the global air traffic simulation model coupled to the ESM, i.e. AirTraf submodel, is needed. And of course it has to be described and validated. The validation refers to standard aircraft applications in this paper, such great circle and time-optimal calculations.

    In the revised manuscript, we will revise the title, abstract, introduction and conclusion to be consistent with what is presented in the paper as follows: the title will be revised as, " **Air traffic simulation** in the Chemistry-Climate Model EMAC 2.41: AirTraf 1.0".

    On page 1, line 9 in Abstract, the text will be revised as, "This study introduces AirTraf (version 1.0)  that performs global air traffic simulations on long time scales, including effects of local weather conditions on the emissions."

    On page 3, final paragraph (line 84 – 87), " **This paper presents a new submodel AirTraf (version 1.0, Yamashita et al., 2015) that performs global**

**air traffic simulations coupled to the Chemistry-Climate model EMAC (Jöckel et al., 2010). This paper technically describes the AirTraf and validates the various components for simple aircraft routings: great circle and time-optimal routings. Eventually, we are aiming at an optimal routing for climate impact reduction. The development described in this paper is a prerequisite for the investigation of climate-optimal routings. The research road map for our study is as follows (Grewe et al., 2014b):** he first step is to investigate...".

On page 26, final paragraph (line 870 – 873), "The fundamental framework of AirTraf has been developed to perform fairly realistic air traffic simulations. AirTraf 1.0  **is ready for more complex routing tasks**. **O**bjective functions corresponding to other routing options will be integrated soon, and AirTraf will be coupled with various submodels of EMAC to evaluate air traffic climate impacts."

- (2) I am also extremely disappointed in the fact that a part of the paper is dedicated in presenting and comparing "great circle routing" calculations. This is nothing new, and no advance in modelling or science presented. This part should be cut down and re-moved from the discussion. The more important difference could come from the fact the Earth is not a perfect sphere or maybe taking into account flight altitude. The table 4 is comparing calculation with decimal and no-decimal data when the difference is in the decimal value.

Reply: The referee is right that a "great circle calculation" is commonly used method. However, we are hesitating to remove the discussion on that part for the following three reasons.

First, the final purpose of the AirTraf is to investigate "optimal routing for climate impact reduction." We will compare AirTraf simulation results among several aircraft routing options. As a climate-optimized route will be evaluated in the light of the detour that would be necessary to avoid "climate-sensitive" areas with respect to the reference (trade-off), i.e. great circle or time-optimal route. Thus, the great circle routing option is used as reference of our comparisons (note that the great circle is the optimal solution for "minimum flight distance"). In addition, we would like to refer to a future Air Traffic Management system, which aims at having aircraft fly more direct routes, so called user-preferred routes without being constrained to Air Traffic Services routes and waypoints any longer. These future user-preferred routes would be great circle segments in the ideal case (without wind). Hence, AirTraf is developed with the objective to evaluate routing options for the future and the great circle is still an important route in reality. We think that a thorough assessment of the great circle routing module should be made in this paper to demonstrate its ability to generate the routes and working well if coupled to the ESM. The "great circle calculation" is suitable for the validation of AirTraf, because it is the widely used method (the benchmark test of the great circle calculation is described on page 12 – 13, Sect. 3.1.2).

Second, the above-mentioned assessment of the great circle routing module is also indispensable to showing the correct implementation and applicability of the genetic algorithm (GA) approach. Because the validated great circle routing module provides the analytical solution ($f_{true}$ = 25,994.0 s) for the benchmark test of flight trajectory optimization with GA (i.e. the single-objective optimization for minimization of flight time from MUC to JFK). This point is described on page 16 line 530, "...the $f_{true}$ equals the flight time along the great circle from MUC to JFK at FL290: $f_{true}$ = 25,994.0 s calculated by Eq. (23) with $h_i$ = FL290 for $i$ = 1, 2,···, 101." That the GA reproduces the analytical solution is an important milestone towards other routing optimizations. The part of the great circle routing module supports the discussion of the flight trajectory optimization with GA. Hence, the description of the great circle routing module should be included in this paper.

Last, we would like to stress that AirTraf submodel, which contains the combination of a routing module (including GA) with an Earth System Model, is unique. That is, the great circle routing module described in the paper is a unique model, which works coupling with the ESM. For example, a flight trajectory consists of

waypoints arranged by the waypoint index $i$ ($i = 1, 2, \cdots, n_{wp}$). The geographical and meteorological values, which are used regarding the great circle calculation (e.g. latitude, longitude, altitude, temperature, wind speeds), are provided by the ESM to individual waypoint $i$. It is important to show correctly how the great circles are calculated through waypoints in the ESM. For this, Eqs. 21 – 27 (on page 11 – 12) include the terms with the index $i$.

As the referee #2 noted, an influence of the asymmetric nature of the Earth is an interesting topic. However, we think that this is a separate study. On page 5 line 135, we describe the assumption for AirTraf (version 1.0) as, "a spherical Earth is assumed (radius is $R_E$ = 6,371 km)," corresponding to the ESM. On page 11 in section 3.1, Eqs. 22 and 23 present in detail how to take into account the flight altitude in AirTraf. This part is included in the revised manuscript.

In addition, as the referee #2 pointed out, the decimal and no-decimal data are compared in Table 4. This is indeed a very important point, which we completely overlooked. We will revise Table 4: on column 4, "**$d_{MTS}$, km; 6,481.1; 10,875.0; 16,312.1; 8,895.6; 13,343.4**". On column 6, "**$\Delta d_{eq23, MTS}$, %; –0.0005; –0.0028; – 0.0036; –0.0008; –0.0019**". On column 7, "**$\Delta d_{eq22, MTS}$, %; 0.0000; 0.0000; 0.0000; 0.0000; 0.0000**". We will also revise the caption in Table 4 as, "...column 4 ($d_{MTS}$) shows the result calculated with the Movable type scripts (MTS),  **using the Haversine formula with a spherical Earth radius of $R_E$ = 6,371 km**."

Related to this matter, we will revise the manuscript as follows: on page 1 line 18, "The first test showed that the great circle calculations were accurate to  **–0.004 %**...". On page 11 line 354, we will revise the word "Harvesine formula" into "**Haversine** formula." On page 13 line 406, "The results showed that $\Delta d_{eq23,eq22}$ and $\Delta d_{eq23,MTS}$ varied between −0.0036 and −0.0008 %, and between  **−0.0036** and  **−0.0005** %, **respectively, while $\Delta d_{eq22,MTS}$ showed 0.0 %** ." On page 13 line 408, "The great circle distances calculated by Eqs. (22) and (23) were accurate to  **–0.004 %**...". On page 25 line 832, "The accuracy of the results was  **–0.004 %**." On page 26 line 876, we will add the text as, "**The authors thank Mr. Chris Veness for providing great circle distances that have been calculated with the Movable type script.**"

- (3) Concerning the "optimisation routing" for flying time the validation over the North Atlantic is interesting but what would happen with a case of congested space or restricted space (military)? Please do tests in different part of the world or at different season.

  Reply: We think that the topics, which the referee #2 noted here, are important and interesting. However, we think that they are application studies which would probably use AirTraf, but which are beyond the scope of this technical documentation and first evaluation. The aim of this paper is to introduce, describe and validate the AirTraf submodel, as replied to the comment (1) above. We believe that this paper shows a substantial comparison of AirTraf simulation results to other studies to validate the model.

- (4) Moreover I am unsure of the complete philosophy of the inclusion of the "optimisation" module in the ECHAM5/MESSY model. I understand well the impact of local weather and composition on the impact the aircraft routing will have on climate change. However I am short in understanding the need of the online optimisation as I don't see the effect of "climate optimal routing" on the climate model – would a simple offline calculation not enough to determine this potential "climate optimal routing" (the day the full module will be ready) as well as making the "optimisation" easier to be adapted to other climate-chemistry model output?

  Reply: As replied to the comment (1) above, our final purpose is to investigate the mitigation gain of the climate impact by climate-optimal routing. We would like to make clear that it is not our final purpose only to find climate-optimal flight trajectories for a specific weather condition. This was achieved, e.g. in Grewe et al., 2014. We eventually want to go one step further and apply an optimization on a daily basis for daily changing weather situations. To investigate then the mitigation gain, multi-annual (long-term) simulations are

required (e.g. for ten years). In the simulations over the ten years, each flight trajectory is optimized with respect to a selected aircraft routing option, considering local weather conditions, and emissions are released. AirTraf can perform such air traffic simulations with the inclusion of the on-line optimization module and the optimal routes will change day by day. We think that the inclusion of the optimization module in EMAC is an appropriate approach for our purpose.

[Reference] Grewe, V., Champougny, T., Matthes, S., Frömming, C., Brinkop, S., Søvde, O. A., Irvine, E. A., and Halscheidt, L.: Reduction of the air traffic's contribution to climate change: A REACT4C case study, Atmospheric Environment, 94, 616–625, 2014a.

- (5) Finally I am unhappy with the fact that the only simple "time optimal routing" (optimising only for one variable) the weather situation if fixed for the entire flight. What would happen in the case of multi optimisation when you have to trade-off between time, fuel use, and different emissions? Could you comment on the impact on contrail formation from long flights? "-For all routing options, local weather conditions provided by EMAC at $t = 1$ (i.e. at the departure day and time of the aircraft) are used to calculate the flight trajectory. The conditions are assumed to be constant during the flight trajectory calculation-"making the model as simple as an offline module but complicated as an inside module of an already complex model?

Reply: In this paper, we would like to confirm whether AirTraf works well and is fit for our purpose. Particularly, the ability of the optimization module (GA) to optimize flight routes must be confirmed. For this, we tested the simple "time-optimal routing." The referee actually points at many interesting future investigations, which are far beyond the scope of this paper. As soon as we really start with climate optimized trajectories in EMAC/AirTraf, we will investigate whether it is necessary to re-optimize the trajectory during long flights. It is clear that a weather forecast, which would be required to optimize not only for time $t = 1$, is not feasible within the climate simulation. To cover all effects, such as $NO_x$ effects, an offline calculation on the other hand is not feasible.

In addition, the contrail formation is one of the important factors on climate impacts. For example, Schumann, et al. 2011 noted in the literature: "…contrails are expected to cause the largest contribution to global radiative forcing of the Earth-atmosphere system, and hence, the largest contribution to aviation-induced global climate change…", and "Contrails and thin cirrus in general warm the Earth atmosphere by reducing terrestrial (longwave, LW) radiation loss into space and may cool the Earth atmosphere by reflecting part of the solar (short-wave, SW) radiation back to space. During night, contrails are always warming. The largest climate impact by contrails comes from thick, wide, long and long-lasting contrails. Hence, with respect to climate, optimal routes during night are those which form contrails with minimum longwave warming. During day time, contrails may cool. This may be the case for thick contrails, over dark and cool surfaces, in particular in the morning and evening times when cirrus is more reflective than during mid day. Hence, with respect to minimum contrail warming impact, optimal routes may be those causing contrails with maximum shortwave cooling."

Those contrail effects will be considered as one of the routing options in AirTraf, by coupling with another submodel of EMAC. AirTraf on-line simulation (coupled to the ESM) is a suited model for taking these complicated effects into account on long time scales and this is a difference from off-line models. In this context, as the referee #2 noted, local weather conditions are assumed to be constant during flight trajectory optimization. We think that this assumption is appropriate to perform such AirTraf on-line simulation for long-term to reduce the computational costs.

[Reference] Schumann, U., Graf, K., and Mannstein, H.: Potential to reduce the climate impact of aviation by flight level changes, in: 3rd AIAA Atmospheric and Space Environments Conference, AIAA paper, vol. 3376, pp. 1–22, 2011.

---

## Author Comment (AC3) · 8 Jul 2016

We are most grateful to the referee #3 for the very helpful and encouraging comments on the original version of our manuscript. Here are our replies:

**1 Introduction:**

- The manuscript is well structured, and different aspects of AirTraf are explained by a nice equilibrium of description and examples. The motivation of the work is reasonably well explained. Figures and tables are informative. There is a substantial comparison with results from other studies to give confidence in the results obtained here.

    Implementing aircraft routing strategies in a general circulation model or a numerical weather prediction model is not an easy task. Arriving at the status as described here in the manuscript is already a considerable achievement. However, as the tool is not finished, one wonders whether it is useful to describe the tool in its current status (with only 2 of the 7 optimization options implemented, fuel consumption due to climbing not included, the meteorological fields in the optimization are the ones at the start of the flight,...).

    Publishing the manuscript now shows the status of the work. It makes clear that for specific options the optimization works, and it can trigger discussion with other researchers/institutes on the approaches chosen (is the optimization working well, could other optimization routines be faster,...).

    Reply: We thank the referee #3 for the positive comments. As the referee pointed out, this paper shows the current status of AirTraf. Nevertheless, we think that it is useful to publish AirTraf v.1.0, for several reasons:
    - Our final purpose is to investigate an optimization strategy of aircraft routing for minimizing the climate impact of aircraft emissions and show its mitigation gain for future. We should make clear that this paper introduces the AirTraf submodel in its basic version, technically describes and validates the various components for first, simple aircraft routings (great circle and time-optimal). Eventually, we are aiming at an optimal routing for climate impact reduction. This will be a separate study, which requires a couple of developments beforehand, amongst which the present study documents one of them.
    - The validation refers to standard aircraft applications in this paper, such as great circle and time-optimal calculations. These two options are appropriate to confirm whether AirTraf works well and is fit for the purpose. This is a big step for the AirTraf development.
    - For our purpose, multi-annual (long-term) simulations are required in EMAC: computationally expensive simulations are required. Hence, in the current model we simplify AirTraf to reduce the computational costs, e.g. we concentrate on the cruise mission only.
    - The related issue is discussed in the reply to "2 Principal remarks, Work in progress."

- I think the manuscript is worth publishing, but is should be considerably improved in several ways. A list of principal remarks is given below, followed by a list of more specific comments. I hope the authors will take them into consideration, and if not give a sound argumentation why they do not.

    Reply: We are grateful to the referee #3 for the useful comments and suggestions that have helped us to improve our manuscript. As indicated in the responses that follow, we have addressed all the comments and suggestions.

**2 Principal remarks**

- **Work in progress:** The manuscript describes a submodel in MESSy which works, but is not finished yet (only 2 of the 7 optimization options are in place). Why not waiting until all the work is finished? One has to guarantee that this manuscript remains valid and worth all the work once the remaining parts come into place, and that this document is therefore worth publishing.

    Reply: The major reasons are replied in "1 Introduction." As replied in "1 Introduction", the currently documented status is a prerequisite for the investigation of climate-optimal routings. Additional reasons are:
    - The GA optimization module is an important part of AirTraf for our purpose. Therefore, we made a thorough assessment of the GA optimization and its performance using the time-optimal option in

this paper. If a new objective function corresponding to other routing options is developed, basically, only the objective function *f* (shown in Eq. (28), on page 15 line 485) is changed. The AirTraf framework validated in the paper is, thanks to its modular structure, unchanged. Therefore, the current status is a big step for AirTraf development.

− The manuscript is not only about the "routing options", but an important and integral part describes the overall structure of the coupling between a "routing module" and a chemistry-climate model. This is a major achievement and unique.

- **Language:** There is a lot of improvement needed for the language. The use of articles (a/an/the/none) should be improved. Specific expressions (e.g., "trajectories as longitude vs altitude, trajectories as location" or "number of $n_p$",...) should be modified.

Reply: Thank you so much. We will recheck and modify articles. Please see the revised manuscript. The modifications of the specific expressions are as follows:

["trajectories as longitude vs altitude, trajectories as location"]
We will change the expression "trajectories as longitude vs altitude, trajectories as location" into "trajectories in the vertical cross-section, trajectories projected onto the Earth":
− On page 14 line 450, "...the  **projection onto the Earth** (bottom) with three control points (CPs, black circles) and the  **vertical cross-section** (top) with five CPs."
− On page 15 line 475, "...B-spline curve with the five CPs  **in the vertical cross-section** (bold solid line, Fig. 7 top)...".
− On page 17 line 553, "...the true-optimal solution  **in the vertical cross-section** are plotted...".
− On page 34 in the caption of Figure 7, "Geometry definition of flight trajectory  **in the vertical cross-section** (top) and  **projected onto the Earth** (bottom)."
− On page 36 in the caption of Figure 9, "...explored trajectories (solid line, black) from MUC to JFK  **in the vertical cross-section** (top) and  **projected onto the Earth** (bottom)."
− On page 41 in the caption of Figure 14, "...explored trajectories (black lines) between MSP and AMS  **in the vertical cross-section** (top) and  **projected onto the Earth** (bottom)."
− On page 45 in the caption of Figure 18, "...the trajectories  **in the vertical cross-section** (top) and  **projected onto the Earth** (bottom)."
− On page 1 (Supplementary material) in the caption of Figure S1, "...explored trajectories (black lines) between JFK and MUC  **in the vertical cross-section** (top) and  **projected onto the Earth** (bottom),...".
− On page 2 (Supplementary material) in the caption of Figure S2, "...explored trajectories (black lines) between SEA and AMS  **in the vertical cross-section** (top) and  **projected onto the Earth** (bottom),...".

["number of $n_p$"]
We will change the expression "number of $n_p$" into "value of $n_p$" in the revised manuscript. We also reply to this modification in the following sections: "p 17, l 569 and 570" and "p 19, l 618."

- **CP in trajectories:** Concerning the treatment of CP points, I have several questions.

- (1) As an example, 3 CPs have been used for the geographical location, and 5 for the altitude. Is this fixed? Do all flights use the same number of CPs?

Reply: Yes. All flights use 3 CPs for the geographical location and 5 for the altitude (as shown in Fig. 7 on page 34). This is now explicitly clarified in the revised text.

- (2) For the 103 flights, which were primarily zonal, rectangles around the CPs could be described by using a

range in latitude and longitude. How is the choice around the CPs when flights cross the equator, e.g., at an angle of 45°? What if flights go from low to high latitudes and defining regions whit fixed ranges in longitude makes them very different in size?

Reply: This is a very important issue for the AirTraf development. In AirTraf version 1.0, the domain size was determined by referring to the literature: Irvine, E. A., et al., "Characterizing North Atlantic weather patterns for climate-optimal aircraft routing," Meteorological applications, 20, 80−93 (2013). They show the many types of flight trajectories between London and New York for different weather conditions. We focused on trans-Atlantic flights in this paper, therefore the current definition of domain size works very well for the trajectory optimizations.

As the referee pointed out, if flights cross the equator (at an angle of 45°) or if flights go from low to high latitudes with almost similar longitude values, the domains are variously shaped in size on the basis of the geometry definitions of the flight trajectory (as described in Sect. 3.2.2 on page 14). This probably increases the computational demand for the trajectory optimization. Nevertheless, the current treatment of the domains is applicable to those flights and trajectory optimization works well. In fact, we have confirmed this issue by test simulations using 1,840 global flight plans including such flights. To improve the computational efficiency of the optimization, we will work on an improvement of the definition of domain size for the next version.

We also reply to this issue in the answer to the referee comment of "p 17, l 554−555."

· (3) For a given trajectory (which is a B-spline curve), how are the waypoints found? Are they equally spaced along that trajectory between the CPs? I am wondering whether it is possible to find explicit expressions for equidistant waypoints on a B-spline curve?

Reply: The referee is right. In AirTraf, the 3$^{rd}$ order B-spline curve is used to generate the waypoints. If CPs are given, a parameter $t$, which is the parameter of the 3$^{rd}$ order B-spline basis functions, is assigned with values between 0 and 1 between the CPs. Here, $t$ is equally spaced along the "basis functions" (i.e., equally spaced between $0 \leq t \leq 1$). After that, the coordinates of the waypoints of the trajectory are determined by summation of the basis functions, corresponding to the equidistant $t$. Therefore, this can not ensure that the waypoints are equally spaced along the trajectory. We reply to this issue in the answer to the referee comment of "p 14, l 464".

· (4) In the example used, 3 CPs were used for the geographical location, 5 CPs for the altitude, and 101 waypoints. However, the condition $(101 − 1)\mathrm{modulo}(5 + 1) = 0$ is not fulfilled. One also gets the impression that the waypoints for the altitude and longitude are not located at the same place (although the manuscript confirms it actually is). Could this be clarified?

Reply: As described on page 14 line 464, the condition is $\mathrm{mod}(n_{\mathrm{wp}} − 1, \underline{n}_{\mathrm{CPloc}} + 1) = 0$. This is only used for the location. Here, $n_{\mathrm{wp}} = 101$ and $n_{\mathrm{CPloc}} = 3$. Therefore, $\mathrm{mod}(101 − 1, 3 + 1) = 0$ is fulfilled. In addition, to clarify the location of waypoints for the altitude and longitude, we will revise the text: on page 15 line 474−478, "A flight trajectory is also represented by a B-spline curve **(3$^{rd}$-order)** with the five CPs  **in the vertical cross-section** (bold solid line, Fig. 7 top) and then waypoints are generated along the trajectory **in such a way that the longitude of the waypoints is the same as that for the flight trajectory projected onto the Earth.** " We also reply to this issue in the referee comment of "p 15, l 476−478."

· **GA algorithm:** This algorithm is explained to some detail, but I suggest that all terms used should be explained to some extent (e.g., mating pool). One should also be informed on how the final solution is derived from the population in the last generation. Finally, the abstract uses some terminology related to the optimization routine (e.g., population), which are too technical to be mentioned in the abstract.

Reply: We will add a section "Appendix; Glossary" after the section "7. Conclusions", where we explain the optimization terminologies: on page 26, **"Appendix; Glossary; Table A1 shows a glossary explaining several terminologies of the GA optimization. The terms from the glossary are written in italics in the text." In Table A1, we will add the explanations, "Table A1. Glossary of terms. Population: A set of solutions. A Genetic Algorithm starts its search with an initial population (a random set of solutions).; Generation: One iteration of a Genetic Algorithm.; Rank: A ranking assigned to each solution to evaluate a relative merit in a population. A *rank* expresses the number of solutions that are superior to a solution.; Fitness: A value assigned to each solution to emphasize superior solutions and eliminate inferior solutions in a population. *Fitness* = 1/*rank*.; Mating pool: A storage space for solutions."** We will refer to those terms in the text in italics. Many variables are modified. Therefore, we will show the modifications in the revised manuscript. Related to this, we will revise the text: on page 2 line 21 in Abstract, "The dependence of **the** optimal solutions on **the** initial  **set of solutions (called population)** was analyzed." On page 15 line 491, "A solution with **a** higher *fitness* value (i.e., **a** smaller *rank* value) has a higher probability of being copied into a *mating pool*."

In addition, we will add the text to inform on how the final solution is obtained from the optimization: on page 16 line 517, "..., GA quits the optimization and an optimal solution **showing the best *f* of the whole *generation*** is output...".

·   **Abstract, introduction, conclusion:** The abstract is sometimes too much a summing up of what has been done, with vocabulary/terms which have no concrete meaning without a concrete context. There is also much more overlap between these three parts (abstract, introduction, and conclusion) than needed. The abstract should be written differently, and considerably improved.

Reply: By following the remarks and the list of specific comments of the referee #3, we revise the abstract, introduction and conclusion. Please see the revised manuscript.

·   **Sensitivity:** In the approach followed here, quite some assumptions and simplifications are introduced. It would be useful to give the reader an idea of the impact of these assumptions on the results. A list of some of the assumptions is:

Reply: Firstly, we would like to make clear again that our final purpose of AirTraf is to investigate an "optimization strategy" of aircraft routing for minimizing the climate impact of aircraft emissions and to show its mitigation gain for the future. It is not our purpose to find detailed flight trajectories. The aim of this paper is to introduce the AirTraf submodel in its basic version, technically describe and validate the various components for first, simple aircraft routings (great circle and time-optimal), in order to confirm whether AirTraf works well and is fit for our purpose. Eventually, we are aiming at an optimal routing for climate impact reduction. This will be a separate study, which requires a couple of additional developments beforehand, amongst which the present study is one of them. In addition, multi-annual (long-term) simulations are required for our purpose (e.g. for ten years) coupled with the Earth System Model: computationally expensive simulations are required. We therefore think that our assumptions are appropriate to perform such AirTraf on-line simulations for long-term periods to reduce the computational costs.

As the referee pointed out, they are all interesting points and might be a future option. However, they are beyond the scope of this paper and we cannot explore all sensitivities. A couple of specific points are as follows:

·   (1) line 274 : $dh(t)/dt = 0$ in Eq. (3).

Reply: Looking at the AirTraf trajectories, there is an altitude change visible, but it appears over a long distance and a long period of time. We evaluated $dh/dt$ of the time-optimal flight trajectories for the three selected airport pairs (listed in Table 8 on page 57). The averages of $dh/dt$ (absolute value, ms$^{-1}$) for the individual flights were: 0.0 (JFK to MUC); 0.0 (MUC to JFK); 0.0 (MSP to AMS); 0.32 (AMS to MSP); 0.24 (SEA to AMS); and 0.13 (AMS to SEA). We therefore conclude that the impact of the zero-assumption is not

a big issue, the more as in AirTraf 1.0, we use so far only a small number of vertical GA control points (shown in Fig. 7 on page 34). If the number of control points increases, the influence of climb/descent rates ($dh/dt$) will increase. This could be an aspect for a next version of AirTraf.

To clarify our assumptions, we will revise the text: on page 9 line 273−275, " **For a cruise flight phase, both altitude and speed changes are negligible. Hence, $dh/dt = 0$ as well as $dV_{TAS}/dt = 0$ is assumed in AirTraf (version 1.0) and Eq. (3) becomes the typical cruise equilibrium equation: $Thr_i = D_i$ at waypoint $i$.**"

· (2) $M$ is set constant. Can this be varied slightly? Or have pilots only a very small envelope of allowed or possible speeds?

Reply: The constant Mach number, $M = 0.82$, is the officially published cruise Mach number of an A330-301 by Eurocontrol in 2011. It is appropriate for the aim of this paper to perform AirTraf simulations for simple conditions, including a constant $M$. On page 5 line 136, we describe the assumption for AirTraf (version 1.0) as, "The aircraft performance model of Eurocontrol's Base of Aircraft Data (BADA Revision 3.9, Eurocontrol, 2011) is used with a constant Mach number $M$...". As the referee noted, a change of Mach number is an interesting topic. However, this will be a separate study. In addition, pilots are not allowed to change flight speed freely in the actual flight operations. The speed is indicated (controlled) from the air traffic management side.

· (3) What if weather not just from $t = 1$ is taken, but from the whole period of the flight?

Reply: The referee actually points out the important and interesting topic. However, this is a separate study, which would probably use AirTraf, but which is beyond the scope of this technical documentation and first evaluation. On page 7 line 201, we describe the assumption for AirTraf (version 1.0) as, "...local weather conditions provided by EMAC at $t = 1$ (i.e. at the departure day and time of the aircraft) are used to calculate the flight trajectory. The conditions are assumed to be constant during the flight trajectory calculation." Note that a weather forecast, which would be required to optimize not only for time $t = 1$, is not feasible within a climate simulation.

· (4) Leaving out the ascent and descend phase of the flight: how does this impact the optimization?

Reply: For our final purpose described in the reply to "1 Introduction" and "Sensitivity", it is appropriate to concentrate on the cruise mission only in AirTraf (version 1.0). On page 5 line 140, we describe the assumption for AirTraf (version 1.0) as, "Only the cruise flight phase is considered, while ground operations, take off, landing and any other flight phases are unconsidered." It is maybe worth to mention that the cruise has a larger climatic impact than the other parts of the operation, since the cruise has a longer operation time. Moreover, there are other attempts to reduce emissions during ground operation (taxiing etc.), which are not connected to routing. In any case, there are not much "re-routing" options between ground operations and reaching the cruise altitude.

· **Mathematical formulas:** The mathematical expressions should be improved.
· (1) In mathematical formulas, variables longer than one letter should be written straight.

Reply: We will recheck all variables and modify them with straight letters. Many variables are modified; therefore, we will show the modifications in the revised manuscript.

· (2) A lot of indices should be straight letters : $V_{ground}$, $V_{wind}$, ...

Reply: We will recheck all indices and modify them with straight letters. Many indices are modified;

therefore, we will show the modifications in the revised manuscript.

- (3) After every equation, there should be a ","  or ".", depending on the function of the equation in the sentence.

  Reply: We will add a "," after **Eqs. (1)–(8), Eqs. (11)–(22), Eqs. (24)–(27) and Eq. (29).** We will show the modifications in the revised manuscript.

- (4) Names of trigonometric formulas should not be italic : sin, cos, ...

  Reply: We will modify the all names of trigonometric formulas into normal straight letters: for "sin," **Eq. (21)** is modified; for "cos," **Eqs. (4), (21) and (23)** are modified; and for "arctan," **Eq. (21)** is modified. We will show the modifications in the revised manuscript.

- **Climate model, long/short time scales:** Why is this tool implemented in a climate model? To my opinion, the tool could also have been build such that it uses off-line 3-hourly meteo fields over the range of time it has flights which should be optimized : one thinks over a range of 1 to 10 days. The meteo data might come from a NWP, or a climate model.
    Maybe the authors want to show that it is possible to have such a tool on-line in a NWP or GCM. However, in that case, I would have chosen for a NWP as that is the place where, if the tool is operationally used, might be most appropriate. What was the reason that the authors made the choice of implementing it in a climate model?
    A reason I can imagine is that one could do tests like : how would the optimal routing be in a year 2100 climate, when global climate is considerably different from nowadays?

  Reply: Our final purpose is to investigate the mitigation gain of the climate impact by climate-optimal routing. We would like to make clear that it is not our purpose to find climate-optimal flight trajectories (or optimal flight trajectories corresponding to a selected routing option, e.g. fastest routes) for a specific weather condition. For this, an Earth System Modeling (ESM) is not necessary and this indeed has been achieved, e.g. by Grewe et al., 2014. We eventually want to go one step further and apply an optimization on a daily basis for daily changing weather situations. To investigate then the mitigation gain, multi-annual (long-term) simulations are required (e.g. for ten years). In the simulations over the ten years, each flight trajectory is optimized with respect to a selected aircraft routing option, considering local weather conditions. The released emissions directly ($CO_2$, $H_2O$) and indirectly ($NO_x$) modify the radiative forcing and therefore the climate. Off-line pre-calculated routes would be inconsistent in such an approach. AirTraf can perform these air traffic simulations with the inclusion of the on-line optimization module and the optimal routes will change day by day. In addition, AirTraf can use the framework of EMAC to assess routing options, e.g. surface temperature changes or changes in the background chemical conditions of the atmosphere ten years later corresponding to the selected routing option, by coupling with other submodels of EMAC. The main point is the interactive coupling, i.e. the on-line re-routing immediately affects the climate model (via air traffic emissions). An on-line feedback cannot be replaced by an off-line approach. We think that the implementation of AirTraf on-line in EMAC is appropriate approach for our purpose. This reply it related to the reply to "p4 l 115."

  [Reference] Grewe, V., Champougny, T., Matthes, S., Frömming, C., Brinkop, S., Søvde, O. A., Irvine, E. A., and Halscheidt, L.: Reduction of the air traffic's contribution to climate change: A REACT4C case study, Atmospheric Environment, 94, 616–625, 2014a.

- **Benchmarks:** Is proving that the great circle option works well worth publishing and/or mentioning in an abstract? In addition, I think that the word benchmark puts more importance on a test than it actually deserves.

  Reply: We understand the referee comment. The "great circle calculation" is a commonly used method.

However, we are hesitating to remove the descriptions of the great circle for the following three reasons:

First, the final purpose of AirTraf is to investigate "optimal routing for climate impact reduction." We will compare AirTraf simulation results among several aircraft routing options. As a climate-optimized route will be evaluated in the light of the detour that would be necessary to avoid "climate-sensitive" areas with respect to the reference (trade-off), i.e. "great circle" or time-optimal route. Thus, the great circle routing option is used as reference for our comparisons (note that the great circle is the optimal solution for "minimum flight distance"). In addition, we would like to refer to a future Air Traffic Management system, which aims at having aircraft fly more direct routes, so called user-preferred routes without being constrained to Air Traffic Services routes and waypoints any longer. These future user-preferred routes would be great circle segments in the ideal case (without wind). Hence, AirTraf is developed with the objective to evaluate routing options for the future and the great circle is still an important route in reality. We think that a thorough assessment of the great circle routing module should be made in this paper to demonstrate its ability to generate the routes and working well if coupled to the ESM. The "great circle calculation" is suitable for the validation of AirTraf, because it is a widely used method (the benchmark test of the great circle calculation is described on page 12–13, Sect. 3.1.2). We believe that the result of the assessment is worth publishing.

Second, the above-mentioned assessment of the great circle routing module is also indispensable to show the correct implementation and applicability of the genetic algorithm (GA) approach. Because the validated great circle routing module provides the analytical solution ($f_{true}$ = 25,994.0 s) for the benchmark test of flight trajectory optimization with GA (i.e. the single-objective optimization for minimization of flight time from MUC to JFK). This point is described on page 16 line 530, "...the $f_{true}$ equals the flight time along the great circle from MUC to JFK at FL290: $f_{true}$ = 25,994.0 s calculated by Eq. (23) with $h_i$ = FL290 for $i$ = 1, 2,···, 101." The result that the GA reproduces the analytical solution is an important milestone towards other routing optimizations.

Last but not least, we would like to stress that the AirTraf submodel, which embeds a routing module (including GA) into an Earth System Model, is unique. The great circle routing module described in the paper is used to show that the coupled system works well. For example, a flight trajectory consists of waypoints arranged by the waypoint index $i$ ($i$ = 1, 2,···, $n_{wp}$). The geographical and meteorological values, which are used for the great circle calculation (e.g. latitude, longitude, altitude, temperature, wind speed), are provided by the ESM at the individual waypoints $i$. It is important to show that the great circles are calculated correctly by waypoints through the ESM domain. For this, Eqs. (21)–(27) (on page 11–12) include the terms with the index $i$. Hence, the description of the great circle routing module should be included.

In addition, we understand the referee comment on the word "benchmark." Nevertheless, we are hesitating to change the word. The tests are performed to confirm the correct performance of the code, which we believe is unique and new, and thus to measure the reliability of the code. We think that those tests are indeed important "benchmark tests."

- **Size of the document:** The files are so large (30 MB) that people will have problems printing the documents. To my opinion it is mainly related to the figures which show different flight trajectories. I assume that the figures contain all the information from all trajectories, while a large central part of the figure is just black. These figures should be made in such a way that they become much smaller in size, without loosing their precision.

Reply: As the referee pointed out, the file size is large. We will make those figures become much smaller in size with almost the current precision and replace them in the revised manuscript: **Figs. 9, 14a, 14b, 18a to 18d, S1a, S1b, S2a** and **S2b** are modified.

**3  Comments on the text**
**Page 1**
- **p 1, l 1–5** : The sequence of the first three sentences is a bit strange. I would even skip the first sentence (as it says the same as the first 7 words of sentence 3).

Reply: We will remove the first sentence: on page 1 line 1, "" Concerning this, we will rephrase the text: on page 1 line 3, "Reducing

 **anthropogenic** climate impact from aviation emissions and...".

- **p 1, l 3–6** : "building a climate-friendly", "for a sustainable development", "is an important approach". It makes me wonder whether this is not a too optimistic view on aviation.

  Reply: We agree. The sustainable development of commercial aviation might be optimistic. However, if we want to have a sustainable development of commercial aviation, we need to have a reduction of aviation emissions and a climate-friendly air transportation system.

- **p 1, l 9** : "stable" gas. This is not precise enough.

  Reply: We will delete the word "stable" in the sentence: on page 1 line 9, "$CO_2$ is a long-lived  gas, while...".

- **p 1, l 9** : "vary regionally". I would rather use something like "inhomogeneous distribution".

  Reply: We will rephrase the text: on page 1 line 9, "...non-$CO_2$ emissions are short-lived and  **are inhomogeneously distributed.**"

- **p 1, l 11** : "on long time scales". I assume that the tool takes into account climate impacts on long time scale, via e.g. the CCFs. However, the tool itself is an optimization of only the flights planned within the next few days. There should be no confusion about these very different aspects.

  Reply: In this sentence, we just wanted to say that AirTraf can perform "long-term" simulations, i.e. not only a few days but also more than ten years (arbitrary duration of simulations). The word "on long time scales" seems to be confusing. We will revise the text: on page 1 line 9–11, "This study introduces AirTraf (version 1.0)  that performs global air traffic simulations , including effects of local weather conditions on the emissions." In AirTraf, we apply an optimization on a daily basis for daily changing weather situations. To investigate the mitigation gain of the climate impact by climate-optimal routing, multi-annual (long-term) simulations are required (e.g. for ten years). In the simulations over the ten years, each flight trajectory is optimized with respect to a selected aircraft routing option, considering local weather conditions. Along the optimized flight path, emissions are released. AirTraf can perform such long-term air traffic simulations with the inclusion of the on-line optimization module and the optimal routes will change day by day.

- **p 1, l 15** : were → are (because you describe the functioning of a tool).

  Reply: We will revise the text: on page 1 line 15, "Fuel use and emissions  **are** calculated by...". In the same way, we will revise the text: on page 1 line 16, "The flight trajectory optimization  **is** performed by a Genetic Algorithm...".

- **p 1, l 15** : DLR. This abbreviation should be explained.

  Reply: We will revise the text: on page 1 line 15, "...and **Deutsches Zentrum für Luft- und Raumfahrt (DLR)** fuel flow method."

- **p 1, l 16–17** : "with respect to routing options" : vague.

  Reply: We will revise the text: on page 1 line 16, "...performed by a Genetic Algorithm (GA) with respect to **a selected** routing option."

- **p 1, l 17–18** : "two benchmark tests ... for great circle and time routing options" : sounds a bit strange → "benchmark tests ... for the great circle and time routing options".

Reply: We will revise the text: on page 1 line 17, "...,  benchmark tests were performed for **the** great circle and flight time routing options."

- **p 1, l 19** : "by other published code" : vague, and inappropriate language for an abstract.

  Reply: We will revise the text: on page 1 line 19, "...calculated by  **the Movable type script**."

- **p 1, l 20** : "optimal solution" → "optimal solution found by the algorithm" (distinguish whether it relates to the real optimal solution, or to the best estimate found by the optimization routine).

  Reply: We will revise the text: on page 1 line 20, "...the optimal solution **found by the algorithm** sufficiently converged to...".

**Page 2**

- **p 2, l 22** : "initial population" : as such, this is too technical for an abstract. I suggest to skip this from the abstract, or one could also choose to describe a bit better the optimization algorithm/methodology in the abstract.

  Reply: Please see the reply to the referee comment: "GA algorithm."

- **p 2, l 22–23** : "We found that the influence was small (around 0.01 %)" : I suggest to combine this into one sentence with the former sentence.

  Reply: We will revise the sentences: on page 2 line 21–23, "The dependence of optimal solutions on the initial  **set of solutions (called population)** was analyzed **and**  the influence was small (around 0.01 %)."

- **p 2, l 24** : "function evaluations", "generation sizing" : too technical for an abstract.

  Reply: We will add explanations and revise the sentence: on page 2 line 24, "The trade-off between the accuracy of GA optimizations and  **computational costs** is investigated and the appropriate population and generation **(one iteration of GA)** sizing is discussed."

- **p 2, l 27** "one-day AirTraf simulations are demonstrated ..." : vague.

  Reply: We will remove the word "one-day" in the sentence: on page 2 line 26, "Finally,  AirTraf simulations are demonstrated with...". Related to this, we will revise the text: on page 2 line 31, "The consistency check for the  AirTraf simulations...". We will also revise the text: on page 4 line 106, "In Sect. 4,  AirTraf simulations are demonstrated  **with** the two options **for a typical winter day (called one-day AirTraf simulations)** and the results are discussed."

- **p 2, l 27** : specific winter day → typical winter day.

  Reply: We will revise the text: on page 2 line 27, "...with the great circle and the flight time routing options for a  **typical** winter day." In the same way, we will revise the text: on page 18 line 599, "The simulation was performed for one  **typical** winter day..."; on page 25 line 844, "AirTraf simulations were demonstrated in EMAC (on-line) for a  **typical** winter day...".

- **p 2, l 29** : "for the two options" : it is a long time ago that these were mentioned. So maybe express them explicitly again.

Reply: We are hesitating to express them explicitly again, because the corresponding word "the great circle and the flight time routing option" are mentioned on page 2 line 27. We think that this is not far from line 29. Nevertheless, we will add the text to express the word more clearly: on page 2 line 29, "...AirTraf simulates the air traffic properly for the two **routing** options."

- **p 2, l 30** : for all airport pairs : too vague for an abstract.

  Reply: We will revise the text: on page 2 line 30, "...for  **103** airport pairs...".

- **p 2, l 30–31** : "reflecting" local weather → taking into account (?).

  Reply: We will revise the text: on page 2 line 30, "...airport pairs,  **taking** local weather conditions **into account**."

- **p 2, l 31** : verified → confirmed.

  Reply: We will revise the text: on page 2 line 31, "...the one-day AirTraf simulations  **confirmed** that...".

- **p 2, l 32** : "comparable to reference data" : too vague.

  Reply: We will revise the text: on page 2 line 31–32, "...calculated flight time, fuel consumption, $NO_x$ emission index and aircraft weights  **show a good agreement with reference data.**"

- **p 2, l 34** : "with increasing the number " → "with the increasing number".

  Reply: We will revise the text: on page 2 line 34, "**With the increasing number** of aircraft, the air traffic's contribution...".

- **p 2, l 35** : "a major problem" : too vague.

  Reply: We will revise the text: on page 2 line 35, "...the air traffic's contribution to climate change becomes **an**  **important** problem."

- **p 2, l 35** : "At present" → Nowadays, currently, ... .

  Reply: We will revise the text: on page 2 line 35, " **Nowadays**, aircraft emission...".

- **p 2, l 35–37** : aircraft emission impacts contribute 4.9 % of total anthropogenic radiative forcing : skip "impacts", as radiative forcing is an impact; 4.9 → to 4.9 ; of total → "of the total".

  Reply: We will revise the text: on page 2 line 35–37, "..., aircraft emission  (this includes still uncertain aviation-induced cirrus cloud effects) contribute**s** approximately **to** 4.9 % (with a range of 2-14 %, which is a 90 % likelihood range) of **the** total anthropogenic radiative forcing...".

- **p 2, l 39** : will grow → might grow.

  Reply: We will revise the text: on page 2 line 39, "An Airbus forecast shows that the world air traffic  **might** grow...".

- **p 2, l 40** : the value of 4.9 % → a value of 4.9 %.

  Reply: We will revise the text: on page 2 line 40, "..., while Boeing forecasts  **a** value of 4.9 % over the

same period."

- **p 2, l 41** : indicates → implies.

  Reply: We will revise the text: on page 2 line 41, "This  **implies** a further increase of aircraft emissions...".

- **p 2, l 41–42** : "and therefore environmental impacts from aviation increase" : try to avoid to have twice "increase" in this sentence.

  Reply: We will revise the text: on page 2 line 41–42, " This  **implies** a further increase of aircraft emissions and therefore environmental impacts from aviation  **rise**."

- **p 2, l 42–43** : This sentence sounds more positive than one can possibly defend.

  Reply: We will reply to the comment in the above section: "p 1, l 3–6".

- **p 2, l 47** : contrail → contrails.

  Reply: We will revise the text: on page 2 line 47, "The emissions also induce cloudiness via the formation of contrail**s**, contrail-cirrus...".

- **p 2, l 49** : depends → depends partially.

  Reply: We will revise the text: on page 2 line 49, "The climate impact induced by aircraft emissions depends **partially** on...".

- **p 2, l 49–51** : What follows behind the ":" is not an explanation from what is said before ":".

  Reply: We will revise the sentences: on page 2 line 49–50, "The climate impact induced by aircraft emissions depends on local weather condition**:**.  **That is, the impact** depends on...".

- **p 2, l 50** : geographic → geographical (both are possible).

  Reply: We will revise the word "geographic" into the "geographical" in the revised manuscript: on page 2 line 50, "...on geographic**al** location (latitude and longitude) and..."; on page 14 line 449, "...the geographic**al** location..."; on page 34 in the caption of Fig. 7, "...and as geographic**al** location...".

- **p 2, l 51–p3, l 59** : "... and affect the atmosphere from minutes to centuries." Minutes probably refers to the time scale for disappearance of some chemical perturbations. However, every appearance (even if it is only a few minutes) of a GHG, has a century-timescale effect. Although I think I understand what the authors want to say, I think that the whole paragraph is rather inaccurate, and should be rewritten more precisely.

  Reply: In this paragraph, we just wanted to focus on atmospheric composition changes, not on the climate changes, which the referee addressed. We will add the word "on the atmospheric composition" into the text to make clear what we want to say here: on page 2 line 51–53, "In addition, the impact **on the atmospheric composition** has different timescales: chemical effects induced by the aircraft emissions have a range of life-times and affect the atmosphere from minutes to centuries. $CO_2$ has  long perturbation life-time**s** in the order of decades to centuries."

**Page 3**
- **p 3, l 61** : "150 km horizontally" : maybe distinguish two directions (is it perpendicular to the flight path, or along the flight path). Isn't this 150 km much too specific? Isn't there a very broad spectrum?

Reply: The mean length of 150 km is from Gierens and Spichtinger (2000). The study showed that: "The mean path length is about 150 km with a standard deviation of 250 km." Therefore, we will refer the original reference in the text and revise the sentence to make clear that point: on page 3 line 61, "...extend a few 100 m vertically and  **about** 150 km  **along a flight path (with a standard deviation of 250 km)** with a large spatial and temporal variability (**Gierens et al., 2000,** Spichtinger et al., 2003)." This modification is also related to our reply to the comment (1) of referee #1.

- **p 3, l 63** : There "are" two options ... : this sounds very optimistic.

  Reply: We will revise the text: on page 3 line 63, "**The measures to counteract the climate impact induced by aircraft emissions can be classified into two categories**: technological and operational  **measures**,...".

- **p 3, l 64** : "approaches" → measures.

  Reply: We will revise the word "approaches" into "measures": on page 3 line 64, "...: technological and operational  **measures**,...". In the same way, we will revise the word "approach" into "measure" in the manuscript: on page 1 line 6, "...is an important  **measure** for climate impact reduction...".

- **p 3, l 69** : "... are optimized with respect to time and economic costs." : if both are taken into account, how are they weighted?

  Reply: In this paper, we would like to show that AirTraf works well and is fit for our purpose. Particularly, the ability of the optimization module (GA) to optimize flight routes must be confirmed. For this, we tested the simple "time-optimal routing." The referee actually points at the interesting future investigation, which is far beyond the scope of this paper. Generally, airlines have own evaluation functions, such as cost index, which uses weight factors on fuel, time, etc., in order to optimize the whole aircraft operating system. This kind of data is almost impossible to get from airlines and depends on their individual strategy.

- **p 3, l 69** : "fuel, crew, operating costs" : isn't fuel part of the operating costs?

  Reply: We will revise the text: on page 3 line 69, "...economic costs (fuel, crew, **other** operating costs)...".

- **p 3, l 72** : "systematic routing changes" : reading this, one gets the impression that there are different options. However, later it is reduced to just "i.e. flight altitude change". I suggest to just say "systematic flight altitude changes".

  Reply: We will revise the text: on page 3 line 72, "Earlier studies investigated the effect of **systematic**  **flight altitude changes** on the climate impact...".

- **p 3, l 74** : has a strong effect on the reduction of the climate impact → has a strong impact on climate. (From the original formulation it is not clear whether the increase or the decrease in flight altitude leads to a reduction of the climate impact.)

  Reply: We understand the referee comment. Nevertheless, we are hesitating to change the text. The four studies referred here showed clearly that the changed altitude has a strong effect on the reduction of the climate impact. However, the studies were performed with respect to different flight plans, different climate impact metrics and different duration of simulations (i.e. atmospheric conditions). We think that it is not appropriate to describe whether the increase or the decrease in flight altitude leads to a reduction of the climate impact. More studies are needed before generalizing that point.

- **p 3, l 74–77** : "the" climate-optimized routing → climate-optimized routing.

Reply: We will revise the text: on page 3 line 74–77, "A number of studies have investigated the potential of applying  **climate-optimized routing** for real flight data. Matthes et al. (2012) and Sridhar et al. (2013) addressed weather-dependent trajectory optimization using real flight routes and showed a large potential of  **climate-optimized routing**."

- **p 3, l 79** : "the" climate sensitive regions → climate-sensitive regions.

  Reply: We will revise the text: on page 3 line 79, "...by considering regions described as  **climate-sensitive regions** and...".

- **p 3, l 80** : "This study" → "That study".

  Reply: We will revise the text: on page 3 line 80, " **That** study reported...".

- **p 3, l 81** : by only small increase → by only a small increase.

  Reply: We will revise the text: on page 3 line 81, "...can be achieved by only **a** small increase in economic costs...".

- **p 3, l 80–81** : This study reported: "large reductions ..." → That study reported that large reductions ...

  Reply: We will revise the text: on page 3 line 80–81, " **That study reported that large reductions** in the climate impact of up to 25 % can be achieved by only **a** small increase in economic costs of less than 0.5%."

- **p 3, l 82** : useful : is useful what one wants to express?

  Reply: We just want to express that the climate-optimized routing is effective to reduce the climate impact. Therefore, we will revise the text: on page 3 line 82, "The climate-optimized routing therefore seems to be **an**  **effective** routing option **for the climate impact reduction,**...".

- **p 3, l 85–86** : The current study wants apparently to investigate something (how much the climate impact of aircraft emissions can be reduced) that already has been investigated before (see lines 80–81: large reductions in the climate impact of up to 25 % can be achieved). One should be more specific of what the current study will do extra with respect to the former study.

  Reply: Our final purpose (yet beyond the scope of the present manuscript) is to investigate the mitigation gain of climate-optimal routing. We would like to stress that the mere construction of climate-optimal flight trajectories for a specific weather condition is not our goal. The latter has been achieved, e.g. by Grewe et al., 2014. We eventually want to go one step further and apply an optimization on a daily basis for daily changing weather situations. To investigate then the mitigation gain, multi-annual (long-term) simulations with full feedback from the re-routed air traffic emissions are required (e.g. for ten years). In such simulations over at least the ten years, each flight trajectory is optimized with respect to a selected aircraft routing option, considering local weather conditions. The air traffic emissions are released into the ESM atmosphere and modify its chemical composition. AirTraf can perform such air traffic simulations with the inclusion of the on-line optimization module and the optimal routes will change day by day. This is an important difference to former studies.

  As the referee pointed out, the subject of this paper (line 84–85) seems to be confusing. We make clear that this paper introduces the AirTraf submodel in its basic version, and technically describes and validates the various components for first, simple aircraft routings (great circle and time-optimal). Eventually, we are aiming at an optimal routing for climate impact reduction. This will be a separate study, which requires a couple of additional developments beforehand, amongst which the present study is only one of

them.

Here, we will revise the sentences: on page 3, final paragraph (line 84–87), " **This paper presents the new submodel AirTraf (version 1.0, Yamashita et al., 2015) that performs global air traffic simulations coupled to the Chemistry-Climate model EMAC (Jöckel et al., 2010). This paper technically describes AirTraf and validates the various components for simple aircraft routings: great circle and time-optimal routings. Eventually, we are aiming at an optimal routing for climate impact reduction. The development described in this paper is a prerequisite for the investigation of climate-optimal routings. The research road map for our study is as follows (Grewe et al., 2014b):** the first step is to investigate...".

- **p 3, l 84–87** : Do you mean by "this study" = "this manuscript"? Or is "this study" broader? After reading the manuscript, I have the impression that line 84–85 is not what is answered by this manuscript.

  Reply: We agree. We will reply this point in the section above: "p 3, l 85–86."

- **p 3, l 87** : The first step "is" → The first step "was".

  Reply: We will revise the text: on page 3 line 87, "The first step  **was** to investigate...".

- **p 3, l 87–89** : The first step is to investigate specific past weather situations, in particular the climate impact of locally released aircraft emissions → The first step was to investigate the influence of specific weather situations on the climate impact of aircraft emissions.

  Reply: As the referee described, this correction makes the sentence more clearly. Thank you very much. We will revise the text: on page 3 line 87–89, "**The first step was to investigate the influence of specific weather situations on the climate impact of aircraft emissions** (Matthes et al., 2012, Grewe et al., 2014b)."

- **p 3, l 89** : "The resulting data are ..." : too vague. Maybe one could say : "This results in climate cost functions ...".

  Reply: Thank you very much. We will revise the text: on page 3 line 89, " **This results in climate cost functions** (CCFs, Frömming et al., 2013, Grewe et al., 2014a, Grewe et al., 2014b) that identify...".

- **p 3, l 91** : Why is $CO_2$ in this list? I can understand that the impact of adding $CO_2$ depends on the altitude, but this comes a bit unexpected after formulating earlier that $CO_2$ is well-mixed.

  Reply: We will delete the word "$CO_2$" in the sentence: on page 3 line 91, "...climate sensitive regions with respect to  $O_3$ , $CH_4$ , $H_2O$ and contrails."

- **p 3, l 91** : "They are specific climate metrics, i.e. climate impact per unit of emission" → "per unit amount of emission."

  Reply: We will revise the text: on page 3 line 91, "They are specific climate metrics, i.e. climate impact **per unit amount of emission**,...".

- **p 3, l 92** : "and are used ..." → "will/might be used".

  Reply: We will revise the text: on page 3 line 92, "...climate impact **per unit amount of emission**, and  **will be used** for optimal aircraft routings."

- **p 4, l 92** : "In a further step, weather proxies are identified for the specific weather situations." It is not clear whether this has been done.

  Reply: This has not been done. To clarify this point, we will revise the text: on page 4 line 92, "In a further step, weather proxies  **will be** identified for the specific weather situations,...".

- **p 4, l 102–104** : "A benchmark test for the great circle routing option is performed and ..." : the part before and after the "and" actually express more or less the same.

  Reply: As the referee noted, that part can be reduced. Therefore, we will revise the text: on page 4 line 102–104, "A benchmark test  **provides a comparison of** resulting great circle distances  **with** those calculated by  **the Movable type script (MTS, Movable type script, 2014)**."

- **p 4, l 103** : "by other published code" : too vague.

  Reply: We will revise the text: on page 4 line 103, "...calculated by  **the Movable type script (MTS, Movable type script, 2014).**" Related to this, we will also revise the text: on page 12 line 401, "...calculated with  **MTS**."

- **p 4, l 103–104** : "Another ... also ..." : I suggest to skip one of these words.

  Reply: We will remove the word "also" from the sentence. In addition, we will revise the text by considering the reply to the comment on "p 4, l 103–105": "Another benchmark test is **also**  **compares**...".

- **p 4, l 103–105** : I would transform this into one sentence.

  Reply: We will transform this into one sentence. We will revise the text: on page 4 line 103–105, "Another benchmark test  **compares the** optimal solution  to the true-optimal solution."

- **p 4, l 105–106** : This sentence is too technical with "population" and "generation sizing".

  Reply: We will add explanations to the words: on page 4 line 105, "The dependence of optimal solutions on the initial *populations* **(a technical terminology set in italics is explained in the glossary in Appendix)** is examined...". On page 4 line 106, "...appropriate *population* and *generation* sizing is discussed." This reply is related to the reply to "GA algorithm".

- **p 4, l 107** : "consistency" is too general. One has not enough background information at this point in the text to understand this.

  Reply: We will rephrase the text: on page 4 line 107, "Section 5 verifies **whether**  **the AirTraf simulations are consistent with reference data** and...".

- **p 4, l 108** : "states" : I suggest to use another word.

  Reply: We will revise the text: on page 4 line 108, "...and Sect. 6  **describes** the code availability."

- **p 4, l 112–116** : This paragraph should be rewritten.

Reply: We will rephrase this paragraph (line 112–116): on page 4 line 112–116, "~~AirTraf was developed as a submodel of EMAC (Jöckel et al., 2010). This is reasonable, because we perform global air traffic simulations on long time scales considering local weather conditions. Geographic location and altitude at which emissions are released should be also considered. In addition, various submodels of EMAC can be used to evaluate climate impacts. Therefore, EMAC is a well suited development environment for AirTraf.~~ **AirTraf was developed as a submodel of EMAC (Jöckel et al., 2010) to eventually assess routing options with respect to climate. This requires a framework, where we can optimize routings everyday and assess them with respect to climate changes. EMAC provides an ideal framework, since it includes various submodels, which actually evaluate climate impact, and it simulates local weather situations on long time scales. As stated above, we were focusing on the development of this model. A publication on the assessment of routing changes will be published as well.**"

- **p 4, l 112** : "reasonable" : I think this is not enough as a motivation.

  Reply: We will rephrase this paragraph to make clear the motivation. Please see the reply to the comment: "p 4, l 112–116".

- **p 4, l 113** : "because we perform global air traffic simulations on long time scales considering local weather conditions." : I think this is a vague argumentation.

  Reply: We will rephrase this paragraph. Please see the reply to the comment: "p 4, l 112–116".

- **p 4, l 114** : "geographic location and altitude at which emissions are released should be also considered" : vague.

  Reply: This part is already explained in Introduction: on page 2 line 49–50, "The climate impact induced by aircraft emissions depends on local weather conditions: it depends on geographic location (latitude and longitude) and altitude at which the emissions are released (except for $CO_2$) and time." We will rephrase this paragraph. Please see the reply to the comment: "p 4, l 112–116".

- **p 4, l 115** : This is maybe the main reason why the effort is done to implement AirTraf in a climate model, and not just in a NWP, or using off-line available weather forecasts. So make this more explicit, and give examples of which climate impacts can be evaluated.

  Reply: Yes. We need the framework of EMAC to assess routing options. By following the referee comment, we will rephrase this paragraph. Please see the reply to the comment: "p 4, l 112–116".

- **p 4, l 117** : Explain what "entries" are.

  Reply: We will rephrase the word "entries" into "parameters" to make clear the meaning of the word: on page 4 line 117, "...AirTraf  **parameters** are read in messy_initialize,...". In addition, we will modify Fig. 2 and its caption: on page 30 in Fig. 2, "AirTraf  **parameters**"; and in the caption, "...AirTraf  **parameters** are input in the initialization phase."

- **p 4, l 121–124** : This sentence should be improved. You have to put "here PE is synonym to MPI task" possibly between brackets. I am also not sure whether "while" is the most appropriate word to use here.

  Reply: As the referee noted, we will put "here PE is synonym to MPI task" between brackets. In addition, we will remove "while" and transform the sentence into two sentences: on page 4 line 121–124, "the one-day flight plan is decomposed for a number of processing elements **(PEs, here PE is synonym to MPI task)**, so that each PE has a similar work load. **A** whole flight trajectory between an airport pair is handled by the same PE." Related to this modification, we will also modify the caption of Fig. 3: on page 31 in Fig. 3, "A one-day flight plan is distributed among many processing elements (PEs) in messy_init_memory

(blue).**, while a A** whole trajectory of an airport pair is handled by the same PE...".

- **p 4, l 125** : I think one should be more specific about what a "time loop" is : isn't rather meant "time step"?

  Reply: We used the word "time loop" according to the following publication, which is one of the basic documents about on the ECHAM5/MESSy Atmospheric Chemistry (EMAC) model: "Jöckel, P., Sander, R., Kerkweg, A., Tost, H., and Lelieveld, J.: Technical Note: The Modular Earth Submodel System (MESSy) - a new approach towards Earth System Modeling, Atmos. Chem. Phys., 5, 433-444, doi:10.5194/acp-5-433-2005, 2005." AirTraf is developed as a submodel of EMAC. Therefore, we think that the word "time loop" is helpful for readers (specifically EMAC users) to understand the flowchart of the AirTraf.

- **p 4, l 125–126** : Thus, naturally short-term and long-term simulations consider the local weather conditions for every flight in EMAC. I think this should be explained more clearly.

  Reply: We will revise the sentence: on page 4 line 125–126, "Thus,  **both** short-term and long-term simulations  **can take into account** the local weather conditions for every flight ...".

- **p 4, l 126–127** : "(AirTraf continuously treats overnight flights)" : this is not logically related to the sentence it is attached to. What is meant by this? Because the weather patterns used in AirTraf are the ones at the time of take-off, it seems to me that there is no large complexity about it. Is it therefore still worth mentioning?

  Reply: We agree. The one-day flight plan includes many flight schedules on a single day. Some international (long-distance) flights fly over two days. For example, NH215 departs at MUC on 21:35 and arrives at Tokyo on 15:50 + 1day. We wanted to say here that AirTraf simulates such flights correctly. Indeed, we have been asked about this issue many times so far. Therefore, we believe that it is still worth mentioning.
      Further, from the comment (4) of the referee #1, we will modify the text "(AirTraf continuously treats overnight flights)" into "(AirTraf continuously treats overnight flights **with arrival on the next day**)." After that, the modified text will be moved from the current position to an appropriate position in the manuscript, which is related logically: on page 4 line 125, "Thus,  **both** short-term and long-term simulations  **can take into account** the local weather conditions for every flight ."; and on page 7 line 225, "$pos_{new}$ and $pos_{old}$ are stored in the memory and the aircraft continues the flight from $pos_{new}$ = 2.3 at the next time step **(AirTraf continuously treats overnight flights with arrival on the next day)**."

**Page 5**

- **p 5, l 131–132** : What is meant by these "global fields"? Give examples.

  Reply: This means "three dimensional emission fields" and we call this "global fields" in the paper. We will add the text to make clear this point: on page 5 line 131–132, "...the calculated flight trajectories and global fields **(three dimensional emission fields)** are output (Fig. 2, rose red). The results are gathered from all PEs for output ."

- **p 5, l 132–134** : What is meant by the sentence "Other evaluation models ... on the climate impact"? I suggest to make this more concrete.

  Reply: We just wanted to say that other objective functions (or other evaluation models) will be integrated into AirTraf in order to assess routing options on climate impact reduction. However, this is not necessary for our argument here. Therefore, we will modify the sentence: on page 132–134, "**T**he output  **will be** used to evaluate the reduction potential of the routing option on the climate impact."

- **p 5, l 135–136** : "$R_E$ = 6371 km" : I don't know whether this level of detail should be mentioned in the manuscript.

Reply: We believe that this information is important, because great circle distances can vary considerably with differences of $R_E$. Concerning this issue, we will revise the caption of Table 4 from the comment (2) of the referee #2 as "...column 4 ($d_{\text{MTS}}$) shows the result calculated with the Movable type scripts (MTS),  **using the Haversine formula with a spherical Earth radius of $R_E$ = 6,371 km**."

- **p 5, l 137–138** : The Mach number is a ($\rightarrow$ "the") velocity divided by a ($\rightarrow$ "the") speed of sound.

  Reply: We will revise the text: on page 5 line 137–138, "...the Mach number is  **the** velocity divided by  **the** speed of sound."

- **p 5, l 138** : "true air speed" $\rightarrow$ "the true air speed". Maybe add to the sentence : "When an aircraft flies at a constant Mach number". Isn't "vary along flight trajectories" enough? I don't think that "latitude, longitude, altitude and time" should be added. If one really wants to be more specific, I would rather add temperature and wind speed as factors modifying the true air speed and ground speed.

  Reply: By following the referee comment, we will revise the text: on page 5 line 138, " **When an aircraft flies at a constant Mach number, the** true air speed $V_{\text{TAS}}$ and ground speed $V_{\text{ground}}$ vary along the flight trajectories ."

- **p 5, l 142** : limits rates $\rightarrow$ limit rates.

  Reply: We will correct the word: on page 5 line 142, "...and **limits rates** of aircraft climb...".

- **p 5, l 142** : Explain "semi-circular rule", and "sector demand analysis".

  Reply: We will modify the words to explain them clearly: on page 5 line 142, "...such as the semi-circular rule **(the basic rule for flight level)** and limits rates of aircraft climb and descent, are disregarded. However, a  **workload** analysis of **air traffic controllers** can be performed on the basis of the output data."

- **p 5, l 144** : "mention" : I do not think this is the appropriate wording.

  Reply: We will revise the text: on page 5 line 144, "The following sections  **describe** the used models briefly...".

- **p 5, l 149** : What is meant by "interactions with human influences"?

  Reply: This means the influence coming from anthropogenic emissions. AirTraf describes one of them. We will rephrase the text: on page 5 line 149, "...and their interaction with oceans, land and  influences **coming from anthropogenic emissions** (Jöckel et al., 2010)."

- **p 5, l 153** : T42L31ECMWF-resolution $\rightarrow$ T42L31ECMWF resolution

  Reply: We will revise the word: on page 5 line 153, "...in the **T42L31ECMWF resolution**,...". On page 18 line 599, "...in the **T42L31ECMWF resolution**."

- **p 5, l 159** : Can it exist out of more than one day? On page 6, line 163 : "Any arbitrary number of flight plans is applicable to AirTraf". So one can give flight plans for many days at once?

  Reply: As the referee noted, this point is not clear what we mean by the phrase "one-day flight plan." As shown in Fig. 3 on page 31, the one-day flight plan, which includes many flight schedules on a single day, is used in AirTraf. This flight plan is reused for simulations longer than two days, as described on page 8 line

240. To clarify this point, we will add a short description the first time we use the phrase "one-day flight plan": on page 4 line 121, "As shown in Fig. 3, the one-day flight plan**, which includes many flight schedules of a single day,** is decomposed for..." (this reply is related to the comment (3) of the referee #1).

- **p 5, l 160** : of A330-301 → of an A330-301 aircraft.

  Reply: We will revise the word in the revised manuscript: on page 5 line 160, "...the primary data of **an** A330-301 **aircraft** used...". The caption of Table 1 on page 51, "Primary data of **Airbus** A330-301 **aircraft** and...".

- **p 5, l 162** : a departure time → the departure time.

  Reply: We will revise the word: on page 5 line 162, "...latitude/longitude of the airports, and  **the** departure time."

- **p 5, l 162** : as values [-90,90] → as values in the range [-90,90].

  Reply: We will add the text "in the range" in the revised manuscript: on page 5 line 162, "The latitude and longitude coordinates are given as values **in the range** $[-90, 90]$ and...".

**Page 6**

- **p 6, l 164** : the data are required → these data are required.

  Reply: We will revise the word: on page 6 line 164, "...;  **these** data are required to calculate...".

- **p 6, l 165** : "As for ..." → "Concerning ...".

  Reply: We will revise the text: on page 6 line 165, " **Concerning** the engine performance data,...".

- **p 6, l 166** : flows (plural) while index (singular).

  Reply: Thank you so much. We will revise the text: on page 6 line 166, "...reference fuel **flows** $f_{ref}$ (in kg(fuel)s$^{-1}$) and...".

- **p 6, l 168** : What is meant by an "overall" weight factor?

  Reply: The word "overall" means "passenger/freight/mail". we will add this text: on page 6 line 168, "An overall **(passenger/freight/mail)** weight load factor is also provided...". On page 51 at the line with OLF in Table 1, "ICAO overall **(passenger/freight/mail)** weight load factor **in 2008**$^{d}$".

- **p 6, l 171** : are described "here" step by step.

  Reply: We will add the word "here" in the revised manuscript: on page 6 line 171, "The calculation procedures in the AirTraf integration are described **here** step by step."

- **p 6, l 172** : a flight status → the flight status.

  Reply: We will revise the text: on page 6 line 172, "... **the** flight status of all flights is initialized...".

- **p 6, l 178** : moving aircraft position → aircraft position calculation.

  Reply: We will revise the word "moving aircraft position" into "aircraft position calculation" in the revised manuscript: on page 6 line 178, "...fuel/emissions calculation,  **aircraft position calculation** and gathering global emissions." Further, on page 30 in the Fig. 1 (bold-black box, light blue),

" **Aircraft position calc.**" On page 32 in the caption of Fig. 4, "(c)  **aircraft position calculation**."

- **p 6, l 182–183** : differ to → differ from.

  Reply: We will revise the text: on page 6 line 182–183, "...fuel (might differ  **from** $H_2O$,...".

- **p 6, l 184** : can be used → can currently be used.

  Reply: We will add the word "currently" in the revised manuscript: on page 6 line 184, "...the great circle and the flight time routing options can **currently** be used."

- **p 6, l 187** : for a selected option → for the selected option.

  Reply: We will revise the text: on page 6 line 187, "...a single-objective minimization problem is solved for  **the** selected option...".

- **p 6, l 191–194** : Why adding these sentences? It makes the text confusing. In addition, it is not well defined how an optimization might work when one optimizes according to two criteria (time and cost). One should also mention then how to weight or compare both (trade-off between them).

  Reply: We have a reason why we added the sentence. Here, we would like to show clearly that a time-optimal route is different from a wind-optimal route. In this paper, we optimize flight trajectories with respect to "time" by taking into account wind effects. These routes are the time-optimal routes, not the wind-optimal routes, because the objective function is different between the time-optimal and the wind-optimal routing options, as described on page 6 line 191–194. We have seen situations many times that people assumed the time-optimal route including wind effects as "the wind-optimal route." To avoid this situation, we distinguish the routes clearly.
  To explain this better, we will revise the text: on page 6 line 191–196, "Generally, a wind-optimal route means an economically optimal flight route taking the most advantageous wind pattern into account. This route minimizes total costs with respect to time, **fuel** and **other** economic costs , i.e. it has multiple objectives.  AirTraf  **will provide**  the flight time and the fuel routing options separately to investigate trade-offs (conflicting scenarios) among different routing options. " This reply is related to the reply to "p 3, l 69".

- **p 6, l 197** : The CCF is → The CCFs are.

  Reply: We will revise the text: on page 6 line 197, "The CCF**s**  **are** provided by the...".

- **p 6, l 199** : "total" climate impacts versus "some" aviation emissions : this sounds strange.

  Reply: We will remove the word "total" from the text: on page 6 line 199, "...and estimates  climate impacts due to some aviation emissions."

**Page 7**

- **p 7, l 211** : $n_{wp-1}$ → $n_{wp} - 1$.

  Reply: Thank you so much. We will correct the text: on page 7 line 211, "...the flight segment index ($i = 1, 2, ..., $  $\boldsymbol{n_{wp} - 1}$)."

- **p 7, l 212–213** : calculation/calculation/calculate : try to vary the wording more.

Reply: We will revise the text: on page 7 line 212–213, "$_x$ $_2$ **Next, fuel use, NO$_x$ and H$_2$O emissions are calculated by the dedicated module (Fig. 2, light orange); this module comprises a total energy model based on the BADA methodology (Schaefer, 2012) and the DLR fuel flow method (Deidewig et al., 1996, see Sects. 2.5 and 2.6 for more details).**"

- **p 7, l 218–219** : corresponding to time steps → corresponding to "the" time steps.

  Reply: We will add the word "the" in the sentence: on page 7 line 218–219, "...along the flight trajectory corresponding to **the** time steps of EMAC (Fig. 4c)."

- **p 7, l 219–220** : "present" and "previous" is a bit vague : isn't it the position at the beginning of a time step of EMAC, and at the end of a time step?

  Reply: Thank you so much. We will revise the text: on page 7 line 219–220, "...aircraft position parameters pos$_{new}$ and pos$_{old}$ are introduced to indicate  **the** present **position (at the end of the time step)** and previous position **(at the beginning of the time step)** of the aircraft along the flight trajectory."

- **p 7, l 220** : "a" present and previous position → "the" present and previous position.

  Reply: We will revise the text: on page 7 line 220, "...aircraft position parameters pos$_{new}$ and pos$_{old}$ are introduced to indicate  **the** present and previous position...".

- **p 7, l 221** : by real numbers of the waypoint index → by real numbers as a function of the waypoint index.

  Reply: We will revise the text: on page 7 line 221, "They are expressed by real numbers **as a function** of the waypoint index...".

- **p 7, l 224** : I would rather say : "This means that the aircraft moves 100% of the distance between $i = 1$ and $i = 2$, and 30 % of the distance between $i = 2$ and $i = 3$ in one time step."

  Reply: Thank you so much. We will revise the text: on page 7 line 224, "This means that the aircraft moves **100% of the distance between $i = 1$ and $i = 2$, and** 30 % of the distance between $i = 2$ and $i = 3$ in one time step."

- **p 7, l 233** : is used → are used.

  Reply: We will revise the text: on page 7 line 233, "...the coordinates of the $(i+1)^{th}$ waypoint  **are** used to find the...".

- **p 7, l 233** : This is a little bit inaccurate (see also Fig. 4). Assess the impact of this inaccuracy.

  Reply: Unfortunately, we do not understand the referee comment. In this sentence, we describe how to gather the aircraft emissions for the case NO$_{x, i}$, as example. This treatment is the same for the cases NO$_{x, i-2}$ and NO$_{x, i-1}$: as shown in Fig. 4d on page 32, for the fraction of NO$_{x, i-2}$, the coordinates of the $(i-1)^{th}$ waypoint is used to find the nearest grid point. Nevertheless, we improve the caption of Fig. 4: on page 32 in the caption of Fig. 4, "...(d) Gathering global emissions; the fraction of NO$_{x, i}$ corresponding to the  **flight segment $i$** is mapped onto the nearest grid box."

**Page 8**
- **p 8, l 237** : "If $t \geq 2$ of the day" : express this better.

Reply: We will revise the text: on page 8 line 237, "**Once the status becomes 'in-f light'**, the departure check is false **in subsequent time steps ($t \geq 2$) and...**".

- **p 8, l 239** : without recalculating flight trajectory and fuel emissions → without recalculating the flight trajectory or fuel emissions.

  Reply: Thank you so much. We will revise the text: on page 8 line 239, "...the aircraft moves to the new air craft position without recalculating **the** flight trajectory  **or** fuel/emissions."

- **p 8, l 240–241** : "For more than two consecutive days simulations" → "For simulations longer than two days".

  Reply: Thank you so much. We will revise the text: on page 8 line 240–241, "For **simulations**  **longer than two**  **days** , the same flight plan...".

- **p 8, l 243** : Twice "calculation".

  Reply: We will remove the first "calculation" in the sentence: on page 8 line 243, "The  methodologies of the fuel/emissions calculation module (Fig. 2, light orange) are described."

- **p 8, l 246** : are used → is used.

  Reply: Thank you so much. We will revise the word "are" into "is" in the revised manuscript: on page 8 line 246, "A total energy model based on the BADA methodology and the DLR fuel flow method  **is** used."

- **p 8, l 246–247** : the first trip fuel estimation → a first trip fuel estimation.

  Reply: We will correct the text: on page 8 line 246–247, "The fuel use calculation consists of the following two steps:  **a** first rough trip fuel estimation and...".

- **p 8, l 247** : the second fuel calculation : a bit vague. Maybe mention that it is more detailed.

  Reply: We will add the word "detailed" in the text: on page 8 line 247, "... **a** first rough trip fuel estimation and the second **detailed** fuel calculation...". Related to this issue, we will add the word "detailed" into the text in Fig. 2 (dashed box, light orange): on page 30, "2nd **detailed** fuel calc.".

- **p 8, l 256** : mean flight altitude of the flight → mean altitude of the flight.

  Reply: We will remove the first "flight" from the sentence: on page 8 line 256, "$F_{\mathrm{BADA}}$ is calculated by inter polating the BADA data (assuming nominal weight) to the mean  altitude of the flight...".

- **p 8, l 260** : it is assumed as → it is assumed to be.

  Reply: Thank you so much. We will revise the text: on page 8 line 260, "It is assumed  **to be** 3 % of the FUEL$_{\mathrm{trip}}$...".

**Page 9**

- **p 9, l 274–275** : "For an aircraft in cruise ..." : express this better.

  Reply: Please see the reply to the referee comment: "Sensitivity (1)."

- **p 9, line 276–278** : One should have a "," or a "." after most of the formula.

Reply: As the referee pointed out, we will recheck the all equations and add "," or "." after most of them. We will reply to this issue in the section of "Mathematical formulas (3)."

- **p 9, line 280** : The numerical value of $\rho_i$ is not given in Table (2) (as for $S$, $C_{D0}$ and $C_{D1}$ in Table 1).

  Reply: The referee is right. We will revise and add the text: on page 9 line 280, "The performance parameters ($S$, $C_{D0}$ and $C_{D2}$) **are given in Table 1,**  $\rho_i$ **is the air density (Table 2)** . **and** $V_{\text{TAS},i}$ **is calculated at every waypoint (Table 2).**"

- **p 9, l 281** : a fuel flow → the fuel flow.

  Reply: We will revise the text: on page 9 line 281, "...and  **the** fuel flow of the aircraft...".

- **p 9, l 282** : I suggest to skip "for jet aircraft".

  Reply: We will skip the text "for jet aircraft" in the sentence: on page 9 line 282, "...calculated assuming a cruise flight :".

- **p 9, l 283–284** : "," after the equations.

  Reply: We will add "," after Eqs. (7) and (8). We will reply to this issue in the section of "Mathematical formulas (3)."

- **p 9, l 287** : Oneday : I suggest to find another name for this variable in the manuscript. In addition, its units in Table 1 should be "sec day$^{-1}$ ".

  Reply: We agree. We will change the name for the variable "Oneday" into the "SPD" (the Seconds Per Day) throughout the revised manuscript: Eq. (9) on page 9 line 287, "FUEL$_i$ = $F_{cr,i}$ ($ETO_{i+1}$ − $ETO_i$)  **SPD**". Further, on page 9 line 288, "...is converted into seconds by multiplying  **with Seconds Per Day** (**SPD,** Table **1**)." On page 12 line 383−385 in Eqs. (26) and (27), "$V_{\text{ground,i−1}}$ ×  **SPD** (denominator)" and "$FT$ = (ETO$_{\text{nwp}}$ − ETO$_1$) ×  **SPD**." On page 51 in Table 1, "(Parameter)  **SPD**; (Value) 86,400; (Unit) s **day$^{-1}$** ; (Description) Time (Julian date) ×  **SPD** = Time (s)." On page 52 in Table 2, description of row 15, "$FT$ = (ETO$_{\text{nwp}}$ − ETO$_1$) ×  **SPD**."

- **p 9, l 289** : "reflects" → "incorporates" or "is impacted by".

  Reply: We will revise the text: on page 9 line 289, "The FUEL$_i$  **incorporates** the tail/head winds effect...".

- **p 9, l 290** : ($m$) → ($m_i$).

  Reply: We will revise the text: on page 9 line 290, "The relation between the FUEL$_i$ and the aircraft weight ($m_i$) is...".

- **p 9, l 294** : next to the last → at the one but last.

  Reply: Thank you so much. We will revise the text: on page 9 line 294, "...the aircraft weight  **at the one but last** waypoint...".

- **p 9, l 296–297** : I do not think this last sentence gives new information. Or formulate it nicer.

  Reply: We agree. We will remove the last sentence in the revised manuscript: on page 9 line 296−297, "

".

**Page 10**

- **p 10, l 302** : first → First.

  Reply: We will revise the text: on page 10 line 302, "The calculation procedure follows four steps: **F**irst, the reference fuel flow...".

- **p 10, l 310–311** : corresponding sea level values → corresponding values at sea level.

  Reply: Thank you so much. We will revise the text: on page 10 line 310−311, "$P_0$ and $T_0$ are the corresponding  values **at sea level**...".

- **p 10, l 314–315** : "," after equations.

  Reply: We will add "," after Eqs. (14) and (15). We will reply to this issue in the section of "Mathematical formulas (3)."

- **p 10, l 327** : "... and $q_i$ is the specific humidity at $h_i$ " : mention units of $q_i$ (kg kg$^{-1}$, g kg$^{-1}$, ...).

  Reply: We will add the unit in the sentence: on page 10 line 327, "...and $q_i$ **(in kg(H$_2$O)(kg(air))$^{-1}$)** is the specific humidity at $h_i$...".

- **p 10, l 329** : pre-calculated → calculated.

  Reply: We will modify the word: on page 10 line 329, "...using the **calculated** FUEL$_i$...".

- **p 10, l 330–331** : "," after equations. I do not think it is a good idea to have variables whit names as NO$_{x,i}$ and H$_2$O$_i$. I would rather use names like $m_{NOx}$.

  Reply: We will add "," after Eqs. (19) and (20). We will reply to this issue in the section of "Mathematical formulas (3)." Further, we understand the referee comment. Nevertheless, we are hesitating to change the variable names, because "$m$" is already used for the aircraft weight, as described on page 9 line 290. Maybe the names are not the best ones, however, we think that the "NO$_{x,i}$" and "H$_2$O$_i$" show clearly that these emissions are calculated for the $i^{th}$ flight segment.

**Page 11**

- **p 11, l 339** : one-day → one day of.

  Reply: From the reply to the referee comment on "p2, line 27," we will define the word "one-day AirTraf simulation": on page 4 line 106, "In Sect. 4,  AirTraf simulations are demonstrated  **with** the two options **for a typical winter day (called one-day AirTraf simulations)** and the results are discussed." Therefore, we will also use the word here.

- **p 11, l 343** : works → works only.

  Reply: We will add the word "only" in the sentence: on page 11 line 343, "The current aircraft routing module (Fig. 2, light green) works **only** with respect to the great circle and...".

- **p 11, l 351** : arctan, sin, cos, ... should not be italic.

  Reply: We will modify the all names of trigonometric formulas into normal straight letters in the revised

manuscript. We will reply to this issue in the section of "Mathematical formulas (4)."

- **p 11, l 351** : "," after equation.

  Reply: We will add "," after Eq. (21). We will reply to this issue in the section of "Mathematical formulas (3)."

- **p 11, l 362** : Why mentioning "km" here? Better to write on line 355 : $d_i$ (km).

  Reply: The "km" is described here for the flight altitude "$h_i$" (not for the great circle distance $d_i$), because Table 2 shows the unit of $h$ is "m". To clarify this, we will add the text in the sentence: on page 11 line 362, "...(**h is used in** km in Eqs. (22) and (23)) and...".

- **p 11, l 363** : i.e. the → i.e.

  Reply: We will remove the word "the" in the sentence: on page 11 line 363, "...hence the great circle distance between airports, i.e. ...".

- **p 11, l 365** : "based on Polar coordinates"? Explain this better.

  Reply: We think that the word "based on" seems to be confusing. We will revise the text: on page 11 line 365, "...by linear interpolation  **in** Polar coordinates."

- **p 11, l 365** : therefore → in that case.

  Reply: We will revise the word "therefore" into "in that case" in the revised manuscript: on page 11 line 365, "... **in** Polar coordinates.  **In that case**,...".

**Page 12**
- **p 12, l 370** : of the $i^{\text{th}}$ waypoint → at the $i^{\text{th}}$ waypoint.

  Reply: We will change the word "of" into "at" in the revised manuscript: on page 12 line 370, "...the true air speed $V_{\text{TAS}}$ and the ground speed $V_{\text{ground}}$  **at** the $i^{\text{th}}$ waypoint are calculated...".

- **p 12, l 371–372** : "," after equations.

  Reply: We will add "," after Eqs. (24) and (25). We will reply to this issue in the section of "Mathematical formulas (3)."

- **p 12, l 374** : where $M$ is "the" Mach number.

  Reply: We will add the word "the" in the sentence: on page 12 line 374, "...where $M$ is **the** Mach number,...".

- **p 12, l 378–379** : Although it is mentioned that $V_{\text{TAS}}$, $V_{\text{wind}}$ and $V_{\text{ground}}$ are scalars, Eq. (25) on line 372 is actually a vector equation.

  Reply: As described on page 12 line 377−379, the flight direction is firstly calculated for every flight segment. Thereafter, the values of $V_{\text{TAS,i}}$ $V_{\text{wind,i}}$ and $V_{\text{ground,i}}$ "corresponding to the flight direction" are calculated. For example, $V_{\text{ground,i}}$ is a component of the wind vector along the flight direction (i.e. scalar value). Therefore, Eq. (25) on line 372 is a scalar equation.

- **p 12, l 386** : "reflects" : this is not the only aspect which is reflected. I suggest to use "incorporates".

Reply: Thank you so much. We will revise the text: on page 12 line 386, "...and ETO$_i$  **incorporates** the influence of tail/head winds...". In the same way, we will revise the text: on page 21 line 700, "..., which  **incorporates** the influences of both $V_{\text{TAS}}$ and winds...".

- **p 12, l 390** : for the five → for five.

  Reply: We will revise the text: on page 12 line 390, "Great circles were calculated for  five representative routes...".

- **p 12, l 393–395** : 180 → 180˚ (while "deg" on line 397).

  Reply: We will revise the sentence: on page 12 line 393−395, "...the difference in longitude between them was $\Delta\lambda_{\text{airport}} < 180$˚ ; R2 consisted of an airport pair in the northern hemisphere (HND-JFK) with $\Delta\lambda_{\text{airport}} > 180$˚ (discontinuous longitude values...".

- **p 12, l 398** : Missing deg?

  Reply: Thank you so much. We will revise the sentence: on page 12 line 397−398, "..., where $\Delta\lambda_{\text{airport}} = 0$˚ and the difference in latitude was $\Delta\phi_{\text{airport}} /= 0$˚; and R5 was another special route with $\Delta\lambda_{\text{airport}} /= 0$˚ and $\Delta\phi_{\text{airport}} = 0$˚."

- **p 12, l 399** : ";" → ",".

  Reply: We will modify the text: on page 12 line 399, "...as follows: $M = 0.82$**,** $h_i = 0$,...".

**Page 13**
- **p 13, l 403** : varying $n_{\text{wp}}$ in "the range" [2, 100].

  Reply: We will add the text "the range" in the revised manuscript: on page 13 line 403, "...$n_{\text{wp}}$ was analyzed varying $n_{\text{wp}}$ in **the range** [2, 100]."

- **p 13, l 404** : and the MTS → and MTS.

  Reply: We will delete the word "the" in the sentence: on page 13 line 404, "...by Eqs. (22) and (23) and  MTS."

- **p 13, l 406** : I do not think that $\Delta d_{\text{eq23,eq22}}$, etc. are appropriate choices for variable names. As these are difference, I think they should not not have a specific variable name attributed.

  Reply: We understand the referee comment. Nevertheless, we are hesitating to change those variable names. We define the variable name for a flight distance as "*d*", as shown in Table 2, and we use the variable "*d*" consistently in the manuscript: on page 11 Eqs. (22) and (23), on page 15 Eq. (28), etc. We think that the current expressions make sense. This reply is related to the reply to "5 Comments on tables, Table 4."

- **p 13, l 409–410** : "shows" versus "showed".

  Reply: We will revise the text: on page 13 line 409−410, "Figure 6 shows the result of the sensitivity analysis of $n_{\text{wp}}$ on the great circle distance. The results **showed** that the distance...".

- **p 13, l 413** : I would not call it linear interpolation : one goes straight whereas the other follows an arc. Shouldn't you also add that $n_{\text{wp}}$ maybe should depend on the length of the flight?

  Reply: We will remove the word "linear interpolation" in the sentence. This is not necessary for our argument

here: on page 13 line 413, "...when using fewer $n_{\mathrm{wp}}$, ." The referee actually points out the important issue. However, we think that it is more important for readers (specifically AirTraf users) to show a criteria to use Eq. (23). For this, we describe as: on page 13 line 414, "Therefore, $n_{\mathrm{wp}}$ $\geq 20$ is practically desired for the use of Eq. (23)."

- **p 13, l 417** : with respect to the flight time routing option → with respect to the flight time.

  Reply: We will revise the text: on page 13 line 417, "The flight trajectory optimization with respect to the flight time  was...".

- **p 13, l 418** : algorithms → algorithm.

  Reply: We will correct the word: on page 13 line 418, "..., which is a stochastic optimization algorithm."

- **p 13, l 422** : The ARMOGA → ARMOGA.

  Reply: We will revise the text: on page 13 line 422, " ARMOGA will be implemented...".

- **p 13, l 424–425** : With a routing option → For each routing option (except ...). I also suggest to skip "on the selected routing" in the second part of the sentence.

  Reply: We will revise the sentence: on page 13 line 424−425, " **For each** routing option**,** except for the great circle routing option, a single-objective optimization problem  is solved."

- **p 13, l 427** : Explain what an objective function in this context is.

  Reply: The word "objective function" means "evaluation function." The word "objective function" is the technical term (commonly used in GA-optimization terminology). Therefore, we will revise the sentence: on page 13 line 427, "Therefore, various  **evaluation functions (called objective functions)** can easily be adapted...".

- **p 13, l 432-433** : "Is called "an" optimal solution" and "is called "the" true-optimal solution".

  Reply: We will revise the sentence: on page 13 line 432−433, "A solution found in GA is called **an** optimal solution, whereas a solution having the theoretical-optimum of the objective function is called **the** true-optimal solution."

- **p 13, l 434** : Say what is meant by converge : larger initial population, or just more generations?

  Reply: The word "converge" means "becomes close to" in this context. As described on page 13 line 432−433, there are two solutions: an optimal solution and the true-optimal solution. When we solve an optimization problem, we expect that the optimal solution (our solution) "converges" to the true-optimal solution by optimization algorithms. This is what we wanted to say here.

- **p 13, l 435** : Will every flight have the same size for its initial population, and the same number of generations? Is that independent of the length of the flight?

  Reply: This paper aims to confirm the ability of the optimization module (GA) to optimize flight routes. Therefore, we solved the simple time-optimal optimization problem using the common optimization setup (the same size for initial populations and the same number of generations for every flight). We understand that the referee pointed out an important issue. However, this is beyond the scope of this paper. If we could choose the setup individually for every flight, the computational requirements for the trajectory optimization could probably be decreased. However, it is difficult to find an appropriate GA setup for every flight before

solving the optimization problem. As the referee noted, the flight length can be used to adjust the population size and the number of generations for a flight. On the other hand, if a day shows complicated weather situations, GA needs a larger population size and more generations to converge. This issue will be one of our future investigations.

**Page 14**

- **p 14, l 440–441** : I do not think that "definitions" is the appropriate word to be used here.

  Reply: We believe that the word "definitions" is appropriate here. To solve an optimization problem, firstly, one has to define the optimization problem itself concerning variables, ranges of variables, evaluation functions, constraints, etc. Thereafter, one can solve the problem. On page 14, Sect. 3.2.2 describes the definitions of the flight trajectory optimization, which we solve here.

- **p 14, l 441** : of objective functions → of the objective function.

  Reply: We will revise the text: on page 14 line 441, "..., the definition of **the** objective function and the genetic operators."

- **p 14, l 444** : used interchangeably to mean a flight trajectory → used interchangeably to flight trajectory.

  Reply: We will revise the text: on page 14 line 444, "...the term is used interchangeably to  flight trajectory...".

- **p 14, l 445** : $n_{dv} = 11$ should not be here.

  Reply: We will remove the word "$n_{dv} = 11$" in the sentence and modify the text: on page 14 line 445, "...the design variable index $j$ ($j$ = 1, 2,···, $n_{dv}$ ),..". On page 15 line 487, "...where $\boldsymbol{n_{dv}}$ **= 11,** $d_i$ and $V_{\text{ground,i}}$ are calculated...".

- **p 14, l 456** : centering → centered.

  Reply: We will revise the text: on page 14 line 456, "...domains  **centered** around the central points...".

- **p 14, l 463–464** : how are these waypoints calculated? Will the arc lengths be equal?

  Reply: We reply to this issue in the section of "CP in trajectories (3)."

- **p 14, l 458–459 and 470–471** : "GA provided the values" : Do you mean already the final optimal values?

  Reply: Here, we just want to say that the values of the eleven design variables are provided by the GA optimization process. In other words, one does not have to determine the values. In fact, the sentence on page 15 line 479−480 says, "The initial ***population*** operator (Fig. 2, dark green) provides initial values of the eleven design variables as random numbers...". Naturally, GA provides not only initial values, but also the final optimal values regarding the design variables.

- **p 14, l 462** : Explain a little bit more a B-spline curve.

  Reply: We will add the text to specify the curve: on page 14 line 462, "...trajectory is represented by a B-spline curve **(3$^{\text{rd}}$-order)** with the three CPs...". On page 15 line 474, "...trajectory is also represented by a B-spline curve **(3$^{\text{rd}}$-order)** with the...".

- **p 14, l 464** : Are the waypoints on the B-spline curve still equidistant?

Reply: No. The referee is right. We explain this issue in the sections of "CP in trajectories (3) and (4)." Here we will modify the text: on page 14 line 464, "To generate the  **same number of waypoints between the CPs**, $n_{wp}$ was calculated...". Related to this issue, we will delete the text: on page 7 line 206, "...the trajectory consists of waypoints generated  along the trajectory, and flight segments...".

- **p 14, l 461 and 472** : "Here $x_1$ , ... indicate longitudes/latitudes/altitude values". Shouldn't this be mentioned earlier in the paragraphs, i.e. on lines 452 and 466?

  Reply: The referee is right. We will revise the manuscript: on page 14 line 461, "", and on line 452, "...as shown in Fig. 7 (bottom). $x_1$, $x_3$ and $x_5$ **indicate longitudes, while $x_2$, $x_4$ and $x_6$ indicate latitudes.**" On page 14 line 472, "", and on line 466, "...were used (Fig. 7, top). **Here $x_7$ to $x_{11}$ indicate altitude values.**"

**Page 15**
- **p 15, l 477** : where longitude-coordinate of waypoints → where "the" longitude of the waypoints.

  Reply: We will modify the sentence in the revised manuscript. Please see the reply to the referee comment on the "CP in trajectories (4)."

- **p 15, l 476–478** "where longitude-coordinate of waypoints is the same for the two curves." Is this true in the example here? The lon-lat curve contains 3 CPs and thus 4 intervals. The the lon-altitude curve contains 5 CPs and 6 intervals. The number of waypoints is 101, so 100 intervals. This is however not a multiple of 6, so I don't see that the longitude of the waypoints for both B-spline curves are automatically identical.

  Reply: This is true. The longitude of the waypoints for both B-spline curves are identical. A flight trajectory is also represented by a B-spline curve (the lon-altitude curve) and waypoints are generated along the curve. These waypoints are tentative points ($> n_{wp}$). And then, we create actual waypoints on the lon-altitude curve, by interpolating the lon-altitude curve to the longitude-coordinate of the lon-lat curve. We modify the related sentences in the section of "CP in trajectories (4)."

- **p 15, l 479** : "provides initial values by random numbers" : this is too cryptic.

  Reply: As described on page 13 line 418, GA is a stochastic optimization algorithm. Thus, the optimization proceeds using random numbers. Maybe the current sentence is a little bit unclear, therefore we will modify the sentence in the revised manuscript: on page 15 line 479, "The initial ***population*** operator (Fig. 2, dark green) provides initial values of the eleven design variables  **at random within the lower/upper bounds described above**,...".

- **p 15, l 481** : "The operator creates divers solutions defined by a fixed population size $n_p$.": This is a complicated way to say: "The operator creates $n_p$ different solutions (where $n_p$ is the population size)."

  Reply: We agree. We will revise the text: on page 15 line 480−481, "The operator creates  $n_p$ **different solutions (where $n_p$ is the *population* size)**...".

- **p 15, l 481** : "a random set" : do you mean the random set which is just described (then I suggest to use "the"), or is it even another random set? I would put the sentence "GA starts its search with a random set of solutions (population approach)" at the beginning of the paragraph.

  Reply: "a random set" means the random set which is already described. We will move the sentence at the beginning of the paragraph (in this case, the word "a random set" is used). Finally, we will revise the sentence: on page 15 line 479 (at the beginning of the paragraph), "**GA starts its search with a random set**

**of solutions (*population* approach).** The initial ***population*** operator...".

- **p 15, l 483** : By summing the flight time for flight segments → by summing the flight time over all flight segments.

  Reply: We will revise the text: on page 15 line 483, "...for each of the solutions by summing the flight time  **over all** flight segments...".

- **p 15, l 483–484** : "The .. optimization solved here" : too cryptic and vague.

  Reply: We will revise the text: on page 483−484, "The single-objective optimization **problem on the flight time**  **can be written** as follows:".

- **p 15, l 485** : "Minimize" and "Subject to" should not be italic.

  Reply: We will modify the words "Minimize" and "Subject to" with straight letters in the revised manuscript: on page 15 line 485, "** **Minimize**" and "** **Subject to**".

- **p 15, l 490** : What is meant by "solutions that dominate it"?

  Reply: This expression shows an inferior-to-superior relationship among solutions, and is commonly used in GA optimization terminology. In optimization problems, for example, if a solution A is superior to a solution B on an objective function, we can say that the solution A dominates the solution B.

- **p 15, l 489–491** : Why is "rank" written in italic, but "fitness" not?

  Reply: We will add the glossary and refer the word "rank" in italics in the revised manuscript: on page 15 line 489−492, "A ***rank*** of a...was computed by 1/***rank***. A solution...smaller ***rank*** value...". This reply is related to the reply to "GA algorithm".

- **p 15, l 493** : made → makes (because "are identified" on line 488).

  Reply: We will revise the text: on page 15 line 493, "...Sampling Selection (Baker, 1985)  **makes** duplicates...".

- **p 15, l 492** : What is meant by a "mating pool"?

  Reply: We will add the glossary in the revised manuscript to explain the technical term "mating pool". Please see the reply to the referee comment on the "GA algorithm."

- **p 15, l 500** : "This operator was applied to each design value." : Isn't this said already in the sentence before?

  Reply: By following the referee comment, we will delete the sentence and add the word "$n_{dv} = 11$" into the previous sentence: on page 15 line 500−501, "...with $\gamma = (1 + 2\alpha)u_1 - \alpha$ and $j$ varies in $[1, n_{dv}]$ **($n_{dv}$ = 11)**. "

- **p 15, l 504** : "added a disturbance to the child solution." : It does if for both child solutions I presume.

  Reply: The referee comment is correct. We will correct the word "the child solution" into "the child solutions": on page 15 line 504, "...added a disturbance to the child solution**s** by...".

**Page 16**
- **p 16, l 515** : the population of "the" solutions → the population of solutions.

Reply: We will remove the word "the" in the revised manuscript: on page 16 line 515, "...it is expected that the *population* of  solutions is...".

- **p 16, l 517** : "an optimal solution is output." : How is that solution found based on the last generation?

  Reply: We will add the text to inform on how the final solution is obtained from the optimization. Please see the reply to the referee comment on the "GA algorithm."

- **p 16, l 518** : "corresponding to the routing option": I don't think this has to be repeated here.

  Reply: We will remove the word "corresponding to the routing option" in the revised manuscript: on page 16 line 517−518, "..., GA quits the optimization and an optimal solution **showing the best *f* of the whole generation** is output :...".

- **p 16, l 518** : "the best" : one cannot guarantee that it is the best I think.

  Reply: By following the referee comment, we will change the word "the best" into "the superior" in the revised manuscript: on page 16 line 518, "The optimal solution has the  **superior** combination of the...".

- **p 16, l 519** : "naturally" : is this the appropriate wording?

  Reply: We will revise the sentence: on page 16 line 519, "**T**he flight properties of the optimal solution are **also** available...".

- **p 16, l 521–522** : can be applied to any routing option (I thought that was not possible yet in version 1.0?) → could.

  Reply: We agree. We will correct the word "can" into the "could" in the revised manuscript: on page 16 line 521−522, "The flight trajectory optimization methodology described here  **could** be applied to any routing option...".

- **p 16, l 529** : "As $V_{TAS}$ and $V_{ground}$ were set to 898.8 km h$^{-1}$" : Isn't it better to mention first explicitly that we have set $V_{wind} = 0$, and from that it follows that $V_{TAS}$ and $V_{ground}$ are 898.8 km h$^{-1}$ (and not set).

  Reply: By following the referee comment, we will revise the sentence: on page 16 line 529, "$\boldsymbol{V_{wind}}$ **was set to 0 km h$^{-1}$ (no-wind conditions);**  $V_{TAS}$ and $V_{ground}$ were set to 898.8 km h$^{-1}$ (constant) **. Hence,**  $f_{true}$ equals the flight time along the great circle from MUC to JFK at FL290:...".

- **p 16, l 531** : Maybe one should say why flying at FL290 will be faster than at other altitudes. I assume that this depends on the value of $T$. Are the initial and final points at FL290? Mention that $M = 0.82$.

  Reply: To show clearly why flying at FL290 will be faster than at other altitudes, we will add the text in the revised manuscript: on page 16 line 530−531, "...$f_{true}$ equals the flight time along the great circle from MUC to JFK at FL290 **(having its minimum $d_i$ in the range of [FL290, FL410])**: $f_{true}$ = 25,994.0 s...".
      In this benchmark test (off-line), $V_{wind}$ = 0 km h$^{-1}$ and $V_{TAS}$ = $V_{ground}$ = 898.8 km h$^{-1}$ were set, as described on page 16 line 529. Hence, the results do not depend on the values of $T$ and $M$ (see Eqs. (24) and (25)).
      In addition, the initial and final points were at FL290. Table 5 summarizes the calculation conditions for the test. In Table 5, the altitudes of departure (MUC) and arrival airport (JFK) are described as, "alt. = FL290."

- **p 16, l 537** : total 1000 independent → a total of 1000 independent.

Reply: We will revise the text: on page 16 line 537, "...i.e. **a** total **of** 1,000 independent GA simulations...".

- **p 16, l 532–538** : Isn't the first experiment also included in the second setup?

Reply: Yes. To clarify this point, we will modify the text: on page 16 line 532−538, " 10 independent GA simulations from different initial **populations** were performed for each combination of $n_p$ (10, 20,···, 100) and $n_g$ (10, 20,···, 100), i.e. total 1,000 independent GA simulations were performed. " Related to this modification, we will add the text: on page 17 line 559, "...the 10 independent GA simulations **from different initial populations** with $n_p = 100$ and $n_g = 100$."

**Page 17**
- **p 17, l 540** : generation number $n_g$ → number of generations $n_g$.

Reply: We will revise the text: on page 17 line 540, "The influence of the **population** size $n_p$ and the **number of *generations***  $n_g$...". In the same way, we will revise the manuscript as follows: on page 16 line 517, "...computed for a fixed **number of *generations***  $n_g$,...". On page 35 in the caption of Fig. 8, "...and **the number of *generations***  $n_g$." On page 35 in the x-axis label of Fig. 8, " **number of generations $n_g$**". On page 36 in the caption of Fig. 9, "...and **the number of *generations***  $n_g$ **is** 100." On page 38 in the caption of Fig. 11, "...and **the number of *generations***  $n_g$ **is** 100." On page 39 in the caption of Fig. 12, "...and **the number of *generations***  $n_g$." On page 44 in the caption of Fig. 17, "...and **the number of *generations***  $n_g$ **is** 100." On page 55 in Table 5, " **Number of generations**". On page 56 in Table 7, " **Number of generations**". On page 8 (Supplementary material) in the caption of Table S1, "...and **the number of *generations***  $n_g$ = 100." On page 9 (Supplementary material) in the caption of Table S2, "...and **number of *generations***  $n_g$...". On page 9 (Supplementary material) in Table S2, " **Number of generations** $n_g$".

- **p 17, l 541** : Is "confirmed" the appropriate wording?

Reply: We will modify the word: on page 17 line 540−541, "...the convergence properties of GA was  **examined**."

- **p 17, l 542** : sufficiently come close → come sufficiently close.

Reply: We will revise the text: on page 17 line 542, "...the optimal solutions  come **sufficiently** close to the $f_{\text{true}}$...".

- **p 17, l 542, 543, 545** : the $f_{true}$ → $f_{\text{true}}$ .

Reply: We will revise the word: on page 17 line 542, "...close to  $f_{\text{true}}$ with increasing..."; on page 17 line 543, "...closest flight time to  $f_{\text{true}}$ was..."; and on page 17 line 545, "...between the $f_{\text{best}}$ and  $f_{\text{true}}$ was...". In the same way, we will correct the word "the $f_{\text{true}}$" in the revised manuscript: on page 16 line 530, "... $f_{\text{true}}$ equals the flight time..."; on page 17 line 565, "0.01 % of  $f_{\text{true}}$"; on page 17 line 566, "0.001 % of  $f_{\text{true}}$"; and on page 35 in the caption of Fig. 8, "as close to  $f_{\text{true}}$...".

- **p 17, l 545** : $\Delta f$ : you do not need an extra variable name for something you express only once.

Reply: We understand the referee comment. Nevertheless, we are hesitating to remove the variable name. We

use the variable "$\Delta f$" consistently in the manuscript to express the difference in flight time: on page 17 line 564−565; on page 18 line 575, 581, 588−590; on page 39 in the caption of Fig. 12; on page 8 (Supplementary material) in the caption of Table S1, etc. We think that this variable name is reasonable.

- **p 17, l 547** : What is meant by "diversity" of GA optimization?

  Reply: This word "diversity" is one of the performance indices of an optimization algorithm and is used to show whether the algorithm explores solutions widely or not. It is important to confirm the diversity of the algorithm. On page 17 line 549, we confirmed it for our optimization results as, "It is clear that GA explored diverse solutions from MUC to JFK...".

- **p 17, l 547–548** : we focus on the optimization results, which found the best solution → we focus on the optimization setup which gave the best solution.

  Reply: We believe that the word "optimization results" is appropriate here. We performed the optimizations for each combination of $n_p$ (10, 20,···, 100) and $n_g$ (10, 20,···, 100). Here, we say that we focus on the optimization case of $n_p = 100$ and $n_g = 100$; this case includes the best solution $f_{best}$. In fact, Fig. 9 shows the results obtained from this optimization case, which includes all solutions (10,000 trajectories, black lines) and the best solution (red line) explored by GA. Nevertheless, we modify the sentence by following the referee comment: on page 17 line 547−548, "To confirm the diversity of GA optimization, we focus on the optimization  **yielding** the best solution...".

- **p 17, l 548** : "all the solutions" : Are these the $100 \times 100 = 10000$?

  Reply: Yes. Figure 9 shows the 10,000 trajectories explored by GA. Related to this, we will correct the text "1,000" into "10,000" in the revised manuscript: in the captions of Figs. 9 (p 36), 14 (p 41), S1 (Supplementary material, p 1) and S2 (Supplementary material, p 2), " **10,000** explored trajectories (solid line, black)...".

- **p 17, l 548–549** : solutions explored by GA as longitude vs altitude (top) and as location. This should be worded correctly.

  Reply: We will modify the sentence in the revised manuscript: on page 17 line 548−549, "Figure 9 shows all the solutions explored by GA ." We reply to this issue in the section of "Language."

- **p 17, l 552** : "To confirm the difference" : I don't think confirm is appropriate to be used here.

  Reply: We will revise the text: on page 17 line 552, "To  **investigate** the difference between the solutions,...".

- **p 17, l 554–555** : Isn't this conclusion too fast? What if the trajectory is not so zonal, but the trajectory crosses the equator at an angle of 45°: how would the CPs and regions around be defined?

  Reply: We will reply to this issue in the section of "CP in trajectories (2)." We will add the text into the sentence to confine this conclusion with more precision: on page 17 line 554−555, "Therefore, GA is adequate for finding an optimal solution with sufficient accuracy **(in a strict sense, this conclusion is confined to the benchmark test)**."

- **p 17 , l 552** : "confirm" is not appropriate here.

  Reply: (The "p 17, line 552" means probably "p 17, line 557") We will change the word "confirm" into "analyze": on page 17 line 557, "To  **analyze** the dependence of...".

- **p 17, l 552** : To confirm the dependence of optimal solutions on initial populations → To "analyze" the dependence of "the" optimal solution on "the" initial population, ...

  Reply: (The "p 17, line 552" means probably "p 17, line 557") We will revise the text: on page 17 line 557, "To  **analyze** the dependence of **the** optimal solution on **the** initial *populations*,...".

- **p 17, l 552–553** : I don't think one should use words like "best-of-generation".

  Reply: (The "p 17, line 552−553" means probably "p 17, line 557−558") We will remove the word "best-of-generation" in the sentence: on page 17 line 557−558, "...Fig. 11 shows the  flight time vs the number of objective function evaluations...".

- **p 17, l 558–559** : corresponding to → for.

  Reply: We will modify the text: on page 17 line 558−559, "...function evaluations ($= n_p \times n_g$ )  **for** the 10 independent GA simulations...".

- **p 17, l 653** : "there is a small degree of variation in the objective function". Stated like this, it gives the impression that a different objective function is used. Probably, what is meant is that the value of the objective function for the final flight is different.

  Reply: (The "p 17, line 653" means probably "p 17, line 563") By following the referee comment, we will revise the text: on page 17 line 563, "As indicated in Table S1,  **the value of the objective function $f$ (= flight time) is slightly different.**"

- **p 17, l 564** : Writing $f - f_{\text{true}}$ is a bit strange. For me, $f$ and $f_{\text{true}}$ are solutions, i.e. flights defined by $x_1,...x_{11}$. Here, $f$ and $f_{\text{true}}$ seem to indicate the value of the flight time.

  Reply: $f$ (and also $f_{\text{true}}$) means the objective function value for a solution (i.e. a flight trajectory), which is defined by the eleven design variables $x_1, x_2, \cdots, x_{11}$. As Eq. (28) defines, $f$ (also $f_{\text{true}}$) actually indicates the value of the flight time here.

- **p 17, l 569 and 570** : "number of $n_p$ and $n_g$ " and "size of $n_p$ and $n_g$ ". One should use : "the value of $n_p$ ", or "the size of the population", not something hybrid like "the number of $n_p$ ".

  Reply: We will modify the expression: on page 17 line 569, "With  **increased**  $n_p$ and $n_g$, GA  **tends to find** an improved solution."

- **p 17, l 569** : "discover" : I suggest to use a different word.

  Reply: We will change the word "discover" into "find" in the revised manuscript. In addition, we will modify the word "can" into "tends to" to show exactly the meaning of the sentence: on page 17 line 569, "With  **increased**  $n_p$ and $n_g$, GA  **tends to**  **find** an improved solution." As shown in Fig. 11, the optimal solution finally converges with increasing $n_p$ and $n_g$. The word "can" seems to mean that the solution is improved unlimitedly. Therefore, we think that the word "tend to" is appropriate.

- **p 17, l 570** : "is problem dependent, e.g. weather situations" : this should be formulated properly.

  Reply: This sentence on line 570−571 seems to be confusing. We will modify the sentence: on page 17 line 570−572, "...the required size of $n_p$ and $n_g$ is problem-dependent. However, following a simple initial guess for $n_p$ and $n_g$

is a good starting point for their sizing."

- **p 17, l 571** : "estimating appropriate $n_p$ and $n_g$ could be different" : I suggest to formulate this differently.

  Reply: We will reply to the comment in the above section: "p 17, l 570".

**Page 18**
- **p 18, l 573–574** : unclear sentence. What is, e.g., the difference between accuracy of GA optimizations and variation in the optimal solutions? I also had the impression that the impact of the initial population was already studied in Sect. 3.2.5.

  Reply: The word "accuracy of GA optimizations" shows how close a solution converges to the true-optimal solution. On the other hand, a variation in optimal solutions is caused by different initial populations. Because GA is a stochastic optimization algorithm (not a deterministic optimization method, such as the gradient-based method). In addition, the impact of the initial population was studied in Sect. 3.2.5 regarding the results with "$n_p = 100$ and $n_g = 100$." The impact also depends on $n_p$ and $n_g$ and is investigated in Sect. 3.2.6 in detail. Those results are necessary for the population and generation sizing.

- **p 18, l 574** : Skip "calculated".

  Reply: We will remove the word "calculated" in the sentence: on page 18 line 574, "Figure 12 shows the  $\Delta f$ and...".

- **p 18, l 581** : the variation of the $\Delta f$ and the $s_{\Delta f}$ → Skip "the".

  Reply: We will remove the word "the" in the sentence: on page 18 line 581, "Figure 13 shows the variation of  $\Delta f$ and  $s_{\Delta f}$ for all...".

- **p 18, l 582** : the $\Delta f$ → Skip "the".

  Reply: We will remove the word "the" in the sentence: on page 18 line 582, "The symbols and error bars in the figure correspond to  $\Delta f$ and $s_{\Delta f}$,...".

- **p 18, l 589** : that reduction → a reduction.

  Reply: We will correct the text: on page 18 line 589, "Similarly,  **a** reduction of 97 % can be achieved...".

- **p 18, l 591** : "by selecting $n_p$ and $n_g$ for different purposes." This should be formulated differently.

  Reply: Values of $\Delta f$ and $s_{\Delta f}$ are the basis for selecting $n_p$ and $n_g$. As described on page 18 line 586, the enlarged drawing in Fig. 13 shows that if one selects the number of function evaluations (= $n_p \times n_g$) of 800, the large reduction of computational costs of 92 % can be achieved, keeping $\Delta f$ less than 0.05 % ($s_{\Delta f} \approx 0.02$ %), compared to the optimal solution by 10,000 function evaluations. For $n_p \times n_g = 800$, one can select any combination of $n_p$ and $n_g$: for example, $n_p = 10$ and $n_g = 80$; $n_p = 20$ and $n_g = 40$ etc. A user makes his/her own choice on $n_p$ and $n_g$ by referring the values of $\Delta f$ and $s_{\Delta f}$, as shown in Fig. 13. The formulae of $\Delta f$ and $s_{\Delta f}$ are described clearly in the caption of Fig. 13.

  We will add this explanation to the revised manuscript: on page 18 line 586−589, " **The enlarged drawing in Fig. 13 shows that if one selects the number of function evaluations (= $n_p \times n_g$) of 800,** the large reduction  **of computational costs of 92 %** can be achieved, keeping $\Delta f$ less than 0.05 % ($s_{\Delta f} \approx 0.02$ %), compared to the optimal solution obtained by 10,000 function evaluations ($n_p = 100$ and $n_g = 100$). **For $n_p \times n_g = 800$, one can select any combination of $n_p$ and $n_g$: $n_p = 10$ and $n_g = 80$; $n_p = 20$ and $n_g = 40$ etc. A user makes his/her own choice on $n_p$ and $n_g$ by referring the values of $\Delta f$ and $s_{\Delta f}$ shown in Fig. 13.**"

- **p 18, l 595** : for demonstrations → for demonstration.

  Reply: We will correct the text: on page 18 line 595, "...one-day AirTraf simulations were performed in EMAC (on-line) with the respective routing options for demonstrations."

- **p 18, l 596, 598** : Calculation conditions : too vague.

  Reply: We will change the word "Calculation conditions" into "Simulation setup" in the revised manuscript: on page 18 line 596, "4.1  **Simulation setup**". On page 18 line 598, "Table 7 lists the  **setup** for the one-day simulations." On page 56 in the caption of Table 7, "Table 7.  **Setup** for AirTraf one-day simulations."

- **p 18, l 598–599** : simulation"s" and simulation.

  Reply: We will correct the text: on page 18 line 598−599, "Table 7 lists the calculation conditions for the one-day simulations. The simulation **were** performed for...".

- **p 18, l 605** : "On the other hand" → "In addition".

  Reply: We will change the word "On the other hand" into "In addition": on page 18 line 605, " **In addition**, a single one-day simulation was...".

- **p 18, l 606–p19, l 607** : in [FL290, FL410] → in the range of ..

  Reply: (The "p 19, line 607" means probably "p 18, line 607") We will add the text "the range of" in the revised manuscript: on page 18 line 606, "...altitude changes in **the range of** [FL290, FL410]."

- **p 18, l 607** : "and therefore" : I think $V_{ground}$ also varies for other reasons, e.g., due to varying wind speed and direction.

  Reply: We just wanted to say here that the values of $V_{TAS}$ and $V_{ground}$ are different at every waypoint. We will modify the sentence: on page 18 line 607, "For the two options, the Mach number was set to $M = 0.82$ and therefore  **the values of $V_{TAS}$ and $V_{ground}$ were different at every waypoint (Eqs. (24) and (25)).**"

**Page 19**
- **p 19, l 615** : Does "case" refers to just one flight, or to all 103 flights together?

  Reply: The "case" means the one-day simulation including all 103 flights. We will revise the sentence: on page 19 line 614−615, "The one-day simulation required approximately 15 min for  **the** great circle  routing option, while it took approximately 20 hours for  **the**  **flight time routing option**."

- **p 19, l 616** : It is initially unclear what "it" refers to.

  Reply: The word "it" means "the computational time." We will change the word "it" into "this time" in the sentence: on page 19 line 616, "...the computational time is consumed by the trajectory optimizations. Therefore  **this time** can be reduced by...".

- **p 19, l 617** : "right" : This is maybe not the most appropriate wording.

  Reply: We will change the word "right" into "appropriately": on page 19 line 617, "...choosing all GA parameters  **appropriately**, using more PEs,...".

- **p 19, l 618** : by a small → by "using" a small.

  Reply: We will add the "using" in the text: on page 19 line 618, "...a large reduction in computing time of roughly 90 % can be achieved by **using** a small  $n_p$...".

- **p 19, l 618** : "a small number of $n_p$" → "a small value of $n_p$", or "a small population size"

  Reply: We will modify the text: on page 19 line 618, "...a large reduction in computing time of roughly 90 % can be achieved by **using** a small  $n_p$...".

- **p 19, l 619** : with sufficient accuracy → with "still" sufficient accuracy.

  Reply: We will add the word "still" in the text: on page 19 line 619, "...and $n_g$ with **still** suffcient accuracy of the optimizations."

- **p 19, l 620** : I think the title of Sect. 4.2 does not describe well the content : only one airport pair is discussed (Amsterdam - Minneapolis) really in depth. I suggest something more general.

  Reply: In Sect. 4.2, we have focused on the results of three airport pairs and discussed the one. The rest is in the Supplementary material. To make the title more general, we will delete the word "three" in the title: on page 19 line 620, "4.2 Optimal solutions for  selected airport pairs."

- **p 19, l 623** : trajectories : Is meant the final trajectories?

  Reply: Yes. The "trajectories" mean the optimized flight trajectories (final solutions). We will modify the sentence: on page 19 line 623, "...we classified  **those optimized flight** trajectories according to their altitude changes into three categories."

- **p 19, l 627** : we have selected "the" three airport pairs → we have selected three airport pairs.

  Reply: We will remove the word "the" in the sentence: on page 19 line 627, "We have selected  three airport pairs of..."

- **p 19, l 633** : in [FL290,FL410] → in the range of [FL290,FL410].

  Reply: We will add the text "the range of" in the revised manuscript: on page 19 line 633, "...altitude changes in **the range of** [FL290, FL410]."

- **p 19, l 633–634** : "when calculating for the selected solutions" : This should be formulated better.

  Reply: This text seems to be confusing. We will revise the text: on page 19 line 633−634, "Similar results were obtained  for the selected solutions of Type I and III,...".

- **p 19, l 634** : in the supplements → in the supplementary material.

  Reply: We will change the text "in the supplements" into "in the supplementary material" in the revised manuscript: on page 19 line 634, "..., as shown in Figs. S1 and S2 in the  **Supplementary material**." In the same way, we will modify the text: on page 17 line 562, "Table S1 in the  **Supplementary material** shows a summary of...". On page 18 line 583, "...Table S2 in the  **Supplementary material**...". On page 20 line 657, "see  **Supplementary materials**". On page 22 line 719, "are shown in the  **Supplementary material**."

- **p 19, l 638–639** : east and west direction → eastern and western directions.

  Reply: We will revise the text: on page 638−639, "To calculate tail/head winds in  **eastern** and  **western** directions,...".

- **p 19, l 639** : major wind component : What is meant by this?

  Reply: We just wanted to express the wind component, which has a dominant influence on the flight trajectory, to show the relation clearly between the wind fields and optimal flight trajectories. In fact, the contours in Fig. 15 show the zonal wind speed $u$; they do not include $v$ and $w$.

- **p 19, l 640–641** : at the $h$ → at $h$.

  Reply: We will modify the text: on page 19 line 640−641, "...direction at the departure time at  $h$."

**Page 20**

- **p 20, l 646** : Supplements → Supplementary material.

  Reply: We will modify the text: on page 20 line 646, "...take advantages of the wind fields (see  **Supplementary materials**, Figs. S3 and S4)."

- **p 20, l 647** : the behaviour of altitude changes → the behaviour of the altitude changes.

  Reply: We will revise the text: on page 20 line 647, "To understand the behavior of **the** altitude changes of the optimal flight...".

- **p 20, l 647** : Fig. 16 plots → Fig. 16 shows.

  Reply: We will revise the text: on page 20 line 647, "Fig. 16  **shows** the altitude distribution of the true air speed...".

- **p 20, l 650–651** : this means tail winds ($\geq$ 1.0) and head winds ($<$ 1.0) to the flight direction : Formulate better.

  Reply: We will add the text to the sentence: on page 20 line 650−651, "...; this means tail winds $((V_{ground}/V_{TAS}) \geq 1.0)$ and head winds $((V_{ground}/V_{TAS}) < 1.0)$ to the flight direction."

- **p 20, l 655, 662** : "reflects" → "takes into account", or "accounts for".

  Reply: We will revise the word: on page 20 line 655, "...GA correctly  **takes into account** the weather conditions and...". On page 20 line 662, "...GA correctly  **takes into account** weather conditions for the...".

- **p 20, l 658** : confirmed → compared. Skip "quantitatively".

  Reply: We will revise the text: on page 20 line 658, "Next, we  **compared** the resulting flight **times**  for the selected solutions."

- **p 20, l 659** : as indicated → as shown.

  Reply: We will revise the text: on page 20 line 659, "As  **shown** in Table 8...".

- **p 20, l 659–662** : decreased → is lower.

  Reply: We will revise the sentences: on page 20 line 659−662, "As  **shown** in Table 8, the flight time  **is lower** for the time-optimal case compared to the great circle cases. In addition, the flight time  **is lower** for the eastbound time-optimal flight trajectories compared to that for the westbound time-optimal flight trajectories."

- **p 20, l 664** : "sufficiently" : I think this is a bit vague.

  Reply: (The "p 20, line 664" means probably "p 20, line 666") We will delete the word: on page 20 line 666, "...the solutions  converged to each optimal solution."

- **p 20, l 667** : that the reduction → a reduction.

  Reply: We will revise the text: on page 20 line 667, "It is also clear from Fig. 17 that  **a** reduction in...".

- **p 20, l 668** : "sizing" → "reducing" or "choosing properly".

  Reply: We will revise the text: on page 20 line 668, "...the reduction in computing time can be achieved by  **choosing properly** $n_p$ and $n_g$...".

- **p 20, l 671** : This is not a nice first sentence for a paragraph.

  Reply: We will modify the sentence: on page 20 line 671, "**Next, the** one-day  simulation**s results** for 103 trans-Atlantic flights are  **analyzed**."

- **p 20, l 673–674** : trans-Atlantic Ocean → Atlantic ocean.

  Reply: We will remove the word "trans-" in the text: on page 20 line 673−674, "...flight trajectories congregated around 50° N over the Atlantic Ocean to take advantage...".

- **p 20, l 675** : of "the" region → of "that" region.

  Reply: We will revise the text: on page 20 line 675, "...the westbound time-optimal flight trajectories were located to the north and south of  **that** region...".

**Page 21**

- **p 21, l 681** : plot → show.

  Reply: We will revise the text: on page 21 line 681, "Figures 19a and 19b  **show** the...".

- **p 21, l 683** : with linear fitted lines : be more precise.

  Reply: We will modify the text: on page 21 line 683, "...with linear  lines **fitted by the Least Squares algorithm**." Related to this issue, we will also modify the text: on page 18 line 586, "...**Least Squares** algorithm...".

- **p 21, l 683** : increased → is higher.

  Reply: We will revise the text: on page 21 line 683, "Figure 19a shows that $V_{TAS}$  **is higher** at low altitudes."

- **p 21, l 688–689** : which had high $V_{TAS}$ values → with high $V_{TAS}$ values.

  Reply: We will revise the text: on page 21 line 688−689, "GA successfully found the flight trajectories **with** high $V_{TAS}$ values as time-optimal flights."

- **p 21, l 691** : increases → is larger.

  Reply: We will revise the text: on page 21 line 691, "...time-optimal case (solid line, red)  **is larger** between...".

- **p 21, l 696** : increases → is larger.

  Reply: We will revise the text: on page 21 line 696, "...time-optimal case (solid line, blue) is distributed widely in altitude and  **is larger** between".

- **p 21, l 700** : Supplement → Supplementary material.

  Reply: We will modify the text: on page 21 line 700, "...is shown in the  **Supplementary material** (Fig. S7)...".

- **p 21, l 703** : correctly selected the airspace : improve this formulation.

  Reply: We will modify the sentence: on page 21 line 703, "Therefore,  **the trajectories found by GA through altitude changes passed areas, which correctly lead to larger $V_{\text{ground}}$.**"

- **p 21, l 705** : This behaviour of altitude changes → These altitude changes.

  Reply: We will revise the text: on page 21 line 705, " **These** altitude changes affect the...".

- **p 21, l 705** : affects the variation in fuel consumptions → affects the fuel consumption.

  Reply: We will revise the text: on page 21 line 705, "**These** altitude changes affect the  fuel consumption...".

- **p 21, l 705** : the terms are used interchangeably to mean fuel flows : improve the formulation.

  Reply: We will improve the text: on page 21 line 705, "...affect the  fuel consumption (the term  **is** used interchangeably to  fuel flow)."

- **p 21, l 708** : increases → is higher.

  Reply: We will revise the text: on page 21 line 708, "The results show that the fuel consumption  **is higher** at low altitudes...".

- **p 21, l 708** : the mean value → the mean value of the fuel consumption.

  Reply: We will revise the text: on page 21 line 709, "In addition, the mean value **of the fuel consumption** for the time-optimal case is high...".

- **p 21, l 714** : increases → is higher.

Reply: We will revise the text: on page 21 line 714, "...the mean value for the eastbound time-optimal case  **is higher** owing to its low mean flight altitude...".

**Page 22**

- **p 22, l 718** : corresponding to "the 103" individual flights.

  Reply: We will revise the text: on page 22 line 718, "Figure 21 shows the flight time corresponding to **the 103** individual flights...".

- **p 22, l 718–719** : the similar figures → similar figures.

  Reply: We will revise the text: on page 22 line 718−719, "...( similar figures for the fuel use, NO$_x$ and H$_2$O emissions are shown...".

- **p 22, l 720** : showed → show.

  Reply: We will revise the text: on page 22 line 720, "The results show that all symbols...".

- **p 22, l 720** : in the right-hand domain : choose a better expression.

  Reply: We will rephrase the text: on page 22 line 720, "...all symbols lay  **on the right side of the 1:1 solid line**."

- **p 22, l 721** : decreases → is lower. Put "for all airport pairs" at the end of the sentence.

  Reply: We will revise the sentence: on page 22 line 721, "...the flight time for the time-optimal flights  **is lower**  compared to that for the great circle flights **for all airport pairs**."

- **p 22, l 723 and 725** : increased → increases.

  Reply: We will revise the sentences: on page 22 line 722−725, "The total value  **is** certainly minimal for the time-optimal case, while in relative terms the value  **increases** by +1.5 %, +2.5 %, +2.9 % and +2.9 % for the great circle cases at FL290, FL330, FL370 and FL410, respectively. Regarding the total value of fuel use, Table 11 indicates that the value  **increases** by +5.4%."

- **p 22, l 740–741** : "Consistency" : just by reading the section title, it is not clear what is meant by this.

  Reply: We will change the section title: on page 22 line 740, "5  **Verification of** the AirTraf simulations."

- **p 22, l 742** : were → are.

  Reply: We will revise the text: on page 22 line 742, "...the one-day simulation results described in Sect. 4  **are** compared to reference data...".

- **p 22, l 742–743** : The data → Data.

  Reply: We will revise the text: on page 22 line 742−743, "**D**ata obtained under similar conditions...".

- **p 22, l 744** : "they" is ambiguous.

  Reply: We will revise the text: on page 22 line 742−744, "**D**ata obtained under similar conditions (aircraft/engine types, flight conditions, weather situations, etc.) were selected for the comparison, although

**the conditions** are not completely the same as the calculation conditions for the one-day simulations."

**Page 23**

- **p 23, l 723** : I would not say explicitly that the table shows "a comparison".

  Reply: (The "p 23, line 723" means probably "p 23, line 749") We will revise the text: on page 23 line 749, "...Table 12 shows  the flight time  **for** the seven time-optimal flight trajectories simulated by AirTraf and three reference data...".

- **p 23, l 758 and 764** : literature → write the correct reference.

  Reply: We will revise the text: on page 23 line 758, "...(see Fig. 3 in  **Irvine et al. (2013)**)." On page 23 line 764, "...(see Tables 2 and 3 in  **Grewe et al. (2014a)**)."

- **p 23, l 758** : indicated → indicates.

  Reply: This part will be deleted. Please see the reply to the comment below: "p 23, l 758–759 / 764–765".

- **p 23, l 758–759 / 764–765** : Is it worth mentioning this?

  Reply: As the referee pointed out, those sentences are not necessary here. Therefore, we will revise the sentences: on page 23 line 758–759, "". On page 23 line 764–765, "". Related to this issue, we will modify the text: on page 23 line 765–767, "The  flight times  **between** the seven airport pairs  **are** close **to the reference data** and the variation shows **a** good agreement with the trend of the flight **times** for westbound **trans-Atlantic** flights in winter **due to westerly jets streams**, as indicated from the reference data."

- **p 23, l 764** : "indicate" : I don't think this is the appropriate word.

  Reply: This part will be deleted. Please see the reply to the comment above: "p 23, l 758–759 / 764–765".

- **p 23, l 765** : "close" → "close to".

  Reply: We will revise the text: on page 23 line 765, "The  flight times  **between** the seven airport pairs  **are** close **to the reference data** and the variation shows...".

- **p 23. l 769** : reference data → the reference data.

  Reply: We will revise the text: on page 23 line 769, "...using the mean fuel consumption value of 103 flights and **the** reference data,...".

- **p 23, l 774** : indication : shouldn't one use a different word?

  Reply: We will remove the word "indication" and revise the text: on page 23 line 774, "...the **overall** load factor of the worldwide air traffic  was used (Table 1)."

- **p 23, l 778** : decreased → is lower.

  Reply: We will revise the text: on page 23 line 778, "Table 13 shows that the obtained mean $EINO_x$ value  **is lower** at high altitudes...".

- **p 23, l 783** : installed → contains.

  Reply: We will revise the text: on page 23 line 783, "The 2GE051  **utilizes** the new 1862M39 combustor,...".

**Page 24**
- **p 24, l 787** : "duplicates" : What is meant by this?

  Reply: We just wanted to say here as, "estimates" or "simulates." We will revise the text: on page 24 line 787, "AirTraf  **simulates realistic** fuel consumptions...".

- **p 24, l 790** : for 103 flights → for "the" 103 flights.

  Reply: We will revise the text: on page 24 line 790, "Here the obtained $m_1$ and $m_{nwp}$ for **the** 103 flights were compared...".

- **p 24, l 792** : safety flight operations → flight operations safety.

  Reply: We will revise the text: on page 24 line 792, "...to provide  **flight operations safety**, and...".

- **p 24, l 794** : constrains to → constraints

  Reply: We will revise the text: on page 24 line 794, "...no model that **constrains**  the structural  weight **limits** was included in AirTraf."

- **p 24, l 800–801** : This sentence should be improved.

  Reply: We will improve the sentence: on page 24 line 800–801, "For these 15 flights, actual flight planning data  indicate  higher flight altitudes to increase  **the** fuel mileage,  **leading to the decrease in** $m_1$."

- **p 24, l 802** : to prevent "the" structural damage → to prevent structural damage.

  Reply: We will revise the text: on page 24 line 802, "To prevent  structural damage to the landing gear...".

- **p 24, l 803** : "aircraft has" → "aircraft have" or "an aircraft has".

  Reply: We will revise the text: on page 24 line 803, "...**an** aircraft has to reduce the total weight...".

- **p 24, l 803** : "to reduce below" → "to reduce until" or "to bring below".

  Reply: We will revise the text: on page 24 line 803, "...**an** aircraft has to reduce the total weight  **until** MLW prior to landing."

- **p 24, l 808** : Why not using ≤?

  Reply: We will revise the text: on page 24 line 808, "This always satisfies the third constraint ZFW ≤ MZFW."

- **p 24, l 806, 810** : of A330-301 → of an A330-301 aircraft.

  Reply: We will revise the word in the revised manuscript: on page 24 line 806, "The MZFW of **an** A330-301

**aircraft** is...". On page 24 line 810, "...minimum operational weight of **an** A330-301 **aircraft** in the...".

- **p 24, l 812** : more → higher.

  Reply: We will revise the text: on page 24 line 812, "..., all the $m_{\text{nwp}}$ (open circle) were  **higher** than the MLOW."

- **p 24, l 814** : Skip "calculations".

  Reply: We will remove the word "calculations": on page 24 line 814, "...AirTraf simulates fairly good fuel use ."

- **p 24, l 816** : an submodel → a submodel.

  Reply: We will revise the text: on page 24 line 816, "AirTraf is published for the first time as  **a** submodel of the Modular Earth Submodel System...".

- **p 24, l 817** : "applied" : shouldn't it be "used"?

  Reply: We will revise the text: on page 24 line 817, "The MESSy is continuously further developed and  **used** by a consortium of institutions."

**Page 25**

- **p 25, l 823–824** : What is meant by this sentence?

  Reply: This sentence is not necessary for our argument here. Therefore, we will delete the sentence: on page 25 line 822−825, "Some improvements will be performed and AirTraf 1.0 will be updated for the latest version of the code.  The status information for AirTraf including the licence conditions will be available at the website."

- **p 25, l 829** : the benchmark test → a benchmark test.

  Reply: We will revise the text: on page 25 line 829, "First,  **a** benchmark test was performed...".

- **p 25, l 831–832** : by other published code : this is too vague.

  Reply: We will revise the text: on page 25 line 831–832, "...calculated by  **MTS**."

- **p 25, l 832** : the benchmark test → a benchmark test.

  Reply: We will revise the text: on page 25 line 832, "Second,  **a** benchmark test was performed...".

- **p 25, l 836** : dependence on the initial population.

  Reply: We will revise the text: on page 25 line 836, "The dependence of the optimal solution on **the** initial *population* was investigated...".

- **p 25, l 835 and 838** : The fact that both values are 0.01 % is maybe not a good sign. I would think that you want the second one to be much smaller than the first one.

  Reply: The referee pointed out a very important issue. However, these values are sufficiently small and the performance of GA is well enough to find an optimal solution. In fact, we showed in Fig. 21 that GA found

the trajectories for all airport pairs; the trajectories could decrease flight time compared to the great circle flights. This performance is sufficient for our purpose. In fact, the second "0.01 %" is actually smaller than what we expected. As replied to the referee comment in the section of "p 18, l 573–574", GA is a stochastic optimization algorithm. Hence, optimal solutions calculated from different initial populations are not always the same.

Regarding the performance of GA, Deb, K., (1991) reported that "the welded beam structure is a practical design problem (minimization of the total cost $f$) that is often used as a bench-mark problem in testing different optimization techniques." Rekliatis, G. V., et al., (1983) studied this test and reported the optimal solution of $f^* = 2.38$. Deb, K., (1991) performed 3 independent GA calculations with different initial populations to this problem: the obtained (optimal) solution was $f = 2.43$ (the best among the three solutions), $f = 2.59$ and $f = 2.49$. The difference in the total cost between the $f$ (the best solution: 2.43) and $f^*$ was $\Delta f = f - f^* = 0.05$ (2.1 % of $f^*$). $\Delta f$ also ranged from 0.05 to 0.21 (2.1 to 8.8 % of $f^*$). This shows that both values "0.01 %" are indeed small. Of course, the performance of GA largely depends on the optimization problem and GA parameters. Therefore, we analyzed the performance on our trajectory optimization problem with our setting in Sects. 3.2.4 and 3.2.5.

[Reference]
Rekliatis, G. V., et al., Engineering Optimization Methods and Applications, Wiley, New York, 1983.
Deb, K., "Optimal design of a welded beam via genetic algorithms," AIAA Journal, 29 (11), 1991.

**Page 26**

- **p 26, l 860 and 866** : Please be more specific about what "reference data" is.

  Reply: We will revise the text: on page 26 line 860, "The consistency of the one-day simulations was verified with reference data **(published in earlier studies and BADA)** of flight time...". On page 26 line 866, "The mean $EINO_x$ values were in the same range as the reference  **values of earlier studies.**"

- **p 26, l 862** : close → (very) similar.

  Reply: We will revise the text: on page 26 line 862, "...the reference data showed that the values were  **similar** and...".

- **p 26, l 869** : fuel use calculation model → fuel use model.

  Reply: We will revise the text: on page 26 line 869, "Thus, AirTraf comprises a sufficiently good fuel use  model."

- **p 26, l 871** : "is sufficient" : But some things do not work yet?

  Reply: We will revise the sentence: on page 26 line 871, "AirTraf 1.0  **is ready for more complex routing tasks**."

- **p 26, l 871** : "a" reduction potential → "the" reduction potential.

  Reply: This part will be deleted. Please see the reply to the comment above: "p 26, l 871".

**4 Remarks on figures**

- **Figure 1** : I presume parts of this are already done in other optimized studies. Mention what is already done, what is part of this manuscript, and what shall be done in the future.

  Reply: By following the comment (2) of the referee #1, we will remove Fig. 1 (on page 29).

- **Figure 7** : Bizarre first sentence in caption. Consisting of → determined by. $\Delta\lambda_{airport}$ → $\Delta\lambda_{\text{airport}}$.

Reply: We will revise the caption: on page 34 in the caption of Fig. 7, "Geometry definition of flight traject-ory  **in the vertical cross-section** (top) and  **projected onto the Earth** (bottom). The bold solid line indicates a trajectory from MUC to JFK. •: control points  **determined by** design variables....which divide the  $\Delta\lambda_{\mathbf{airport}}$ into four equal parts...the coordinates divide the  $\Delta\lambda_{\mathbf{airport}}$ into six equal parts."

- **Figure 8** : Conclusions/observations/interpretations should not be written in figure captions. I would not use the word "discovers".

Reply: We will revise the caption: on page 35 in the caption of Fig. 8, "Figure 8. Optimal solutions  varying with the *population* size $n_p$ and **the number of *generations***  $n_g$. $\Delta f$ means the difference in flight time between the optimal solution $f$ and the true-optimal solution $f_{\text{true}}$ **(= 25,994.0 s)**. The $\Delta f$ (in %) is calculated as $(\Delta f / f_{\text{true}}) \times 100$.  The flight time of the best solution is $f_{\text{best}} = 25,996.6$ s (for $n_p = 100$ and $n_g = 100$, $\Delta f < 3.0$ s (less than 0.01 %))."

- **Figure 9** : Change the first sentence. "The population size $n_p = 100$ ..." : This is not a good sentence. Replace "=" by "is".

Reply: We will revise the caption: on page 36 in the caption of Fig. 9, "**10,000** explored trajectories (solid line, black) from MUC to JFK  **in the vertical cross-section** (top) and  **projected onto the Earth** (bottom). The *population* size  $n_p$ **is** 100 and **the number of *generations***  $n_g$ **is** 100." In the same way, we will revise the caption: on page 38 in the caption of Fig. 11, "The *population* size  $n_p$ **is** 100 and **the number of *generations***  $n_g$ **is** 100." On page 44 in the caption of Fig. 17, "**The *population*** size  $n_p$ **is** 100 and **the number of *generations***  $n_g$ **is** 100."

- **Figure 10** : Skip "Comparison of".

Reply: We will remove the word "Comparison of" in the caption: on page 37 in the caption of Fig. 10, "Figure 10. **T**rajectories for the best solution (red line) and the true-optimal solution (dashed line, black)."

- **Figure 11** : Don't use expressions like "Best-of-generation". "vs function evaluations" → "vs number of function evaluations". $f_{true}$ → $f_{\text{true}}$. Change "On the ... and ...".

Reply: We will revise the caption: on page 38 in the caption of Fig. 11, "**F**light time vs **number of** function evaluations...and the true-optimal solution $f_{\mathbf{true}}$...is calculated as $(\Delta f /  f_{\mathbf{true}})$...".

- **Figure 17** : Don't use expressions like "Best-of-generation".

Reply: We will revise the caption: on page 44 in the caption of Fig. 17, "**F**light time (in %) vs **number of** function evaluations...".

- **Figure 22** : Shouldn't one have as unit for the emissions : kg(fuel) m$^{-2}$s$^{-1}$? The figures are 2-hourly averages. However, the ranges are not clear from just mentioning 14:00:00, 16:00:00, 18:00:00, 20:00:00. Is it 14:00:00–16:00:00, 16:00:00–18:00:00, ..., or rather 12:00:00–14:00:00, 14:00:00–16:00:00, ...

Reply: By following the comment (8) of the referee #1, we will remove Fig. 22 (on page 49).

**5  Comments on tables**

- **Table 1** 101.325 → 101,325. Why is there "(jet)" at the end of the line with $C_{f1}$? There should be a small space between "kg" and "min". I would not give a variable the name "Oneday". $P_0$ and $T_0$ are not total pressure or temperature, but reference pressure and temperature.

  Reply: Thank you so much. We will correct the value: on page 51 at the line with $P_0$ in Table 1, " **101,325**." Eurocontrol, 2011 publishes the thrust specific fuel consumption coefficient for jet, turboprop and piston engines. The word "jet" means "jet engines". We will modify the line: at the end of the line with $C_{f1}$, "...(jet **engines**)[a]". As the referee pointed out, we will add a space between "kg" and "min": at the line with $C_{f1}$, "**kg min$^{-1}$kN$^{-1}$**." Regarding the variable name "Oneday", please see the reply to the referee comment on "p 9, l 287." In addition, we will correct the word on $P_0$ and $T_0$: at the line with $P_0$ and $T_0$, " **Reference** pressure" and " **Reference** temperature".

- **Table 2** : $n_{wp-1} \rightarrow n_{wp} - 1$.

  Reply: Thank you so much. We will correct the text: on page 52 in the caption of Table 2, "..., flight segments ($i = 1, 2,..., $  $\boldsymbol{n_{wp} - 1}$)."

- **Table 4** : I think it makes no sense to introduce all these new variable names. Put in the heading (first row) of the table just : "Eq. (22)", "Eq. (23)", ...

  Reply: We understand the referee comment. Nevertheless, we are hesitating to change those variable names. We define the variable name for a flight distance as "$d$", as shown in Table 2, and we use the variable "$d$" consistently in the manuscript: on page 11 Eqs. (22) and (23), on page 15 Eq. (28), etc. We think that the current expressions are reasonable.

- **Table 5** : For population size and generation number : "· · ·" → ". . .".

  Reply: We will modify and add the variable names "$n_p$" and "$n_g$" in the Table: on page 55 in Table 5, "Population size, $\boldsymbol{n_p}$, 10,20,**...**,100" and "**Number of generations**, $\boldsymbol{n_g}$, 10,20,**...**,100". This reply is related to the reply to "p 17, l 540." In addition, we will add the text at the line with design variable: on page 55 in Table 5, "Design variable, $n_{dv}$, 11 **(6 locations and 5 altitudes)**." Related to this, we will modify the text: on page 56 in Table 7, "Design variable, $n_{dv}$, 11 **(6 locations and 5 altitudes)**."

- **Table 9** : "that of" → "average of". Why "medium"?

  Reply: We will modify the caption of Table 9: on page 58 in the caption of Table 9, "Eastbound:  **average** of 52 eastbound flights; Westbound:  **average** of 51 westbound flights; and Total:  **average** of 103 flights." In the same way, we will modify the caption of Table 10: on page 58 in the caption of Table 10, "Eastbound:  **average** of 52 eastbound flights; Westbound:  **average** of 51 westbound flights; and Total:  **average** of 103 flights." Unfortunately, we didn't understand the comment: Why "medium".

- **Table 12** : Skip "Comparison of".

  Reply: We will remove the word "Comparison of" in the caption: on page 60 in the caption of Table 12, "**T**he flight time for time-optimal flight trajectories from one-day AirTraf simulations...".

- **Table 14** : "Constraints on" → "Constraints from". Why not just using $\geq$ and $\leq$? Why have on all the four lines A330-301 after some "." at the end of the line?

  Reply: We will revise Table 14: on page 62 in the caption of Table 14, "Constraints  **from** the structural  weight **limits** (MTOW, MLW and MZFW) and one specific  weight **limit** (MLOW)...". In column 1, "$m_1$ $\leq$ MTOW; $m_{nwp}$ $\leq$ MLW; Zero fuel weight $\leq$ MZFW; and $m_{nwp}$ $\geq$ MLOW." In column 3, "Maximum take-off weight; Maximum landing weight; Maximum zero fuel weight.

MZFW = OEW + MPL. ; and Planned minimum operational weight in the international standard atmosphere.[b] MLOW = 1.2 × OEW. .[b]"

Related to this, we will change the word "limit weights" into "weight limits" in the revised manuscript: on page 24 line 791, "three structural  weight**s** **limits**..."; on page 24 line 792, "...,and one specified  weight **limit**..."; on page 24 line 793, "...and the four  weight**s** **limits**..."; on page 24 line 794, "...constrains to the structural  weight**s** **limits**..."; on page 24 line 797, "...with the  weight **limits**..."; on page 26 line 867, "...the three structural  weight**s** **limits** and one specified  weight **limit** of..."; on page 26 line 868, "...the values satisfied the four  weight**s** **limits** and..."; on page 50 in the caption of Fig. 23, "Comparison of aircraft weights with structural  weight**s** **limits** (MTOW and MLW) and one specified  weight **limit** (MLOW)"; on page 62 in column 2 of Table 14, "**W**eight **limit**, kg".

**6  Supplementary material**

- Fig. S1 and S2 : including "the" time-optimal flight trajectories.

  Reply: We will add the word "the" in the caption: on page 1 (Supplementary material) in the caption of Fig. S1, "...(bottom), including **the** time-optimal flight trajectories...". On page 2 (Supplementary material) in the caption of Fig. S2, "...(bottom), including **the** time-optimal flight trajectories...". In the same way, we will add the word: on page 41 in the caption of Fig. 14, "...(bottom), including **the** time-optimal flight trajectories...".

- Fig. S3 and S4 : Skip "Comparison of".

  Reply: We will remove the word "Comparison of" in the caption: on page 3 (Supplementary material) in the caption of Fig. S3, "**T**rajectories for the time-optimal...". On page 3 (Supplementary material) in the caption of Fig. S4, "**T**rajectories for the time-optimal...". In the same way, we will remove the word: on page 42 in the caption of Fig. 15, "**T**rajectories for the time-optimal...".

- Fig. S7 : Skip "that".

  Reply: We will remove the word "that" in the caption: on page 6 (Supplementary material) in the caption of Fig. S7, "Linear fits of the time-optimal (solid line, red (eastbound) and blue (westbound)) and  of the great circle...". In the same way, we will remove the word: on page 46 in the caption of Fig. 19, "Linear fits of the time-optimal (solid line, red (eastbound) and blue (westbound)) and  of the great circle...".

---

## Author Response (AR1)

Dear Dr. Jason Williams,

We are most grateful to you and the reviewers for the helpful comments on the original version of our manuscript. We have taken all the comments into account and submit a revised version of our paper here. Please find attached the comments of the referees and our replies (available also on-line) together with the revised manuscript with highlighted modifications.

Please note:

- Figure 1 and Figure 22 in the original manuscript have been deleted according to the suggestions by referee #1.
- Many equations are highlighted. However, the modifications are just to be modified from "italic letters" to "straight letters" according to the suggestions by referee #3.
- Figure 2, Figure 8, Figures 12a and 12b have been modified according to the suggestions by referees, however they are not highlighted due to some technical issues with "latexdiff." These modifications are all described in the following replies.
- We add a section "Appendix; Glossary" after the section "7. Conclusions", where we explain the several terminologies of the GA optimization. The terms from the glossary are written in italics in the text.
- A lack of information, e.g. a name of journal, volumes, pages, etc. is added in the section "References". However they are not highlighted due to some technical issues with "latexdiff."

Thank you very much again for your guiding the editorial process of our manuscript. We are looking forward to hearing from you.

Yours sincerely, Hiroshi Yamashita (on behalf of all co-authors) We are most grateful to the referee #1 for the very helpful and encouraging comments on the original version of our manuscript. Here are our replies:

Summary: This paper describes a new model which will eventually be used in calculating the climate impact of aircraft routes. There are several different parts to the model, which are detailed in the paper, including generating the route either by calculating a great circle or time-optimal route (the two constraints which are described and tested here), calculating the fuel use along the route, and the emissions for example of water vapour and NOx along the route. A thorough assessment is made of the model and its ability to generate the routes and calculate the various parameters; the model performs well and appears to be fit for purpose. The paper is generally clear and the different components of the model are well-described. My only major concern regards the vertical flight profiles, please see the major comment below. I recommend the paper for publication after revision.

Reply: We thank the referee #1 for these positive comments. We will reply to your major concern regarding the vertical flight profiles in the "General comment" section.

General comment: In the calculation of the time-optimal flights, the flight altitude is allowed to vary freely between FL290 and FL410. Some of the resulting time-optimal flight profiles display significant altitude changes during the flight, as shown in Figure 14 (b), where the flight altitude profile along the flight is 'm' shaped (i.e. increases, decreases, increases and then decreases again). This is in contrast to the familiar stepped profiles, where the aircraft altitude increases are done as step climbs when enough fuel has been burned off, or alternatively a gradual increase in height to a maximum cruising altitude, followed by a descent. It is difficult to see how (or why) an aircraft would do this 'm' profile in real life, given air traffic constraints, for example. Given how unusual these profiles are, some justification or explanation for why these profiles are allowed in this study should be given, as well as a comment on how realistic it would be for an aircraft to fly this profile.

Reply: In this paper, we have confirmed that the 'm' shaped flight profile effectively takes advantages of the wind fields and leads to the time-optimal solution (please see on page 20 line 647 – 657).

As the referee #1 pointed out, AirTraf allows aircraft to vary flight altitudes freely between FL290 and FL410. Here, the AirTraf submodel is used to investigate an optimization strategy of aircraft routing for minimizing the climate impact of aircraft emissions and to show its mitigation gain for the future. The regions with high climate impacts, e.g. regions where contrail form, are often very shallow (vertically). In order to investigate how such regions can be avoided more flexibility in the routing options is required. Hence, in this approach it is necessary for aircraft to have a high flexibility for flight profiles to explore widely the possibility of minimizing climate impact by aircraft routing.

If the optimization strategy is found, it will be proven by a more realistic air traffic simulation model, considering realistic air traffic constraints. The "m" shaped flight profile will be modified to adapt to the constraints (probably stepped profiles). The development of the realistic air traffic simulation model is addressed by research groups of DLR-Hamburg and DLR-Braunschweig in the DLR Project WeCare.

We will add this information in the revised manuscript: on page 14 line 472, "Here  $x_7$  to  $x_{11}$ -indicate altitudevalues. Note that these values vary freely between FL290 and FL410 to explore widely the possibility of minimizing climate impact by aircraft routing." On page 14 line 466, "…were used (Fig. 7, top). Here  $x_7$ to  $x_{11}$  indicate altitude values." This modification is related to our reply to the comment "p 14, 1 461 and 472" of referee #3.

Further, we will add the text: on page 19 line 635, "..., while that for west-bound showed large altitude changes, i.e. it climbed, descended, climbed and **then descended** again."

• Minor comments:

(1) p3 L61 – the Spichtinger et al (2003) study referenced by the authors analyses the vertical distribution of ice-supersaturated regions. The mean length of 150 km is from Gierens and Spichtinger (2000), as stated in the Spichtinger paper.

Reply: Thank you very much. We will refer the paper in the revised manuscript: on page 3 line 61, "...extend a few 100 m vertically and around about 150 km horizontally along a flight path (with a standard deviation of 250 km) with a large spatial and temporal variability (Gierens et al., 2000, Spichtinger et al., 2003)." This modification is related to our reply to the comment "p 3, l 61" of referee #3. We will also add the paper to References in the revised manuscript: on page 27, "Gierens, K., and Spichtinger, P.: On the size distribution of ice-supersaturated regions in the upper troposphere and lowermost stratosphere, Annales Geophysicae, vol. 18, No. 4, 499–504, 2000."

• (2) p3, final paragraph (L84 – 99). As I understand it, the aim of the study presented in the paper is to introduce, describe and validate the AirTraf model, not to investigate 'how much the climate impact ... can be reduced by aircraft routing' – that is a separate study which would use AirTraf. This paragraph is therefore confusing to the reader, and there is extra detail here which is not all necessary to understand this paper. Please rephrase the aims of the study to be consistent with what is presented in the paper, remove unnecessary detail about future studies and I also suggest removing Figure 1 which is not needed here.

Reply: As the referee #1 noted, this paragraph is confusing. On the other hand, we think that the information of this paragraph is helpful for readers to understand the motivation and background for the AirTraf development. To improve the manuscript, we will remove Fig. 1 and rephrase the aims of this study: on page 3, final paragraph (line 84 – 99),

"This paper presents the new submodel AirTraf (version 1.0, Yamashita et al., 2015) that performs global air traffic simulations coupled to the Chemistry-Climate model EMAC (Jöckel et al., 2010). This paper technically describes AirTraf and validates the various components for simple aircraft routings: great circle and time-optimal routings. Eventually, we are aiming at an optimal routing for climate impact reduction. The development described in this paper is a prerequisite for the investigation of climate-optimized routings. The research road map for our study is as follows (Grewe et al., 2014b):-Tthe first step was to investigate...". This modification is related to our reply to the comment "p 3, l 85–86" of referee #3.

• (3) p4 L121, p5 L159 and caption of Figure 3 – "one-day flight plan". It is not clear what you mean by this phrase (it sounds like you are referring to a single flight on a single day, rather than many flights on a single day). It would be helpful to give a short description the first time you use the phrase.

Reply: We will add the text in the revised manuscript: on page 4 line 121, "As shown in Fig. 3, the one-day flight plan, which includes many flight schedules of a single day, is decomposed for a number of processing elements (PEs)."

• (4) p4 L126 – "AirTraf continuously treats overnight flights". What does this mean?

Reply: Some international (long-distance) flights fly over two days. For example, NH215 departs at MUC on 21:35 and arrives at Tokyo on 15:50 + 1day. AirTraf can simulate the flight correctly. We will rewrite the text in the revised manuscript: on page 4 line 125, "Thus, <del>naturally both</del> short-term and long-term simulations <del>consider</del> can take into account the local weather conditions for every flight in EMAC (AirTraf continuously treats overnight flights with arrival on the next day)."

Further, from the referee #3 comment on "p 4, l 126 – 127", the text of the sentence "(AirTraf continuously treats overnight flights **with arrival on the next day**)" will be moved from the current position to an appro priate position in the manuscript, which is related logically: finally, on page 4 line 125, "Thus, <del>naturally both</del> short-term and long-term simulations <del>consider</del> **can take into account** the local weather conditions for every

flight in EMAC (AirTraf continuously treats overnight flights with arrival on the next day)"; and on page 7 line 223, "...the Estimated Time Over (ETO, Table .2) (AirTraf continuously treats overnight flights with arrival on the next day)."

• (5) p7 L201 – "local weather conditions provided by EMAC". Specifically, the wind field is used?

Reply: Specifically, temperature and wind fields are used here to calculate a flight trajectory. On pages 6 - 8 in section 2.4, we describe the overview of calculation procedures briefly. Thus, we describe on page 7 line 201 as, "For all routing options, local weather conditions provided by EMAC at t = 1 (i.e. at the departure day and time of the aircraft) are used to calculate the flight trajectory."

In the following section, this trajectory calculation method is described in detail. For great circle routing option, on page 12 line 375 in section 3.1.1, "Temperature  $T_i$  and three dimensional wind components  $(u_i, v_i, w_i)$  of the  $i^{th}$  waypoint are available from the EMAC model fields at t = 1." For the time-optimal routing option, on page 15 line 487, "... where  $d_i$  and  $V_{ground,i}$  are calculated by Eqs. (23) and (25), respectively ( $V_{TAS,i}$  and  $V_{wind,i}$  are calculated as described in Sect. 3.1.1)."

• (6) p8 L260 – You assume that the sum of the alternate, reserve and extra fuel is 3% of the total fuel. Is there any justification for this number? I acknowledge that this kind of data is almost impossible to get from airlines, but have other studies used a similar number, for instance?

Reply: According to general fuel planning regulations, e.g. JAR-OPS  $1.255^{(1)}$ , an additional 3% of the total fuel is considered as contingency fuel in the fuel planning assuming an en-route alternate aerodrome can be found on any mission whereas alternate, final reserve, additional and extra fuel are neglected as their contribution to the overall fuel amount is very small on long-haul flights. Although the fuel planning process of Air Traf, which is described on page 8 - 9 in section 2.5, is not exactly the same as JAR-OPS 1.255, the 3% as sumption (calculated by Eq. (2) on page 8) as the entire reserve fuel is not far from reality.

Further, we will delete the sentence related to this matter: on page 8 line 265, "**A refined fuel estimation will be employed for calculating m\_{nwp} in future.**" will be deleted in the revised manuscript, since the sentence is not necessary for our argument here.

[1] The Joint Aviation Authorities Committee, "Joint Aviation Requirements: JAR-OPS 1, Commercial Air Transportation (Aeroplanes)", 1-D-4.

(7) p20 L647 – 656. The explanation of why the flight altitude profiles are optimal is that the flight changes altitude to benefit from changes to the true airspeed and to increase its tailwind or reduce its headwind. The argument is currently not well supported by the figures (Figure 16, and S5 and S6) which show the altitude distribution of the true airspeed and tailwind indicator. The variations in these quantities at flight altitude are hard to see, since the vertical scale on the plots is 0 – 15 km. The case might be made much clearer simply by re-plotting these figures with a limited altitude range (i.e. only plot the range of altitudes relevant to the aircraft), and re-scaling the colour bar.

Reply: We think that the referee's suggestion is right. On this matter, we have a reason why we used the vertical scale on the plots as 0 - 15 km. In Figs. 16, S5 and S6, we would like to show clearly that we start with the trajectory at FL290 and concentrate on the cruise mission only. In fact, we optimize flight trajectories within the general cruise flight altitude of commercial aircraft in [FL290, FL410], as shown in Fig. 7 (top), and the altitude of the airports are located at FL290 (not ground at 0 ft). We have seen situations many times that people assumed the start/end point of the time-optimal flight trajectories (in Fig. 16) as "the ground at 0 ft," when we plotted the same results in the range of altitude relevant to the aircraft. To avoid this situation, we plotted these figures in 0 - 15 km including the ground (just like Figs. 9, 14 and 18).

• (8) p22 L727 – 729 and Figure 22. "The maps show the time-optimal case has low values of the fuel use"

(compared to the great circle case). The great circle case at FL290 clearly has a lower fuel use, as shown in Table 11. However, I do not think this is clear from Figure 22; the flights in the time optimal case are spread over a larger area than in the great circle case therefore it is difficult to assess objectively whether the fuel use is higher or lower in the time-optimal case. I do not think that this figure adds any weight to your argument. I suggest removing it.

Reply: As the referee #1 suggested, we will remove Fig. 22 in the revised manuscript. In addition, we will remove the sentences related to Fig. 22: on page 22 line 726 - 729, "To confirm this intuitively, Fig. 22 shows-the global distribution maps of the fuel use (in kg(fuel)box-1s-1, 2 hour averages) for these cases. The maps show that the time-optimal case has low values of the fuel use. On the other hand, Table 11 indicates that the fuel use decreased...".

• (9) p25 L824. I cannot find AirTraf or any status information for it on the list of submodels on the MESSy website (accessed on 24/02/2016).

Reply: On the basis of the MESSy Consortium Steering Group Policy, a status information for a new submodel is generally provided on the MESSy website after its publication. Nevertheless, we have provided the status information for AirTraf on the website. In the revised manuscript, we will rephrase the sentence related to this matter: on page 25 line 824, "The status information for AirTraf including the licence conditions **is** will be available at the website."

• (10) Figure 15, 16, S4 – S6 – Please add units to the colour bar and/or text.

Reply: Thank you very much. We will add units in the captions for Figs. 15, 16, S3 – S6. In Figs. 15, S3 and S4, we will add the unit in the captions as, "The contours show the zonal wind speed (*u* in ms-1)." In Figs. 16, S5 and S6, we will add the unit in the captions as, "Altitude distributions of the true air speed  $V_{TAS}$  in ms-1 (a and b)." The wind indicator is dimensionless quantity.

• (11) Table 8. It is difficult to compare the flight time for the time-optimal with the great circle at different altitudes, since the mean flight altitude of the time-optimal flights is given in m and the altitude of the great circle flights in feet. Please add either the mean flight altitude in feet for the time-optimal flights, or the flight altitude in m for the great circle flights to aid the comparison.

Reply: Thank you very much. In the revised manuscript, we will add the mean flight altitude in feet for the time-optimal flights on column 6 in Table 8: "Mean flight altitude h, m (in ft); 8,841 (29,005); 8,839 (29,000); 8,839 (29,000); 10,002 (32,815); 10,829 (35,527); 9,311 (30,546)."

• (12) Table 11, Caption. 'sum of flight time, fuel use, NOx and H2O emissions...'. This implies that the table shows the quantity flight time + fuel use + NOx + H2O, when in fact they are displayed separately. Please rephrase.

Reply: Thank you very much. In the revised manuscript, we will remove the word "**Sum of**" from the caption: on page 59 in Table 11, we will rewrite the caption as "**Flight time, fuel use, NO**x and H2O emissions for the time-optimal and the great circle cases...".

We are most grateful to the referee #2 for the very helpful and encouraging comments on the original version of our manuscript. Here are our replies:

This paper presents a development of "module" adapted to the climate chemistry model ECHAM5/MESSY in order to calculate the climate impact of aircraft routes. Only one part of the module needed has been included in the model and presented in this paper: the part generating the route and only in the case of great circle (simple) or time-optimal route (optimisation). From these two routing the module calculates fuel use, and some emissions ( $H_2O$  and  $NO_x$  only), these parameter are assessed with real data. The module is tested over one winter day data over the North Atlantic corridor. In its present form I unfortunately cannot recommend the publication of the paper in Geoscientific Model Development for several reasons that I will be listing. I would strongly recommend the editor to request a severe revision before publication. The timeoptimal calculation module may be of interest for modellers. The optimisation module description as well as the size of the population to be included in the optimisation to converge toward optimal time may be presented in a revised paper.

Reply: We are grateful to the referee #2 for the critical comments and useful suggestions that have helped us to improve our manuscript. As indicated in the responses that follow, we have addressed all the comments and suggestions. We now state in the introduction that this development is a prerequisite for the investigation of climate-optimal routings. So that the motivation for this development is clear. And we are deleting this overall objective from other text passages, since we agree that they are misleading. We will reply to this point in the following (1). As the referee #2 noted, the descriptions of the time-optimal calculation module and the population sizing are included in the revised manuscript, as originally described.

(1) My first problem is the presentation of the subject within most of the article (title, abstract and even structure of the manuscript). The focus seems to be in the "optimal routing for climate impact reduction" when you check the paper, however the reader is disappointed as the presented module is not doing that at all – only optimising for travel time. The manuscript needs to be reshaped completely to acknowledge that fact.

Reply: As the referee #2 pointed out, the subject of this paper seems to be confusing. We should make clear that this paper introduces AirTraf submodel in its basic version, technically describes and validates the various components for first, simple aircraft routings (great circle and time-optimal). Eventually, we are aiming at an optimal routing for climate impact reduction. This will be a separate study, which requires a couple of developments beforehand, amongst which the present study is one of them. Here, we would like to make clear that the final purpose of the AirTraf is not to find "fastest routes." For this, an Earth System Model (ESM) is not necessary. There are even better tools to answer this question. However, to find climate-optimal routes, the global air traffic simulation model coupled to the ESM, i.e. AirTraf submodel, is needed. And of course it has to be described and validated. The validation refers to standard aircraft applications in this paper, such great circle and time-optimal calculations.

In the revised manuscript, we will revise the title, abstract, introduction and conclusion to be consistent with what is presented in the paper as follows: the title will be revised as, "<del>Climate Assessment Platform of Different Aircraft Routing Strategies</del> **Air traffic simulation** in the Chemistry-Climate Model EMAC 2.41: AirTraf 1.0".

On page 1, line 9 in Abstract, the text will be revised as, "This study introduces AirTraf (version 1.0) forclimate impact evaluations that performs global air traffic simulations on long time scales, including effects of local weather conditions on the emissions."

On page 3, final paragraph (line 84 - 87), "This study aims to investigate how much the climate impact of aircraft emissions can be reduced by aircraft routing. Here, we present a new assessment platform AirTraf-(version 1.0, Yamashita et al., 2015) that is a global air traffic submodel coupled to the Chemistry-Climate-model EMAC (Jöckel et al., 2010). Figure 1 shows the research road map for this study (Grewe et al., 2014b). This paper presents the new submodel AirTraf (version 1.0, Yamashita et al., 2015) that performs

global air traffic simulations coupled to the Chemistry-Climate model EMAC (Jöckel et al., 2010). This paper technically describes AirTraf and validates the various components for simple aircraft routings: great circle and time-optimal routings. Eventually, we are aiming at an optimal routing for climate impact reduction. The development described in this paper is a prerequisite for the investigation of climate-optimized routings. The research road map for our study is as follows (Grewe et al., 2014b):-Tthe first step was to investigate...".

On page 26, final paragraph (line 870 – 873), "The fundamental framework of AirTraf has been developed to perform fairly realistic air traffic simulations. AirTraf 1.0 is sufficient to investigate a reduction potential of aircraft routings on air traffic climate impacts is ready for more complex routing tasks. AirTraf is coupled with various submodels of EMAC to evaluate the impacts, and oObjective functions corresponding to other routing options will be integrated soon, and AirTraf will be coupled with various submodels of EMAC to evaluate air traffic climate impacts."

(2) I am also extremely disappointed in the fact that a part of the paper is dedicated in presenting and comparing "great circle routing" calculations. This is nothing new, and no advance in modelling or science presented. This part should be cut down and re-moved from the discussion. The more important difference could come from the fact the Earth is not a perfect sphere or maybe taking into account flight altitude. The table 4 is comparing calculation with decimal and no-decimal data when the difference is in the decimal value.

Reply: The referee is right that a "great circle calculation" is commonly used method. However, we are hesitating to remove the discussion on that part for the following three reasons.

First, the final purpose of the AirTraf is to investigate "optimal routing for climate impact reduction." We will compare AirTraf simulation results among several aircraft routing options. As a climate-optimized route will be evaluated in the light of the detour that would be necessary to avoid "climate-sensitive" areas with respect to the reference (trade-off), i.e. great circle or time-optimal route. Thus, the great circle routing option is used as reference of our comparisons (note that the great circle is the optimal solution for "minimum flight distance"). In addition, we would like to refer to a future Air Traffic Management system, which aims at having aircraft fly more direct routes, so called user-preferred routes without being constrained to Air Traffic Services routes and waypoints any longer. These future user-preferred routes would be great circle segments in the ideal case (without wind). Hence, AirTraf is developed with the objective to evaluate routing options for the future and the great circle is still an important route in reality. We think that a thorough assessment of the great circle routing module should be made in this paper to demonstrate its ability to generate the routes and working well if coupled to the ESM. The "great circle calculation" is suitable for the validation of AirTraf, because it is the widely used method (the benchmark test of the great circle calculation is described on page 12 - 13, Sect. 3.1.2).

Second, the above-mentioned assessment of the great circle routing module is also indispensable to showing the correct implementation and applicability of the genetic algorithm (GA) approach. Because the validated great circle routing module provides the analytical solution ( $f_{true} = 25,994.0$  s) for the benchmark test of flight trajectory optimization with GA (i.e. the single-objective optimization for minimization of flight time from MUC to JFK). This point is described on page 16 line 530, "...the  $f_{true}$  equals the flight time along the great circle from MUC to JFK at FL290:  $f_{true} = 25,994.0$  s calculated by Eq. (23) with  $h_i =$  FL290 for  $i = 1, 2, \cdots$ , 101." That the GA reproduces the analytical solution is an important milestone towards other routing optimizations. The part of the great circle routing module supports the discussion of the flight trajectory optimization with GA. Hence, the description of the great circle routing module should be included in this paper.

Last, we would like to stress that AirTraf submodel, which contains the combination of a routing module (including GA) with an Earth System Model, is unique. That is, the great circle routing module described in the paper is a unique model, which works coupling with the ESM. For example, a flight trajectory consists of

waypoints arranged by the waypoint index i ( $i = 1, 2, \dots, n_{wp}$ ). The geographical and meteorological values, which are used regarding the great circle calculation (e.g. latitude, longitude, altitude, temperature, wind speeds), are provided by the ESM to individual waypoint i. It is important to show correctly how the great circles are calculated through waypoints in the ESM. For this, Eqs. 21 - 27 (on page 11 - 12) include the terms with the index i.

As the referee #2 noted, an influence of the asymmetric nature of the Earth is an interesting topic. However, we think that this is a separate study. On page 5 line 135, we describe the assumption for AirTraf (version 1.0) as, "a spherical Earth is assumed (radius is  $R_E$  = 6,371 km)," corresponding to the ESM. On page 11 in section 3.1, Eqs. 22 and 23 present in detail how to take into account the flight altitude in AirTraf. This part is included in the revised manuscript.

In addition, as the referee #2 pointed out, the decimal and no-decimal data are compared in Table 4. This is indeed a very important point, which we completely overlooked. We will revise Table 4: on column 4, "dMTS, km; 6,481.1; 10,875.0; 16,312.1; 8,895.6; 13,343.4". On column 6, " $\Delta d_{eq23, MTS}$ , %; –0.0005; –0.0028; – 0.0036; –0.0008; –0.0019". On column 7, " $\Delta d_{eq22, MTS}$ , %; 0.0000; 0.0000; 0.0000; 0.0000". We will also revise the caption in Table 4 as, "...column 4 (dMTS) shows the result calculated with the Movable Type scripts (MTS), which output only integer values using the Haversine formula with a spherical Earth radius of  $R_E = 6,371$  km."

Related to this matter, we will revise the manuscript as follows: on page 1 line 18, "The first test showed that the great circle calculations were accurate to within –0.004 %…". On page 11 line 354, we will revise the word "Harvesine formula" into "Haversine formula." On page 13 line 406, "The results showed that both  $\Delta d_{eq23,eq22}$  and  $\Delta d_{eq23,MTS}$  varied between –0.0036 and –0.00085 %, and between –0.0435 –0.0036 and 0.0054 –0.0005 %, respectively, while  $\Delta d_{eq22,MTS}$  showed 0.0 % and between –0.0463 and 0.0046 %." On page 13 line 408, "The great circle distances calculated by Eqs. (22) and (23) were accurate to within –0.004 %…". On page 25 line 832, "The accuracy of the results was within –0.004 %." On page 26 line 876, we will add the text as, "The authors thank Mr. Chris Veness for providing great circle distances that have been calculated with the Movable tType script."

• (3) Concerning the "optimisation routing" for flying time the validation over the North Atlantic is interesting but what would happen with a case of congested space or restricted space (military)? Please do tests in different part of the world or at different season.

Reply: We think that the topics, which the referee #2 noted here, are important and interesting. However, we think that they are application studies which would probably use AirTraf, but which are beyond the scope of this technical documentation and first evaluation. The aim of this paper is to introduce, describe and validate the AirTraf submodel, as replied to the comment (1) above. We believe that this paper shows a substantial comparison of AirTraf simulation results to other studies to validate the model.

• (4) Moreover I am unsure of the complete philosophy of the inclusion of the "optimisation" module in the ECHAM5/MESSY model. I understand well the impact of local weather and composition on the impact the aircraft routing will have on climate change. However I am short in understanding the need of the online optimisation as I don't see the effect of "climate optimal routing" on the climate model – would a simple offline calculation not enough to determine this potential "climate optimal routing" (the day the full module will be ready) as well as making the "optimisation" easier to be adapted to other climate-chemistry model output?

Reply: As replied to the comment (1) above, our final purpose is to investigate the mitigation gain of the climate impact by climate-optimal routing. We would like to make clear that it is not our final purpose only to find climate-optimal flight trajectories for a specific weather condition. This was achieved, e.g. in Grewe et al., 2014. We eventually want to go one step further and apply an optimization on a daily basis for daily changing weather situations. To investigate then the mitigation gain, multi-annual (long-term) simulations are

required (e.g. for ten years). In the simulations over the ten years, each flight trajectory is optimized with respect to a selected aircraft routing option, considering local weather conditions, and emissions are released. AirTraf can perform such air traffic simulations with the inclusion of the on-line optimization module and the optimal routes will change day by day. We think that the inclusion of the optimization module in EMAC is an appropriate approach for our purpose.

[Reference] Grewe, V., Champougny, T., Matthes, S., Frömming, C., Brinkop, S., Søvde, O. A., Irvine, E. A., and Halscheidt, L.: Reduction of the air traffic's contribution to climate change: A REACT4C case study, Atmospheric Environment, 94, 616–625, 2014a.

(5) Finally I am unhappy with the fact that the only simple "time optimal routing" (optimising only for one variable) the weather situation if fixed for the entire flight. What would happen in the case of multi optimisation when you have to trade-off between time, fuel use, and different emissions? Could you comment on the impact on contrail formation from long flights? "-For all routing options, local weather conditions provided by EMAC at t = 1 (i.e. at the departure day and time of the aircraft) are used to calculate the flight trajectory. The conditions are assumed to be constant during the flight trajectory calculation-"making the model as simple as an offline module but complicated as an inside module of an already complex model?

Reply: In this paper, we would like to confirm whether AirTraf works well and is fit for our purpose. Particularly, the ability of the optimization module (GA) to optimize flight routes must be confirmed. For this, we tested the simple "time-optimal routing." The referee actually points at many interesting future investigations, which are far beyond the scope of this paper. As soon as we really start with climate optimized trajectories in EMAC/AirTraf, we will investigate whether it is necessary to re-optimize the trajectory during long flights. It is clear that a weather forecast, which would be required to optimize not only for time t = 1, is not feasible within the climate simulation. To cover all effects, such as NOx effects, an offline calculation on the other hand is not feasible.

In addition, the contrail formation is one of the important factors on climate impacts. For example, Schumann, et al. 2011 noted in the literature: "...contrails are expected to cause the largest contribution to global radiative forcing of the Earth-atmosphere system, and hence, the largest contribution to aviation-induced global climate change...", and "Contrails and thin cirrus in general warm the Earth atmosphere by reducing terrestrial (longwave, LW) radiation loss into space and may cool the Earth atmosphere by reflecting part of the solar (short-wave, SW) radiation back to space. During night, contrails are always warming. The largest climate impact by contrails comes from thick, wide, long and long-lasting contrails. Hence, with respect to climate, optimal routes during night are those which form contrails, over dark and cool surfaces, in particular in the morning and evening times when cirrus is more reflective than during mid day. Hence, with respect to minimum contrail warming impact, optimal routes may be those causing contrails with maximum shortwave cooling."

Those contrail effects will be considered as one of the routing options in AirTraf, by coupling with another submodel of EMAC. AirTraf on-line simulation (coupled to the ESM) is a suited model for taking these complicated effects into account on long time scales and this is a difference from off-line models. In this context, as the referee #2 noted, local weather conditions are assumed to be constant during flight trajectory optimization. We think that this assumption is appropriate to perform such AirTraf on-line simulation for long-term to reduce the computational costs.

[Reference] Schumann, U., Graf, K., and Mannstein, H.: Potential to reduce the climate impact of aviation by flight level changes, in: 3rd AIAA Atmospheric and Space Environments Conference, AIAA paper, vol. 3376, pp. 1–22, 2011.

We are most grateful to the referee #3 for the very helpful and encouraging comments on the original version of our manuscript. Here are our replies:

**1 Introduction:**

٠

The manuscript is well structured, and different aspects of AirTraf are explained by a nice equilibrium of description and examples. The motivation of the work is reasonably well explained. Figures and tables are informative. There is a substantial comparison with results from other studies to give confidence in the results obtained here.

Implementing aircraft routing strategies in a general circulation model or a numerical weather prediction model is not an easy task. Arriving at the status as described here in the manuscript is already a considerable achievement. However, as the tool is not finished, one wonders whether it is useful to describe the tool in its current status (with only 2 of the 7 optimization options implemented, fuel consumption due to climbing not included, the meteorological fields in the optimization are the ones at the start of the flight,...).

Publishing the manuscript now shows the status of the work. It makes clear that for specific options the optimization works, and it can trigger discussion with other researchers/institutes on the approaches chosen (is the optimization working well, could other optimization routines be faster,...).

Reply: We thank the referee #3 for the positive comments. As the referee pointed out, this paper shows the current status of AirTraf. Nevertheless, we think that it is useful to publish AirTraf v.1.0, for several reasons:

- Our final purpose is to investigate an optimization strategy of aircraft routing for minimizing the climate impact of aircraft emissions and show its mitigation gain for future. We should make clear that this paper introduces the AirTraf submodel in its basic version, technically describes and validates the various components for first, simple aircraft routings (great circle and time-optimal). Eventually, we are aiming at an optimal routing for climate impact reduction. This will be a separate study, which requires a couple of developments beforehand, amongst which the present study documents one of them.

- The validation refers to standard aircraft applications in this paper, such as great circle and timeoptimal calculations. These two options are appropriate to confirm whether AirTraf works well and is fit for the purpose. This is a big step for the AirTraf development.

– For our purpose, multi-annual (long-term) simulations are required in EMAC: computationally expensive simulations are required. Hence, in the current model we simplify AirTraf to reduce the computational costs, e.g. we concentrate on the cruise mission only.

- The related issue is discussed in the reply to "2 Principal remarks, Work in progress."

• I think the manuscript is worth publishing, but is should be considerably improved in several ways. A list of principal remarks is given below, followed by a list of more specific comments. I hope the authors will take them into consideration, and if not give a sound argumentation why they do not.

Reply: We are grateful to the referee #3 for the useful comments and suggestions that have helped us to improve our manuscript. As indicated in the responses that follow, we have addressed all the comments and suggestions.

**2 Principal remarks**

**Work in progress:** The manuscript describes a submodel in MESSy which works, but is not finished yet (only 2 of the 7 optimization options are in place). Why not waiting until all the work is finished? One has to guarantee that this manuscript remains valid and worth all the work once the remaining parts come into place, and that this document is therefore worth publishing.

Reply: The major reasons are replied in "1 Introduction." As replied in "1 Introduction", the currently documented status is a prerequisite for the investigation of climate-optimal routings. Additional reasons are:

- The GA optimization module is an important part of AirTraf for our purpose. Therefore, we made a thorough assessment of the GA optimization and its performance using the time-optimal option in

this paper. If a new objective function corresponding to other routing options is developed, basically, only the objective function f (shown in Eq. (28), on page 15 line 485) is changed. The AirTraf framework validated in the paper is, thanks to its modular structure, unchanged. Therefore, the current status is a big step for AirTraf development.

- The manuscript is not only about the "routing options", but an important and integral part describes the overall structure of the coupling between a "routing module" and a chemistry-climate model. This is a major achievement and unique.

• **Language:** There is a lot of improvement needed for the language. The use of articles (a/an/the/none) should be improved. Specific expressions (e.g., "trajectories as longitude vs altitude, trajectories as location" or "number of  $n_p$ ",...) should be modified.

Reply: Thank you so much. We will recheck and modify articles. Please see the revised manuscript. The modifications of the specific expressions are as follows:

["trajectories as longitude vs altitude, trajectories as location"]

We will change the expression "trajectories as longitude vs altitude, trajectories as location" into "trajectories in the vertical cross-section, trajectories projected onto the Earth":

- On page 14 line 449, "...the geographic location **projection onto the Earth** (bottom) with three control points (CPs, black circles) and the longitude vs altitude vertical cross-section (top) with five CPs."

- On page 15 line 475, "...B-spline curve with the five CPs as longitude vs altitude in the vertical cross-section (bold solid line, Fig. 7 top)...".

- On page 17 line 553, "...the true-optimal solution as longitude vs altitude in the vertical cross-section are plotted...".

- On page 34 in the caption of Figure 7, "Geometry definition of flight trajectory as longitude vs altitude in **the vertical cross-section** (top) and as geographic location **projected onto the Earth** (bottom)."

- On page 36 in the caption of Figure 9, "...explored trajectories (solid line, black) from MUC to JFK as longitude vs altitude in the vertical cross-section (top) and as location projected onto the Earth (bottom)."

- On page 41 in the caption of Figure 14, "...explored trajectories (black lines) between MSP and AMS as longitude vs altitude in the vertical cross-section (top) and as location projected onto the Earth (bottom)."

– On page 45 in the caption of Figure 18, "...the trajectories as longitude vs altitude in the vertical cross-section (top) and as location projected onto the Earth (bottom)."

- On page 1 (Supplementary material) in the caption of Figure S1, "...explored trajectories (black lines) between JFK and MUC as longitude vs altitude in the vertical cross-section (top) and as location projected onto the Earth (bottom),...".

- On page 2 (Supplementary material) in the caption of Figure S2, "...explored trajectories (black lines) between SEA and AMS as longitude vs altitude in the vertical cross-section (top) and as location projected onto the Earth (bottom),...".

["number of  $n_p$ "]

We will change the expression "number of  $n_p$ " into "value of  $n_p$ " in the revised manuscript. We also reply to this modification in the following sections: "p 17, l 569 and 570" and "p 19, l 618."

- **CP** in trajectories: Concerning the treatment of CP points, I have several questions.
- (1) As an example, 3 CPs have been used for the geographical location, and 5 for the altitude. Is this fixed? Do all flights use the same number of CPs?

Reply: Yes. All flights use 3 CPs for the geographical location and 5 for the altitude (as shown in Fig. 7 on page 34). This is now explicitly clarified in the revised text.

• (2) For the 103 flights, which were primarily zonal, rectangles around the CPs could be described by using a

range in latitude and longitude. How is the choice around the CPs when flights cross the equator, e.g., at an angle of 45°? What if flights go from low to high latitudes and defining regions whit fixed ranges in longitude makes them very different in size?

Reply: This is a very important issue for the AirTraf development. In AirTraf version 1.0, the domain size was determined by referring to the literature: Irvine, E. A., et al., "Characterizing North Atlantic weather patterns for climate-optimal aircraft routing," Meteorological applications, 20, 80–93 (2013). They show the many types of flight trajectories between London and New York for different weather conditions. We focused on trans-Atlantic flights in this paper, therefore the current definition of domain size works very well for the trajectory optimizations.

As the referee pointed out, if flights cross the equator (at an angle of 45°) or if flights go from low to high latitudes with almost similar longitude values, the domains are variously shaped in size on the basis of the geometry definitions of the flight trajectory (as described in Sect. 3.2.2 on page 14). This probably increases the computational demand for the trajectory optimization. Nevertheless, the current treatment of the domains is applicable to those flights and trajectory optimization works well. In fact, we have confirmed this issue by test simulations using 1,840 global flight plans including such flights. To improve the computational efficiency of the optimization, we will work on an improvement of the definition of domain size for the next version.

We also reply to this issue in the answer to the referee comment of "p 17, l 554–555."

• (3) For a given trajectory (which is a B-spline curve), how are the waypoints found? Are they equally spaced along that trajectory between the CPs? I am wondering whether it is possible to find explicit expressions for equidistant waypoints on a B-spline curve?

Reply: The referee is right. In AirTraf, the 3rd order B-spline curve is used to generate the waypoints. If CPs are given, a parameter *t*, which is the parameter of the 3rd order B-spline basis functions, is assigned with values between 0 and 1 between the CPs. Here, *t* is equally spaced along the "basis functions" (i.e., equally spaced between  $0 \le t \le 1$ ). After that, the coordinates of the waypoints of the trajectory are determined by summation of the basis functions, corresponding to the equidistant *t*. Therefore, this can not ensure that the waypoints are equally spaced along the trajectory. We reply to this issue in the answer to the referee comment of "p 14, l 464".

(4) In the example used, 3 CPs were used for the geographical location, 5 CPs for the altitude, and 101 waypoints. However, the condition (101 - 1)modulo(5 + 1) = 0 is not fulfilled. One also gets the impression that the waypoints for the altitude and longitude are not located at the same place (although the manuscript confirms it actually is). Could this be clarified?

Reply: As described on page 14 line 464, the condition is  $mod(n_{wp} - 1, \underline{n_{CPloc}} + 1) = 0$ . This is only used for the location. Here,  $n_{wp} = 101$  and  $n_{CPloc} = 3$ . Therefore, mod(101 - 1, 3 + 1) = 0 is fulfilled. In addition, to clarify the location of waypoints for the altitude and longitude, we will revise the text: on page 15 line 474–478, "A flight trajectory is also represented by a B-spline curve (**3**rd-**order**) with the five CPs as longitude vs altitude in **the vertical cross-section** (bold solid line, Fig. 7 top) and then waypoints are generated along the trajectory **in such a way that the longitude of the waypoints is the same as that for the flight trajectory projected onto the Earth.** Note, GA creates trajectories represented by two B-splines, one latitude vs-longitude and one longitude vs altitude, where longitude-coordinate of waypoints is the same for the two-curves." We also reply to this issue in the referee comment of "p 15, l 476–478."

• **GA algorithm:** This algorithm is explained to some detail, but I suggest that all terms used should be explained to some extent (e.g., mating pool). One should also be informed on how the final solution is derived from the population in the last generation. Finally, the abstract uses some terminology related to the optimization routine (e.g., population), which are too technical to be mentioned in the abstract.

Reply: We will add a section "Appendix; Glossary" after the section "7. Conclusions", where we explain the optimization terminologies: on page 26, "Appendix; Glossary; Table A1 shows a glossary explaining several terminologies of the GA optimization. The terms from the glossary are written in italics in the text." In Table A1, we will add the explanations, "Table A1. Glossary of terms. Population: A set of solutions. A Genetic Algorithm starts its search with an initial *population* (a random set of solutions).; Generation: One iteration of a Genetic Algorithm.; Rank: A ranking assigned to each solution to evaluate a relative merit in a *population*. A *rank* expresses the number of solutions that are superior to a solution; Fitness: A value assigned to each solution to emphasize superior solutions and eliminate inferior solutions in a *population*. *Fitness* = 1/*rank*.; Mating pool: A storage space for solutions." We will refer to those terms in the text in italics. Many variables are modified. Therefore, we will show the modifications in the revised manuscript. Related to this, we will revise the text: on page 2 line 21 in Abstract, "The dependence of the optimal solutions on the initial *populations* set of solutions (called population) was analyzed." On page 15 line 491, "A solution with a higher *fitness* value (i.e., a smaller *rank* value) has a higher probability of being copied into a *mating pool*."

In addition, we will add the text to inform on how the final solution is obtained from the optimization: on page 16 line 517, "..., GA quits the optimization and an optimal solution **showing the best** f **of the whole** *generation* is output...".

• **Abstract, introduction, conclusion:** The abstract is sometimes too much a summing up of what has been done, with vocabulary/terms which have no concrete meaning without a concrete context. There is also much more overlap between these three parts (abstract, introduction, and conclusion) than needed. The abstract should be written differently, and considerably improved.

Reply: By following the remarks and the list of specific comments of the referee #3, we revise the abstract, introduction and conclusion. Please see the revised manuscript.

**Sensitivity:** In the approach followed here, quite some assumptions and simplifications are introduced. It would be useful to give the reader an idea of the impact of these assumptions on the results. A list of some of the assumptions is:

Reply: Firstly, we would like to make clear again that our final purpose of AirTraf is to investigate an "optimization strategy" of aircraft routing for minimizing the climate impact of aircraft emissions and to show its mitigation gain for the future. It is not our purpose to find detailed flight trajectories. The aim of this paper is to introduce the AirTraf submodel in its basic version, technically describe and validate the various components for first, simple aircraft routings (great circle and time-optimal), in order to confirm whether AirTraf works well and is fit for our purpose. Eventually, we are aiming at an optimal routing for climate impact reduction. This will be a separate study, which requires a couple of additional developments beforehand, amongst which the present study is one of them. In addition, multi-annual (long-term) simulations are required for our purpose (e.g. for ten years) coupled with the Earth System Model: computationally expensive simulations are required. We therefore think that our assumptions are appropriate to perform such AirTraf on-line simulations for long-term periods to reduce the computational costs.

As the referee pointed out, they are all interesting points and might be a future option. However, they are beyond the scope of this paper and we cannot explore all sensitivities. A couple of specific points are as follows:

• (1) line 274 : dh(t)/dt = 0 in Eq. (3).

Reply: Looking at the AirTraf trajectories, there is an altitude change visible, but it appears over a long distance and a long period of time. We evaluated dh/dt of the time-optimal flight trajectories for the three selected airport pairs (listed in Table 8 on page 57). The averages of dh/dt (absolute value, ms-1) for the individual flights were: 0.0 (JFK to MUC); 0.0 (MUC to JFK); 0.0 (MSP to AMS); 0.32 (AMS to MSP); 0.24 (SEA to AMS); and 0.13 (AMS to SEA). We therefore conclude that the impact of the zero-assumption is not

a big issue, the more as in AirTraf 1.0, we use so far only a small number of vertical GA control points (shown in Fig. 7 on page 34). If the number of control points increases, the influence of climb/descent rates (dh/dt) will increase. This could be an aspect for a next version of AirTraf.

To clarify our assumptions, we will revise the text: on page 9 line 273–275, "In AirTraf (version 1.0), dh/dt = 0 is assumed and  $V_{\text{FAS}}$  is calculated at every waypoint (Table 2). For an aircraft in cruise, Eq. (3) becomes  $Thr_i = D_i$  at waypoint *i*. For a cruise flight phase, both altitude and speed changes are negligible. Hence, dh/dt = 0 as well as  $dV_{\text{TAS}}/dt = 0$  is assumed in AirTraf (version 1.0) and Eq. (3) becomes the typical cruise equilibrium equation:  $Thr_i = D_i$  at waypoint *i*."

• (2) *M* is set constant. Can this be varied slightly? Or have pilots only a very small envelope of allowed or possible speeds?

Reply: The constant Mach number, M = 0.82, is the officially published cruise Mach number of an A330-301 by Eurocontrol in 2011. It is appropriate for the aim of this paper to perform AirTraf simulations for simple conditions, including a constant M. On page 5 line 136, we describe the assumption for AirTraf (version 1.0) as, "The aircraft performance model of Eurocontrol's Base of Aircraft Data (BADA Revision 3.9, Eurocontrol, 2011) is used with a constant Mach number M...". As the referee noted, a change of Mach number is an interesting topic. However, this will be a separate study. In addition, pilots are not allowed to change flight speed freely in the actual flight operations. The speed is indicated (controlled) from the air traffic management side.

• (3) What if weather not just from t = 1 is taken, but from the whole period of the flight?

Reply: The referee actually points out the important and interesting topic. However, this is a separate study, which would probably use AirTraf, but which is beyond the scope of this technical documentation and first evaluation. On page 7 line 201, we describe the assumption for AirTraf (version 1.0) as, "…local weather conditions provided by EMAC at t = 1 (i.e. at the departure day and time of the aircraft) are used to calculate the flight trajectory. The conditions are assumed to be constant during the flight trajectory calculation." Note that a weather forecast, which would be required to optimize not only for time t = 1, is not feasible within a climate simulation.

• (4) Leaving out the ascent and descend phase of the flight: how does this impact the optimization?

Reply: For our final purpose described in the reply to "1 Introduction" and "Sensitivity", it is appropriate to concentrate on the cruise mission only in AirTraf (version 1.0). On page 5 line 140, we describe the assumption for AirTraf (version 1.0) as, "Only the cruise flight phase is considered, while ground operations, take off, landing and any other flight phases are unconsidered." It is maybe worth to mention that the cruise has a larger climatic impact than the other parts of the operation, since the cruise has a longer operation time. Moreover, there are other attempts to reduce emissions during ground operation (taxiing etc.), which are not connected to routing. In any case, there are not much "re-routing" options between ground operations and reaching the cruise altitude.

- **Mathematical formulas:** The mathematical expressions should be improved.
- (1) In mathematical formulas, variables longer than one letter should be written straight.

Reply: We will recheck all variables and modify them with straight letters. Many variables are modified; therefore, we will show the modifications in the revised manuscript.

• (2) A lot of indices should be straight letters :  $V_{\text{ground}}$ ,  $V_{\text{wind}}$ , ...

Reply: We will recheck all indices and modify them with straight letters. Many indices are modified;

therefore, we will show the modifications in the revised manuscript.

• (3) After every equation, there should be a "," or ".", depending on the function of the equation in the sentence.

Reply: We will add a "," after **Eqs. (1)–(8), Eqs. (11)–(22), Eqs. (24)–(27) and Eq. (29).** We will show the modifications in the revised manuscript.

• (4) Names of trigonometric formulas should not be italic : sin, cos, ...

Reply: We will modify the all names of trigonometric formulas into normal straight letters: for "sin," **Eq. (21)** is modified; for "cos," **Eqs. (4), (21) and (23)** are modified; and for "arctan," **Eq. (21)** is modified. We will show the modifications in the revised manuscript.

• **Climate model, long/short time scales:** Why is this tool implemented in a climate model? To my opinion, the tool could also have been build such that it uses off-line 3-hourly meteo fields over the range of time it has flights which should be optimized : one thinks over a range of 1 to 10 days. The meteo data might come from a NWP, or a climate model.

Maybe the authors want to show that it is possible to have such a tool on-line in a NWP or GCM. However, in that case, I would have chosen for a NWP as that is the place where, if the tool is operationally used, might be most appropriate. What was the reason that the authors made the choice of implementing it in a climate model?

A reason I can imagine is that one could do tests like : how would the optimal routing be in a year 2100 climate, when global climate is considerably different from nowadays?

Reply: Our final purpose is to investigate the mitigation gain of the climate impact by climate-optimal routing. We would like to make clear that it is not our purpose to find climate-optimal flight trajectories (or optimal flight trajectories corresponding to a selected routing option, e.g. fastest routes) for a specific weather condition. For this, an Earth System Modeling (ESM) is not necessary and this indeed has been achieved, e.g. by Grewe et al., 2014. We eventually want to go one step further and apply an optimization on a daily basis for daily changing weather situations. To investigate then the mitigation gain, multi-annual (long-term) simulations are required (e.g. for ten years). In the simulations over the ten years, each flight trajectory is optimized with respect to a selected aircraft routing option, considering local weather conditions. The released emissions directly ( $CO_2$ ,  $H_2O$ ) and indirectly ( $NO_x$ ) modify the radiative forcing and therefore the climate. Off-line pre-calculated routes would be inconsistent in such an approach. AirTraf can perform these air traffic simulations with the inclusion of the on-line optimization module and the optimal routes will change day by day. In addition, AirTraf can use the framework of EMAC to assess routing options, e.g. surface temperature changes or changes in the background chemical conditions of the atmosphere ten years later corresponding to the selected routing option, by coupling with other submodels of EMAC. The main point is the interactive coupling, i.e. the on-line re-routing immediately affects the climate model (via air traffic emissions). An online feedback cannot be replaced by an off-line approach. We think that the implementation of AirTraf on-line in EMAC is appropriate approach for our purpose. This reply it related to the reply to "p4 l 115."

[Reference] Grewe, V., Champougny, T., Matthes, S., Frömming, C., Brinkop, S., Søvde, O. A., Irvine, E. A., and Halscheidt, L.: Reduction of the air traffic's contribution to climate change: A REACT4C case study, Atmospheric Environment, 94, 616–625, 2014a.

• **Benchmarks:** Is proving that the great circle option works well worth publishing and/or mentioning in an abstract? In addition, I think that the word benchmark puts more importance on a test than it actually deserves.

Reply: We understand the referee comment. The "great circle calculation" is a commonly used method.

However, we are hesitating to remove the descriptions of the great circle for the following three reasons:

First, the final purpose of AirTraf is to investigate "optimal routing for climate impact reduction." We will compare AirTraf simulation results among several aircraft routing options. As a climate-optimized route will be evaluated in the light of the detour that would be necessary to avoid "climate-sensitive" areas with respect to the reference (trade-off), i.e. "great circle" or time-optimal route. Thus, the great circle routing option is used as reference for our comparisons (note that the great circle is the optimal solution for "minimum flight distance"). In addition, we would like to refer to a future Air Traffic Management system, which aims at having aircraft fly more direct routes, so called user-preferred routes without being constrained to Air Traffic Services routes and waypoints any longer. These future user-preferred routes would be great circle segments in the ideal case (without wind). Hence, AirTraf is developed with the objective to evaluate routing options for the future and the great circle is still an important route in reality. We think that a thorough assessment of the great circle routing module should be made in this paper to demonstrate its ability to generate the routes and working well if coupled to the ESM. The "great circle calculation" is suitable for the validation of AirTraf, because it is a widely used method (the benchmark test of the great circle calculation is described on page 12–13, Sect. 3.1.2). We believe that the result of the assessment is worth publishing.

Second, the above-mentioned assessment of the great circle routing module is also indispensable to show the correct implementation and applicability of the genetic algorithm (GA) approach. Because the validated great circle routing module provides the analytical solution ( $f_{true} = 25,994.0$  s) for the benchmark test of flight trajectory optimization with GA (i.e. the single-objective optimization for minimization of flight time from MUC to JFK). This point is described on page 16 line 530, "...the  $f_{true}$  equals the flight time along the great circle from MUC to JFK at FL290:  $f_{true} = 25,994.0$  s calculated by Eq. (23) with  $h_i = FL290$  for i = 1, 2,…, 101." The result that the GA reproduces the analytical solution is an important milestone towards other routing optimizations.

Last but not least, we would like to stress that the AirTraf submodel, which embeds a routing module (including GA) into an Earth System Model, is unique. The great circle routing module described in the paper is used to show that the coupled system works well. For example, a flight trajectory consists of waypoints arranged by the waypoint index *i* ( $i = 1, 2, \dots, n_{wp}$ ). The geographical and meteorological values, which are used for the great circle calculation (e.g. latitude, longitude, altitude, temperature, wind speed), are provided by the ESM at the individual waypoints *i*. It is important to show that the great circles are calculated correctly by waypoints through the ESM domain. For this, Eqs. (21)–(27) (on page 11–12) include the terms with the index *i*. Hence, the description of the great circle routing module should be included.

In addition, we understand the referee comment on the word "benchmark." Nevertheless, we are hesitating to change the word. The tests are performed to confirm the correct performance of the code, which we believe is unique and new, and thus to measure the reliability of the code. We think that those tests are indeed important "benchmark tests."

• **Size of the document:** The files are so large (30 MB) that people will have problems printing the documents. To my opinion it is mainly related to the figures which show different flight trajectories. I assume that the figures contain all the information from all trajectories, while a large central part of the figure is just black. These figures should be made in such a way that they become much smaller in size, without loosing their precision.

Reply: As the referee pointed out, the file size is large. We will make those figures become much smaller in size with almost the current precision and replace them in the revised manuscript: **Figs. 9, 14a, 14b, S1a, S1b, S2a** and **S2b** are modified.

**3** Comments on the text**

**Page 1**

**p** 1, l 1–5 : The sequence of the first three sentences is a bit strange. I would even skip the first sentence (as it says the same as the first 7 words of sentence 3).

Reply: We will remove the first sentence: on page 1 line 1, "Aviation contributes to anthropogenic climateimpact through various emissions." Concerning this, we will rephrase the text: on page 1 line 3, "Reducing the anthropogenic climate impact from aviation emissions and...".

• **p** 1, **l** 3–6 : "building a climate-friendly", "for a sustainable development", "is an important approach". It makes me wonder whether this is not a too optimistic view on aviation.

Reply: We agree. The sustainable development of commercial aviation might be optimistic. However, if we want to have a sustainable development of commercial aviation, we need to have a reduction of aviation emissions and a climate-friendly air transportation system.

• **p 1, l 9** : "stable" gas. This is not precise enough.

Reply: We will delete the word "stable" in the sentence: on page 1 line 9, "CO2 is a long-lived <del>and stable</del> gas, while...".

• **p 1, l 9** : "vary regionally". I would rather use something like "inhomogeneous distribution".

Reply: We will rephrase the text: on page 1 line 9, "...non-CO2 emissions are short-lived and <del>vary regionally</del> **are inhomogeneously distributed.**"

• **p 1, l 11** : "on long time scales". I assume that the tool takes into account climate impacts on long time scale, via e.g. the CCFs. However, the tool itself is an optimization of only the flights planned within the next few days. There should be no confusion about these very different aspects.

Reply: In this sentence, we just wanted to say that AirTraf can perform "long-term" simulations, i.e. not only a few days but also more than ten years (arbitrary duration of simulations). The word "on long time scales" seems to be confusing. We will revise the text: on page 1 line 9–11, "This study introduces AirTraf (version 1.0) for climate impact evaluations that performs global air traffic simulations on long time scales, including effects of local weather conditions on the emissions." In AirTraf, we apply an optimization on a daily basis for daily changing weather situations. To investigate the mitigation gain of the climate impact by climate-optimal routing, multi-annual (long-term) simulations are required (e.g. for ten years). In the simulations over the ten years, each flight trajectory is optimized with respect to a selected aircraft routing option, considering local weather conditions. Along the optimized flight path, emissions are released. AirTraf can perform such long-term air traffic simulations with the inclusion of the on-line optimization module and the optimal routes will change day by day.

• **p 1, l 15** : were  $\rightarrow$  are (because you describe the functioning of a tool).

Reply: We will revise the text: on page 1 line 15, "Fuel use and emissions were **are** calculated by...". In the same way, we will revise the text: on page 1 line 16, "The flight trajectory optimization was **is** performed by a Genetic Algorithm...".

• **p 1, l 15** : DLR. This abbreviation should be explained.

Reply: We will revise the text: on page 1 line 15, "...and **Deutsches Zentrum für Luft- und Raumfahrt (DLR)** fuel flow method."

• **p 1, l 16–17** : "with respect to routing options" : vague.

Reply: We will revise the text: on page 1 line 16, "…performed by a Genetic Algorithm (GA) with respect to **a selected** routing option<del>s</del>."

• **p 1, l 17–18** : "two benchmark tests ... for great circle and time routing options" : sounds a bit strange → "benchmark tests ... for the great circle and time routing options".

Reply: We will revise the text: on page 1 line 17, "..., two benchmark tests were performed for **the** great circle and flight time routing options."

• **p 1, l 19** : "by other published code" : vague, and inappropriate language for an abstract.

Reply: We will revise the text: on page 1 line 19, "...calculated by other published code the Movable Type script."

• **p 1, l 20** : "optimal solution"  $\rightarrow$  "optimal solution found by the algorithm" (distinguish whether it relates to the real optimal solution, or to the best estimate found by the optimization routine).

Reply: We will revise the text: on page 1 line 20, "...the optimal solution **found by the algorithm** sufficiently converged to...".

**Page 2**

• **p 2**, **l 22** : "initial population" : as such, this is too technical for an abstract. I suggest to skip this from the abstract, or one could also choose to describe a bit better the optimization algorithm/methodology in the abstract.

Reply: Please see the reply to the referee comment: "GA algorithm."

• **p** 2, **l** 22–23 : "We found that the influence was small (around 0.01 %)" : I suggest to combine this into one sentence with the former sentence.

Reply: We will revise the sentences: on page 2 line 21–23, "The dependence of the optimal solutions on the initial populations set of solutions (called population) was analyzed and We found that the influence was small (around 0.01 %)."

• **p** 2, **l** 24 : "function evaluations", "generation sizing" : too technical for an abstract.

Reply: We will add explanations and revise the sentence: on page 2 line 23, "The trade-off between the accuracy of GA optimizations and the number of function evaluations computational costs is clarified and the appropriate population and generation (one iteration of GA) sizing is discussed."

• **p** 2, **l** 27 "one-day AirTraf simulations are demonstrated ..." : vague.

Reply: We will remove the word "one-day" in the sentence: on page 2 line 26, "Finally, <del>one-day</del> AirTraf simulations are demonstrated with...". Related to this, we will revise the text: on page 2 line 31, "The consistency check for the <del>one-day</del> AirTraf simulations...". We will also revise the text: on page 4 line 106, "In Sect. 4, <del>one-day</del> AirTraf simulations are demonstrated <del>for</del> **with** the two options **for a typical winter day** (called **one-day** AirTraf simulations) and the results are discussed."

• **p 2**, **l 27** : specific winter day  $\rightarrow$  typical winter day.

Reply: We will revise the text: on page 2 line 27, "...with the great circle and the flight time routing options for a specific **typical** winter day." In the same way, we will revise the text: on page 18 line 599, "The simulation was performed for one specific **typical** winter day..."; on page 25 line 844, "AirTraf simulations were demonstrated in EMAC (on-line) for a specific **typical** winter day...".

• **p** 2, **l** 29 : "for the two options" : it is a long time ago that these were mentioned. So maybe express them explicitly again.

Reply: We are hesitating to express them explicitly again, because the corresponding word "the great circle and the flight time routing option" are mentioned on page 2 line 27. We think that this is not far from line 29. Nevertheless, we will add the text to express the word more clearly: on page 2 line 29, "...AirTraf simulates the air traffic properly for the two **routing** options."

• **p 2**, **l 30** : for all airport pairs : too vague for an abstract.

Reply: We will revise the text: on page 2 line 30, "...for all the 103 airport pairs...".

• **p** 2, **l** 30–31 : "reflecting" local weather  $\rightarrow$  taking into account (?).

Reply: We will revise the text: on page 2 line 30, "...airport pairs, reflecting taking local weather conditions into account."

• **p** 2, **l** 31 : verified  $\rightarrow$  confirmed.

Reply: We will revise the text: on page 2 line 31, "...the one-day AirTraf simulations verified confirmed that...".

• **p** 2, **l** 32 : "comparable to reference data" : too vague.

Reply: We will revise the text: on page 2 line 31–32, "...calculated flight time, fuel consumption,  $NO_x$  emission index and aircraft weights are comparable to show a good agreement with reference data."

• **p** 2, **l** 34 : "with increasing the number " $\rightarrow$  "with the increasing number".

Reply: We will revise the text: on page 2 line 34, **"With the increasing number** of aircraft, the air traffic's contribution...".

• **p** 2, **l** 35 : "a major problem" : too vague.

Reply: We will revise the text: on page 2 line 35, "...the air traffic's contribution to climate change becomes **an** major **important** problem."

• **p** 2, **l** 35 : "At present"  $\rightarrow$  Nowadays, currently, ....

Reply: We will revise the text: on page 2 line 35, "At present Nowadays, aircraft emission...".

• **p** 2, **l** 35–37 : aircraft emission impacts contribute 4.9 % of total anthropogenic radiative forcing : skip "impacts", as radiative forcing is an impact; 4.9 → to 4.9 ; of total → "of the total".

Reply: We will revise the text: on page 2 line 35–37, "..., aircraft emission impacts (this includes still uncertain aviation-induced cirrus cloud effects) contributes approximately to 4.9 % (with a range of 2-14 %, which is a 90 % likelihood range) of **the** total anthropogenic radiative forcing...".

• **p 2, l 39** : will grow  $\rightarrow$  might grow.

Reply: We will revise the text: on page 2 line 38, "An Airbus forecast shows that the world air traffic will **might** grow...".

• **p** 2, **l** 40 : the value of 4.9 %  $\rightarrow$  a value of 4.9 %.

Reply: We will revise the text: on page 2 line 40, "..., while Boeing forecasts the a value of 4.9 % over the

same period."

• **p** 2, **l** 41 : indicates  $\rightarrow$  implies.

Reply: We will revise the text: on page 2 line 41, "This indicates implies a further increase of aircraft emissions...".

• **p** 2, **l** 41–42 : "and therefore environmental impacts from aviation increase" : try to avoid to have twice "increase" in this sentence.

Reply: We will revise the text: on page 2 line 41–42, " This indicates implies a further increase of aircraft emissions and therefore environmental impacts from aviation increase rise."

• **p** 2, **l** 42–43 : This sentence sounds more positive than one can possibly defend.

Reply: We will reply to the comment in the above section: "p 1, l 3–6".

• **p** 2, **l** 47 : contrail  $\rightarrow$  contrails.

Reply: We will revise the text: on page 2 line 47, "The emissions also induce cloudiness via the formation of contrails, contrail-cirrus...".

• **p 2**, **l 49** : depends  $\rightarrow$  depends partially.

Reply: We will revise the text: on page 2 line 49, "The climate impact induced by aircraft emissions depends **partially** on...".

• **p** 2, **l** 49–51 : What follows behind the ":" is not an explanation from what is said before ":".

Reply: We will revise the sentences: on page 2 line 49–50, "The climate impact induced by aircraft emissions depends **partially** on local weather conditions<del>:</del>. **it That is, the impact** depends on...".

• **p** 2, **l** 50 : geographic  $\rightarrow$  geographical (both are possible).

Reply: We will revise the word "geographic" into the "geographical" in the revised manuscript: on page 2 line 50, "...on geographical location (latitude and longitude) and...".

• **p** 2, **l** 51–**p**3, **l** 59 : "... and affect the atmosphere from minutes to centuries." Minutes probably refers to the time scale for disappearance of some chemical perturbations. However, every appearance (even if it is only a few minutes) of a GHG, has a century-timescale effect. Although I think I understand what the authors want to say, I think that the whole paragraph is rather inaccurate, and should be rewritten more precisely.

Reply: In this paragraph, we just wanted to focus on atmospheric composition changes, not on the climate changes, which the referee addressed. We will add the word "on the atmospheric composition" into the text to make clear what we want to say here: on page 2 line 51-53, "In addition, the impact **on the atmospheric composition** has different timescales: chemical effects induced by the aircraft emissions have a range of life-times and affect the atmosphere from minutes to centuries.  $CO_2$  has a long perturbation life-times in the order of decades to centuries."

**Page 3**

• **p** 3, **l** 61 : "150 km horizontally" : maybe distinguish two directions (is it perpendicular to the flight path, or along the flight path). Isn't this 150 km much too specific? Isn't there a very broad spectrum?

Reply: The mean length of 150 km is from Gierens and Spichtinger (2000). The study showed that: "The mean path length is about 150 km with a standard deviation of 250 km." Therefore, we will refer the original reference in the text and revise the sentence to make clear that point: on page 3 line 61, "...extend a few 100 m vertically and <del>around</del> **about** 150 km <del>horizontally</del> **along a flight path (with a standard deviation of 250 km)** with a large spatial and temporal variability (**Gierens et al., 2000,** Spichtinger et al., 2003)." This modification is also related to our reply to the comment (1) of referee #1.

• **p** 3, **l** 63 : There "are" two options ... : this sounds very optimistic.

Reply: We will revise the text: on page 3 line 63, "**The measures to counteract the climate impact induced by aircraft emissions can be classified into two categories**: technological and operational <del>approaches</del> **measures**,...".

• **p** 3, **l** 64 : "approaches"  $\rightarrow$  measures.

Reply: We will revise the word "approaches" into "measures": on page 3 line 64, "…: technological and operational approaches **measures**,…". In the same way, we will revise the word "approach" into "measure" in the manuscript: on page 1 line 6, "…is an important approach **measure** for climate impact reduction…".

• **p 3**, **l 69** : "... are optimized with respect to time and economic costs." : if both are taken into account, how are they weighted?

Reply: In this paper, we would like to show that AirTraf works well and is fit for our purpose. Particularly, the ability of the optimization module (GA) to optimize flight routes must be confirmed. For this, we tested the simple "time-optimal routing." The referee actually points at the interesting future investigation, which is far beyond the scope of this paper. Generally, airlines have own evaluation functions, such as cost index, which uses weight factors on fuel, time, etc., in order to optimize the whole aircraft operating system. This kind of data is almost impossible to get from airlines and depends on their individual strategy.

• **p** 3, **l** 69 : "fuel, crew, operating costs" : isn't fuel part of the operating costs?

Reply: We will revise the text: on page 3 line 69, "...economic costs (fuel, crew, other operating costs)...".

• **p** 3, **l** 72 : "systematic routing changes" : reading this, one gets the impression that there are different options. However, later it is reduced to just "i.e. flight altitude change". I suggest to just say "systematic flight altitude changes".

Reply: We will revise the text: on page 3 line 72, "Earlier studies investigated the effect of **systematic** routing changes, i.e. **flight altitude changes**, on the climate impact...".

• **p** 3, **l** 74 : has a strong effect on the reduction of the climate impact  $\rightarrow$  has a strong impact on climate. (From the original formulation it is not clear whether the increase or the decrease in flight altitude leads to a reduction of the climate impact.)

Reply: We understand the referee comment. Nevertheless, we are hesitating to change the text. The four studies referred here showed clearly that the changed altitude has a strong effect on the reduction of the climate impact. However, the studies were performed with respect to different flight plans, different climate impact metrics and different duration of simulations (i.e. atmospheric conditions). We think that it is not appropriate to describe whether the increase or the decrease in flight altitude leads to a reduction of the climate impact. More studies are needed before generalizing that point.

• **p** 3, **l** 74–77 : "the" climate-optimized routing → climate-optimized routing.

Reply: We will revise the text: on page 3 line 74–77, "A number of studies have investigated the potential of applying the climate-optimized routing for real flight data. Matthes et al. (2012) and Sridhar et al. (2013) addressed weather-dependent trajectory optimization using real flight routes and showed a large potential of the climate-optimized routing."

• **p** 3, **l** 79 : "the" climate sensitive regions  $\rightarrow$  climate-sensitive regions.

Reply: We will revise the text: on page 3 line 79, "...by considering regions described as the climate-sensitive regions and...".

• **p** 3, **l** 80 : "This study"  $\rightarrow$  "That study".

Reply: We will revise the text: on page 3 line 80, "This That study reported...".

• **p** 3, **l** 81 : by only small increase → by only a small increase.

Reply: We will revise the text: on page 3 line 81, "...can be achieved by only **a** small increase in economic costs...".

• **p** 3, **l** 80–81 : This study reported: "large reductions ..." → That study reported that large reductions ...

Reply: We will revise the text: on page 3 line 80–81, "<del>This</del> **That study reported that large reductions** in the climate impact of up to 25 % can be achieved by only **a** small increase in economic costs of less than 0.5%."

• **p** 3, **l** 82 : useful : is useful what one wants to express?

Reply: We just want to express that the climate-optimized routing is effective to reduce the climate impact. Therefore, we will revise the text: on page 3 line 82, "The climate-optimized routing therefore seems to be **an a useful effective** routing option **for the climate impact reduction,**...".

• **p** 3, **l** 85–86 : The current study wants apparently to investigate something (how much the climate impact of aircraft emissions can be reduced) that already has been investigated before (see lines 80–81: large reductions in the climate impact of up to 25 % can be achieved). One should be more specific of what the current study will do extra with respect to the former study.

Reply: Our final purpose (yet beyond the scope of the present manuscript) is to investigate the mitigation gain of climate-optimal routing. We would like to stress that the mere construction of climate-optimal flight trajectories for a specific weather condition is not our goal. The latter has been achieved, e.g. by Grewe et al., 2014. We eventually want to go one step further and apply an optimization on a daily basis for daily changing weather situations. To investigate then the mitigation gain, multi-annual (long-term) simulations with full feedback from the re-routed air traffic emissions are required (e.g. for ten years). In such simulations over at least the ten years, each flight trajectory is optimized with respect to a selected aircraft routing option, considering local weather conditions. The air traffic emissions are released into the ESM atmosphere and modify its chemical composition. AirTraf can perform such air traffic simulations with the inclusion of the on-line optimization module and the optimal routes will change day by day. This is an important difference to former studies.

As the referee pointed out, the subject of this paper (line 84–85) seems to be confusing. We make clear that this paper introduces the AirTraf submodel in its basic version, and technically describes and validates the various components for first, simple aircraft routings (great circle and time-optimal). Eventually, we are aiming at an optimal routing for climate impact reduction. This will be a separate study, which requires a couple of additional developments beforehand, amongst which the present study is only one of them.

Here, we will revise the sentences: on page 3, final paragraph (line 84–87), "This study aims toinvestigate how much the climate impact of aircraft emissions can be reduced by aircraft routing. Here, wepresent a new assessment platform AirTraf (version 1.0, Yamashita et al., 2015) that is a global air trafficsubmodel coupled to the Chemistry-Climate model EMAC (Jöckel et al., 2010). Figure 1 shows the researchroad map for this study (Grewe et al., 2014b) This paper presents the new submodel AirTraf (version 1.0, Yamashita et al., 2015) that performs global air traffic simulations coupled to the Chemistry-Climate model EMAC (Jöckel et al., 2010). This paper technically describes AirTraf and validates the various components for simple aircraft routings: great circle and time-optimal routings. Eventually, we are aiming at an optimal routing for climate impact reduction. The development described in this paper is a prerequisite for the investigation of climate-optimized routings. The research road map for our study is as follows (Grewe et al., 2014b): Tthe first step was to investigate...".

• **p** 3, **l** 84–87 : Do you mean by "this study" = "this manuscript"? Or is "this study" broader? After reading the manuscript, I have the impression that line 84–85 is not what is answered by this manuscript.

Reply: We agree. We will reply this point in the section above: "p 3, l 85–86."

• **p** 3, **l** 87 : The first step "is"  $\rightarrow$  The first step "was".

Reply: We will revise the text: on page 3 line 87, "The first step is was to investigate...".

• **p** 3, **l** 87–89 : The first step is to investigate specific past weather situations, in particular the climate impact of locally released aircraft emissions → The first step was to investigate the influence of specific weather situations on the climate impact of aircraft emissions.

Reply: As the referee described, this correction makes the sentence more clearly. Thank you very much. We will revise the text: on page 3 line 87–89, "...: **the first step was to investigate the influence of specific weather situations on the climate impact of aircraft emissions** (Matthes et al., 2012, Grewe et al., 2014b)."

• **p** 3, **l** 89 : "The resulting data are ..." : too vague. Maybe one could say : "This results in climate cost functions ...".

Reply: Thank you very much. We will revise the text: on page 3 line 89, "The resulting data are called **This results in climate cost functions** (CCFs, Frömming et al., 2013, Grewe et al., 2014a, Grewe et al., 2014b) that identify...".

• **p** 3, **l** 91 : Why is CO2 in this list? I can understand that the impact of adding CO2 depends on the altitude, but this comes a bit unexpected after formulating earlier that CO2 is well-mixed.

Reply: We will delete the word "CO2" in the sentence: on page 3 line 91, "...climate sensitive regions with respect to  $CO_2$ ,  $O_3$ ,  $CH_4$ ,  $H_2O$  and contrails."

• **p** 3, **l** 91 : "They are specific climate metrics, i.e. climate impact per unit of emission" → "per unit amount of emission."

Reply: We will revise the text: on page 3 line 91, "They are specific climate metrics, i.e. climate impacts **per unit amount of emission**,...".

• **p** 3, **l** 92 : "and are used ..."  $\rightarrow$  "will/might be used".

Reply: We will revise the text: on page 3 line 92, "...climate impacts **per unit amount of emission**, and <del>are</del> **will be used** for optimal aircraft routings."

**Page 4**

**p 4**, **l 92** : "In a further step, weather proxies are identified for the specific weather situations." It is not clear whether this has been done.

Reply: This has not been done. To clarify this point, we will revise the text: on page 4 line 92, "In a further step, weather proxies are **will be** identified for the specific weather situations,...".

• **p** 4, **l** 102–104 : "A benchmark test for the great circle routing option is performed and ..." : the part before and after the "and" actually express more or less the same.

Reply: As the referee noted, that part can be reduced. Therefore, we will revise the text: on page 4 line 102–104, "A benchmark test for the great circle routing option is performed and provides a comparison of resulting great circle distances are compared to with those calculated by other published code the Movable Type script (MTS, Movable Type script, 2014)."

• **p** 4, **l** 103 : "by other published code" : too vague.

Reply: We will revise the text: on page 4 line 103, "...calculated by other published code the Movable Type script (MTS, Movable Type script, 2014)." Related to this, we will also revise the text: on page 12 line 401, "...calculated with the Movable type script (MTS, Movable type script, 2014) MTS."

• p 4, l 103–104 : "Another ... also ..." : I suggest to skip one of these words.

Reply: We will remove the word "also" from the sentence. In addition, we will revise the text by considering the reply to the comment on "p 4, l 103–105": "Another benchmark test is **also** performed for the flight time-routing option. **compares**...".

• **p** 4, **l** 103–105 : I would transform this into one sentence.

Reply: We will transform this into one sentence. We will revise the text: on page 4 line 103–105, "Another benchmark test is also performed for the flight time routing option. compares the optimal solution iscompared to the true-optimal solution."

• **p** 4, **l** 105–106 : This sentence is too technical with "population" and "generation sizing".

Reply: We will add explanations to the words: on page 4 line 105, "The dependence of optimal solutions on the initial *populations* (a technical terminology set in italics is explained in the glossary in Appendix) is examined...". On page 4 line 106, "...appropriate *population* and *generation* sizing is discussed." This reply is related to the reply to "GA algorithm".

• **p 4**, **l 107** : "consistency" is too general. One has not enough background information at this point in the text to understand this.

Reply: We will rephrase the text: on page 4 line 107, "Section 5 verifies whether the consistency for the AirTraf simulations are consistent with reference data and...".

• **p** 4, **l** 108 : "states" : I suggest to use another word.

Reply: We will revise the text: on page 4 line 108, "...and Sect. 6 states describes the code availability."

• **p** 4, **l** 112–116 : This paragraph should be rewritten.

Reply: We will rephrase this paragraph (line 112–116): on page 4 line 112–116, "AirTraf was developed as asubmodel of EMAC (Jöckel et al., 2010). This is reasonable, because we perform global air traffic simulations on long time scales considering local weather conditions. Geographic location and altitude at which emissions are released should be also considered. In addition, various submodels of EMAC can be used to evaluateclimate impacts. Therefore, EMAC is a well suited development environment for AirTraf. AirTraf was developed as a submodel of EMAC (Jöckel et al., 2010) to eventually assess routing options with respect to climate. This requires a framework, where we can optimize routings everyday and assess them with respect to climate changes. EMAC provides an ideal framework, since it includes various submodels, which actually evaluate climate impact, and it simulates local weather situations on long time scales. As stated above, we were focusing on the development of this model. A publication on the climate assessment of routing changes will be published as well."

• **p** 4, **l** 112 : "reasonable" : I think this is not enough as a motivation.

Reply: We will rephrase this paragraph to make clear the motivation. Please see the reply to the comment: "p 4, l 112–116".

• **p** 4, **l** 113 : "because we perform global air traffic simulations on long time scales considering local weather conditions." : I think this is a vague argumentation.

Reply: We will rephrase this paragraph. Please see the reply to the comment: "p 4, l 112–116".

• **p 4**, **l 114** : "geographic location and altitude at which emissions are released should be also considered" : vague.

Reply: This part is already explained in Introduction: on page 2 line 49–50, "The climate impact induced by aircraft emissions depends on local weather conditions: it depends on geographic location (latitude and longitude) and altitude at which the emissions are released (except for  $CO_2$ ) and time." We will rephrase this paragraph. Please see the reply to the comment: "p 4, l 112–116".

• **p 4**, **l 115** : This is maybe the main reason why the effort is done to implement AirTraf in a climate model, and not just in a NWP, or using off-line available weather forecasts. So make this more explicit, and give examples of which climate impacts can be evaluated.

Reply: Yes. We need the framework of EMAC to assess routing options. By following the referee comment, we will rephrase this paragraph. Please see the reply to the comment: "p 4, l 112–116".

• **p 4, l 117** : Explain what "entries" are.

Reply: We will rephrase the word "entries" into "parameters" to make clear the meaning of the word: on page 4 line 117, "...AirTraf <del>entries</del> **parameters** are read in messy\_initialize,...". In addition, we will modify Fig. 2 and its caption: on page 30 in Fig. 2, "AirTraf <del>entries</del> **parameters**"; and in the caption, "...AirTraf <del>entries</del> **parameters** are input in the initialization phase."

• **p 4**, **l 121–124** : This sentence should be improved. You have to put "here PE is synonym to MPI task" possibly between brackets. I am also not sure whether "while" is the most appropriate word to use here.

Reply: As the referee noted, we will put "here PE is synonym to MPI task" between brackets. In addition, we will remove "while" and transform the sentence into two sentences: on page 4 line 121–124, "the one-day flight plan, which includes many flight schedules of a single day, is decomposed for a number of processing elements (PEs), here PE is synonym to MPI task), so that each PE has a similar work load., while a A whole flight trajectory between an airport pair is handled by the same PE." Related to this modification, we will also modify the caption of Fig. 3: on page 31 in Fig. 3, "A one-day flight plan is distributed among many processing elements (PEs) in messy\_init\_memory (blue)., while a A whole

trajectory of an airport pair is handled by the same PE...".

• **p** 4, **l** 125 : I think one should be more specific about what a "time loop" is : isn't rather meant "time step"?

Reply: We used the word "time loop" according to the following publication, which is one of the basic documents about on the ECHAM5/MESSy Atmospheric Chemistry (EMAC) model: "Jöckel, P., Sander, R., Kerkweg, A., Tost, H., and Lelieveld, J.: Technical Note: The Modular Earth Submodel System (MESSy) - a new approach towards Earth System Modeling, Atmos. Chem. Phys., 5, 433-444, doi:10.5194/acp-5-433-2005, 2005." AirTraf is developed as a submodel of EMAC. Therefore, we think that the word "time loop" is helpful for readers (specifically EMAC users) to understand the flowchart of the AirTraf.

• **p** 4, **l** 125–126 : Thus, naturally short-term and long-term simulations consider the local weather conditions for every flight in EMAC. I think this should be explained more clearly.

Reply: We will revise the sentence: on page 4 line 125–126, "Thus, <del>naturally **both**</del> short-term and long-term simulations <del>consider</del> **can take into account** the local weather conditions for every flight <del>in EMAC</del>...".

• **p** 4, **l** 126–127 : "(AirTraf continuously treats overnight flights)" : this is not logically related to the sentence it is attached to. What is meant by this? Because the weather patterns used in AirTraf are the ones at the time of take-off, it seems to me that there is no large complexity about it. Is it therefore still worth mentioning?

Reply: We agree. The one-day flight plan includes many flight schedules on a single day. Some international (long-distance) flights fly over two days. For example, NH215 departs at MUC on 21:35 and arrives at Tokyo on 15:50 + 1day. We wanted to say here that AirTraf simulates such flights correctly. Indeed, we have been asked about this issue many times so far. Therefore, we believe that it is still worth mentioning.

Further, from the comment (4) of the referee #1, we will modify the text "(AirTraf continuously treats overnight flights)" into "(AirTraf continuously treats overnight flights with arrival on the next day)." After that, the modified text will be moved from the current position to an appropriate position in the manuscript, which is related logically: on page 4 line 125, "Thus, <del>naturally</del> both short-term and long-term simulations <del>consider</del> can take into account the local weather conditions for every flight <del>in EMAC (AirTraf continuously treats overnight flights with arrival on the next day)</del>."; and on page 7 line 223, "...the Estimated Time Over (ETO, Table 2) (AirTraf continuously treats overnight flights with arrival on the next day)."

**Page 5**

• **p** 5, **l** 131–132 : What is meant by these "global fields"? Give examples.

Reply: This means "three dimensional emission fields" and we call this "global fields" in the paper. We will add the text to make clear this point: on page 5 line 131–132, "...the calculated flight trajectories and global fields **(three dimensional emission fields)** are output (Fig. 2, rose red). The results are gathered from all PEs for output-of global fields."

• **p** 5, **l** 132–134 : What is meant by the sentence "Other evaluation models ... on the climate impact"? I suggest to make this more concrete.

Reply: We just wanted to say that other objective functions (or other evaluation models) will be integrated into AirTraf in order to assess routing options on climate impact reduction. However, this is not necessary for our argument here. Therefore, we will modify the sentence: on page 132–134, "Other evaluation models, e.g. climate metric models, can easily be integrated into AirTraf and hence tThe output is will be used to eval uate the reduction potential of the routing option on the climate impact."

• **p** 5, **l** 135–136 : " $R_E = 6371$  km" : I don't know whether this level of detail should be mentioned in the manuscript.

Reply: We believe that this information is important, because great circle distances can vary considerably with differences of  $R_E$ . Concerning this issue, we will revise the caption of Table 4 from the comment (2) of the referee #2 as "…column 4 ( $d_{\text{MTS}}$ ) shows the result calculated with the Movable Type scripts (MTS), which output only integer values using the Haversine formula with a spherical Earth radius of  $R_E$  = 6,371 km."

• **p** 5, **l** 137–138 : The Mach number is a ( $\rightarrow$  "the") velocity divided by a ( $\rightarrow$  "the") speed of sound.

Reply: We will revise the text: on page 5 line 137–138, "...the Mach number is **a the** velocity divided by **a the** speed of sound."

• **p** 5, **l** 138 : "true air speed" → "the true air speed". Maybe add to the sentence : "When an aircraft flies at a constant Mach number". Isn't "vary along flight trajectories" enough? I don't think that "latitude, longitude, altitude and time" should be added. If one really wants to be more specific, I would rather add temperature and wind speed as factors modifying the true air speed and ground speed.

Reply: By following the referee comment, we will revise the text: on page 5 line 138, "Therefore When an aircraft flies at a constant Mach number, the true air speed  $V_{TAS}$  and ground speed  $V_{ground}$  vary along the flight trajectories-corresponding to a given latitude, longitude, altitude and time."

• **p** 5, **l** 142 : limits rates  $\rightarrow$  limit rates.

Reply: We will correct the word: on page 5 line 142, "...and **limits rates** of aircraft climb...".

• **p** 5, **l** 142 : Explain "semi-circular rule", and "sector demand analysis".

Reply: We will modify the words to explain them clearly: on page 5 line 142, "…such as the semi-circular rule **(the basic rule for flight level)** and limit<del>s</del> rates of aircraft climb and descent, are disregarded. However, a <del>sector demand</del> **workload** analysis of **air traffic controllers** can be performed on the basis of the output data."

• **p** 5, **l** 144 : "mention" : I do not think this is the appropriate wording.

Reply: We will revise the text: on page 5 line 144, "The following sections mention **describe** the used models briefly...".

• **p** 5, **l** 149 : What is meant by "interactions with human influences"?

Reply: This means the influence coming from anthropogenic emissions. AirTraf describes one of them. We will rephrase the text: on page 5 line 149, "...and their interaction with oceans, land and human influences **coming from anthropogenic emissions** (Jöckel et al., 2010)."

• **p** 5, **l** 153 : T42L31ECMWF-resolution  $\rightarrow$  T42L31ECMWF resolution

Reply: We will revise the word: on page 5 line 153, "...in the **T42L31ECMWF resolution**,...". On page 18 line 599, "...in the **T42L31ECMWF resolution**."

• **p** 5, **l** 159 : Can it exist out of more than one day? On page 6, line 163 : "Any arbitrary number of flight plans is applicable to AirTraf". So one can give flight plans for many days at once?

Reply: As the referee noted, this point is not clear what we mean by the phrase "one-day flight plan." As shown in Fig. 3 on page 31, the one-day flight plan, which includes many flight schedules on a single day, is used in AirTraf. This flight plan is reused for simulations longer than two days, as described on page 8 line 240. To clarify this point, we will add a short description the first time we use the phrase "one-day flight

plan": on page 4 line 121, "As shown in Fig. 3, the one-day flight plan, which includes many flight schedules of a single day, is decomposed for..." (this reply is related to the comment (3) of the referee #1).

• **p** 5, **l** 160 : of A330-301 → of an A330-301 aircraft.

Reply: We will revise the word in the revised manuscript: on page 5 line 160, "...the primary data of **an** A330-301 **aircraft** used...". The caption of Table 1 on page 51, "Primary data of **Airbus** A330-301 **aircraft** and...".

• **p** 5, **l** 162 : a departure time  $\rightarrow$  the departure time.

Reply: We will revise the word: on page 5 line 162, "...latitude/longitude of the airports, and **a the** departure time."

• **p** 5, **l** 162 : as values [-90,90] → as values in the range [-90,90].

Reply: We will add the text "in the range" in the revised manuscript: on page 5 line 162, "The latitude and longitude coordinates are given as values **in the range** [–90, 90] and...".

**Page 6**

• **p 6, l 164** : the data are required  $\rightarrow$  these data are required.

Reply: We will revise the word: on page 6 line 164, "...; the these data are required to calculate...".

• **p 6, l 165** : "As for ..." → "Concerning ...".

Reply: We will revise the text: on page 6 line 165, "As for Concerning the engine performance data,...".

• **p 6, l 166** : flows (plural) while index (singular).

Reply: Thank you so much. We will revise the text: on page 6 line 166, "...reference fuel **flows**  $f_{ref}$  (in kg(fuel)s-1) and...".

• **p 6, l 168** : What is meant by an "overall" weight factor?

Reply: The word "overall" means "passenger/freight/mail". we will add this text: on page 6 line 168, "An overall **(passenger/freight/mail)** weight load factor is also provided...". On page 51 at the line with OLF in Table 1, "ICAO overall **(passenger/freight/mail)** weight load factor **in 2008**d".

• **p** 6, **l** 171 : are described "here" step by step.

Reply: We will add the word "here" in the revised manuscript: on page 6 line 171, "The calculation procedures in the AirTraf integration are described **here** step by step."

• **p 6, l 172** : a flight status  $\rightarrow$  the flight status.

Reply: We will revise the text: on page 6 line 172, "...a the flight status of all flights is initialized...".

• **p** 6, **l** 178 : moving aircraft position  $\rightarrow$  aircraft position calculation.

Reply: We will revise the word "moving aircraft position" into "aircraft position calculation" in the revised manuscript: on page 6 line 178, "…fuel/emissions calculation, <del>moving aircraft position</del> **aircraft position cal culation** and gathering global emissions." Further, on page 30 in the Fig. 2 (bold-black box, light blue), "Move aircraft position Aircraft position calc." On page 32 in the caption of Fig. 4, "(c) Moving aircraft po

**sition aircraft position calculation."**

• **p 6, l 182–183** : differ to  $\rightarrow$  differ from.

Reply: We will revise the text: on page 6 line 182–183, "...fuel (might differ to **from** H2O, if alternative fuel options can be used), contrail and CCF**s**...".

• **p 6, l 184** : can be used  $\rightarrow$  can currently be used.

Reply: We will add the word "currently" in the revised manuscript: on page 6 line 184, "...the great circle and the flight time routing options can **currently** be used."

• **p 6, l 187** : for a selected option  $\rightarrow$  for the selected option.

Reply: We will revise the text: on page 6 line 187, "...a single-objective minimization problem is solved for **a the** selected option...".

• **p** 6, **l** 191–194 : Why adding these sentences? It makes the text confusing. In addition, it is not well defined how an optimization might work when one optimizes according to two criteria (time and cost). One should also mention then how to weight or compare both (trade-off between them).

Reply: We have a reason why we added the sentence. Here, we would like to show clearly that a time-optimal route is different from a wind-optimal route. In this paper, we optimize flight trajectories with respect to "time" by taking into account wind effects. These routes are the time-optimal routes, not the wind-optimal routes, because the objective function is different between the time-optimal and the wind-optimal routing options, as described on page 6 line 191–194. We have seen situations many times that people assumed the time-optimal route including wind effects as "the wind-optimal route." To avoid this situation, we distinguish the routes clearly.

To explain this better, we will revise the text: on page 6 line 191–196, "Generally, a wind-optimal route means an economically optimal flight route taking the most advantageous wind pattern into account. This route minimizes total costs with respect to time, **fuel** and **other** economic costs (fuel, crew and others), i.e. it has multiple objectives. On the other hand, AirTraf distinguishes will provide between the flight time and the fuel routing options separately to investigate trade-offs (conflicting scenarios) among different routing options. Thus, the time-optimal route is not always the same as the wind-optimal route." This reply is related to the reply to "p 3, 1 69".

• **p 6, l 197** : The CCF is  $\rightarrow$  The CCFs are.

Reply: We will revise the text: on page 6 line 197, "The CCFs is are provided by the...". Related to this, we will modify Fig. 2: on page 30 in Fig. 2 (light green), "CCF  $\rightarrow$  CCFs".

• **p** 6, **l** 199 : "total" climate impacts versus "some" aviation emissions : this sounds strange.

Reply: We will remove the word "total" from the text: on page 6 line 199, "…and estimate<del>s total</del> climate impacts due to some aviation emissions (see Sect. 1). Thus, the best trajectory for minimum CCFs will be calculated."

**Page 7**

• **p** 7, **l** 211 :  $n_{wp-1} \rightarrow n_{wp} - 1$ .

Reply: Thank you so much. We will correct the text: on page 7 line 211, "...the flight segment index ( $i = 1, 2, ..., n_{wp-1} n_{wp} - 1$ )."

**p** 7, l 212–213 : calculation/calculation/calculate : try to vary the wording more.

Reply: We will revise the text: on page 7 line 212–215, "Next, the fuel/emissions calculation linked to thefuel/emissions calculation module (Fig. 2, light orange) calculates fuel use,  $NO_x$  and  $H_2O$  emissions by using a total energy model based on the BADA methodology (Schaefer, 2012) and the DLR fuel flow method-(Deidewig et al., 1996, see Sects. 2.5 and 2.6 for more details) Next, fuel use,  $NO_x$  and  $H_2O$  emissions are calculated by the dedicated module (Fig. 2, light orange); this module comprises a total energy model based on the BADA methodology (Schaefer, 2012) and the DLR fuel flow method (Deidewig et al., 1996, see Sects. 2.5 and 2.6 for more details)."

• **p** 7, **l** 218–219 : corresponding to time steps  $\rightarrow$  corresponding to "the" time steps.

Reply: We will add the word "the" in the sentence: on page 7 line 218–219, "...along the flight trajectory corresponding to **the** time steps of EMAC (Fig. 4c)."

• **p** 7, **l** 219–220 : "present" and "previous" is a bit vague : isn't it the position at the beginning of a time step of EMAC, and at the end of a time step?

Reply: Thank you so much. We will revise the text: on page 7 line 219–220, "…aircraft position parameters  $pos_{new}$  and  $pos_{old}$  are introduced to indicate **a the** present **position (at the end of the time step)** and previous position **(at the beginning of the time step)** of the aircraft along the flight trajectory."

• **p** 7, **l** 220 : "a" present and previous position  $\rightarrow$  "the" present and previous position.

Reply: We will revise the text. Please see the reply to the comment above: "p 7, l 219–220".

• **p** 7, **l** 221 : by real numbers of the waypoint index  $\rightarrow$  by real numbers as a function of the waypoint index.

Reply: We will revise the text: on page 7 line 221, "They are expressed by real numbers **as a function** of the waypoint index...".

• **p** 7, **l** 224 : I would rather say : "This means that the aircraft moves 100% of the distance between *i* = 1 and *i* = 2, and 30 % of the distance between *i* = 2 and *i* = 3 in one time step."

Reply: Thank you so much. We will revise the text: on page 7 line 224, "This means that the aircraft moves **100% of the distance between** i = 1 and i = 2, and 30 % of the distance between i = 2 and i = 3 in one time step."

• **p** 7, **l** 233 : is used  $\rightarrow$  are used.

Reply: We will revise the text: on page 7 line 233, "...the coordinates of the  $(i+1)^{th}$  waypoint is are used to find the...".

• **p** 7, **l** 233 : This is a little bit inaccurate (see also Fig. 4). Assess the impact of this inaccuracy.

Reply: Unfortunately, we do not understand the referee comment. In this sentence, we describe how to gather the aircraft emissions for the case  $NO_{x, i}$ , as example. This treatment is the same for the cases  $NO_{x, i-2}$  and  $NO_{x, i-1}$  as shown in Fig. 4d on page 32, for the fraction of  $NO_{x, i-2}$ , the coordinates of the  $(i-1)^{th}$  waypoint is used to find the nearest grid point. Nevertheless, we improve the caption of Fig. 4: on page 32 in the caption of Fig. 4, "...(d) Gathering global emissions; the fraction of  $NO_{x, i}$  corresponding to the EMAC–grid–box flight segment *i* is mapped onto the nearest grid box."

• **p** 8, **l** 237 : "If  $t \ge 2$  of the day" : express this better.

Reply: We will revise the text: on page 8 line 237, "If  $t \ge 2$  of the day (i.e. **oOnce the status becomes 'in-f** light'), the departure check is false in subsequent time steps ( $t \ge 2$ ) and...".

• **p** 8, **l** 239 : without recalculating flight trajectory and fuel emissions  $\rightarrow$  without recalculating the flight trajectory or fuel emissions.

Reply: Thank you so much. We will revise the text: on page 8 line 239, "...the aircraft moves to the new air craft position without recalculating **the** flight trajectory **and or** fuel/emissions."

**p** 8, l 240–241 : "For more than two consecutive days simulations" → "For simulations longer than two days".

Reply: Thank you so much. We will revise the text: on page 8 line 240–241, "For **simulations** more **longer than two** <del>consecutive</del> **days** <del>simulations</del>, the same flight plan...".

• **p 8, l 243** : Twice "calculation".

Reply: We will remove the first "calculation" in the sentence: on page 8 line 243, "The calculation methodologies of the fuel/emissions calculation module (Fig. 2, light orange) are described."

• **p 8, l 246** : are used  $\rightarrow$  is used.

Reply: Thank you so much. We will revise the word "are" into "is" in the revised manuscript: on page 8 line 246, "A total energy model based on the BADA methodology and the DLR fuel flow method are is used."

• **p 8, l 246–247** : the first trip fuel estimation  $\rightarrow$  a first trip fuel estimation.

Reply: We will correct the text: on page 8 line 246–247, "The fuel use calculation consists of the following two steps: the **a** first rough trip fuel estimation and...".

• **p** 8, **l** 247 : the second fuel calculation : a bit vague. Maybe mention that it is more detailed.

Reply: We will add the word "detailed" in the text: on page 8 line 247, "...the **a** first rough trip fuel estimation and the second **detailed** fuel calculation...". Related to this issue, we will add the word "detailed" into the text in Fig. 2 (dashed box, light orange): on page 30, "2nd **detailed** fuel calc.".

• **p** 8, **l** 256 : mean flight altitude of the flight  $\rightarrow$  mean altitude of the flight.

Reply: We will remove the first "flight" from the sentence: on page 8 line 256, " $F_{BADA}$  is calculated by inter polating the BADA data (assuming nominal weight) to the mean flight altitude of the flight...".

• **p 8, l 260** : it is assumed as  $\rightarrow$  it is assumed to be.

Reply: Thank you so much. We will revise the text: on page 8 line 260, "It is assumed as to be 3 % of the  $FUEL_{trip...}$ ".

**Page 9**

• **p** 9, **l** 274–275 : "For an aircraft in cruise ..." : express this better.

Reply: Please see the reply to the referee comment: "Sensitivity (1)."

• **p** 9, line 276–278 : One should have a "," or a "." after most of the formula.

Reply: As the referee pointed out, we will recheck the all equations and add "," or "." after most of them. We will reply to this issue in the section of "Mathematical formulas (3)."

• **p** 9, line 280 : The numerical value of  $\rho_i$  is not given in Table (2) (as for *S*,  $C_{D0}$  and  $C_{D1}$  in Table 1).

Reply: The referee is right. We will revise and add the text: on page 9 line 280, "The performance parameters (S,  $C_{D0}$  and  $C_{D2}$ ) are given in Table 1, and the air density  $\rho_i$  is the air density (Table 2) are given in Tables 1 and 2. and  $V_{\text{TAS},i}$  is calculated at every waypoint (Table 2)."

• **p** 9, **l** 281 : a fuel flow  $\rightarrow$  the fuel flow.

Reply: We will revise the text: on page 9 line 281, "...and a the fuel flow of the aircraft...".

• **p** 9, **l** 282 : I suggest to skip "for jet aircraft".

Reply: We will skip the text "for jet aircraft" in the sentence: on page 9 line 282, "...calculated assuming a cruise flight for jet aircraft:".

• **p** 9, **l** 283–284 : "," after the equations.

Reply: We will add "," after Eqs. (7) and (8). We will reply to this issue in the section of "Mathematical formulas (3)."

• **p** 9, **l** 287 : Oneday : I suggest to find another name for this variable in the manuscript. In addition, its units in Table 1 should be "sec day-1".

Reply: We agree. We will change the name for the variable "Oneday" into the "SPD" (the Seconds Per Day) throughout the revised manuscript: Eq. (9) on page 9 line 287, "FUEL*i* =  $F_{cr,i}$  ( $ETO_{i+1} - ETO_i$ ) <del>Oneday</del> **SPD**". Further, on page 9 line 288, "...is converted into seconds by multiplying <del>Oneday</del> **with Seconds Per Day** (**SPD**, Table 21)." On page 12 line 383–385 in Eqs. (26) and (27), " $V_{ground,i-1} \times$ <del>Oneday</del> **SPD** (denominator)" and " $FT = (ETO_{nwp} - ETO_1) \times$ <del>Oneday</del> **SPD**." On page 51 in Table 1, "(Parameter) <del>Oneday</del> **SPD**; (Value) 86,400; (Unit) s **day**-1; (Description) Time (Julian date) × <del>Oneday</del> **SPD** = Time (s)." On page 52 in Table 2, description of row 15, " $FT = (ETO_{nwp} - ETO_1) \times$ <del>Oneday</del> **SPD**."

• **p 9, l 289** : "reflects" → "incorporates" or "is impacted by".

Reply: We will revise the text: on page 9 line 289, "The FUELi reflects incorporates the tail/head winds effect...".

• **p 9, l 290** :  $(m) \rightarrow (m_i)$ .

Reply: We will revise the text: on page 9 line 290, "The relation between the FUELi and the aircraft weight  $(\mathbf{m}_i)$  is...".

• **p** 9, **l** 294 : next to the last  $\rightarrow$  at the one but last.

Reply: Thank you so much. We will revise the text: on page 9 line 294, "...the aircraft weight next to the last at the one but last waypoint...".

• **p** 9, **l** 296–297 : I do not think this last sentence gives new information. Or formulate it nicer.

Reply: We agree. We will remove the last sentence in the revised manuscript: on page 9 line 296–297, "As the aircraft weight is pre-calculated in this module, it reduces during the flight as fuel is burnt, corresponding to the time steps of EMAC."

**Page 10**

•  $p \ 10, l \ 302$  : first  $\rightarrow$  First.

Reply: We will revise the text: on page 10 line 302, "The calculation procedure follows four steps: **fF**irst, the reference fuel flow...".

• **p** 10, **l** 310–311 : corresponding sea level values  $\rightarrow$  corresponding values at sea level.

Reply: Thank you so much. We will revise the text: on page 10 line 310–311, " $P_0$  and  $T_0$  are the corresponding sea level values at sea level...".

• **p 10, l 314–315** : "," after equations.

Reply: We will add "," after Eqs. (14) and (15). We will reply to this issue in the section of "Mathematical formulas (3)."

• **p 10, l 327** : "... and  $q_i$  is the specific humidity at  $h_i$  ": mention units of  $q_i$  (kg kg-1, g kg-1, ...).

Reply: We will add the unit in the sentence: on page 10 line 327, "...and  $q_i$  (in kg(H2O)(kg(air))-1) is the specific humidity at  $h_i$ ...".

• **p 10, l 329** : pre-calculated  $\rightarrow$  calculated.

Reply: We will modify the word: on page 10 line 329, "...using the pre-calculated FUELi...".

• **p 10, l 330–331** : "," after equations. I do not think it is a good idea to have variables whit names as NOx,i and H2Oi. I would rather use names like *m*NOx.

Reply: We will add "," after Eqs. (19) and (20). We will reply to this issue in the section of "Mathematical formulas (3)." Further, we understand the referee comment. Nevertheless, we are hesitating to change the variable names, because "*m*" is already used for the aircraft weight, as described on page 9 line 290. Maybe the names are not the best ones, however, we think that the "NO*x*,*i*" and "H2O*i*" show clearly that these emissions are calculated for the *i*th flight segment.

**Page 11**

**p 11, l 339** : one-day  $\rightarrow$  one day of.

Reply: From the reply to the referee comment on "p2, line 27," we will define the word "one-day AirTraf simulation": on page 4 line 106, "In Sect. 4, <del>one-day</del> AirTraf simulations are demonstrated <del>for</del> **with** the two options **for a typical winter day (called one-day AirTraf simulations)** and the results are discussed." Therefore, we will also use the word here.

• **p 11, l 343** : works  $\rightarrow$  works only.

Reply: We will add the word "only" in the sentence: on page 11 line 343, "The current aircraft routing module (Fig. 2, light green) works **only** with respect to the great circle and...".

• **p** 11, **l** 351 : arctan, sin, cos, ... should not be italic.

Reply: We will modify the all names of trigonometric formulas into normal straight letters in the revised manuscript. We will reply to this issue in the section of "Mathematical formulas (4)."

• **p 11, l 351** : "," after equation.

Reply: We will add "," after Eq. (21). We will reply to this issue in the section of "Mathematical formulas (3)."

• **p 11, l 362** : Why mentioning "km" here? Better to write on line 355 : *di* (km).

Reply: The "km" is described here for the flight altitude " $h_i$ " (not for the great circle distance  $d_i$ ), because Table 2 shows the unit of h is "m". To clarify this, we will add the text in the sentence: on page 11 line 362, "...( $h_i$  is used in km in Eqs. (22) and (23)) and...".

• **p 11, l 363** : i.e. the  $\rightarrow$  i.e.

Reply: We will remove the word "the" in the sentence: on page 11 line 363, "…hence the great circle distance between airports, i.e. <del>the</del>…".

• **p 11, l 365** : "based on Polar coordinates"? Explain this better.

Reply: We think that the word "based on" seems to be confusing. We will revise the text: on page 11 line 365, "...by linear interpolation based on in Polar coordinates."

• **p 11, l 365** : therefore  $\rightarrow$  in that case.

Reply: We will revise the word "therefore" into "in that case" in the revised manuscript: on page 11 line 365, "...based on in Polar coordinates. Therefore In that case,...".

**Page 12**

• **p 12, l 370** : of the  $i^{th}$  waypoint  $\rightarrow$  at the  $i^{th}$  waypoint.

Reply: We will change the word "of" into "at" in the revised manuscript: on page 12 line 370, "...the true air speed  $V_{\text{TAS}}$  and the ground speed  $V_{\text{ground}}$  of **at** the  $i^{\text{th}}$  waypoint are calculated...".

• **p** 12, **l** 371–372 : "," after equations.

Reply: We will add "," after Eqs. (24) and (25). We will reply to this issue in the section of "Mathematical formulas (3)."

• **p 12**, **l 374** : where *M* is "the" Mach number.

Reply: We will add the word "the" in the sentence: on page 12 line 374, "...where *M* is **the** Mach number,...".

• **p 12**, **l 378–379** : Although it is mentioned that *V*TAS, *V*wind and *V*ground are scalars, Eq. (25) on line 372 is actually a vector equation.

Reply: As described on page 12 line 377–379, the flight direction is firstly calculated for every flight segment. Thereafter, the values of  $V_{\text{TAS},i}$   $V_{\text{wind},i}$  and  $V_{\text{ground},i}$  "corresponding to the flight direction" are calculated. For example,  $V_{\text{ground},i}$  is a component of the wind vector along the flight direction (i.e. scalar value). Therefore, Eq. (25) on line 372 is a scalar equation.

• **p 12, l 386** : "reflects" : this is not the only aspect which is reflected. I suggest to use "incorporates".

Reply: Thank you so much. We will revise the text: on page 12 line 386, "...and  $\text{ETO}_i$  reflects incorporates the influence of tail/head winds...". In the same way, we will revise the text: on page 21 line 700, "..., which reflects incorporates the influences of both  $V_{\text{TAS}}$  and winds...".

• **p 12, l 390** : for the five  $\rightarrow$  for five.

Reply: We will revise the text: on page 12 line 390, "Great circles were calculated for the five representative routes...".

• **p 12, l 393–395** : 180 → 180° (while "deg" on line 397).

Reply: We will revise the sentence: on page 12 line 393–395, "...the difference in longitude between them was  $\Delta \lambda_{airport} < 180^{\circ}$  (in **deg**); R2 consisted of an airport pair in the northern hemisphere (HND-JFK) with  $\Delta \lambda_{airport} > 180^{\circ}$  (discontinuous longitude values...".

• p 12, l 398 : Missing deg?

Reply: Thank you so much. We will revise the sentence: on page 12 line 397–398, "..., where  $\Delta \lambda_{airport} = 0^{\circ}$  and the difference in latitude was  $\Delta \phi_{airport} /= 0^{\circ} \frac{deg}{deg}$ ; and R5 was another special route with  $\Delta \lambda_{airport} /= 0^{\circ} \frac{deg}{deg}$ ; and R5 was another special route with  $\Delta \lambda_{airport} /= 0^{\circ} \frac{deg}{deg}$ ; and R5 was another special route with  $\Delta \lambda_{airport} /= 0^{\circ} \frac{deg}{deg}$ ; and R5 was another special route with  $\Delta \lambda_{airport} /= 0^{\circ} \frac{deg}{deg}$ ; and R5 was another special route with  $\Delta \lambda_{airport} /= 0^{\circ} \frac{deg}{deg}$ ; and R5 was another special route with  $\Delta \lambda_{airport} /= 0^{\circ} \frac{deg}{deg}$ ; and R5 was another special route with  $\Delta \lambda_{airport} /= 0^{\circ} \frac{deg}{deg}$ ; and R5 was another special route with  $\Delta \lambda_{airport} /= 0^{\circ} \frac{deg}{deg}$ ; and R5 was another special route with  $\Delta \lambda_{airport} /= 0^{\circ} \frac{deg}{deg}$ ; and R5 was another special route with  $\Delta \lambda_{airport} /= 0^{\circ} \frac{deg}{deg}$ ; and R5 was another special route with  $\Delta \lambda_{airport} /= 0^{\circ} \frac{deg}{deg}$ ; and R5 was another special route with  $\Delta \lambda_{airport} /= 0^{\circ} \frac{deg}{deg}$ ; and R5 was another special route with  $\Delta \lambda_{airport} /= 0^{\circ} \frac{deg}{deg}$ ; and R5 was another special route with  $\Delta \lambda_{airport} /= 0^{\circ} \frac{deg}{deg}$ ; and R5 was another special route with  $\Delta \lambda_{airport} /= 0^{\circ} \frac{deg}{deg}$ ; and R5 was another special route with  $\Delta \lambda_{airport} /= 0^{\circ} \frac{deg}{deg}$ ; and R5 was another special route with  $\Delta \lambda_{airport} /= 0^{\circ} \frac{deg}{deg}$ ; and R5 was another special route with  $\Delta \lambda_{airport} /= 0^{\circ} \frac{deg}{deg}$ ; and R5 was another special route with  $\Delta \lambda_{airport} /= 0^{\circ} \frac{deg}{deg}$ ; and R5 was another special route with  $\Delta \lambda_{airport} /= 0^{\circ} \frac{deg}{deg}$ ; and R5 was another special route with  $\Delta \lambda_{airport} /= 0^{\circ} \frac{deg}{deg}$ .

• **p 12, l 399** : ";"  $\rightarrow$  ",".

Reply: We will modify the text: on page 12 line 399, "...as follows: M = 0.82;,  $h_i = 0,...$ ".

**Page 13**

**p 13, l 403** : varying *n*wp in "the range" [2, 100].

Reply: We will add the text "the range" in the revised manuscript: on page 13 line 403, "... $n_{wp}$  was analyzed varying  $n_{wp}$  in **the range** [2, 100]."

• **p 13, l 404** : and the MTS  $\rightarrow$  and MTS.

Reply: We will delete the word "the" in the sentence: on page 13 line 404, "...by Eqs. (22) and (23) and the MTS."

• **p** 13, **l** 406 : I do not think that  $\Delta d_{eq^{23},eq^{22}}$ , etc. are appropriate choices for variable names. As these are difference, I think they should not not have a specific variable name attributed.

Reply: We understand the referee comment. Nevertheless, we are hesitating to change those variable names. We define the variable name for a flight distance as "d", as shown in Table 2, and we use the variable "d" consistently in the manuscript: on page 11 Eqs. (22) and (23), on page 15 Eq. (28), etc. We think that the current expressions make sense. This reply is related to the reply to "5 Comments on tables, Table 4."

• p 13, l 409–410 : "shows" versus "showed".

Reply: We will revise the text: on page 13 line 409–410, "Figure 6 shows the result of the sensitivity analysis of  $n_{wp}$  on the great circle distance. The results **showed** that the distance...".

• **p** 13, **l** 413 : I would not call it linear interpolation : one goes straight whereas the other follows an arc. Shouldn't you also add that  $n_{wp}$  maybe should depend on the length of the flight?

Reply: We will remove the word "linear interpolation" in the sentence. This is not necessary for our argument here: on page 13 line 413, "...when using fewer  $n_{wp}$ , as a result of the linear interpolation." The referee actually points out the important issue. However, we think that it is more important for readers (specifically AirTraf users) to show a criteria to use Eq. (23). For this, we describe as: on page 13 line 414, "Therefore,  $n_{wp} \ge 20$  is practically desired for the use of Eq. (23)."

• **p 13, l 417** : with respect to the flight time routing option  $\rightarrow$  with respect to the flight time.

Reply: We will revise the text: on page 13 line 417, "The flight trajectory optimization with respect to the flight time routing option was...".

• **p 13, l 418** : algorithms  $\rightarrow$  algorithm.

Reply: We will correct the word: on page 13 line 418, "..., which is a stochastic optimization algorithms."

• **p 13, l 422** : The ARMOGA  $\rightarrow$  ARMOGA.

Reply: We will revise the text: on page 13 line 422, "The ARMOGA will be implemented...".

• **p 13, l 424–425** : With a routing option  $\rightarrow$  For each routing option (except ...). I also suggest to skip "on the selected routing" in the second part of the sentence.

Reply: We will revise the sentence: on page 13 line 424–425, "<del>With a</del> **For each** routing option (except for the great circle routing option), a single-objective optimization problem on the selected routing option is solved."

• **p 13, l 427** : Explain what an objective function in this context is.

Reply: The word "objective function" means "evaluation function." The word "objective function" is the technical term (commonly used in GA-optimization terminology). Therefore, we will revise the sentence: on page 13 line 427, "Therefore, various objective evaluation functions (called objective functions) can easily be adapted...".

• **p 13, l 432-433** : "Is called "an" optimal solution" and "is called "the" true-optimal solution".

Reply: We will revise the sentence: on page 13 line 432–433, "A solution found in GA is called **an** optimal solution, whereas a solution having the theoretical-optimum of the objective function is called **the** true-optimal solution."

• **p 13, l 434** : Say what is meant by converge : larger initial population, or just more generations?

Reply: The word "converge" means "becomes close to" in this context. As described on page 13 line 432–433, there are two solutions: an optimal solution and the true-optimal solution. When we solve an optimization problem, we expect that the optimal solution (our solution) "converges" to the true-optimal solution by optimization algorithms. This is what we wanted to say here.

• **p 13**, **l 435** : Will every flight have the same size for its initial population, and the same number of generations? Is that independent of the length of the flight?

Reply: This paper aims to confirm the ability of the optimization module (GA) to optimize flight routes. Therefore, we solved the simple time-optimal optimization problem using the common optimization setup (the same size for initial populations and the same number of generations for every flight). We understand that the referee pointed out an important issue. However, this is beyond the scope of this paper. If we could choose the setup individually for every flight, the computational requirements for the trajectory optimization
could probably be decreased. However, it is difficult to find an appropriate GA setup for every flight before solving the optimization problem. As the referee noted, the flight length can be used to adjust the population size and the number of generations for a flight. On the other hand, if a day shows complicated weather situations, GA needs a larger population size and more generations to converge. This issue will be one of our future investigations.

**Page 14**

• **p 14, l 440–441** : I do not think that "definitions" is the appropriate word to be used here.

Reply: We believe that the word "definitions" is appropriate here. To solve an optimization problem, firstly, one has to define the optimization problem itself concerning variables, ranges of variables, evaluation functions, constraints, etc. Thereafter, one can solve the problem. On page 14, Sect. 3.2.2 describes the definitions of the flight trajectory optimization, which we solve here.

• **p 14, l 441** : of objective functions  $\rightarrow$  of the objective function.

Reply: We will revise the text: on page 14 line 441, "..., the definition of **the** objective function and the genetic operators."

• **p** 14, **l** 444 : used interchangeably to mean a flight trajectory  $\rightarrow$  used interchangeably to flight trajectory.

Reply: We will revise the text: on page 14 line 444, "...the term is used interchangeably to mean a flight trajectory...".

• **p 14, l 445** : *n*dv = 11 should not be here.

Reply: We will remove the word " $n_{dv} = 11$ " in the sentence and modify the text: on page 14 line 445, "...the design variable index j ( $j = 1, 2, \dots, n_{dv}; n_{dv} = 11$ ),...". On page 15 line 487, "...where  $n_{dv} = 11$ ,  $d_i$  and  $V_{ground,i}$  are calculated...".

• **p 14, l 456** : centering  $\rightarrow$  centered.

Reply: We will revise the text: on page 14 line 456, "...domains centering centered around the central points...".

• **p 14, l 463–464** : how are these waypoints calculated? Will the arc lengths be equal?

Reply: We reply to this issue in the section of "CP in trajectories (3)."

• p 14, l 458–459 and 470–471 : "GA provided the values" : Do you mean already the final optimal values?

Reply: Here, we just want to say that the values of the eleven design variables are provided by the GA optimization process. In other words, one does not have to determine the values. In fact, the sentence on page 15 line 479–480 says, "The initial *population* operator (Fig. 2, dark green) provides initial values of the eleven design variables as random numbers...". Naturally, GA provides not only initial values, but also the final optimal values regarding the design variables.

• **p 14, l 462** : Explain a little bit more a B-spline curve.

Reply: We will add the text to specify the curve: on page 14 line 462, "...trajectory is represented by a B-spline curve (**3rd-order**) with the three CPs...". On page 15 line 474, "...trajectory is also represented by a B-spline curve (**3rd-order**) with the...".

• **p 14, l 464** : Are the waypoints on the B-spline curve still equidistant?

Reply: No. The referee is right. We explain this issue in the sections of "CP in trajectories (3) and (4)." Here we will modify the text: on page 14 line 464, "To generate the waypoints at even intervals same number of waypoints between the CPs,  $n_{wp}$  was calculated...". Related to this issue, we will delete the text: on page 7 line 206, "...the trajectory consists of waypoints generated at even intervals along the trajectory, and flight segments...".

• **p 14, l 461 and 472** : "Here *x*1 , ... indicate longitudes/latitudes/altitude values". Shouldn't this be mentioned earlier in the paragraphs, i.e. on lines 452 and 466?

Reply: The referee is right. We will revise the manuscript: on page 14 line 461, "Here  $x_4$ ,  $x_3$  and  $x_5$  indicate longitudes, while  $x_2$ ,  $x_4$  and  $x_6$  indicate latitudes.", and on line 452, "...as shown in Fig. 7 (bottom).  $x_1$ ,  $x_3$  and  $x_5$  indicate longitudes, while  $x_2$ ,  $x_4$  and  $x_6$  indicate latitudes." On page 14 line 472, "Here  $x_7$  to  $x_{41}$  indicate altitude values.", and on line 466, "...were used (Fig. 7, top). Here  $x_7$  to  $x_{11}$  indicate altitude values."

**Page 15**

• **p** 15, **l** 477 : where longitude-coordinate of waypoints  $\rightarrow$  where "the" longitude of the waypoints.

Reply: We will modify the sentence in the revised manuscript. Please see the reply to the referee comment on the "CP in trajectories (4)."

• **p 15, l 476–478** "where longitude-coordinate of waypoints is the same for the two curves." Is this true in the example here? The lon-lat curve contains 3 CPs and thus 4 intervals. The the lon-altitude curve contains 5 CPs and 6 intervals. The number of waypoints is 101, so 100 intervals. This is however not a multiple of 6, so I don't see that the longitude of the waypoints for both B-spline curves are automatically identical.

Reply: This is true. The longitude of the waypoints for both B-spline curves are identical. A flight trajectory is also represented by a B-spline curve (the lon-altitude curve) and waypoints are generated along the curve. These waypoints are tentative points (>  $n_{wp}$ ). And then, we create actual waypoints on the lon-altitude curve, by interpolating the lon-altitude curve to the longitude-coordinate of the lon-lat curve. We modify the related sentences in the section of "CP in trajectories (4)."

• **p 15**, **l 479** : "provides initial values by random numbers" : this is too cryptic.

Reply: As described on page 13 line 418, GA is a stochastic optimization algorithm. Thus, the optimization proceeds using random numbers. Maybe the current sentence is a little bit unclear, therefore we will modify the sentence in the revised manuscript: on page 15 line 479, "The initial *population* operator (Fig. 2, dark green) provides initial values of the eleven design variables by random numbers at random within the lower/upper bounds described above,...".

• **p 15**, **l 481** : "The operator creates divers solutions defined by a fixed population size *np*.": This is a complicated way to say: "The operator creates *np* different solutions (where *np* is the population size)."

Reply: We agree. We will revise the text: on page 15 line 480–481, "The operator creates diverse solutionsdefined by a fixed population size  $n_p$   $n_p$  different solutions (where  $n_p$  is the *population* size)...".

• **p 15**, **l 481** : "a random set" : do you mean the random set which is just described (then I suggest to use "the"), or is it even another random set? I would put the sentence "GA starts its search with a random set of solutions (population approach)" at the beginning of the paragraph.

Reply: "a random set" means the random set which is already described. We will move the sentence at the beginning of the paragraph (in this case, the word "a random set" is used). Finally, we will revise the

sentence: on page 15 line 479 (at the beginning of the paragraph), "**GA starts its search with a random set of solutions** (*population* **approach**). The initial *population* operator...".

• **p 15, l 483** : By summing the flight time for flight segments  $\rightarrow$  by summing the flight time over all flight segments.

Reply: We will revise the text: on page 15 line 482, "...for each of the solutions by summing the flight time for over all flight segments...".

• **p 15, l 483–484** : "The .. optimization solved here" : too cryptic and vague.

Reply: We will revise the text: on page 483–484, "The single-objective optimization **problem on the flight time** solved here is **can be written** as follows:".

• **p 15, l 485** : "Minimize" and "Subject to" should not be italic.

Reply: We will modify the words "Minimize" and "Subject to" with straight letters in the revised manuscript: on page 15 line 485, "*Minimize* Minimize" and "*Subject to* Subject to".

• **p 15, l 490** : What is meant by "solutions that dominate it"?

Reply: This expression shows an inferior-to-superior relationship among solutions, and is commonly used in GA optimization terminology. In optimization problems, for example, if a solution A is superior to a solution B on an objective function, we can say that the solution A dominates the solution B.

• **p 15, l 489–491** : Why is "rank" written in italic, but "fitness" not?

Reply: We will add the glossary in Appendix and refer the word "rank" in italics in the revised manuscript: on page 15 line 489–492, "A *rank* of a...was computed by 1/*rank*. A solution...smaller *rank* value...". This reply is related to the reply to "GA algorithm".

• **p 15**, **l 493** : made  $\rightarrow$  makes (because "are identified" on line 488).

Reply: We will revise the text: on page 15 line 493, "...Sampling Selection (Baker, 1985) made makes duplicates...".

• **p 15, l 492** : What is meant by a "mating pool"?

Reply: We will add the glossary in the revised manuscript to explain the technical term "mating pool". Please see the reply to the referee comment on the "GA algorithm."

• **p 15, l 500** : "This operator was applied to each design value." : Isn't this said already in the sentence before?

Reply: By following the referee comment, we will delete the sentence and add the word " $n_{dv} = 11$ " into the previous sentence: on page 15 line 500–501, "...with  $\gamma = (1 + 2\alpha)u_1 - \alpha$  and j varies in  $[1, n_{dv}]$  ( $n_{dv} = 11$ ). This operator was applied to each design variable;  $n_{dv} = 11$ ."

• **p 15**, **l 504** : "added a disturbance to the child solution." : It does if for both child solutions I presume.

Reply: The referee comment is correct. We will correct the word "the child solution" into "the child solutions": on page 15 line 504, "...added a disturbance to the child solutions by...".

• **p 16, l 515** : the population of "the" solutions  $\rightarrow$  the population of solutions.

Reply: We will remove the word "the" in the revised manuscript: on page 16 line 515, "…it is expected that the *population* of the solutions is…".

• **p 16, l 517** : "an optimal solution is output." : How is that solution found based on the last generation?

Reply: We will add the text to inform on how the final solution is obtained from the optimization. Please see the reply to the referee comment on the "GA algorithm."

• **p 16, l 518** : "corresponding to the routing option": I don't think this has to be repeated here.

Reply: We will remove the word "corresponding to the routing option" in the revised manuscript: on page 16 line 517-518, "..., GA quits the optimization and an optimal solution **showing the best** *f* **of the whole** *generation* is output corresponding to the routing option."

• **p 16, l 518** : "the best" : one cannot guarantee that it is the best I think.

Reply: By following the referee comment, we will change the word "the best" into "the superior" in the revised manuscript: on page 16 line 518, "The optimal solution has the **best superior** combination of the…".

• **p 16, l 519** : "naturally" : is this the appropriate wording?

Reply: We will revise the sentence: on page 16 line 519, "Naturally, tThe flight properties of the optimal solution are **also** available...".

• **p 16, l 521–522** : can be applied to any routing option (I thought that was not possible yet in version 1.0?)  $\rightarrow$  could.

Reply: We agree. We will correct the word "can" into the "could" in the revised manuscript: on page 16 line 521–522, "The flight trajectory optimization methodology described here <del>can</del> **could** be applied to any routing option…".

• **p 16, l 529** : "As  $V_{\text{TAS}}$  and  $V_{\text{ground}}$  were set to 898.8 km h-1" : Isn't it better to mention first explicitly that we have set  $V_{\text{wind}} = 0$ , and from that it follows that  $V_{\text{TAS}}$  and  $V_{\text{ground}}$  are 898.8 km h-1 (and not set).

Reply: By following the referee comment, we will revise the sentence: on page 16 line 529, " $V_{wind}$  was set to **0** km h-1 (no-wind conditions); As  $V_{TAS}$  and  $V_{ground}$  were set to 898.8 km h-1 (constant) under no-wind-conditions,. Hence, the  $f_{true}$  equals the flight time along the great circle from MUC to JFK at FL290:...".

• **p 16**, **l 531** : Maybe one should say why flying at FL290 will be faster than at other altitudes. I assume that this depends on the value of *T*. Are the initial and final points at FL290? Mention that *M* = 0.82.

Reply: To show clearly why flying at FL290 will be faster than at other altitudes, we will add the text in the revised manuscript: on page 16 line 530–531, "... $f_{true}$  equals the flight time along the great circle from MUC to JFK at FL290 (having its minimum  $d_i$  in the range of [FL290, FL410]):  $f_{true} = 25,994.0$  s...".

In this benchmark test (off-line),  $V_{wind} = 0 \text{ km h}^{-1}$  and  $V_{TAS} = V_{ground} = 898.8 \text{ km h}^{-1}$  were set, as described on page 16 line 529. Hence, the results do not depend on the values of *T* and *M* (see Eqs. (24) and (25)).

In addition, the initial and final points were at FL290. Table 5 summarizes the calculation conditions for the test. In Table 5, the altitudes of departure (MUC) and arrival airport (JFK) are described as, "alt. = FL290."

• **p 16, l 537** : total 1000 independent  $\rightarrow$  a total of 1000 independent.

Reply: We will revise the text: on page 16 line 537, "...i.e. a total of 1,000 independent GA simulations...".

• **p 16, l 532–538** : Isn't the first experiment also included in the second setup?

Reply: Yes. To clarify this point, we will modify the text: on page 16 line 532–538, "With regard to the dependence of the optimal solutions on initial populations, 10 independent GA simulations from differentinitial populations were performed. In these simulations, both  $n_p$  and  $n_g$  were set to 100, while othercalculation conditions were set as shown in Table 5. In the same way, to discuss an appropriate  $n_p$  and  $n_g$ sizing, 10 independent GA simulations from different initial **populations** were performed for each combination of  $n_p$  (10, 20,…, 100) and  $n_g$  (10, 20,…, 100), i.e. total 1,000 independent GA simulations were performed. Other calculation conditions were also set as shown in Table 5." Related to this modification, we will add the text: on page 17 line 559, "...the 10 independent GA simulations **from different initial populations** with  $n_p = 100$  and  $n_q = 100$ ."

**Page 17**

• **p** 17, **l** 540 : generation number  $n_g \rightarrow$  number of generations  $n_g$ .

Reply: We will revise the text: on page 17 line 540, "The influence of the **population** size  $n_p$  and the **number** of **generations** number  $n_g$ ...". In the same way, we will revise the manuscript as follows: on page 16 line 517, "...computed for a fixed **number of generations** number  $n_g$ ,...". On page 35 in the caption of Fig. 8, "...and the number of **generations** number  $n_g$ ." On page 35 in the x-axis label of Fig. 8, "generation number  $n_g$  number of **generations**  $n_g$ ". On page 36 in the caption of Fig. 9, "...and the number of **generations**  $n_g$ ". On page 36 in the caption of Fig. 9, "...and the number of **generations**  $n_g$  is 100." On page 38 in the caption of Fig. 11, "...and the number of **generations**  $n_g$ ". On page 39 in the x-axis label of Figs. 12a and 12b, "generation number  $n_g$  number of **generations**  $n_g$ ". On page 39 in the caption of Fig. 12, "...and the number of **generations**  $n_g$ ". On page 39 in the caption of Fig. 12, "...and the number of **generations**  $n_g$ ". On page 36 in Table 7, "Generation number  $n_g$ ." On page 44 in the caption of Fig. 17, "...and the number of **generations**". On page 8 (Supplementary material) in the caption of Table 51, "...and the number of **generations**  $n_g = 100$   $n_g$  is 100." On page 9 (Supplementary material) in the caption of Table 52, "...and number of **generations**  $n_g$ ".

• **p 17, l 541** : Is "confirmed" the appropriate wording?

Reply: We will modify the word: on page 17 line 540–541, "...the convergence properties of GA was confirmed examined."

• **p** 17, **l** 542 : sufficiently come close  $\rightarrow$  come sufficiently close.

Reply: We will revise the text: on page 17 line 542, "...the optimal solutions sufficiently come sufficiently close to the  $f_{true}$ ...".

• **p 17, l 542, 543, 545** : the  $f_{true} \rightarrow f_{true}$ .

Reply: We will revise the word: on page 17 line 542, "...close to the  $f_{true}$  with increasing..."; on page 17 line 543, "...closest flight time to the  $f_{true}$  was..."; and on page 17 line 545, "...between the  $f_{best}$  and the  $f_{true}$  was...". In the same way, we will correct the word "the  $f_{true}$ " in the revised manuscript: on page 16 line 530, "...the  $f_{true}$  equals the flight time..."; on page 17 line 565, "0.01 % of the  $f_{true}$ "; and on page 17 line 566, "0.001 % of the  $f_{true}$ ".

• **p** 17, **l** 545 :  $\Delta f$  : you do not need an extra variable name for something you express only once.

Reply: We understand the referee comment. Nevertheless, we are hesitating to remove the variable name. We use the variable " $\Delta f$ " consistently in the manuscript to express the difference in flight time: on page 17 line 564–565; on page 18 line 575, 581, 588–590; on page 39 in the caption of Fig. 12; on page 8 (Supplementary material) in the caption of Table S1, etc. We think that this variable name is reasonable.

• **p 17**, **l 547** : What is meant by "diversity" of GA optimization?

Reply: This word "diversity" is one of the performance indices of an optimization algorithm and is used to show whether the algorithm explores solutions widely or not. It is important to confirm the diversity of the algorithm. On page 17 line 549, we confirmed it for our optimization results as, "It is clear that GA explored diverse solutions from MUC to JFK...".

• **p** 17, **l** 547–548 : we focus on the optimization results, which found the best solution  $\rightarrow$  we focus on the optimization setup which gave the best solution.

Reply: We believe that the word "optimization results" is appropriate here. We performed the optimizations for each combination of  $n_p$  (10, 20,…, 100) and  $n_g$  (10, 20,…, 100). Here, we say that we focus on the optimization case of  $n_p = 100$  and  $n_g = 100$ ; this case includes the best solution  $f_{\text{best}}$ . In fact, Fig. 9 shows the results obtained from this optimization case, which includes all solutions (10,000 trajectories, black lines) and the best solution (red line) explored by GA. Nevertheless, we modify the sentence by following the referee comment: on page 17 line 547–548, "To confirm the diversity of GA optimization, we focus on the optimization results, which found yielding the best solution...".

• **p 17, l 548** : "all the solutions" : Are these the 100 × 100 = 10000?

Reply: Yes. Figure 9 shows the 10,000 trajectories explored by GA. Related to this, we will correct the text "1,000" into "10,000" in the revised manuscript: in the captions of Figs. 9 (p 36), 14 (p 41), S1 (Supplementary material, p 1) and S2 (Supplementary material, p 2), "<del>1,000</del> **10,000** explored trajectories (solid line, black)...".

• **p 17**, **l 548–549** : solutions explored by GA as longitude vs altitude (top) and as location. This should be worded correctly.

Reply: We will modify the sentence in the revised manuscript: on page 17 line 548–549, "Figure 9 shows all the solutions explored by GA-as longitude vs altitude (top) and as location (bottom)."

• **p** 17, **l** 552 : "To confirm the difference" : I don't think confirm is appropriate to be used here.

Reply: We will revise the text: on page 17 line 552, "To confirm investigate the difference between the solutions,...".

• **p 17, l 554–555** : Isn't this conclusion too fast? What if the trajectory is not so zonal, but the trajectory crosses the equator at an angle of 45°: how would the CPs and regions around be defined?

Reply: We will reply to this issue in the section of "CP in trajectories (2)." We will add the text into the sentence to confine this conclusion with more precision: on page 17 line 554–555, "Therefore, GA is adequate for finding an optimal solution with sufficient accuracy **(in a strict sense, this conclusion is confined to the benchmark test)**."

• **p** 17 , **l** 552 : "confirm" is not appropriate here.

Reply: (The "p 17, line 552" means probably "p 17, line 557") We will change the word "confirm" into "analyze": on page 17 line 557, "To <del>confirm</del> **analyze** the dependence of...".

• **p** 17, **l** 552 : To confirm the dependence of optimal solutions on initial populations  $\rightarrow$  To "analyze" the dependence of "the" optimal solution on "the" initial population, ...

Reply: (The "p 17, line 552" means probably "p 17, line 557") We will revise the text: on page 17 line 557, "To <del>confirm</del> **analyze** the dependence of **the** optimal solution<del>s</del> on **the** initial **populations**,...".

• p 17, l 552–553 : I don't think one should use words like "best-of-generation".

Reply: (The "p 17, line 552–553" means probably "p 17, line 557–558") We will remove the word "best-of-generation" in the sentence: on page 17 line 557–558, "...Fig. 11 shows the best-of-generation flight time vs the number of objective function evaluations...". In the same way, we will remove the word "best-of-generation": on page 20 line 664, "...Fig. 17 shows the best-of-generation flight time vs the number of objective function evaluations...".

• **p 17, l 558–559** : corresponding to  $\rightarrow$  for.

Reply: We will modify the text: on page 17 line 558–559, "...function evaluations (=  $n_p \times n_g$ ) correspondingto for the 10 independent GA simulations...".

• **p 17**, **l 653** : "there is a small degree of variation in the objective function". Stated like this, it gives the impression that a different objective function is used. Probably, what is meant is that the value of the objective function for the final flight is different.

Reply: (The "p 17, line 653" means probably "p 17, line 563") By following the referee comment, we will revise the text: on page 17 line 563, "As indicated in Table S1, there is a small degree of variation in the objective function f (= flight time) the value of the objective function f (= flight time) is slightly different."

• **p** 17, **l** 564 : Writing  $f - f_{true}$  is a bit strange. For me, f and  $f_{true}$  are solutions, i.e. flights defined by  $x_1, ..., x_{11}$ . Here, f and  $f_{true}$  seem to indicate the value of the flight time.

Reply: f (and also  $f_{true}$ ) means the objective function value for a solution (i.e. a flight trajectory), which is defined by the eleven design variables  $x_1, x_2, \dots, x_{11}$ . As Eq. (28) defines, f (also  $f_{true}$ ) actually indicates the value of the flight time here.

• **p** 17, **l** 569 and 570 : "number of  $n_p$  and  $n_g$ " and "size of  $n_p$  and  $n_g$ ". One should use : "the value of  $n_p$ ", or "the size of the population", not something hybrid like "the number of  $n_p$ ".

Reply: We will modify the expression: on page 17 line 569, "With an **increased** in number of  $n_p$  and  $n_g$ , GA can discover **tends to find** an improved solution."

• **p 17, l 569** : "discover" : I suggest to use a different word.

Reply: We will change the word "discover" into "find" in the revised manuscript. In addition, we will modify the word "can" into "tends to" to show exactly the meaning of the sentence: on page 17 line 569, "With an **increased** in number of  $n_p$  and  $n_g$ , GA can tends to discover find an improved solution." As shown in Fig. 11, the optimal solution finally converges with increasing  $n_p$  and  $n_g$ . The word "can" seems to mean that the solution is improved unlimitedly. Therefore, we think that the word "tend to" is appropriate.

• **p** 17, **l** 570 : "is problem dependent, e.g. weather situations" : this should be formulated properly.

Reply: This sentence on line 570-571 seems to be confusing. We will modify the sentence: on page 17 line

570–572, "...the required size of  $n_p$  and  $n_g$  is problem-dependent, e.g. weather situations, and thereforeestimating appropriate  $n_p$  and  $n_g$  could be different. However, following a simple initial guess for  $n_p$  and  $n_g$  is a good starting point for their sizing."

• **p** 17, **l** 571 : "estimating appropriate *np* and *ng* could be different" : I suggest to formulate this differently.

Reply: We will reply to the comment in the above section: "p 17, l 570".

**Page 18**

• **p 18, l 573–574** : unclear sentence. What is, e.g., the difference between accuracy of GA optimizations and variation in the optimal solutions? I also had the impression that the impact of the initial population was already studied in Sect. 3.2.5.

Reply: The word "accuracy of GA optimizations" shows how close a solution converges to the true-optimal solution. On the other hand, a variation in optimal solutions is caused by different initial populations. Because GA is a stochastic optimization algorithm (not a deterministic optimization method, such as the gradient-based method). In addition, the impact of the initial population was studied in Sect. 3.2.5 regarding the results with " $n_p = 100$  and  $n_g = 100$ ." The impact also depends on  $n_p$  and  $n_g$  and is investigated in Sect. 3.2.6 in detail. Those results are necessary for the population and generation sizing.

• **p 18, l 574** : Skip "calculated".

Reply: We will remove the word "calculated" in the sentence: on page 18 line 574, "Figure 12 shows the calculated  $\Delta f$  and...".

• **p 18, l 581** : the variation of the  $\Delta f$  and the  $s_{\Delta f} \rightarrow \text{Skip}$  "the".

Reply: We will remove the word "the" in the sentence: on page 18 line 581, "Figure 13 shows the variation of the  $\Delta f$  and the  $s_{\Delta f}$  for all...".

• **p 18, l 582** : the  $\Delta f \rightarrow \text{Skip "the"}$ .

Reply: We will remove the word "the" in the sentence: on page 18 line 582, "The symbols and error bars in the figure correspond to the  $\Delta f$  and  $s_{\Delta f}$ ,...".

• **p 18, l 589** : that reduction  $\rightarrow$  a reduction.

Reply: We will correct the text: on page 18 line 589, "Similarly, that a reduction of 97 % can be achieved...".

• **p** 18, **l** 591 : "by selecting *np* and *ng* for different purposes." This should be formulated differently.

Reply: Values of  $\Delta f$  and  $s_{\Delta f}$  are the basis for selecting  $n_p$  and  $n_g$ . As described on page 18 line 586, the enlarged drawing in Fig. 13 shows that if one selects the number of function evaluations (=  $n_p \times n_g$ ) of 800, the large reduction of computational costs of 92 % can be achieved, keeping  $\Delta f$  less than 0.05 % ( $s_{\Delta f} \approx 0.02$  %), compared to the optimal solution by 10,000 function evaluations. For  $n_p \times n_g$  = 800, one can select any combination of  $n_p$  and  $n_g$ : for example,  $n_p$  = 10 and  $n_g$  = 80;  $n_p$  = 20 and  $n_g$  = 40 etc. A user makes his/her own choice on  $n_p$  and  $n_g$  by referring the values of  $\Delta f$  and  $s_{\Delta f}$ , as shown in Fig. 13. The formulae of  $\Delta f$  and  $s_{\Delta f}$  are described clearly in the caption of Fig. 13.

We will add this explanation to the revised manuscript: on page 18 line 586–589, "As shown in the enlarged drawing in Fig. 13, The enlarged drawing in Fig. 13 shows that if one selects the number of function evaluations (=  $n_p \times n_g$ ) of 800, the large reduction in number of function evaluations of computational costs of 92 % can be achieved, keeping  $\Delta f$  less than 0.05 % ( $s_{\Delta f} \approx 0.02$  %), compared to the optimal solution obtained by 10,000 function evaluations ( $n_p = 100$  and  $n_g = 100$ ). For  $n_p \times n_g = 800$ , one can

select any combination of  $n_p$  and  $n_g$ :  $n_p = 10$  and  $n_g = 80$ ;  $n_p = 20$  and  $n_g = 40$  etc. A user makes his/her own choice on  $n_p$  and  $n_g$  by referring the values of  $\Delta f$  and  $s_{\Delta f}$  shown in Fig. 13."

• **p 18, l 595** : for demonstrations  $\rightarrow$  for demonstration.

Reply: We will correct the text: on page 18 line 595, "…one-day AirTraf simulations were performed in EMAC (on-line) with the respective routing options for demonstration<del>s</del>."

• **p 18, l 596, 598** : Calculation conditions : too vague.

Reply: We will change the word "Calculation conditions" into "Simulation setup" in the revised manuscript: on page 18 line 596, "4.1 Calculation conditions **Simulation setup**". On page 18 line 598, "Table 7 lists the calculation conditions setup for the one-day simulations." On page 56 in the caption of Table 7, "Table 7. Calculation conditions Setup for AirTraf one-day simulations."

• **p 18, l 598–599** : simulation"s" and simulation.

Reply: We will correct the text: on page 18 line 598–599, "Table 7 lists the <del>calculation conditions</del> **setup** for the one-day simulations. The simulations <del>was</del> **were** performed for...".

• **p 18, l 605** : "On the other hand"  $\rightarrow$  "In addition".

Reply: We will change the word "On the other hand" into "In addition": on page 18 line 605, "<del>On the other hand</del> **In addition**, a single one-day simulation was...".

• **p 18, l 606–p19, l 607** : in [FL290, FL410] → in the range of ...

Reply: (The "p 19, line 607" means probably "p 18, line 607") We will add the text "the range of" in the revised manuscript: on page 18 line 606, "...altitude changes in **the range of** [FL290, FL410]."

• **p 18, l 607** : "and therefore" : I think *V*ground also varies for other reasons, e.g., due to varying wind speed and direction.

Reply: We just wanted to say here that the values of  $V_{TAS}$  and  $V_{ground}$  are different at every waypoint. We will modify the sentence: on page 18 line 607, "For the two options, the Mach number was set to M = 0.82 and therefore  $-V_{TAS}$  and  $V_{ground}$  varied along the waypoints the values of  $V_{TAS}$  and  $V_{ground}$  were different at every waypoint (Eqs. (24) and (25))."

**Page 19**

**p 19, l 615** : Does "case" refers to just one flight, or to all 103 flights together?

Reply: The "case" means the one-day simulation including all 103 flights. We will revise the sentence: on page 19 line 614–615, "The one-day simulation required approximately 15 min for **a the** great circle case **routing option**, while it took approximately 20 hours for **a the** time-optimal case **flight time routing option**."

• **p 19, l 616** : It is initially unclear what "it" refers to.

Reply: The word "it" means "the computational time." We will change the word "it" into "this time" in the sentence: on page 19 line 616, "...the computational time is consumed by the trajectory optimizations. Therefore it this time can be reduced by...".

• **p 19, l 617** : "right" : This is maybe not the most appropriate wording.

Reply: We will change the word "right" into "properly": on page 19 line 617, "…choosing **properly** all GA parameters <del>right</del>, using more PEs,…".

• **p 19, l 618** : by a small  $\rightarrow$  by "using" a small.

Reply: We will add the "using" in the text: on page 19 line 618, "...a large reduction in computing time of roughly 90 % can be achieved by **using** a small <del>number of</del>  $n_p$ ...".

• **p 19, l 618** : "a small number of  $n_p$ "  $\rightarrow$  "a small value of  $n_p$ ", or "a small population size"

Reply: We will modify the text: on page 19 line 618, "...a large reduction in computing time of roughly 90 % can be achieved by **using** a small number of  $n_p$ ...".

• **p 19, l 619** : with sufficient accuracy  $\rightarrow$  with "still" sufficient accuracy.

Reply: We will add the word "still" in the text: on page 19 line 619, "...and  $n_g$  with still sufficient accuracy of the optimizations."

• **p 19, l 620** : I think the title of Sect. 4.2 does not describe well the content : only one airport pair is discussed (Amsterdam - Minneapolis) really in depth. I suggest something more general.

Reply: In Sect. 4.2, we have focused on the results of three airport pairs and discussed the one. The rest is in the Supplementary material. To make the title more general, we will delete the word "three" in the title: on page 19 line 620, "4.2 Optimal solutions for <del>three</del> selected airport pairs."

• **p 19, l 623** : trajectories : Is meant the final trajectories?

Reply: Yes. The "trajectories" mean the optimized flight trajectories (final solutions). We will modify the sentence: on page 19 line 623, "...we classified <del>the</del> **those optimized flight** trajectories according to their altitude changes into three categories."

• **p 19, l 627** : we have selected "the" three airport pairs  $\rightarrow$  we have selected three airport pairs.

Reply: We will remove the word "the" in the sentence: on page 19 line 627, "We have selected <del>the</del> three airport pairs of..."

• **p 19, l 633** : in [FL290,FL410] → in the range of [FL290,FL410].

Reply: We will add the text "the range of" in the revised manuscript: on page 19 line 633, "...altitude changes in **the range of** [FL290, FL410]."

• **p 19, l 633–634** : "when calculating for the selected solutions" : This should be formulated better.

Reply: This text seems to be confusing. We will revise the text: on page 19 line 633–634, "Similar results were obtained when calculating for the selected solutions of Type I and III,...".

• **p 19, l 634** : in the supplements  $\rightarrow$  in the supplementary material.

Reply: We will change the text "in the supplements" into "in the supplementary material" in the revised manuscript: on page 19 line 634, "..., as shown in Figs. S1 and S2 in the Supplements Supplementary material." In the same way, we will modify the text: on page 17 line 562, "Table S1 in the Supplement Supplementary material shows a summary of...". On page 18 line 583, "...Table S2 in the Supplement

**Supplementary material**...". On page 20 line 657, "see Supplements Supplementary materials". On page 22 line 719, "are shown in the Supplement Supplementary material."

• **p 19, l 638–639** : east and west direction  $\rightarrow$  eastern and western directions.

Reply: We will revise the text: on page 638–639, "To calculate tail/head winds in east eastern and west western directions,...".

• **p 19, l 639** : major wind component : What is meant by this?

Reply: We just wanted to express the wind component, which has a dominant influence on the flight trajectory, to show the relation clearly between the wind fields and optimal flight trajectories. In fact, the contours in Fig. 15 show the zonal wind speed *u*; they do not include *v* and *w*.

• **p 19, l 640–641** : at the  $h \rightarrow$  at h.

Reply: We will modify the text: on page 19 line 640–641, "...direction at the departure time at the *h*."

**Page 20**

• **p 20, l 646** : Supplements → Supplementary material.

Reply: We will modify the text: on page 20 line 646, "...take advantages of the wind fields (see Supplementary Supplementary materials, Figs. S3 and S4)."

• **p 20, l 647** : the behaviour of altitude changes  $\rightarrow$  the behaviour of the altitude changes.

Reply: We will revise the text: on page 20 line 647, "To understand the behavior of **the** altitude changes of the optimal flight...".

• **p 20, l 647** : Fig. 16 plots → Fig. 16 shows.

Reply: We will revise the text: on page 20 line 647, "Fig. 16 <del>plots</del> **shows** the altitude distribution of the true air speed...".

• **p 20, l 650–651** : this means tail winds (≥ 1.0) and head winds (< 1.0) to the flight direction : Formulate better.

Reply: We will add the text to the sentence: on page 20 line 650–651, "...; this means tail winds  $((V_{ground} / V_{TAS}) \ge 1.0)$  and head winds  $((V_{ground} / V_{TAS}) < 1.0)$  to the flight direction."

• **p 20, l 655, 662** : "reflects"  $\rightarrow$  "takes into account", or "accounts for".

Reply: We will revise the word: on page 20 line 655, "...GA correctly reflects takes into account the weather conditions and...". On page 20 line 662, "...GA correctly reflects takes into account weather conditions for the...".

• **p 20, l 658** : confirmed  $\rightarrow$  compared. Skip "quantitatively".

Reply: We will revise the text: on page 20 line 658, "Next, we <del>confirmed</del> **compared** the resulting flight **times** <del>quantitatively</del> for the selected solutions."

• **p 20, l 659** : as indicated  $\rightarrow$  as shown.

Reply: We will revise the text: on page 20 line 659, "As indicated shown in Table 8...".

• **p 20, l 659–662** : decreased  $\rightarrow$  is lower.

Reply: We will revise the sentences: on page 20 line 659–662, "As indicated **shown** in Table 8, the flight time decreased is **lower** for the time-optimal case compared to the great circle cases. In addition, the flight time decreased is **lower** for the eastbound time-optimal flight trajectories compared to that for the westbound time-optimal flight trajectories."

• **p 20, l 664** : "sufficiently" : I think this is a bit vague.

Reply: (The "p 20, line 664" means probably "p 20, line 666") We will delete the word: on page 20 line 666, "...the solutions sufficiently converged to each optimal solution."

• **p 20, l 667** : that the reduction  $\rightarrow$  a reduction.

Reply: We will revise the text: on page 20 line 667, "It is also clear from Fig. 17 that the a reduction in...".

• **p 20, l 668** : "sizing" → "reducing" or "choosing properly".

Reply: We will revise the text: on page 20 line 668, "...the **a** reduction in computing time can be achieved by sizing choosing properly  $n_p$  and  $n_g$ ...".

• **p 20, l 671** : This is not a nice first sentence for a paragraph.

Reply: We will modify the sentence: on page 20 line 671, "**Next, the** one-day AirTraf simulations results for 103 trans-Atlantic flights are discussed analyzed."

• **p 20, l 673–674** : trans-Atlantic Ocean  $\rightarrow$  Atlantic ocean.

Reply: We will remove the word "trans-" in the text: on page 20 line 673–674, "…flight trajectories congregated around 50° N over the trans-Atlantic Ocean to take advantage…".

• **p 20, l 675** : of "the" region  $\rightarrow$  of "that" region.

Reply: We will revise the text: on page 20 line 675, "...the westbound time-optimal flight trajectories were located to the north and south of the that region...".

**Page 21**

**p 21, l 681** : plot  $\rightarrow$  show.

Reply: We will revise the text: on page 21 line 681, "Figures 19a and 19b <del>plot</del> **show** the...".

• **p 21, l 683** : with linear fitted lines : be more precise.

Reply: We will modify the text: on page 21 line 683, "...with linear fitted lines fitted by the Least Squares algorithm." Related to this issue, we will also modify the text: on page 18 line 586, "...<del>least-squares</del>Least Squares algorithm...".

• **p 21, l 683** : increased  $\rightarrow$  is higher.

Reply: We will revise the text: on page 21 line 683, "Figure 19a shows that  $V_{TAS}$  increased is higher at low altitudes."

• **p 21, l 688–689** : which had high  $V_{\text{TAS}}$  values  $\rightarrow$  with high  $V_{\text{TAS}}$  values.

Reply: We will revise the text: on page 21 line 688–689, "GA successfully found the flight trajectories<del>, which had</del> with high  $V_{\text{TAS}}$  values, as time-optimal flights."

• **p 21**, **l 691** : increases  $\rightarrow$  is larger.

Reply: We will revise the text: on page 21 line 691, "...time-optimal case (solid line, red) increases is larger between...".

• **p 21, l 696** : increases  $\rightarrow$  is larger.

Reply: We will revise the text: on page 21 line 696, "...time-optimal case (solid line, blue) is distributed widely in altitude and increases is **larger** between".

• **p 21, l 700** : Supplement → Supplementary material.

Reply: We will modify the text: on page 21 line 700, "...is shown in the Supplementary material (Fig. S7)...".

• **p 21, l 703** : correctly selected the airspace : improve this formulation.

Reply: We will modify the sentence: on page 21 line 703, "Therefore, GA correctly selected the airspace by altitude changes, where  $V_{ground}$  values increased the trajectories found by GA through altitude changes passed areas, which correctly lead to larger  $V_{ground}$ ."

• **p 21, l 705** : This behaviour of altitude changes  $\rightarrow$  These altitude changes.

Reply: We will revise the text: on page 21 line 705, "This behavior of These altitude changes affects the...".

• **p 21, l 705** : affects the variation in fuel consumptions  $\rightarrow$  affects the fuel consumption.

Reply: We will revise the text: on page 21 line 705, "**These** altitude changes affects the <del>variation in</del> fuel consumptions...".

• **p 21, l 705** : the terms are used interchangeably to mean fuel flows : improve the formulation.

Reply: We will improve the text: on page 21 line 705, "...affects the variation in fuel consumptions (the terms are is used interchangeably to mean fuel flows)."

• **p 21, l 708** : increases  $\rightarrow$  is higher.

Reply: We will revise the text: on page 21 line 708, "The results show that the fuel consumption increases is higher at low altitudes...".

• **p** 21, **l** 708 : the mean value  $\rightarrow$  the mean value of the fuel consumption.

Reply: We will revise the text: on page 21 line 709, "In addition, the mean value **of the fuel consumption** for the time-optimal case is high...".

• **p 21**, **l 714** : increases  $\rightarrow$  is higher.

Reply: We will revise the text: on page 21 line 714, "...the mean value for the eastbound time-optimal case <del>increases</del> **is higher** owing to its low mean flight altitude...".

• **p 22, l 718** : corresponding to "the 103" individual flights.

Reply: We will revise the text: on page 22 line 718, "Figure 21 shows the flight time corresponding to **the 103** individual flights...".

• **p** 22, **l** 718–719 : the similar figures  $\rightarrow$  similar figures.

Reply: We will revise the text: on page 22 line 718–719, "...(the similar figures for the fuel use,  $NO_x$  and  $H_2O$  emissions are shown...".

• **p 22, l 720** : showed  $\rightarrow$  show.

Reply: We will revise the text: on page 22 line 720, "The results showed that all symbols...".

• **p** 22, **l** 720 : in the right-hand domain : choose a better expression.

Reply: We will rephrase the text: on page 22 line 720, "...all symbols lay in the right-hand domain on the right side of the 1:1 solid line."

• **p** 22, **l** 721 : decreases  $\rightarrow$  is lower. Put "for all airport pairs" at the end of the sentence.

Reply: We will revise the sentence: on page 22 line 720, "...the flight time for the time-optimal flights <del>decreased</del> **is lower** for all airport pairs compared to that for the great circle flights **for all airport pairs**."

• **p** 22, **l** 723 and 725 : increased  $\rightarrow$  increases.

Reply: We will revise the sentences: on page 22 line 722–725, "The total value <del>was</del> is certainly minimal for the time-optimal case, while in relative terms the value <del>increased</del> **increases** by +1.5 %, +2.5 %, +2.9 % and +2.9 % for the great circle cases at FL290, FL330, FL370 and FL410, respectively. Regarding the total value of fuel use, Table 11 indicates that the value <del>increased</del> **increases** by +5.4%...".

• **p 22, l 740–741** : "Consistency" : just by reading the section title, it is not clear what is meant by this.

Reply: We will change the section title: on page 22 line 740, "5 Consistency check for **Verification of** the AirTraf simulations."

• **p** 22, **l** 742 : were  $\rightarrow$  are.

Reply: We will revise the text: on page 22 line 742, "...the one-day simulation results described in Sect. 4 were **are** compared to reference data...".

• **p** 22, **l** 742–743 : The data → Data.

Reply: We will revise the text: on page 22 line 742–743, "The dData obtained under similar conditions...".

• **p 22, l 744** : "they" is ambiguous.

Reply: We will revise the text: on page 22 line 742-744, "The dData obtained under similar conditions (aircraft/engine types, flight conditions, weather situations, etc.) were selected for the comparison, although they the conditions are not completely the same as the calculation conditions for the one-day simulations."

**p** 23, l 723 : I would not say explicitly that the table shows "a comparison".

Reply: (The "p 23, line 723" means probably "p 23, line 749") We will revise the text: on page 23 line 749, "...Table 12 shows a comparison of the flight time between **for** the seven time-optimal flight trajectories simulated by AirTraf and three reference data...".

• **p 23, l 758 and 764** : literature → write the correct reference.

Reply: We will revise the text: on page 23 line 758, "...(see Fig. 3 in the literature **Irvine et al. (2013)**)." On page 23 line 764, "...(see Tables 2 and 3 in the literature **Grewe et al. (2014a)**)."

• **p** 23, **l** 758 : indicated  $\rightarrow$  indicates.

Reply: This part will be deleted. Please see the reply to the comment below: "p 23, 1758–759 / 764–765".

• p 23, l 758–759 / 764–765 : Is it worth mentioning this?

Reply: As the referee pointed out, those sentences are not necessary here. Therefore, we will revise the sentences: on page 23 line 758–759, "This indicated that the flight time increased for westbound flights on the trans-Atlantic region in winter due to westerly jet streams.". On page 23 line 764–765, "This also indicated the increased flight time for westbound trans-Atlantic flights in winter due to westerly jets streams.". Related to this issue, we will modify the text: on page 23 line 765–767, "The magnitude in flight times of between the seven airport pairs is are close to the reference data and the variation shows a good agreement with the trend of the increased flight times for westbound trans-Atlantic flights in winter due to westerly jet streams, as indicated from the reference data."

• **p 23**, **l 764** : "indicate" : I don't think this is the appropriate word.

Reply: This part will be deleted. Please see the reply to the comment above: "p 23, 1758–759 / 764–765".

• **p 23, l 765** : "close" → "close to".

Reply: We will revise the text: on page 23 line 765, "The magnitude in flight times of **between** the seven airport pairs is are close to the reference data and the variation shows...".

• **p** 23. **l** 769 : reference data  $\rightarrow$  the reference data.

Reply: We will revise the text: on page 23 line 768, "...using the mean fuel consumption value of 103 flights and **the** reference data,...".

• **p** 23, **l** 774 : indication : shouldn't one use a different word?

Reply: We will remove the word "indication" and revise the text: on page 23 line 774, "...the **overall** load factor of the worldwide air traffic indication in 2008 was used (Table 1)."

• **p 23, l 778** : decreased  $\rightarrow$  is lower.

Reply: We will revise the text: on page 23 line 778, "Table 13 shows that the obtained mean  $EINO_x$  value decreased is lower at high altitudes...".

• **p** 23, **l** 783 : installed  $\rightarrow$  contains.

Reply: We will revise the text: on page 23 line 783, "The 2GE051 installed **utilizes** the new 1862M39 combustor,...".

**Page 24**

• **p 24, l 787** : "duplicates" : What is meant by this?

Reply: We just wanted to say here as, "estimates" or "simulates." We will revise the text: on page 24 line 787, "AirTraf <del>duplicates real</del> **simulates realistic** fuel consumptions...".

• **p 24, l 790** : for 103 flights → for "the" 103 flights.

Reply: We will revise the text: on page 24 line 790, "Here the obtained  $m_1$  and  $m_{nwp}$  for **the** 103 flights were compared...".

• **p** 24, **l** 792 : safety flight operations  $\rightarrow$  flight operations safety.

Reply: We will revise the text: on page 24 line 791, "...to provide safety flight operations safety, and...".

• **p 24, l 794** : constrains to  $\rightarrow$  constraints

Reply: We will revise the text: on page 24 line 794, "…no model that **constrains** to the structural <del>limit</del> weight<del>s</del> **limits** was included in AirTraf."

• **p 24, l 800–801** : This sentence should be improved.

Reply: We will improve the sentence: on page 24 line 800–801, "For these 15 flights, actual flight planning data probably indicate altitude changes (generally higher flight altitudes) to increase a the fuel mileage, which decreases leading to the decrease in  $m_1$ ."

• **p** 24, **l** 802 : to prevent "the" structural damage  $\rightarrow$  to prevent structural damage.

Reply: We will revise the text: on page 24 line 802, "To prevent the structural damage to the landing gear...".

• **p 24, l 803** : "aircraft has" → "aircraft have" or "an aircraft has".

Reply: We will revise the text: on page 24 line 803, "...an aircraft has to reduce the total weight...".

• **p 24, l 803** : "to reduce below" → "to reduce until" or "to bring below".

Reply: We will revise the text: on page 24 line 803, "...**an** aircraft has to reduce the total weight <del>below</del> **until** MLW prior to landing."

• **p 24, l 808** : Why not using ≤?

Reply: We will revise the text: on page 24 line 807, "This always satisfies the third constraint ZFW  $\leq \leq$  MZFW."

• **p 24, l 806, 810** : of A330-301 → of an A330-301 aircraft.

Reply: We will revise the word in the revised manuscript: on page 24 line 806, "The MZFW of **an** A330-301 **aircraft** is...". On page 24 line 810, "...minimum operational weight of **an** A330-301 **aircraft** in the...".

• **p 24, l 812** : more  $\rightarrow$  higher.

Reply: We will revise the text: on page 24 line 812, "..., all the  $m_{nwp}$  (open circle) were more higher than the MLOW."

• **p 24, l 814** : Skip "calculations".

Reply: We will remove the word "calculations": on page 24 line 814, "...AirTraf simulates fairly good fuel use calculations."

• **p** 24, **l** 816 : an submodel  $\rightarrow$  a submodel.

Reply: We will revise the text: on page 24 line 816, "AirTraf is published for the first time as an **a** submodel of the Modular Earth Submodel System...".

• **p 24, l 817** : "applied" : shouldn't it be "used"?

Reply: We will revise the text: on page 24 line 817, "The MESSy is continuously further developed and applied **used** by a consortium of institutions."

**Page 25**

**p 25, l 823–824** : What is meant by this sentence?

Reply: This sentence is not necessary for our argument here. Therefore, we will delete the sentence: on page 25 line 822–825, "Some improvements will be performed and AirTraf 1.0 will be updated for the latest version of the code. For example, evaluation functions corresponding to the NO\*, H2O, fuel, contrail and CCF routing options will be added. The status information for AirTraf including the licence conditions **is** available at the website."

• **p 25, l 829** : the benchmark test  $\rightarrow$  a benchmark test.

Reply: We will revise the text: on page 25 line 829, "First, the a benchmark test was performed...".

• **p 25, l 831–832** : by other published code : this is too vague.

Reply: We will revise the text: on page 25 line 831–832, "...calculated by other published code MTS."

• **p 25**, **l 832** : the benchmark test  $\rightarrow$  a benchmark test.

Reply: We will revise the text: on page 25 line 832, "Second, the a benchmark test was performed...".

• **p 25, l 836** : dependence on the initial population.

Reply: We will revise the text: on page 25 line 836, "The dependence of the optimal solutions on **the** initial *populations* was investigated...".

• **p 25, l 835 and 838** : The fact that both values are 0.01 % is maybe not a good sign. I would think that you want the second one to be much smaller than the first one.

Reply: The referee pointed out a very important issue. However, these values are sufficiently small and the

performance of GA is well enough to find an optimal solution. In fact, we showed in Fig. 21 that GA found the trajectories for all airport pairs; the trajectories could decrease flight time compared to the great circle flights. This performance is sufficient for our purpose. In fact, the second "0.01 %" is actually smaller than what we expected. As replied to the referee comment in the section of "p 18, l 573–574", GA is a stochastic optimization algorithm. Hence, optimal solutions calculated from different initial populations are not always the same.

Regarding the performance of GA, Deb, K., (1991) reported that "the welded beam structure is a practical design problem (minimization of the total cost *f*) that is often used as a bench-mark problem in testing different optimization techniques." Rekliatis, G. V., et al., (1983) studied this test and reported the optimal solution of  $f^* = 2.38$ . Deb, K., (1991) performed 3 independent GA calculations with different initial populations to this problem: the obtained (optimal) solution was f = 2.43 (the best among the three solutions), f = 2.59 and f = 2.49. The difference in the total cost between the *f* (the best solution: 2.43) and  $f^*$  was  $\Delta f = f - f^* = 0.05$  (2.1% of  $f^*$ ).  $\Delta f$  also ranged from 0.05 to 0.21 (2.1 to 8.8% of  $f^*$ ). This shows that both values "0.01%" are indeed small. Of course, the performance of GA largely depends on the optimization problem and GA parameters. Therefore, we analyzed the performance on our trajectory optimization problem with our setting in Sects. 3.2.4 and 3.2.5.

**[Reference]**

Rekliatis, G. V., et al., Engineering Optimization Methods and Applications, Wiley, New York, 1983. Deb, K., "Optimal design of a welded beam via genetic algorithms," AIAA Journal, 29 (11), 1991.

**Page 26**

• **p 26, l 860 and 866** : Please be more specific about what "reference data" is.

Reply: We will revise the text: on page 26 line 860, "The consistency of the one-day simulations was verified with reference data **(published in earlier studies and BADA)** of flight time...". On page 26 line 865, "The mean EINOx values were in the same range as the reference  $\frac{data}{data}$  values of earlier studies."

• **p 26, l 862** : close  $\rightarrow$  (very) similar.

Reply: We will revise the text: on page 26 line 862, "...the reference data showed that the values were <del>close</del> **similar** and...".

• **p 26, l 869** : fuel use calculation model  $\rightarrow$  fuel use model.

Reply: We will revise the text: on page 26 line 869, "Thus, AirTraf comprises a sufficiently good fuel use calculation model."

• **p 26, l 871** : "is sufficient" : But some things do not work yet?

Reply: We will revise the sentence: on page 26 line 871, "AirTraf 1.0 is sufficient to investigate a reduction potential of aircraft routings on air traffic climate impacts is ready for more complex routing tasks."

• **p 26, l 871** : "a" reduction potential  $\rightarrow$  "the" reduction potential.

Reply: This part will be deleted. Please see the reply to the comment above: "p 26, l 871".

**4 Remarks on figures**

• **Figure 1** : I presume parts of this are already done in other optimized studies. Mention what is already done, what is part of this manuscript, and what shall be done in the future.

Reply: By following the comment (2) of the referee #1, we will remove Fig. 1 (on page 29).

**Figure 7** : Bizarre first sentence in caption. Consisting of  $\rightarrow$  determined by.  $\Delta \lambda_{airport} \rightarrow \Delta \lambda_{airport}$ .

Reply: We will revise the caption: on page 34 in the caption of Fig. 7, "Geometry definition of flight trajectory as longitude vs altitude in the vertical cross-section (top) and as geographic location projected onto the Earth (bottom). The bold solid line indicates a trajectory from MUC to JFK. •: control points consisting of determined by design variables....which divide the  $\Delta \lambda_{airport} \Delta \lambda_{airport}$  into four equal parts...the coordinates divide the  $\Delta \lambda_{airport} \Delta \lambda_{airport}$  into six equal parts...

• **Figure 8** : Conclusions/observations/interpretations should not be written in figure captions. I would not use the word "discovers".

Reply: We will revise the caption: on page 35 in the caption of Fig. 8, "Figure 8. Optimal solutions are shown varying with the **population** size  $n_p$  and **the number of generations** number  $n_g$ .  $\Delta f$  means the difference in flight time between the optimal solution f and the true-optimal solution  $f_{true}$  (= **25,994.0 s**). The  $\Delta f$  (in %) is calculated as  $(\Delta f / f_{true}) \times 100$ . GA discovers the solutions as close to the  $f_{true}$  (= **25,994.0 s**) with increasing  $n_p$  and  $n_g$ . For each  $n_p$ , the optimal solution shows minimum flight time for  $n_g$  = 100. For each  $n_g$ , the optimal solution shows minimum flight time for  $n_p$  = 100 and  $n_g$  = 100,  $\Delta f < 3.0$  s (less than 0.01 %))."

• **Figure 9** : Change the first sentence. "The population size  $n_p = 100 \dots$ " : This is not a good sentence. Replace "=" by "is".

Reply: We will revise the caption: on page 36 in the caption of Fig. 9, "1,00010,000 explored trajectories (solid line, black) from MUC to JFK as longitude vs altitude in the vertical cross-section (top) and as location projected onto the Earth (bottom). The *population* size  $n_p = 100 n_p$  is 100 and the number of *generations* number  $n_g = 100 n_g$  is 100." In the same way, we will revise the caption: on page 38 in the caption of Fig. 11, "The *population* size  $n_p = 100 n_p$  is 100 and the number  $n_g = 100 n_g$  is 100." On page 44 in the caption of Fig. 17, "The *population* size  $n_p = 100 n_p$  is 100 and the number of *generations* number  $n_g = 100 n_g$  is 100."

• **Figure 10** : Skip "Comparison of".

Reply: We will remove the word "Comparison of" in the caption: on page 37 in the caption of Fig. 10, "Figure 10. <del>Comparison of t</del>Trajectories for the best solution (red line) and the true-optimal solution (dashed line, black)."

• **Figure 11** : Don't use expressions like "Best-of-generation". "vs function evaluations" → "vs number of function evaluations". *f*true → *f*true. Change "On the ... and ...".

Reply: We will revise the caption: on page 38 in the caption of Fig. 11, "Best-of-generation fFlight time vs **number of** function evaluations...and the true-optimal solution  $f_{true}f_{true}$ ...is calculated as  $(\Delta f / f_{true}f_{true})$ ...".

• **Figure 17** : Don't use expressions like "Best-of-generation".

Reply: We will revise the caption: on page 44 in the caption of Fig. 17, "Best-of-generation fFlight time (in %) vs **number of** function evaluations...".

• **Figure 22** : Shouldn't one have as unit for the emissions : kg(fuel) m-2s-1? The figures are 2-hourly averages. However, the ranges are not clear from just mentioning 14:00:00, 16:00:00, 18:00:00, 20:00:00. Is it 14:00:00–16:00:00, 16:00:00–18:00:00, ..., or rather 12:00:00–14:00:00, 14:00:00–16:00:00, ...

Reply: By following the comment (8) of the referee #1, we will remove Fig. 22 (on page 49).

**5** Comments on tables**

**Table 1** 101.325  $\rightarrow$  101,325. Why is there "(jet)" at the end of the line with  $C_{f1}$ ? There should be a small space between "kg" and "min". I would not give a variable the name "Oneday".  $P_0$  and  $T_0$  are not total pressure or temperature, but reference pressure and temperature.

Reply: Thank you so much. We will correct the value: on page 51 at the line with  $P_0$  in Table 1, "101.325 101,325." Eurocontrol, 2011 publishes the thrust specific fuel consumption coefficient for jet, turboprop and piston engines. The word "jet" means "jet engines". We will modify the line: at the end of the line with  $C_{f_1}$ , "...(jet **engines**)a". As the referee pointed out, we will add a space between "kg" and "min": at the line with  $C_{f_1}$ , "**kg** min-1**k**N-1." Regarding the variable name "Oneday", please see the reply to the referee comment on "p 9, l 287." In addition, we will correct the word on  $P_0$  and  $T_0$ : at the line with  $P_0$  and  $T_0$ , "Total Reference pressure" and "Total Reference temperature".

• **Table 2** :  $n_{wp-1} \rightarrow n_{wp} - 1$ .

Reply: Thank you so much. We will correct the text: on page 52 in the caption of Table 2, "..., flight segments  $(i = 1, 2, ..., n_{wp-1} n_{wp} - 1)$ ."

• **Table 4** : I think it makes no sense to introduce all these new variable names. Put in the heading (first row) of the table just : "Eq. (22)", "Eq. (23)", ...

Reply: We understand the referee comment. Nevertheless, we are hesitating to change those variable names. We define the variable name for a flight distance as "d", as shown in Table 2, and we use the variable "d" consistently in the manuscript: on page 11 Eqs. (22) and (23), on page 15 Eq. (28), etc. We think that the current expressions are reasonable.

• **Table 5** : For population size and generation number : " $\cdots$ "  $\rightarrow$  "...".

Reply: We will modify and add the variable names " $n_p$ " and " $n_g$ " in the Table: on page 55 in Table 5, "Population size,  $n_p$ , 10,20,...,100" and "Generation number**Number of generations**,  $n_g$ , 10,20,...,100". This reply is related to the reply to "p 17, l 540." In addition, we will add the text at the lines with "parameter" and "design variable": on page 55 in Table 5, "Parameter; **Description**"; and "Design variable,  $n_{dv}$ , 11 **(6 locations and 5 altitudes)**."

• **Table 9** : "that of" → "average of". Why "medium"?

Reply: We will modify the caption of Table 9: on page 58 in the caption of Table 9, "Eastbound: mean value **average** of 52 eastbound flights; Westbound: that **average** of 51 westbound flights; and Total: that **average** of 103 flights." In the same way, we will modify the caption of Table 10: on page 58 in the caption of Table 10, "Eastbound: mean value **average** of 52 eastbound flights; Westbound: that **average** of 51 westbound flights; and Total: that **average** of 52 eastbound flights." Similarly, we will modify the caption of Table 11: on page 59 in the caption of Table 11, "Eastbound: sum of 52 eastbound flights; Westbound: that **sum** of 51 westbound flights; and Total: that **sum** of 103 flights."

(The "Why "medium"?" is probably the comment for Table 10) We will delete the word "medium" in the caption of Table 10: on page 58 in the caption of Table 10 "..., which is the <del>medium</del> value between...".

• **Table 12** : Skip "Comparison of".

Reply: We will remove the word "Comparison of" in the caption: on page 60 in the caption of Table 12, "<del>Comparison of t</del>The flight time for time-optimal flight trajectories from one-day AirTraf simulations...".

• **Table 14** : "Constraints on"  $\rightarrow$  "Constraints from". Why not just using  $\geq$  and  $\leq$ ? Why have on all the four

lines A330-301 after some "." at the end of the line?

Reply: We will revise Table 14: on page 62 in the caption of Table 14, "Constraints on **from** the structural limits weight limits (MTOW, MLW and MZFW) and one specified limit weight limit (MLOW)...". In column 1, " $m_1 \leq MTOW$ ;  $m_{nwp} \leq MLW$ ; Zero fuel weight  $\leq MZFW$ ; and  $m_{nwp} \geq MLOW$ ." In column 3, "Maximum take-off weight. A330-301; Maximum landing weight. A330-301; Maximum zero fuel weight. MZFW = OEW + MPL. A330-301; and Planned minimum operational weight in the international standard atmosphere.b MLOW = 1.2 × OEW. A330-301.b"

Related to this, we will change the word "limit weights" into "weight limits" in the revised manuscript: on page 24 line 791, "three structural <del>limit</del> weights **limits**..."; on page 24 line 792, "...,and one specified <del>limit</del> weight **limit**..."; on page 24 line 793, "...and the four <del>limit</del> weights **limits**..."; on page 24 line 794, "...constrains to the structural <del>limit</del> weights **limits**..."; on page 24 line 797, "...with the <del>limit</del> weights-**limits**..."; on page 26 line 867, "...the three structural <del>limit</del> weights **limits** and one specified <del>limit</del> weight **limit** of..."; on page 26 line 868, "...the values satisfied the four <del>limit</del> weights **limits** and..."; on page 50 in the caption of Fig. 23, "Comparison of aircraft weights with structural <del>limit</del> weights **limits** (MTOW and MLW) and one specified <del>limit</del> weight **limit** (MLOW)"; on page 62 in column 2 of Table 14, "<del>Limit w</del>Weight **limit**, kg".

**6 Supplementary material**

• Fig. S1 and S2 : including "the" time-optimal flight trajectories.

Reply: We will add the word "the" in the caption: on page 1 (Supplementary material) in the caption of Fig. S1, "...(bottom), including **the** time-optimal flight trajectories...". On page 2 (Supplementary material) in the caption of Fig. S2, "...(bottom), including **the** time-optimal flight trajectories...". In the same way, we will add the word: on page 41 in the caption of Fig. 14, "...(bottom), including **the** time-optimal flight trajectories...".

• Fig. S3 and S4 : Skip "Comparison of".

Reply: We will remove the word "Comparison of" in the caption: on page 3 (Supplementary material) in the caption of Fig. S3, "<del>Comparison of t</del>Trajectories for the time-optimal...". On page 3 (Supplementary material) in the caption of Fig. S4, "<del>Comparison of t</del>Trajectories for the time-optimal...". In the same way, we will remove the word: on page 42 in the caption of Fig. 15, "<del>Comparison of t</del>Trajectories for the time-optimal...".

• Fig. S7 : Skip "that".

Reply: We will remove the word "that" in the caption: on page 6 (Supplementary material) in the caption of Fig. S7, "Linear fits of the time-optimal (solid line, red (eastbound) and blue (westbound)) and that of the great circle...". In the same way, we will remove the word: on page 46 in the caption of Fig. 19, "Linear fits of the time-optimal (solid line, red (eastbound)) and blue (westbound)) and that of the great circle...".

Manuscript prepared for Geosci. Model Dev. with version 2014/07/29 7.12 Copernicus papers of the LATEX class copernicus.cls. Date: 19 July 2016

**Climate Assessment Platform of Different Aircraft Routing Strategies Air Traffic Simulation** in the Chemistry-Climate Model EMAC 2.41: AirTraf 1.0**

Hiroshi Yamashita1, Volker Grewe1,2, Patrick Jöckel1, Florian Linke3, Martin Schaefer4</sup